# A proxy modelling approach to assess the potential of extracting ENSO signal from tropical Pacific planktonic foraminifera

Brett Metcalfe[1,2], Bryan C. Lougheed[1,3], Claire Waelbroeck[1] and Didier M. Roche[1,2]

[1]Laboratoire des Sciences du Climat et de l'Environnement, LSCE/IPSL, CEA-CNRS-UVSQ, Université Paris-Saclay, F-91191 Gif-sur-Yvette, France
[2]Earth and Climate Cluster, Department of Earth Sciences, Faculty of Sciences, VU University Amsterdam, de Boelelaan 1085, 1081 HV, Amsterdam, The Netherlands
[3]Department of Earth Sciences, Uppsala University, Villavägen 16, 75236 Uppsala, Sweden

*Correspondence to*: Brett Metcalfe (brett_metcalfe@outlook.com)

**A complete understanding of past El Niño-Southern Oscillation (ENSO) fluctuations is important for the future predictions of regional climate using climate models. One approach to reconstructing past ENSO dynamics uses planktonic foraminifera as recorders of past climate to assess past spatiotemporal changes in upper ocean conditions. In this paper, we utilise a model of planktonic foraminifera populations, *Foraminifera as Modelled Entities* (FAME), to forward model the potential monthly average $\delta^{18}O_c$ and temperature signal proxy values for *Globigerinoides ruber*, *Globigerinoides sacculifer* and *Neogloboquadrina dutertrei* from input variables covering the period of the instrumental record. We test whether the modelled foraminifera population $\delta^{18}O_c$ and $T_c$ associated with El Niño events statistically differ from the values associated with other climate states. Provided the assumptions of the model are correct, our results indicate that the values of El Niño events can be differentiated from other climate states using these species. Our model computes the proxy values of foraminifera in the water, suggesting that, in theory, water locations for a large portion of the Tropical Pacific should be suitable for differentiating El Niño events from other climate states. However, in practice it may not be possible to differentiate climate states in the sediment record. Specifically, comparison of our model results with the sedimentological features of the Pacific Ocean shows that a large portion of the hydrographically/ecologically suitable water regions coincide with low sediment accumulation rate at the sea floor and/or of sea floor that lie below threshold water depths for calcite preservation.**

## 1. Introduction

### 1.1 El Niño-Southern Oscillation (ENSO)

Predictions of short-term, abrupt changes in regional climate are imperative for improving spatiotemporal precision and accuracy when forecasting future climate. Coupled ocean-atmosphere interactions (wind circulation and sea surface temperature) in the tropical Pacific, collectively known as the El Niño-Southern Oscillation (ENSO) on interannual timescales and the Pacific Decadal Oscillation on decadal timescales, represent global climate's largest source of inter-

annual climate variability (Wang et al., 2017) . Due to ENSO's major socio-economic impacts upon pan-Pacific nations, which, depending on the location, can include flooding, drought and fire risk, it is imperative to have an accurate understanding of both past and future behaviour of ENSO (Trenberth and Otto-Bliesner, 2003; Rosenthal and Broccoli, 2004; McPhaden et al., 2006). The instrumental record of the past century provides important information (that can be translated into the Southern Oscillation Index; SOI), however, detailed oceanographic observations of the components of ENSO (both the El Niño and Southern Oscillation), such as the Tropical Oceans Global Atmosphere (TOGA; 1985-1994) experiment only provide information from the latter half of the twentieth century (Wang et al., 2017). To acquire longer records, researchers must turn to the geological record using various archives that are available from the (pan-)Pacific region. An integrated approach combining palaeoclimate proxies (Ford et al., 2015; Garidel-Thoron et al., 2007; Koutavas et al., 2006; Koutavas and Joanides, 2012; Koutavas and Lynch-Stieglitz, 2003; Leduc et al., 2009; White et al., 2018) and computer models (Zhu et al., 2017a) can help shed light on the triggers of past ENSO events, their magnitude and their spatiotemporal distribution.

## 1.2 Foraminiferal Proxies

The simulation of past ENSO using climate models has been fraught with difficulties due to ENSO's integration into the climate system and the associated feedbacks of ENSO upon model boundary conditions (*e.g.*, SST, $pCO_2$) (Ford et al., 2015). One way to deduce the relative impact and importance of various feedbacks and, in turn, reduce model-dependent noise in our predictions, is to compare model output with proxy data such as foraminifera. Such an approach, however, requires an abundance of reliable spatiotemporal proxy data for the entire Pacific Ocean. The reliability of proxy reconstructions are themselves subject to several unknowns, uncertainties and biases, for instance culture experiments have identified temperature (Lombard et al., 2009, 2011), light (Bé et al., 1982; Bé and Spero, 1981; Lombard et al., 2010; Rink et al., 1998; Spero, 1987; Wolf-Gladrow et al., 1999), carbonate ion concentration ($[CO_3^{2-}]$) (Bijma et al., 2002; Lombard et al., 2010) and ontogenetic changes (Hamilton et al., 2008; Wycech et al., 2018) as variables that drive, alter or induce changes in foraminiferal growth. These variables are important as foraminifera are not passive recorders of environmental conditions such as SST, in that the very ambient environment that researchers wish to reconstruct can modify the foraminiferal population (Mix, 1987; Mulitza et al., 1998). Sensitivity to the variable being reconstructed may increase or decrease the relative contribution of individual ENSO events, due to modulation of the flux to the seafloor, increasing or decreasing the chance of sampling such occurrences, etc. (Mix, 1987; Mulitza et al., 1998). Computation of the influence of biological and vital effects upon physiochemical proxies, such as those based on foraminifera should be a fundamental consideration for any accurate data-model comparison. Recent attempts at circumnavigating proxy related problems have employed isotope-enabled models (Caley et al., 2014; Roche et al., 2014; Zhu et al., 2017a), proxy system models (Evans et al., 2013; Dolman and Laepple, 2018; Jonkers and Kučera, 2017; Roche et al., 2018) or uncertainty analysis (Thirumalai et al., 2013; Fraass and Lowery, 2017; Dolman and Laepple, 2018) to predict both the potential $\delta^{18}O_c$ values in foraminifera and/or the probability of detection of a climatic event. The use of ecophysiological models (Kageyama et al., 2013; Lombard et al.,

2009, 2011) can help circumvent some of the problems associated with a purely mathematical approximation (e.g., Caley et al., 2014) of the translation of an ambient signal into a palaeoclimate proxy. They are not limited to foraminifera and can provide an important way to test whether proxies used for palaeoclimate reconstructions are suitable for the given research question. Several studies have investigated the response of planktonic foraminifera from core material or computed pseudo foraminiferal distributions, their proxy values, and the resultant (likely) distribution of these proxy values with respect to ENSO (e.g., Leduc et al., 2009; Thirumalai et al., 2013; Ford et al., 2015; Zhu et al., 2017).

## 1.3 Aims and Objectives

Here, we investigate whether living planktonic foraminifera can be theoretically used in ENSO reconstructions, differing from previous research (e.g., Thirumalai et al., 2013) by using a foraminiferal growth model, *Foraminifera as modelled entities* (FAME; Roche et al., 2018), to tackle the dynamic seasonal and depth habitat of planktonic foraminifera (Wilke et al., 2006; Steinhardt et al., 2015; Mix, 1987; Mulitza et al., 1998). To be a useful proxy for the reconstruction of ENSO, the resulting proxy values of populations of planktonic foraminifera associated to different climatic states (*i.e.*, El Niño, Neutral, La Niña) should be significantly different from one another. In order to test our research question, '*are the distributions of proxy values associated with El Niño months statistically different from distributions of proxy values associated with neutral or La Niña months?*', our methodology follows a forward modelling approach in which the computed values of the temperature recorded by calcite ($T_c$ - a pseudo temperature aimed at mimicking Mg/Ca) and $\delta^{18}O_c$ are assigned to one of these climatological states. This forward modelling approach does not pre-suppose foraminifera can record ENSO variability (*i.e.*, it asks 'Can we detect?'), which is done when inverting the core top pooled $\delta^{18}O$ or individual foraminiferal $\delta^{18}O$ distributions and using measured values to infer changes in ENSO ('How could we detect?'). Whilst we are principally interested in understanding whether living foraminifera can theoretically reconstruct ENSO (Section 4 and 5), comparison with data requires further analysis. A secondary objective is to compare the output of this approach with secondary factors that further modulate the climatic signal through post-mortem processes. If the foraminifera modulate the original climate signal, then preservation selectively filters which specimens are conserved and bioturbation acts to reorder, thus scrambling the stratigraphic order in which they are recorded by the sediment depth domain, such that the stratigraphic order is no longer directly equivalent to the time domain. Once the sediment is recovered, the researcher acts as a final filter, which is in essence a random picking process (Section 6). We identify regions in the Pacific Ocean where the sedimentation rate may be too low or the water depth too deep (causing dissolution of carbonate sediments) thus potentially preventing the capture and preservation of the foraminiferal signal (Section 7). To aid the reader, only the general methodology is outlined in section 2, with the individual methodologies of each objective (referred to as Experiments 1 to 5) defined in each subsequent section (sections 3 to 7).

## 2. General Methodology

### 2.1 Input variables (Temperature; Salinity; $\delta^{18}O_{sw}$ and $\delta^{18}O_{eq}$)

For input variables, temperature and salinity of the ocean reanalysis data product (Universiteit Hamburg, DE) ORAS4 (Balmaseda et al., 2013) were extracted at one-degree resolution for the tropical Pacific (-20°S to 20°N and 120°E to -70°W), with each single grid cell comprised of data for 42 depth intervals (5 – 5300 m water depth) and 696 months (January 1958 – December 2015). For computation of the oxygen isotope of seawater ($\delta^{18}O_{sw}$), a global 1-degree grid was generated, and each grid cell was classified as belonging to one of 27 distinct ocean regions, as defined by either societal and scientific agencies, for identifying regional $\delta^{18}O_{sw}$ – salinity relationships (LeGrande and Schmidt, 2006). Using the $\delta^{18}O_{sw}$ database of LeGrande and Schmidt (2006) a regional $\delta^{18}O_{sw}$ – salinity relationship was defined, of which the salinity is the salinity measured directly at the isotope sample collection point (included within the database). Two matrices were computed; one giving values of the slope ($m$) and the other of intercept ($c$) of the resultant linear regression equations, and these were used as look-up tables to define the monthly $\delta^{18}O_{sw}$ from the monthly salinity Ocean reanalysis product ORAS4 (Balmaseda et al., 2013), which was used for the calculation of $\delta^{18}O_{eq}$, *i.e.* the expected $\delta^{18}O$ for foraminiferal calcite formed at a certain temperature (Kim and O'Neil, 1997). The $\delta^{18}O_{eq}$ is calculated from a rearranged form of the following temperature equation:

$$T = T_0 - b \cdot (\delta^{18}O_c - \delta^{18}O_{sw}) + a \cdot (\delta^{18}O_c - \delta^{18}O_{sw})^2 \text{ , (1)}$$

Specifically, we used the quadratic approximation (Bemis et al., 1998) of Kim and O'Neil (1997), where $T_0$ = 16.1, a = 0.09, b = 4.64 and converted from V-SMOW to V-PDB using a constant of -0.27 ‰ (Hut, 1987; Roche et al., 2017):

$$\Delta = b^2 - 4a \cdot (T_0 - T_{sw}) \text{, (2)}$$

$$\delta^{18}O_{c,eq} = \frac{-b - \sqrt{\Delta}}{2a} + \delta^{18}O_{sw} - 0.27 \text{ , (3)}$$

The dynamic value of Brand et al. (2014) is not used.

### 2.2 Climate classification

Pan-Pacific meteorological agencies differ in their definition of an El Niño (An and Bong, 2016; 2018), with each country's definition reflecting socio-economic factors. Therefore, for simplicity we use the Oceanic Niño Index (ONI), based upon the Niño 3.4 region (5°N to -5°S, 170°W to 120°W; Supplementary Figure 1) because of the region's importance for interactions between ocean and atmosphere which is a 3-month running mean of SST anomalies in ERSST.v5 (Huang et al., 2017). We utilise a threshold of $\chi \geq +0.5°C$ (where $\chi$ is the value of ONI) as a proxy for El Niño, $-0.5°C \leq \chi \geq +0.5°C$ for neutral climate conditions and $-0.5°C \leq \chi$ for a La Niña in the Oceanic Niño Index. Many meteorological agencies consider that five consecutive months of $\chi \geq +0.5°C$ must occur for the classification of an El Niño event. However, here the only difference is that we consider that any single month falling within our threshold values as representative of El Niño, neutral or La Niña conditions (grey bars in Supplementary Figure 1). This simplification reflects the lifecycle of planktonic foraminifera (~4

weeks) seeing that the population at time step $t$ does not record what happened at $t$-$1$ or what will happen at $t$+$1$. As we are producing the mean population growth weighted $\delta^{18}O$ values, the periods when the ONI threshold is exceeded but an El Niño or La Niña event does not occur (i.e., an 'almost' El Niño or 'almost' La Niña) would be indistinguishable from the build-up and subsequent climb-down of actual El Niño and La Niña events when the foraminiferal values are pooled in the sediment.

Therefore, these 'almost' El Niño or 'almost' La Niña (months that exceed the threshold) are placed within their respective climatological pools as El Niño or La Niña.

Each time-step for the entirety of the Pacific was classified as one of three climate states (El Niño; Neutral; and La Niña) and the corresponding values at each timestep binned into their respective categories for each grid-point. The binned values are

either the input data (Section 3: Experiment 1) or the $\delta^{18}O_c$ and $T_c$ produced by FAME (Section 4: Experiment 2). An Epanechnikov-kernel distribution was first fitted to the binned monthly output of a single climate state (using the fit distribution function fitdist of MatLab), the bandwidth varies between grid-points to provide for an optimal kernel distribution (applying the 'default' option of the function in MatLab). The use of a nonparametric representation (i.e., the kernel distribution) to fit the data, as opposed to other types of distribution (e.g., gaussian), represents a trade-off between

keeping as many parameters constant; mimicking the underlying dataset for a large number of grid points and avoiding making too many assumptions regarding the structure of the underlying data. The conversion of the data from dataset to distribution may induce some small error by: rounding to whole integers; the use of a $\delta^{18}O_{mid\text{-}point}$ which gives an error associated with the bin size ($\pm0.05$ ‰) that is symmetrical close to the distributions measures of central tendency but asymmetrical at the sides; and finally, the associated rounding error at the bin edges within a histogram ($\pm0.005$ ‰).

Subsequently, the shape of any two desired distributions can be compared for statistically significant (dis)similarity using an Anderson-Darling test (1954). For each test, comparison is made between all the values of one climatological state and all the values of another climatological state.

## 3. Experiment 1: Input Parameters

### 3.1. Objective

The resultant values produced by FAME are a modulation of the original input climate signal, therefore it is important to determine to what extent our model has altered the signal and if interpretations we garner from FAME depend upon the models growth rates values (Roche et al., 2018). In Experiment 1 we use a basin-wide statistical test to examine whether the temperature or $\delta^{18}O_{eq}$ values used as input in FAME for a given El Niño population and a given non-El Niño ('Neutral conditions') population can be expected to be significantly different at any given specific location. Where the two

populations exhibit significantly different distributions, ENSO events can potentially be detected by paleoceanographers. However, where the populations do not exhibit significantly different values, then the location represents a poor choice to study ENSO dynamics.

### 3.2. Methodology (Temperature and calculated $\delta^{18}O_{eq}$)

The input datasets of temperature and calculated $\delta^{18}O_{eq}$ underwent the following statistical test (Figure 1): for each grid-point and for every timestep, values were extracted from fixed depths of 5, 149 and 235 m (Supplementary Figure 2). These selected values from discrete-depth intervals were placed into their climatological classifications, and the resultant climatic distributions compared with one another using an Anderson-Darling test in order to compare the (dis)similarity of the resultant climatic distributions. Unlike FAME, which integrates over several depth levels using the computed growth rate, the test of the input datasets was with fixed depths without any growth rate weighting, in order to observe the implications of FAME's dynamic depth habitat. The threshold error (*i.e.*, the difference between the means of each distribution) are for temperature 0.5 °C (Figure 1A) and for $\delta^{18}O_{eq}$ 0.10 ‰ (Figure 1B), these errors should be viewed a guide rather than an implicit rejection of a site.

### 3.3 Results and Discussion

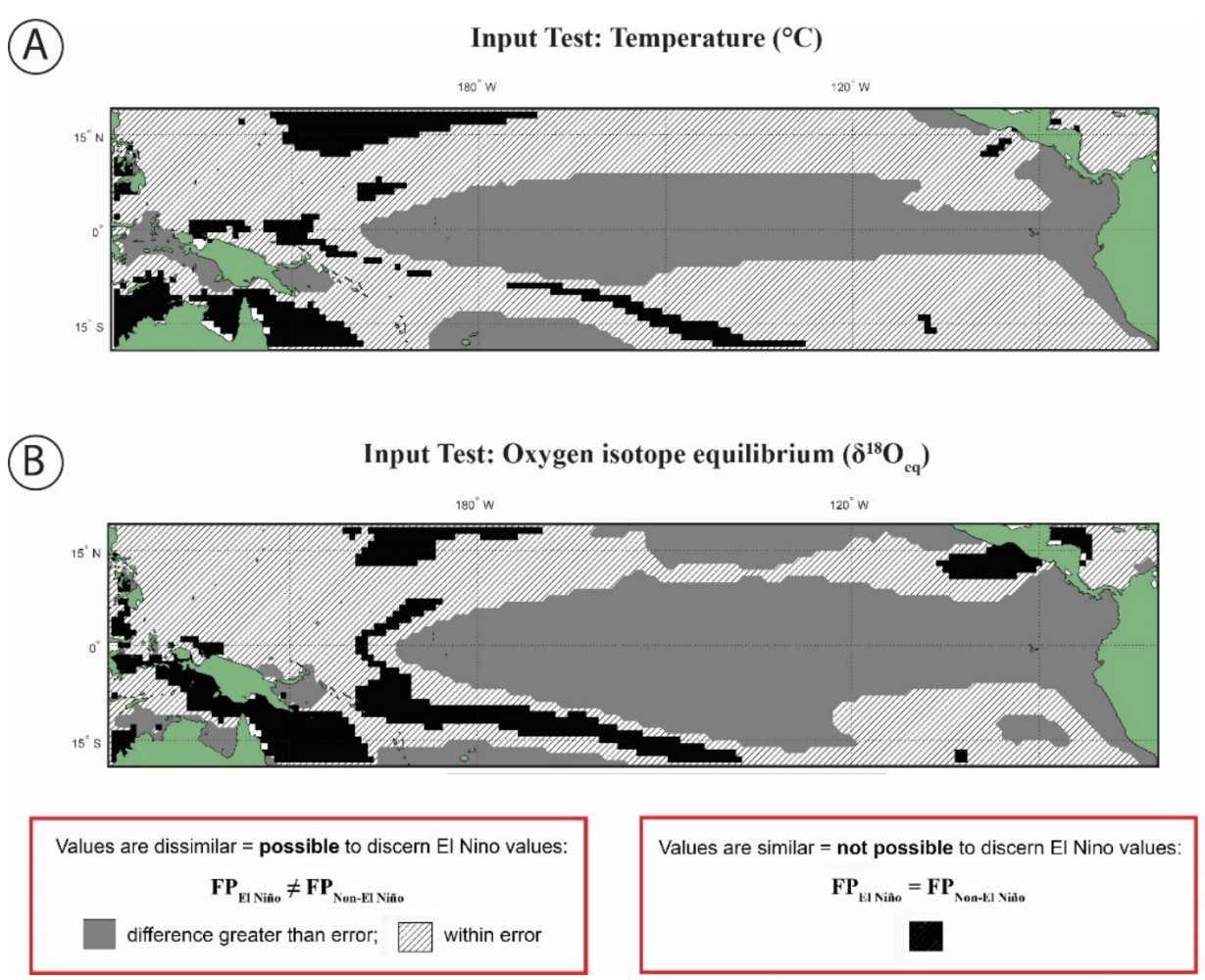

**Figure 1 Anderson-Darling Results for Input datasets of Temperature and Equilibrium δ¹⁸O (δ¹⁸O$_{eq}$). Results of the test in which input variables underwent the same statistical procedure (see section 2.0) as the modelled data for (A) temperature and (B) δ¹⁸O$_{eq}$ values. Here, model input data was extracted for a single depth of ~5 m without any growth weighting applied. Black regions are those grid points in which the null hypothesis (H$_0$), that the El Niño and Non- El Niño (Neutral) foraminifera populations (FP) are not statistically different (FP$_{El Niño}$ = FP$_{Non-El Niño}$), cannot be rejected. Grey regions represent grid points where the H$_1$ hypothesis is accepted, therefore the distributions of the foraminiferal population for El Niño and Non- El Niño can be said to be unique (FP$_{El Niño}$ ≠ FP$_{Non-El Niño}$). The hatched regions represent areas were the H$_1$ hypothesis can be accepted, therefore the distributions of the foraminiferal population for El Niño and Non- El Niño can be said to be unique (FP$_{El Niño}$ ≠ FP$_{Non-El Niño}$), though the difference between the means of tested distribution are less than (A) 0.5°C or (B) 0.1 ‰. For a comparison with three different fixed depths (5; 149; and 235 m) without any growth weighting applied see Supplementary Figure 2.**

The results of the Anderson-Darling test performed on the underlying input dataset (temperature and δ¹⁸O$_{eq}$) for each grid point are presented as either black, grey or hashed. Areas where the population distributions of the two climate states are found to be statistically similar have black grid cells. Regions in which the difference between the two populations are larger

than the potential error, are associated with grey, whereas the regions with differences less than the potential error are represented as hashed regions (Figure 1). The results of this fixed depth, non-FAME, test show that the shallowest depths produce populations that are significantly different both in terms of their mean values and their distributions and are thus suitable water locations for recording ENSO dynamics. In the upper panel of Figure 1, the canonical El Niño 3.4 region is clearly visible at 5 m depth. Though there are marked differences and similarities between the Anderson-Darling results for the other depths of the input data (Supplementary Figure 2).

## 4. Experiment 2: Foraminifera as modelled entities (FAME)

### 4.1 Objective

In Experiment 2 we run FAME on our two input datasets (temperature and oxygen isotope equilibrium). Data-model comparison studies suffer from an inability to directly compare like with like due to differences in (i) the units used *i.e.*, most proxies reconstructing temperature do not directly give values of temperature in degrees C or K but in their own proxy units (e.g., per mil ‰; mmol/mol; species abundance or ratio) necessitating a conversion; and (ii) scales in the time-depth domain, *i.e.*, models give a wealth of information (multiple depth layers and high resolution time slices). Foraminifera as modelled entities (FAME) was developed as an attempt to reduce the error associated with data-model comparisons by: (i) generating simulated-proxy time-series from a climatic input (a reanalysis dataset or climate model output) that can be compared with age-depth values down core; and (ii) to reduce the model information for a given time-slice into a manageable and relevant value using an integration that would make sense from a biological point of view (Roche et al., 2018), approximating the depth integrated growth of foraminifera (e.g., Pracht et al., 2019; Wilke et al., 2006; Steinhardt et al., 2015). FAME uses the temperature and $\delta^{18}O_{eq}$ profiles at each grid cell to compute a time averaged $\delta^{18}O_c$ and $T_c$ for a given species. Using a basin-wide statistical test, we examine whether the $\delta^{18}O_c$ values of a given El Niño foraminifera population ($FP_{EN}$) and a given non-El Niño ('Neutral conditions') foraminifera population ($FP_{NEU}$) can be expected to be significantly different at any given specific location. Where $FP_{EN}$ and $FP_{NEU}$ exhibit significantly different distributions, ENSO events can potentially be detected by paleoceanographers. In cases where $FP_{EN}$ and $FP_{NEU}$ do not exhibit significantly different values, then the chosen species and/or location represent a poor choice to study ENSO dynamics.

### 4.2 Methodology

#### 4.2.1 FAME Model

The FAME model utilises the temperature-growth rate equations of Lombard *et al.* (2009) to simulate temperature-derived growth rate (Kageyama et al., 2013; Lombard et al., 2009, 2011), this growth rate is then used as a weight to produce a growth rate-weighted proxy value (Roche et al., 2018). The original Lombard et al. (2009, 2011) equations are based upon a synthesis of culture studies, pooled together irrespective of experimental design or rationale, therefore they can be

considered to conceptually represent the fundamental niche of a given foraminiferal species, *i.e.* the range in environment that the species can survive. The basic structure of FAME is based upon temperature based Michaelis-Menton kinetics to predict growth rate, described in Lombard et al. (2009), without using the parameters (e.g., light, respiration, food) associated with FORAMCLIM (Lombard et al., 2011). The absence of known values or proxy values for the full set of parameters associated with FORAMCLIM has led us to seek a simplified approach in model parameterisation for FAME (Roche et al., 2018). It is important to note that through reducing the complexity of the problem of modelling foraminifera may lead to some deviation between observed and expected values. Our model assumes that temperature provides the dominant signal to the growth of foraminifera and therefore our results should be seen considering this assumption. Other processes may impact species growth such as mixed layer depth and nutrients.

### 4.2.2 FAME Species selection

Using the MARGO core top $\delta^{18}O_c$ database (Waelbroeck et al., 2005), Roche *et al.* (2018) validated and computed the optimum depth habitat (the depth habitat that exhibits the strongest correlation when comparing FAME $\delta^{18}O_c$ and MARGO $\delta^{18}O_c$) for each species in the MARGO database. Whilst FAME can compute the growth rate of eight foraminiferal species from culture studies (Lombard et al., 2009, 2011; Roche et al., 2018), the limited number of species available for a global core top comparison necessitated a reduction in the number of species modelled (Roche et al., 2018). Here the output of FAME is further restricted to three species that have been the main focus of foraminifera-based studies that have been used to infer ENSO variability, namely the upper ocean dwelling *Globigerinoides sacculifer* and *Globigerinoides ruber*, as well as the thermocline dwelling *Neogloboquadrina dutertrei* (Ford et al., 2015; Koutavas et al., 2006; Koutavas and Joanides, 2012; Koutavas and Lynch-Stieglitz, 2003; Leduc et al., 2009; Sadekov et al., 2013). We use the 1σ values of the observed (MARGO) minus expected (FAME), as computed by Roche et al. (2018) with the MARGO core top $\delta^{18}O_c$ database, as the potential error associated with the FAME model. The MARGO database does not include *N. dutertrei* therefore it is not possible to estimate the FAME – MARGO error as can be done with *G. ruber* and *G. sacculifer* (Roche et al., 2018).

### 4.2.3 FAME Computation

ORAS4 temperature was used as the input variable (see section 2), with the growth rate computations artificially constrained to the upper 60; 100 and 200 m to reflect the presence of photosymbiotic algae in the various foraminiferal species and an extreme value of 400 m. The modelled growth rate was used to compute the monthly depth-weighted oxygen isotope distribution for each species, using the aforementioned computed $\delta^{18}O_{eq}$ for a given latitudinal and longitudinal grid point (Supplementary Figure 3). No correction for species specific disequilibria, such as vital effect, was applied to the $\delta^{18}O_{eq}$ values.

### 4.2.4 Similar or dissimilar populations

A comparison, for each species, of FAME's predicted growth-weighted $\delta^{18}O_c$ and $T_c$ distributions associated with each climate event was done using an Anderson-Darling (AD) test. This statistical test can be used to determine whether or not two distributions can be said to come from the same population. The results of this test are presented in the following way; areas where the population distributions of the two climate states are found to be statistically similar have black grid cells in all panels referring to the Anderson-Darling test results (Figure 2; Supplementary Figures 4-6); areas where the populations distributions of two climate states are found to be statistically distinct are shown in white. For plots including the potential error see Supplementary Figures 4 and 5.

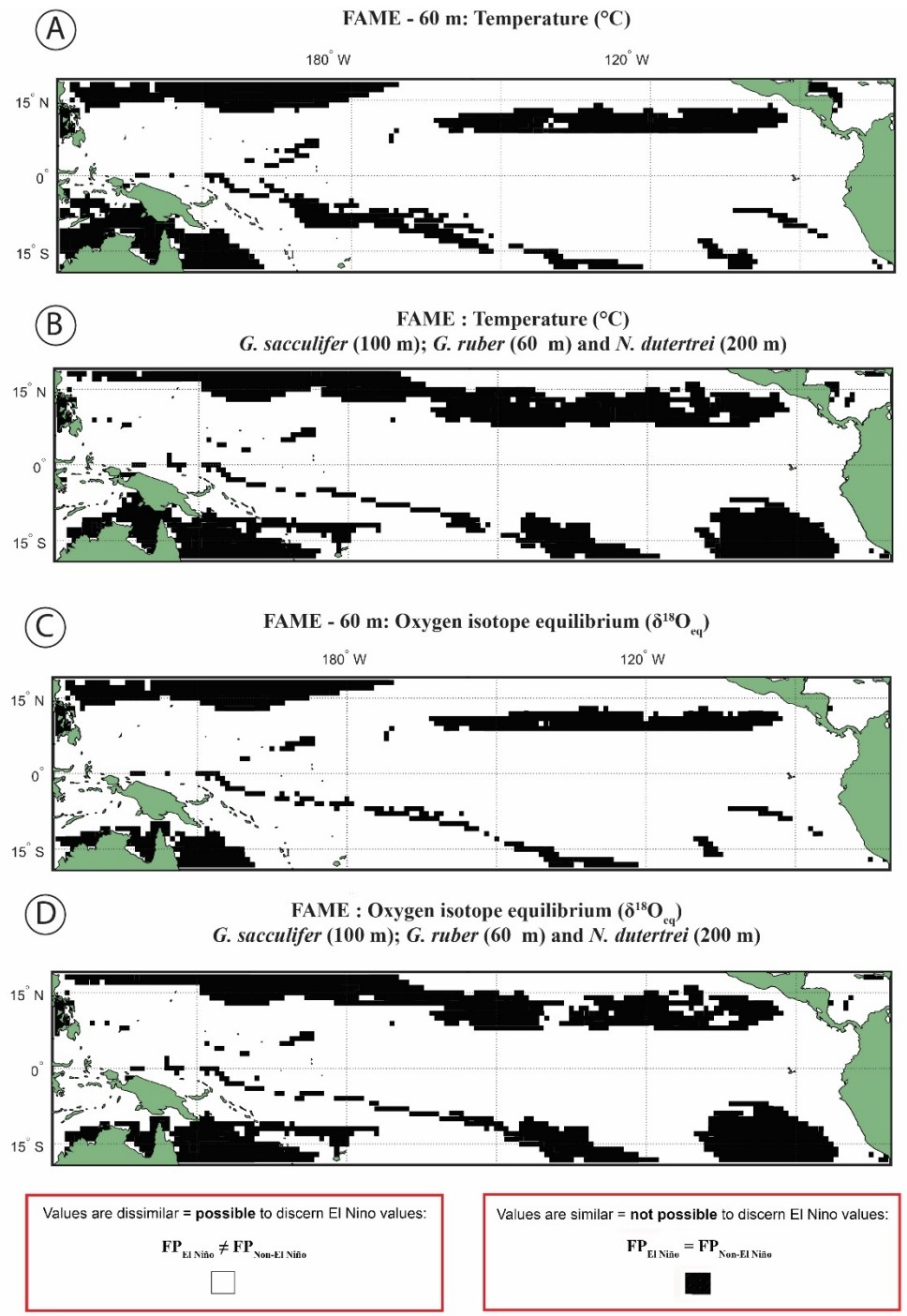

**Figure 2 Anderson-Darling results plotted regionally in which species-specific results are overlain. Panels represent water depth locations where dissimilar and similar values for the two climate states for (a-b) FAME-T$_c$ modelled temperature (c-d) FAME-δ$^{18}$O$_c$ modelled oxygen isotope values recorded in the calcite shells (Tc) occur. Each panel represents the Anderson-Darling test result, the results for *Globigerinoides sacculifer*, *Globigerinoides ruber* and *N. dutertrei* are overlaid with (A and C) cut-off depth of**

60 m and (B and D) species-specific cut-off values. For all panels black areas reflect latitudinal and longitudinal grid points that failed to reject the null hypothesis ($H_0$) and therefore the foraminiferal population (FP) of the El Niño is similar to the Non-El Niño, and therefore the distribution between the neutral climate and El Niño cannot be said to be different ($FP_{El\ Niño} = FP_{Non-El\ Niño}$).

## 4.3 Results

Our results show that much of the Pacific Ocean can be considered to have statistically different population between $FP_{EN}$ and $FP_{NEU}$ for both $\delta^{18}O$ and $T_c$ (Figure 2). We consider that the likely cause for such a remarkable result is due to FAME computing a weighted average and, therefore, the lack of a signal found exclusively within the regions demarked in Figure 1 as El Niño regions could represent how the temperature signal is integrated via an extension of the growth rate; growing season and depth habitat of distinct foraminiferal populations. Taking into account the FAME-$\delta^{18}O_c$ error for *G. ruber* and *G. sacculifer*, we have additionally computed regions in which the difference in oxygen isotopes between the two populations is smaller than the aforementioned error (see section 4.2.2) (Hatching in Supplementary Figure 4), *i.e.* where the mean difference between $FP_{EN}$ and $FP_{NEU}$ is within the error. The hatched regions in Supplementary Figure 4 considerably reduce the areal extent of significant difference between $FP_{EN}$ and $FP_{NEU}$, with the remaining regions aligning with the El Niño 3.4 region (Supplementary Figure 1). It is important to note that this error relates to the model and in reality, the difference between the climate states could be larger or smaller. No such test was performed on the *N. dutertrei* dataset, because of its absence from the MARGO dataset. To further test the model-driven results and to assess if they are still consistent when the depth limitation is varied, the analysis was rerun with depths of 100, 200 and an extreme value of 400 m (Supplementary Figures 4-6). Whilst it is possible to discern differences between the depths, it is important to note that a large percentage of the tropical Pacific remains accessible to palaeoclimate studies. A shallower depth limitation in the model increases the area for the 'warm' species, suggesting that the influence of a reduced variability in temperature or $\delta^{18}O_{eq}$ with a deeper depth limit causes the differences between $FP_{EN}$ and $FP_{NEU}$ to be reduced. Overlaying the results of the Anderson-Darling test for all three species (Figure 2; Supplementary Figures 4-6) per depth for 60, 100 and 200 m highlights the areas where multi-species comparisons could be made. To account for potential differences in depth habitat we make a combination of shallower depth for *G. ruber* and deeper depths for *G. sacculifer* and *N. dutertrei* (Pracht et al., 2019) in the final panels (Figure 2B and 2D).

## 4.4 Discussion

A number of models and modelling studies exist to determine the foraminiferal responses to present (Fraile et al., 2008, 2009; Kageyama et al., 2013; Kretschmer et al., 2017; Lombard et al., 2009, 2011; Roy et al., 2015; Waterson et al., 2016; Žarić et al., 2005, 2006), past (Fraile et al., 2009; Kretschmer et al., 2016) and future (Roy et al., 2015) climate scenarios. Unlike some foraminiferal models, FAME does not include limiting factors such as competition, respiration or predation

variables, because no reliable proxy exists for such parameterisation in the geological record, and therefore aspects such as interspecific competition that may limit the niche width of a species are not computed. By identifying the optimum depth habitat, Roche *et al.* (2018) established the realised niche, *i.e.* the range in environment that the species can be found, for these species for the late Holocene. As these depth constraints (<60 m; <100 m; and <200 m) may induce some variability we opted to include a conservative value of <400 m that grossly exaggerates the potential depth window. It is important to note, however, that as the computation of FAME is based on growth occurring within a temperature window it does not necessarily mean that for a given grid point modelled foraminifera will grow at depths down to 400 m (or whichever cut-off value is used), only that the model in theory can do so (depending if optimal temperature conditions are met). As the optimised depths computed from the MARGO dataset in Roche *et al.* (2018) are shallower, and upper ocean water is more prone to temperature variability, our approach likely dampens both the modelled $\delta^{18}O_c$ and $T_c$. Indeed, the plots testing the input dataset (Section 3; Figure 1) show that our FAME data, in which we allow the possibility for foraminiferal growth down deeper than the depths used in Roche et al. (2018), are a conservative estimate.

## 5. Experiment 3: FAME Variance statistics

In Experiment 3 we examine the variance of the $\delta^{18}O_c$ signal outputted by FAME for *G. sacculifer*. A fundamental problem with proxy records through sampling (Dolman and Laepple, 2018; Pisias and Mix, 1988; Wunsch, 2000; Wunsch and Gunn, 2003) is that they can be confounded by local regional climate, and/or ENSO's teleconnections, that mimic ENSO changes albeit at a different temporal frequency. The results of our Anderson-Darling testing may be unduly influenced by the Pacific decadal variability (PDV), also referred to as the Pacific Decadal Oscillation (PDO) (Pena et al., 2008). In much of the tropical Pacific the ratio of decadal to interannual σSST suggests that they are comparable in magnitude, therefore fluctuations in SST are more obviously apparent outside of the purely canonical regions of ENSO (Wang et al., 2017). It could be that the areas outside of these canonical ENSO regions (Supplementary Figure 1) reflect the PDO (Pena et al., 2008; Wang et al., 2017). The study of ENSO has also focused on whether the variability is entirely in response to ENSO or whether it is dominated by interannual variability (Xie, 1994, 1995; Wang et al 1994, 2010; Thirumalai et al., 2013). Therefore, in order to investigate how the signal may respond to a dynamic depth habitat, variance of the climate timeseries at each grid point was computed. As foraminiferal based ENSO studies reliant have used the spread of the individual foraminifera isotope data (either standard deviation $\sigma(\delta^{18}O_c)$ or its variance) as a measure of the increased variation in SST and, in turn, increased ENSO incidence and/or magnitude (Leduc et al., 2009; Zhu et al., 2017a) this gives us the opportunity to compare our results. For each grid-point both the total variance and the interannual variance ($\sigma^2(\delta^{18}O_c)$) of the FAME timeseries were computed in order to compare our results with previous studies. For the interannual variance, the computation follows the procedure outlined in Zhu et al. (2017a), the mean monthly climatology is subtracted from the dataset, producing monthly anomalies and a linear trend removed (using the detrend function of MatLab 2019a) – the resultant data was left unfiltered (*i.e.*, Zhu et al., 2017a used a 1-2-1 filter). Comparison between the observed variance of

FAME and expected data (Table 1) was done using the nearest grid-cell. However, as foraminifera may drift during their life (van Sebille et al., 2015) a comparison was made with the average variance of a 3 by 3 grid that has the nearest grid-cell to the core location at its centre. A comparison is also made with published iCESM model output for the same core locations (Zhu et al., 2017a).

In a previous study, a  Late Holocene sample (~1.5 ka) MD02-2529 (08°12.33'N 84°07.32'W; 1619 m) of *N. dutertrei* individual foraminifera ( >250 μm fraction) (Leduc et al., 2009) gave a δ¹⁸O standard deviation of  0.38 ‰. Here, the full ~60 year time series (n = 696) of FAME , gives a standard deviation for all species, of between 0.26 and 0.32 ‰ (<60 m depth); between0.20 and 0.29 ‰ (<100 m depth); between 0.20 and 0.25 ‰ (<200 m depth); between 0.20 and 0.24 ‰ (< 400 m depth) (see Table 1). However, these values can vary if the average of the surrounding grid cells is used (see Table 1).

In comparison, the iCESM results have the following standard deviation values, for a Eulerian (fixed) depth of 50 m: 0.4 ‰; Eulerian 100 m: 0.6 ‰; and Lagrangian value of 0.49 ‰. There are three previously analysed samples (Koutavas and Joanides, 2012; Sadekov et al., 2013) located south of core site MD02-2529, these are the Late Holocene (~1.6 ka) samples of V21-30 (01°13'S 89°41'W; 617 m) and (~1.1 ka) V21-29 (01°03'S 89°21'W; 712 m) in which *G. ruber* was measured individually. For these two sites, the measured standard deviation is 0.507 ‰ and 0.510 ‰ for V21-30 and V21-29

respectively (Koutavas and Joanides, 2012). The third core site at a similar location is (~1.6ka) CD38-17P (01°36'04 S 90°25'32W; 2580 m) was not analysed individually, instead replicates of pooled samples of 2 or 3 shells of *N. dutertrei* (Sadekov et al., 2013) were made, and these measured values give a standard deviation of 0.28 ‰. The full ~60 year time series (n = 696) of FAME presented here gives a standard deviation for all species, between 0.33 and 0.41  ‰ (<60 m depth); between 0.27 and 0.40 ‰ (<100 m depth); between 0.25 and 0.35 ‰ (<200 m depth); and between 0.25 and 0.34 ‰ (<400

20   m depth) (see Table 1). Once again, these values can vary if the average of the surrounding grid cells is used (see Table 1). In comparison, the iCESM results have the following standard deviation values, for a Eulerian (fixed) depth of 50 m: 0.53 ‰; Eulerian 100 m: 0.75 ‰; and Lagrangian value of 0.35 ‰.

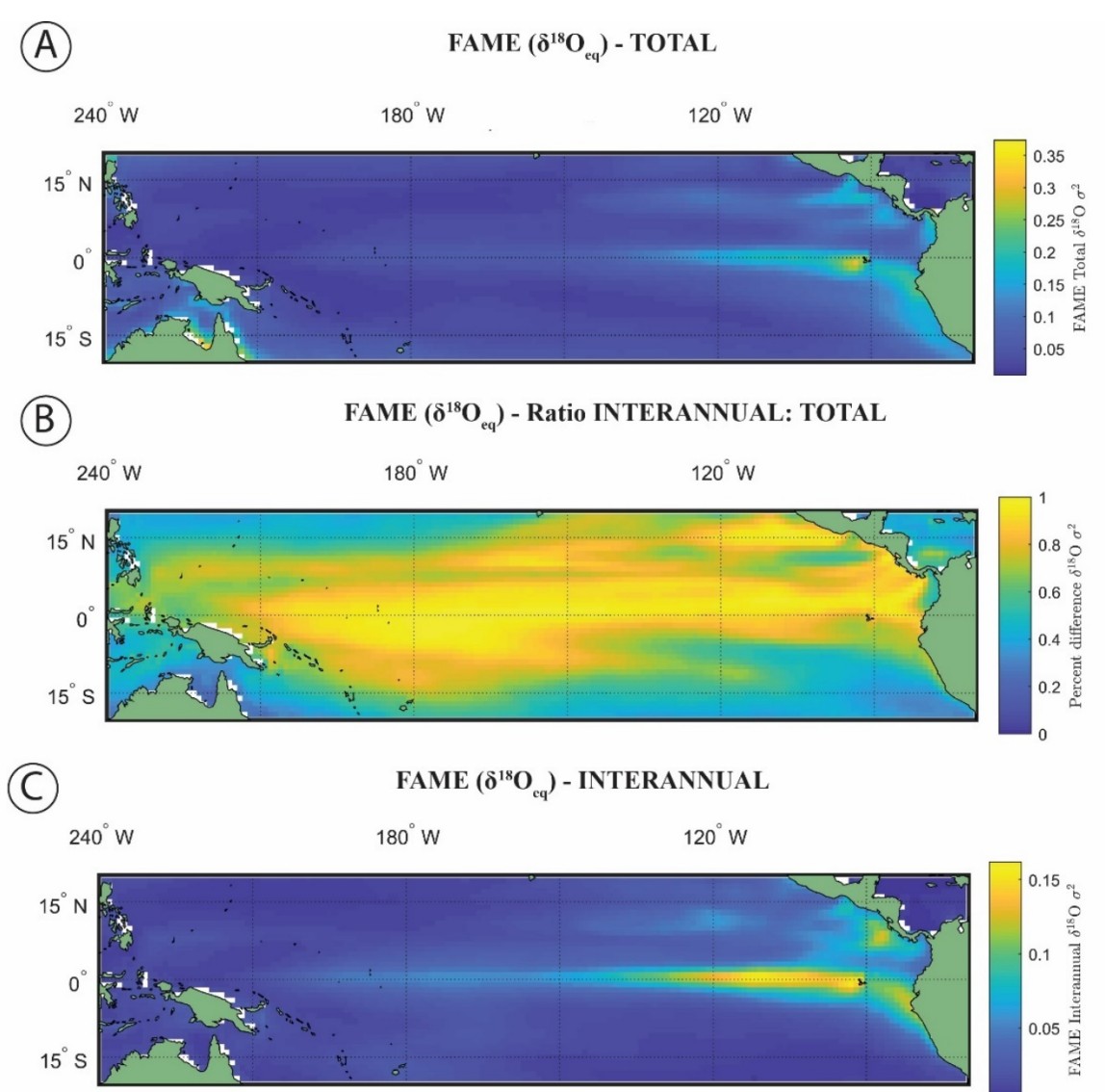

**Figure 3 Total variance and Interannual variance. (a) Total variance of *Globigerinoides sacculifer* $\delta^{18}O_c$, using FAME-$\delta^{18}O_{eq}$ for a cut-off value of 60 m. (b) The ratio of (a) and (c), where (c) is the Interannual variance of the timeseries of (a).**

5    The use of the variance $\sigma^2(\delta^{18}O_c)$, or standard deviation $\sigma(\delta^{18}O_c)$, as an indicator of ENSO is dependent on whether the original climate signal's variance was dominated by interannual variance. Zhu *et al.* (2017) computed the total variance change with and without the annual cycle suggesting that, for some cores the increased assumed ENSO variability at the LGM as deduced by proxy records (Koutavas et al., 2006; Koutavas and Joanides, 2012; Koutavas and Lynch-Stieglitz, 2003) may be purely a by-product of the annual cycle or dominated by it. Computing the ratio between the interannual

10    (Figure 3C) and total variance (Figure 3A) of FAME (Figure 3B; see Table 1) our results have a similarly high ratio of

interannual to total variance as iCESM and SODA reanalysis (Carton et al., 2000a, 2000b; Zhu et al., 2017a). Even in regions in the Eastern Equatorial Pacific wherein the ratio reduces, it is still above > 0.5. Although the values of El Niño can be considered significantly different from other climate states (Section 4), our own analysis using the ratio of total to interannual variance also suggests that much of the variance in the simulated foraminiferal signal is dominated by interannual variance. There are differences in the ratio of total to interannual variance between species and in different regions of the tropical Pacific, however, even with a dynamic depth habitat this ratio is still high (Figure 3; Table 1).

## 6. Experiment 4: FAME Picking Experiment

### 6.1 Objective

In Experiment 4 we perform a series of picking experiments on our FAME output. One source of potential variation in palaeoceanographic analysis is related to the necessity of picking a finite sample for geochemical analysis. The intention with picking is to produce a robust estimate of the population average without necessarily measuring every individual that constitutes a population, however this can bias the result if either a particular event/seasonal/depth-habitat produces a larger flux of individuals. Several 'picking' experiments were performed to determine the variance between picking iterations.

### 6.2 Methodology

FAME is not an individual foraminiferal analysis model it instead computes the average value for a given time step, here it is the average of a single month, therefore with respect to terminology what we are in effect picking is individual 'months' rather than individual 'specimens'. Irrespective of which experiment, 60 months were drawn, with replacement, and the number of Monte Carlo iterations is set at 10,000. No attempt to parameterise for misidentification has been done, as although one could assign a random value to a small percentage of the modelled values (conceptually one can argue that misidentification assigns an incorrect value), the assigned value would require knowledge of the values of co-occurring species. Previous work has highlighted the range in and between co-occurring specimens from different species (e.g., Feldmeijer et al., 2015; Metcalfe et al., 2015; 2019). Therefore, the assumption is made that the 'picker' is taxonomically well-trained and/or has a procedure in which species can be checked taxonomically post-analysis, e.g. photographing all specimens prior to analysis (e.g., Pracht et al., 2019). For picking Experiment-I (Figure 4A) all grid-points have the same selected months per iteration of the Monte Carlo, i.e., there are $10000 \cdot 60$ selected months. This assumes that the picker picks the same months at hypothetical grid point A as they select at grid point B. In Picking Experiment-II (Figure 4B), an individual Monte Carlo was run, *i.e.*, there are $170 \cdot 40 \cdot 10000 \cdot 60$ selected months. This assumes that different months could be selected between hypothetical grid point A and point B. In Picking Experiment-III (Figure 4C), at each grid-point a Monte Carlo was run using the growth rate weighting for each month (*i.e.*, there are $170 \cdot 40 \cdot 10000 \cdot 60$ selected months), this assumes that in periods of higher growth there will be a higher flux of the species and therefore a greater chance of selecting that month. The rationale being that researchers will not pick specimens representing identical time periods between grid

point A and point B. In Picking Experiment-IV (Figure 4D and 4E), the second experiment was re-run but with the addition of two sources of error: The first error is based upon FAME producing the average value for a given time slice, therefore short-term variability in temperature and/or the spread in the population (*i.e.*, variance in depth of an individual; variance in chamber growth per individual), as evidenced by single foraminiferal analysis of sediment trap samples (*e.g.*, Steinhardt et al., 2015), is potentially lost. For each picked month we therefore randomly added between -0.40 and +0.40 ‰ (approximately ±2° C, i.e., for a full range of ~4° C) to its value in intervals of 0.02 ‰. The second error is the analytical error that an individual measurement will have. Machine measurement error is assumed to lie between -0.12 and 0.12 ‰ (in intervals of 0.005 ‰ – the 3rd decimal place is an exaggeration of machine capabilities although it will have repercussions for rounding), the 1σ of within run (as opposed to long-term average) of international stable isotope standards. The intervals of both errors (0.02 ‰ and 0.005 ‰) were chosen to give a similar number (n = 41 and 49) of potential randomly selected error for each picked month. For this experiment the value assigned to each picked month was a (grid-point specific) randomly selected value for both of these errors. The values for within month variability (Figure 4D) and machine error (Figure 4E) are calculated separately and then combined (Figure 4F), as they may have a corresponding or conflicting sign, either 'cancelling' out each other or amplifying the difference.

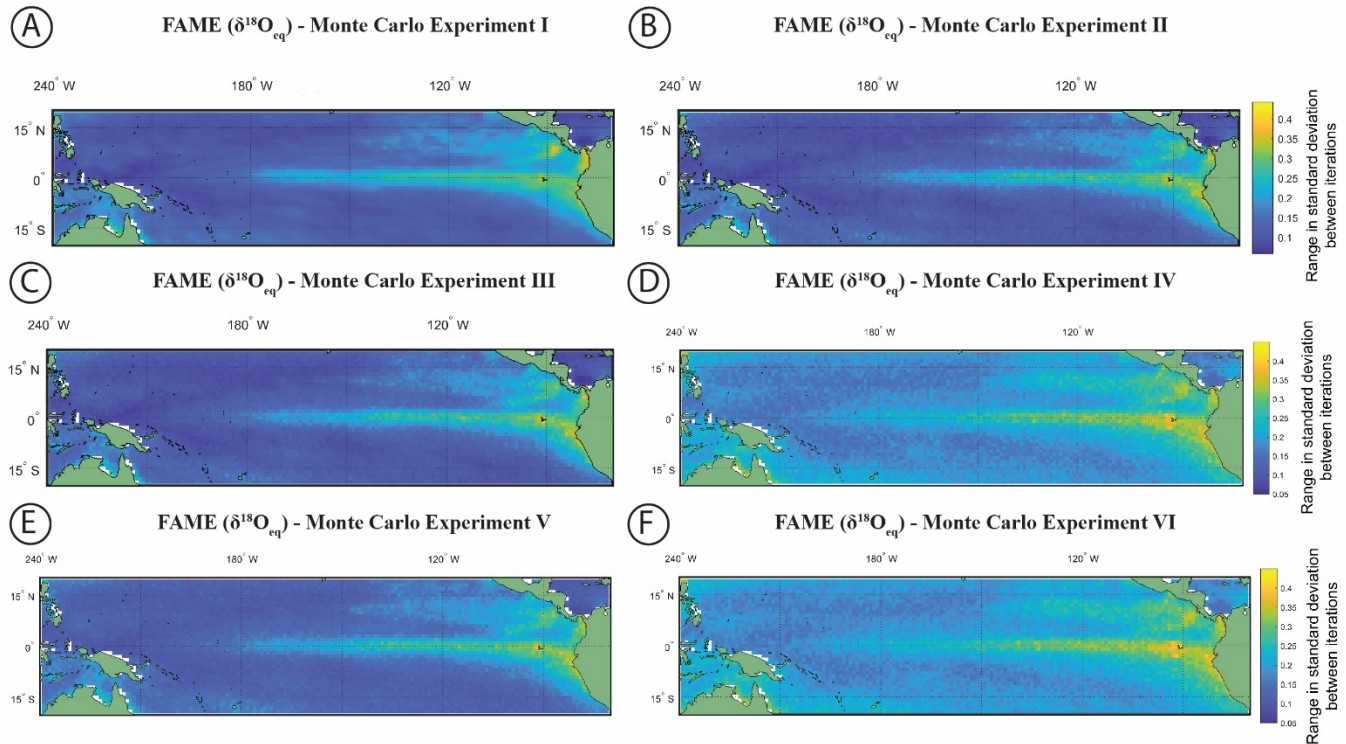

**Figure 4 The range in standard deviation of the Monte-Carlo experiments using FAME-δ¹⁸O$_c$ *G. sacculifer* with a depth cut-off of 60 m. In (a-f) we plot the range in standard deviation obtained by picking 60 months with replacement with 10,000 iterations, the experiments are as follows: (a) the same months were chosen for all grid-points for each iteration of the Monte-Carlo; (b) each**

grid-point has its own randomly selected months for each iteration of the Monte-Carlo; (c) the same as (b) but we weight the values by the total amount of growth per month; (d) the months selected for (c) were re-run but a random variability is added to each month (between -0.4 and 0.4 ‰); (e) the months selected for (b) were re-run but a random measurement error is added to each month (between -0.12 and 0.12 ‰); and (f) the months selected for (b) were re-run but the (d) random variability and (e) measurement error were combined.

## 6.3 Result

The Monte-Carlo experiments (Figure 4A-F) highlight the variation in picking a subset of the months, here 60, from the full timeseries. Given the complexity in reconstructions of trace metal geochemistry (Elderfield and Ganssen, 2000; Nürnberg et al., 1996) the focus of the picking here has been on the $\delta^{18}O_c$. The FAME-$\delta^{18}O_{eq}$ *G. sacculifer* with a depth cut-off of 60 m is plotted here, the values for each grid point is the range in standard deviation (*i.e.*, the maximum standard deviation minus the minimum standard deviation) between iterations of the Monte-Carlo (n= 10,000). The range in standard deviations between iterations is plotted instead of the mean of the standard deviations; with increasing *n* the mean converges toward the sample mean, however as the point of the Monte-Carlo is to generate plausible 'samples' it is more important to take into account the range in possible values which would help to establish the potential variability of subsampling. For the most part, regions with high total variance (Figure 4A) also have a larger range in standard deviations between the iterations 'picked'. It is interesting to note that by changing from the same months picked for each grid-point (Monte-Carlo I: Figure 4A) to varying the months picked between grid-points (Monte-Carlo II: Figure 4B or Monte-Carlo III: Figure 4C) the range goes from 'smooth' to a more noisy dataset. Whilst the values plotted here are not the absolute values (as they are the range in standard deviation for a given grid point for the entire 10,000 iterations), it can be seen that some of the inter-core comparisons could in essence relate to differences in picking, *i.e.* different 'months' picked between grid-points may exacerbate or accentuate differences. Likewise, adding random variability, between -0.4 and 0.4 ‰ (Figure 4D and 4F), may also reduce the differences between areas of high Total variance and low Total variance. Though the values associated with machine error (-0.12 to 0.12 ‰) appear to do little to affect the range (Figure 4E-F). Whilst again the values plotted are not the absolute values, the variability added in an attempt to mimic biological variation of a given time slice increases the range of possible standard deviations in regions with low Total variance (Figure 4D-E). Therefore, understanding the biological variability on shorter timescale (e.g., Steinhardt et al., 2015; Mikis et al., 2019) which, maybe here over exaggerated, may be crucial for understanding discrepancies between cores.

## 7. Experiment 5: Approximation of sediment archives

### 7.1 Objective

In Experiment 5 we compare our FAME results with bathymetric and sedimentological features of the Tropical Pacific. The preceding analysis has focused upon ~60-year reanalysis data, such a comparable resolution would require a core to have a

similar temporal resolution of ~60 years. The hypothetical core should also be above the lysocline to allow for the recovery of a proxy signal equivalent to the original climate signal. At lower sedimentation rates the modification of the original, ambient, climate signal is not limited to just its translation into a foraminiferal proxy signal and the shift in position of sinking foraminifera (van Sebille et al., 2015; Deuser et al., 1981) but can also be affected by the dissolution of calcium

5 carbonate either in the water (Schiebel et al., 2007), at the seafloor, or due to pore fluids; and bioturbation. Much of the deep-sea Pacific is both below the lysocline and has a SAR that is very low (e.g., Hays et al., 1969 at $0.96 \pm 0.43$ cm kyr$^{-1}$) although there are regions that satisfy both bathymetry and enhanced sedimentation (e.g., Koutavas and Lynch-Stieglitz, 2003 at $7.20 \pm 2.82$ cm kyr$^{-1}$). In the following section we investigate where in the tropical Pacific it is possible to extract environmental information with short frequencies from foraminiferal-based proxies, we consider that a core site must be

10 largely unaffected by dissolution (*i.e.*, above the lysocline) so as not to adversely affect the foraminifer population and the sedimentation rate must be high enough to minimise, as much as possible, the disturbance of the downcore temporal record by bioturbation.

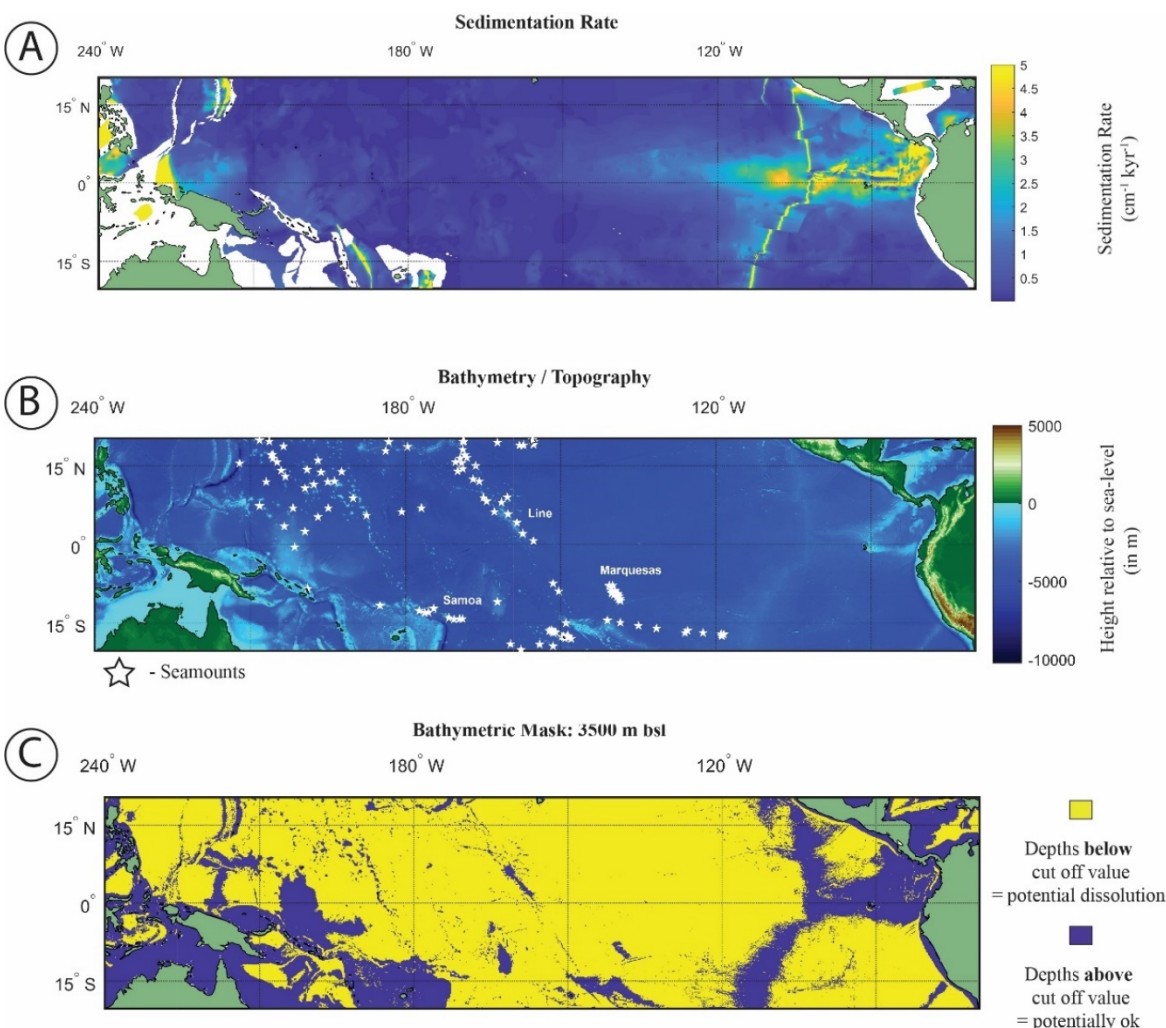

**Figure 5 (A) Map of the sedimentation rate and bathymetry of the Tropical Pacific. (A) Inferred sedimentation rate (Olson et 2016). White regions represent continental shelf. (B) GEBCO map of height relative to 0 m with location of seamounts plotted (white stars). (C) A binary colour map of the GEBCO data, yellow is values below cut-off depth value (3500 m below sea-level (bsl)) and purple above the cut-off depth value. See Supplementary Figure 8 for variation in cut-off values.**

## 7.2 Methodology

### 7.2.1 Dissolution: Cut-off depth rationale

Whilst the presence of water depths in the ocean lacking calcite-rich sediment was described in the earliest work (e.g., Murray and Renaud, 1891; Sverdrup, 1942), overlaying maps of measured surface sediment carbonate percentage with water depth in the Pacific Ocean led Bramlette (1961) to coin the term 'compensation depth' (Wise, 1978). This work highlighted the 'narrow' depths of the CCD in the Central Pacific (4-5000 m). Conceptually Berger (1971) placed three levels in the Pacific ocean that were descriptive of the aspects (e.g., chemical, palaeontological and sedimentological) of the calcite

budget; the saturation depth, demarking supersaturated from undersaturated; the lyscoline, the depth at which dissolution becomes noticeable (Berger 1968, 1971); and compensation depth (Bramlette, 1961), in which supply is compensated through dissolution. The lysocline and carbonate compensation depth (CCD) vary between the different ocean basins; the modern Atlantic Ocean in which deep water forms has a relatively deep CCD as a by-product of being 'young' well ventilated bottom waters whereas the Pacific Ocean (the final portion of the global thermohaline circulation) has a shallower CCD.

### 7.2.2 Dissolution Approximation

Dissolution is approximated by determining if each grid cells depth is above or below the prescribed cut-off value. For much of the equatorial Pacific the lysocline is estimated by a foraminiferal assemblage methodology at ~3800 m (Parker and Berger, 1971), however as the lysocline is where dissolution becomes apparent, ergo it is a sample already visibly degraded, we first set the limit of the water depth mask shallower, at 3500 m bsl. In order to account for potential variability, two further depths were used as cut-off values: 4000 m bsl and 4500 m bsl these depths represent multiple possible depths under which there is the potential for noticeable dissolution (i.e., lysocline) or complete dissolution (i.e., CCD). The bathymetry of the Pacific was extracted from the General Bathymetric Chart of the Oceans GEBCO 2014 30 arc-second grid (version 20150318, www.gebco.net) between -20°S to 20°N and 120°E to -70°W (Figure 5B). A compilation of seamounts was also plotted, as these bathymetric features may provide sufficient height to allow preservation of sediment alongside higher sediment accumulation rates (Batiza, 1982; Clouard and Bonneville, 2005; Hillier, 2007; Koppers et al., 2003; Menard, 1964; Wessel and Lyons, 1997).

### 7.2.3 Bioturbation

If we factor in the sedimentation rate of the Pacific, which in some regions has been estimated to be lower than 1 cm/ka (Blackman and Somayajulu, 1966; Berger, 1969; Hays et al., 1969; Menard, 1964), then dissolution may become further exacerbated. A secondary factor is bioturbation, systematically bioturbated deep-sea sediment cores can produce discrete sediment intervals with foraminifera that have ages spanning many centuries and/or millennia (Berger and Heath, 1968; Lougheed et al., 2018; Peng et al., 1979). In order to model the effect of bioturbation upon the age distribution of discrete core depths, a number of studies have used a diffusion style approach that reduces the parameters down to sediment accumulation rate (SAR) and sediment mixing depth (herein referred to as bioturbation depth, BD) although this may be an artificial division purely driven by mathematical need rather than biological constraints (Boudreau, 1998). The BD has been shown to have a global average of 9.8 ± 4.5 cm (1σ) that is independent of both water depth and sedimentation rate (Boudreau, 1998), likely controlled as a result of the energy efficiency of foraging, *e.g.* deeper burrows may cost more energy to produce than can be offset in extracted food resources, and potential decay in labile food resources with sediment depth.

Following the current available geochronological method (i.e., age-depth method) single specimens that are displaced in depth are assigned the average age of the depth that they were displaced to, which will result in erroneous interpretations of climate variability when analysis such as IFA is applied (Lougheed et al., 2018).To investigate how much temporal signal is integrated into discrete-depth intervals for typical tropical Pacific SAR (Olson et al., 2016; adapted by Lougheed *et al.*, 2018) the single foraminifera sediment accumulation simulator (SEAMUS, Lougheed, 2020) was utilised to bioturbate a climate signal. As it is not possible to carry out a transient bioturbation model with the SAR and BD of the Pacific with only half a century of data (such as the ORAS4 temperature and salinity ocean reanalysis data) a longer highly temporally resolved climate input signal was used, to explore the effect of bioturbation upon a given climate signal. The 0-40,000-year $\delta^{18}O_w$ of NGRIP (North Greenland Ice Core Project Members, 2004; Rasmussen et al., 2014; Seierstad et al., 2014) is considered to be a satisfactory replacement signal to simulate a foraminiferal signal in 10-year timesteps. It must be stressed that the use of the NGRIP timeseries here is purely as an input parameter to investigate the effect of bioturbation upon an oxygen isotope-based climate signal. It is important to stress that by using NGRIP as an input signal for SEAMUS we are neither implying that tropical Pacific cores should have a signal similar to NGRIP, nor that we are translating the NGRIP signal to the tropical Pacific or inferring some kind of causal relationship. As we seek to investigate the effect of bioturbation, no attempt has been made to modulate the input signal's absolute values to mimic expected $\delta^{18}O_c$ values and this is why each plot of the synthetic down core time series retains the use of V-SMOW, despite carbonates being required to be V-PDB (Coplen 1995).

A single parameter was varied whilst all others were kept constant between experiments with SEAMUS. Values of SAR were varied to fixed values of either 1, 2, 5 or 10 cm kyr$^{-1}$ that are representative of typical Pacific SAR. As the oxygen saturation state of the Pacific Ocean bottom waters is above 40 % (Supplementary Figure 9), suggestive that oxygen may not be a limiting factor, values of BD of either 5, 10 or 15cm were used. These values are based upon the global estimate of BD and its error bounds (Boudreau, 1998). For each experiment, the selected values of SAR and BD were kept constant for the entire SEAMUS model run (i.e., the intensity and magnitude of bioturbation was not varied) although in reality SAR and BD may vary temporally depending on local conditions. Each experiment was plotted as a histogram of frequency of age of specimen in BD that represent different thicknesses of sediment (5, 10 and 15 cm) and a timeseries using the computed discrete 1 cm depth median age (Figure 9).

### 7.3 Results and discussion

A factor in the post-mortem preservation of the oceanographic signal in foraminiferal shells is whether the shells can be preserved. Irrespective of the bathymetric cut-off value used for the GEBCO bathymetry data it is evident that much of the canonical El Niño 3.4 region used in oceanography, as well as a large proportion of the Tropical Pacific, is excluded from suitability as a perspective core site (Figure 5B and 5C). Even in regions where bathymetry may be above the cut-off value dissolution may occur. For instance, in regions of high fertility, such as the Eastern Equatorial Pacific, the lysocline was

estimated to be present at ~2800 m (Thunell et al., 1981) or ~3000 m (Berger, 1971; Parker and Berger, 1971). In the EEP region the shallower lyscoline is accompanied by an equally shallower CCD (located at ~3600 m) for which the high fertility/primary production is considered responsible for its shoaling, lowering the $p$H through increased $CO_2$ (Berger et al., 1976). The correspondence between lyscoline depth and CCD depth does not hold true for the entirety of the Pacific, plotting a N-S cross-section from 50°N to 50°S Berger (1971) noted that in the Central Equatorial Pacific, the high fertility region generates a larger zone of dissolution resistant facies even with a shoaled lysocline. A second factor is the sedimentation rate, using a cut off value that has been previously considered sufficiently high enough to outpace bioturbation (*e.g.*, Koutavas and Lynch-Stieglitz, 2003) of 5 cm $kyr^{-1}$, it can be demonstrated that much of the Pacific has an inferred lower sedimentation rate (< 5 cm $kyr^{-1}$; Figure 5A).

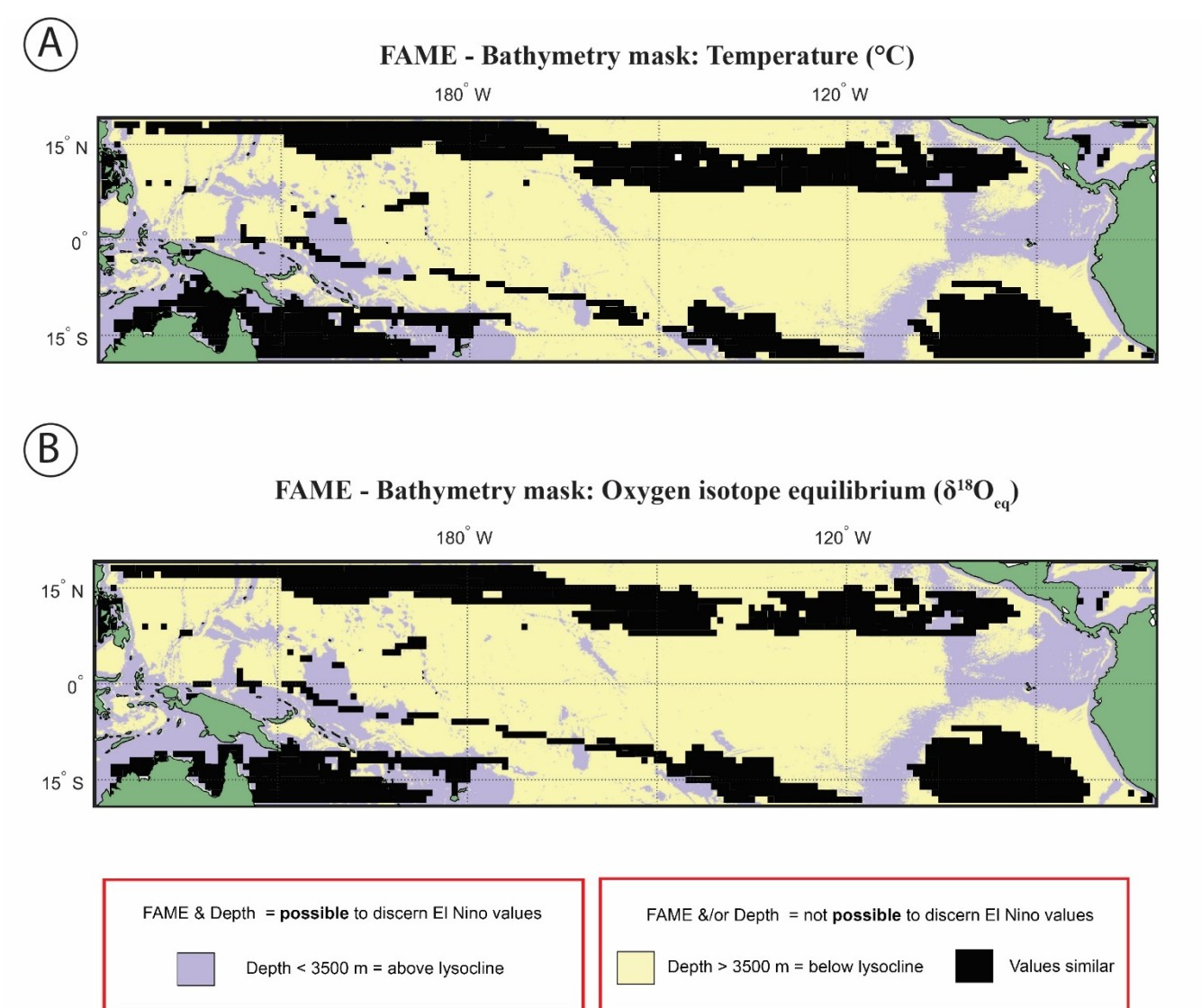

**Figure 6 Overlay between bathymetry and FAME results. The results of the FAME Anderson-Darling test for (A) temperature and (B) oxygen isotope values as input. Locations where the H₁ hypothesis can be accepted, i.e. the distributions can be said to be different (FP$_{El\ Niño}$ ≠ FP$_{Non-El\ Niño}$), are plotted as yellow where the depth is deeper than 3500 m bsl or purple where the depth is shallower than 3500 m bsl (see Figure 2). Purple locations are where our results suggest that the signal of ENSO has different values and the water depth allows for preservation.**

Overlaying the water depth and the SAR with the Anderson-Darling results (Figure 6 and Supplementary Figure 7) highlights that of the total area where $FP_{EN}$ is significantly different from $FP_{NEU}$ (*i.e.* those areas where planktonic foraminiferal flux is suitable for reconstructing past ENSO dynamics), only a small proportion corresponds to areas where the sea floor is both above the CCD (< 3500 mbsl) and SAR is at least 5 cm/ka (Figure 7). However, at certain locations,

near islands or seamounts, the SAR and water depth may be high enough to allow for a signal to be preserved (Figure 5B) that may not be represented here.

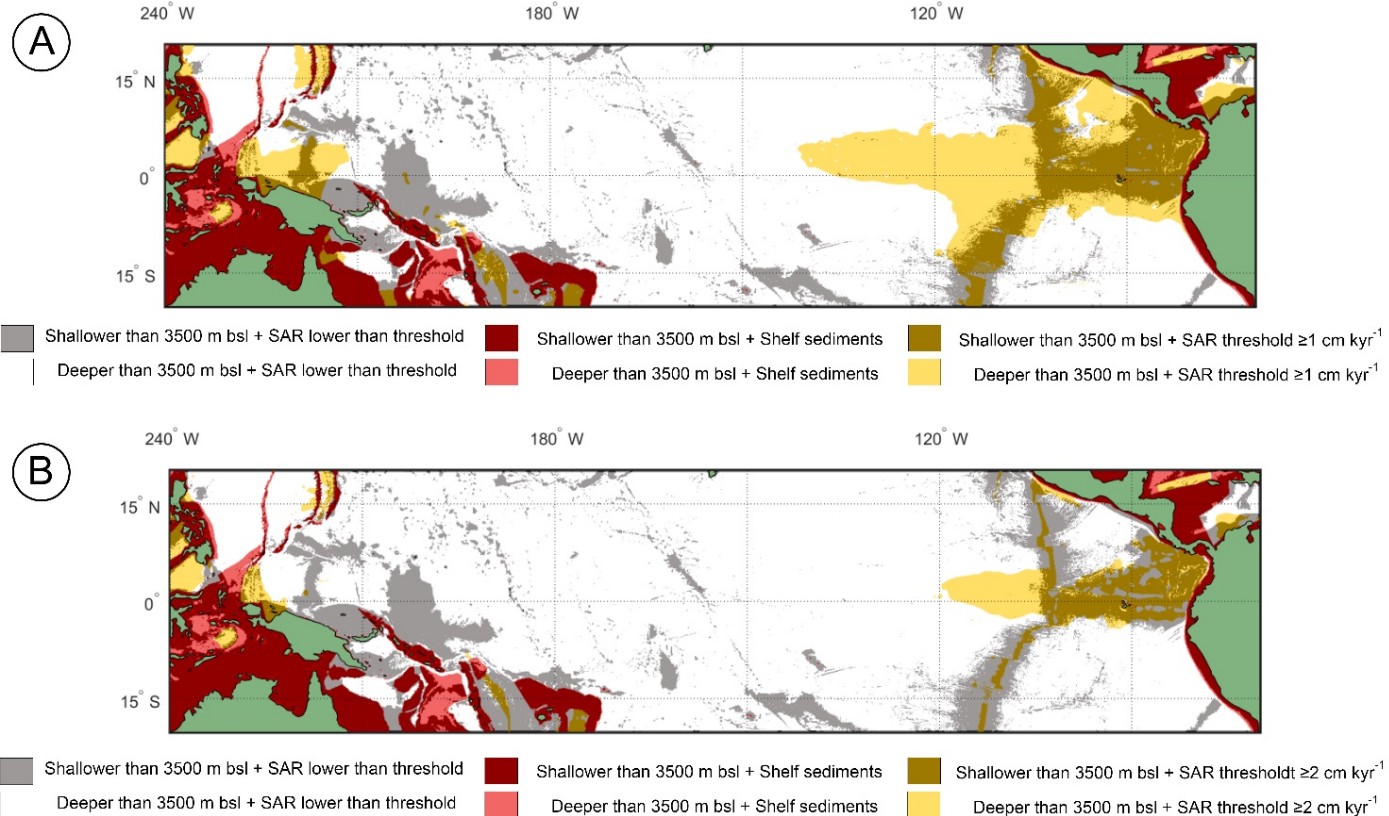

**Figure 7 Overlay between water depth and inferred SAR (Olson et al., 2016). Cut-off limits for bathymetry and SAR are 3500 m below sea-level and (A) ≥1 cm kyr⁻¹ and (B) ≥2 cm kyr⁻¹ respectively. The colours represent the following: Red / Pink: Continental shelf sediments that are (Red) shallower or (Pink) deeper than 3500 mbsl; Grey / White: grid point SAR is lower than SAR threshold and the seafloor depth is (grey) shallower or (white) deeper than 3500 mbsl; Light Yellow/Gold: Light yellow represents areas where the SAR is above the threshold but the water depth is deeper than 3500 mbsl in comparison Gold represents areas where the SAR is above the threshold and the water depth is deeper than 3500 mbsl. The ideal locations are therefore plotted as Gold.**

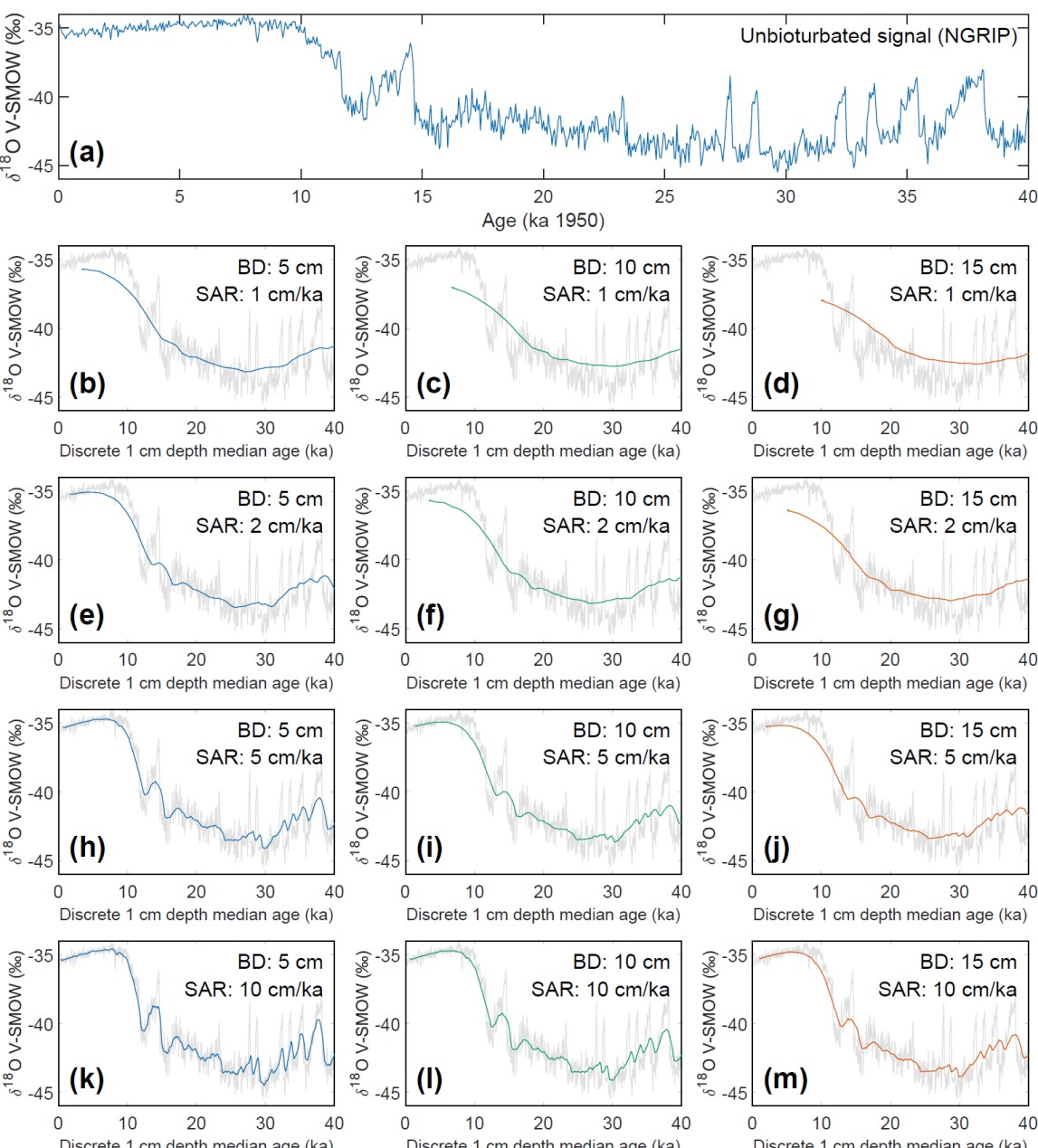

**Figure 8 Output of the bioturbation model SEAMUS. (A) The unbioturbated input signal, NGRIP (North Greenland Ice Core Project Members, 2004; Rasmussen et al., 2014; Seierstad et al., 2014), used in our simulation of bioturbation for different SAR with SEAMUS (Lougheed, 2019). Sediment mixed layer referred to here as bioturbation depth (BD) is fixed at (B, E , H, K) 5 cm, (C, F, I, L) 10 cm and (D, G, J, M) 15 cm for sedimentation accumulation rates (SAR) of (B-D) 1 cm kyr[-1]; (E-G) 2 cm kyr[-1]; (H-J) 5 cm kyr[-1] and (K-M) 10 cm kyr[-1]. The output is plotted as the discrete 1 cm depth median age. In (B-M) grey values represent the unbioturbated input signal, NGRIP. Note, we retain the original units (V-SMOW) of the original timeseries used, no inference between Pacific climate and Greenland is intended by the use of NGRIP.**

The results of the bioturbation simulator SEAMUS, plotted as a time series of the bioturbated 'NGRIP' signal (Figure 8) and as histograms of the probability of finding a particularly pseudo-foraminifera with a given age within the bioturbation depth (Figure 9), highlight the potential single foraminifera depth displacement that occurs with low sedimentation rates (Figure 5). Within a single depth in a core, proxy values largely represent the integrated time signal for that depth, the age of specimens

within the bioturbation depth may vary from a few to tens of thousands of years (Figure 9). A data-model comparison without sufficient knowledge of bioturbation may equate an integrated proxy signal with a climatic signal for an inferred (or measured) average age for the depths in question. For proxies that use an average values (i.e., a pooled foraminiferal signal) or a variance (i.e., individual foraminifera values), the individuals will be based upon a non-uniform distribution in temporal frequency of specimens, i.e., older specimens are few compared to younger specimens. A large proportion of the specimens

in the BD come from years that are 'proximal' (i.e., close to the youngest age) which may give undue confidence that the probability of picking a specimen from these years is higher, however the long-tail of the distribution means that there is an equally high chance of picking a specimen that has come from several thousand years earlier than the discrete-depth's median age. Whilst the temporal integration involved in bioturbation can be problematic for either age-depth modelling (e.g., Lougheed et al., 2018; Lougheed et al., 2020a) or discrete age measurements (e.g., Lougheed et al., 2020b) it will also

integrate the climate signal carried by the individual foraminifera.

If for example the spread in a climate variable, such as temperature, is uniform throughout the integrated time (and the abundance at each temperature value is also uniform) then it could be possible to reproduce a similar temperature distribution in bioturbated cores. Although this would not by definition represent the actual spread in the actual climatic variable for a given time. However, the climate signal is unlikely to be constant, integrating a climatic signal bioturbation

can therefore introduce artefacts inducing the possibility of spurious interpretations. Of course, identification of spurious datapoints are more obvious where the measured distributions over-exaggerate the climate signal (e.g., Wit et al., 2013). Our simulation of a climate signal reveals (Figure 8) the following: a reduction in signal amplitude with low SAR and/or increasing BD; loss of short events at low SAR; a shift in the apparent timing of events with increasing BD; and an apparent increasing 'core-top' age with low SAR and increasing BD (Figure 9). The median age of the bioturbation depth (Figure 9)

is the reason why each timeseries (Figure 8) does not 'start' at 0 age (Keigwin and Guilderson, 2009).

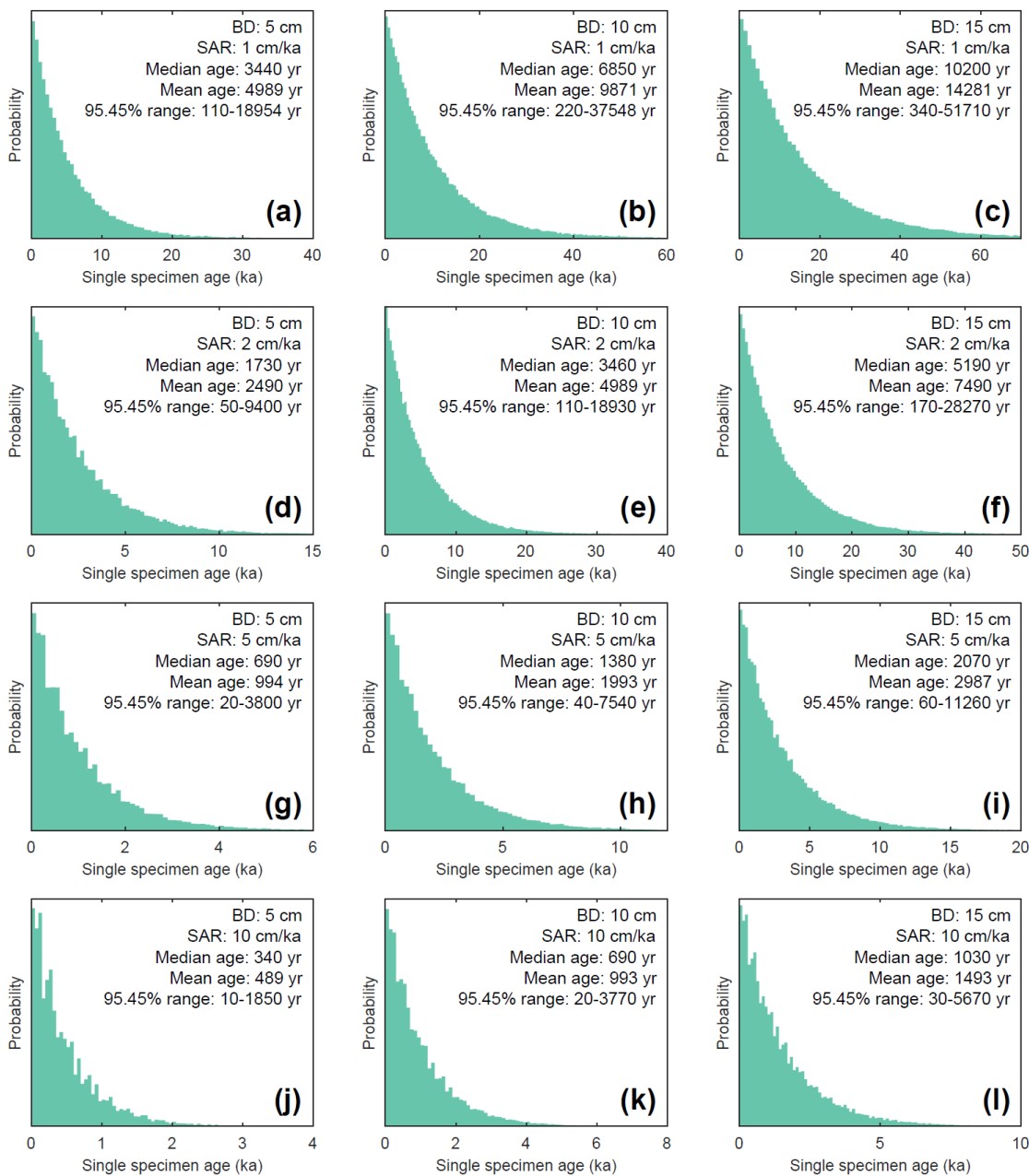

**Figure 9 Histograms of simulated specimen age within the bioturbation depth. The simulated age distribution present within the sediment mixed layer, referred to here as bioturbation depth (BD). BD is fixed at (A, D, G, J) 5 cm, (B, E, H, K) 10 cm and (C, F, I, L) 15 cm for sedimentation accumulation rates (SAR) of (A-C) 1 cm kyr[-1]; (D-F) 2 cm kyr[-1]; (G-I) 5 cm kyr[-1] and (J-L) 10 cm kyr[-1]. The output is plotted as the discrete 1 cm depth median age. Note the size of the BD varies, therefore the simulated age distribution comes from a varying 'core depth'.**

Whilst we are principally interested in understanding whether living foraminifera can theoretically reconstruct ENSO, the results of the sedimentological features, presented here, imply that much of the Pacific Ocean is not suitable for preserving (Figures 5-9) the ENSO signal, despite the possibility of the species of foraminifera in the water having unique values for different climate states (Section 4; Figure 6). In areas where preservation could occur, a hypothetical core could allow for the possible disentanglement of El Niño related signals from the climatic signal, but only in a best-case scenario involving minimal bioturbation, which is unlikely in the case of oxygenated waters. Combined with finite sampling strategies the effects of both dissolution and bioturbation can be further amplified.

## 8. Discussion

### 8.1 Palaeoceanographic Implications

Ecophysiological proxy system models are a mathematical approximation aimed at replicating the proxy signal both as its response to, and modification of, the original target climate signal (e.g., Dees et al., 2015). Linking ecophysiological models to coupled ocean-atmosphere models (e.g., Clement et al., 1999; Zebiak and Cane, 1987); isotope enabled Earth system models (e.g., iCESM; Zhu et al., 2017); or multi-model ensembles with prescribed boundary conditions could be used for the generation of timeseries for testing presumptions in proxy studies. Used a-priori, an explicit forward model can be used to test if it is plausible that the given recording system can record an oceanographic signal to allow robust reconstructions.

A critical presumption in proxy studies is embedded in site selection. Sites selected are presumed to be able to (or not) generate a climate signal, the presumptive answer in such studies is either the feature occurs or did not occur, and if it occurs then it has either enhanced or weakened. Such presumption precludes a scenario in which the feature or oceanographic regime has shifted, passing over or beyond a core site (Weyl, 1978), reacting to the expansion, contraction or shift of certain large scale oceanographic features (e.g., Polar Front, Upwelling) during periods of either warmer than average (e.g., the last interglacial) or colder than average temperatures (e.g. glacial maxima). The analysis of recent El Niño patterns suggests that there are two types of spatially delineated El Niño events: the dateline Central Pacific El Niño and the Eastern Pacific El Niño. Here we have highlighted a way of using models to determine the location where the different climate states could be differentiated. More explicit tests using climate models could be used to optimise sampling design, determine applicable core locations for comparison of proxy values with '*like with like*' oceanographic features (similar to the analysis of Evans et al. (1998) for predicting coral sites), without necessarily the cost of a time-slice project (e.g., CLIMAP, MARGO).

Another test is whether for the same set of environmental conditions two species can record an identical signal. For species with a dynamic depth habitat in which the environmental signal becomes a weighted average of the water column (e.g., Wilke et al., 2006) the likelihood of species recording the same environmental signal becomes less plausible. This is, in

brief, the rationale for the development of FAME, the same climate signal seen through the view of species-specific proxies will give a fractured view constrained by each species ecophysiological constraints (Mix, 1987; Roche et al., 2018). FAME is not the first proxy system model, instead it expands upon previous studies that have either approximated a foraminiferal signal either by weighting of ecological (seasonal or depth) preferences or by assuming that foraminifera record a fixed depth in the water column. What can be seen as contradictory proxy reconstructions can therefore be viewed as the prevailing or dominant conditions at a given location at the time when environmental conditions overlap ecological constraints for a given species. Reconstructions of the past climate (LGM-Holocene) of the Pacific have for instance inferred a relatively weaker Walker circulation, a displaced ITCZ and equatorial cooling (Koutavas and Lynch-Stieglitz, 2003); both a reduction (Koutavas and Lynch-Stieglitz, 2003) and intensification (Dubois et al., 2009) in eastern equatorial Pacific upwelling; and both weakened (Leduc et al., 2009) and strengthened ENSO variability (Koutavas and Joanides, 2012; Sadekov et al., 2013). However, a number of the inferences are contentious, for instance the reduction in upwelling in this region (Koutavas and Lynch-Stieglitz, 2003) is contradicted by Dubois et al. (2009), who used alkenones (*i.e.*, $U_{37}^{K'}$ ratios) to suggest an upwelling intensification. Whilst the $U_{37}^{K'}$ proxy has problems within coastal upwelling sites (Kienast et al., 2012) it does not discount their claim, especially considering that δ$^{18}$O records can themselves be influenced by salinity upon the δ$^{18}$O$_{sw}$ component (Rincón-Martínez et al., 2011) and the potential influence of carbonate ion concentration ([CO$_3^{2-}$]) upon foraminiferal δ$^{18}$O$_c$ (de Nooijer et al., 2009; Spero et al., 1997; Spero and Lea, 1996). The discrepancies in reconstructed climate between marine cores are worth noting, as ultimately it is from proxies that inferences are made about past climate (Trenberth and Otto-Bliesner, 2003; Rosenthal and Broccoli, 2004). Such inferences have suggested that the past climate of the Pacific region (from the geologically recent too deep time) has been in an: El Niño state (Koutavas et al., 2002; Stott et al., 2002; Koutavas and Lynch-Stieglitz, 2003); permanent El Niño state (Huber and Caballero, 2003); Super El Niño state (Stott et al., 2002); La Niña state (Andreasen et al., 2001; Beaufort et al., 2001; Martinez et al., 2003); or a different climatic state altogether (Pisias and Mix, 1997; Feldberg and Mix, 2003). Ultimately the possibility of a marine sediment archive being able to reconstruct ENSO dynamics comes down to several fundamentals besides whether the signal can or cannot be preserved (*i.e.*, whether the core site has either too low SAR, too high BD or a water depth not conducive to calcite preservation): the time-period captured by the sediment intervals (a combination of SAR and bioturbation); the frequency and intensity of ENSO events; the foraminiferal abundance during ENSO and non-ENSO conditions; as well as what the proxy is recording. Reconstructions of the past can benefit from inclusion within conceptual frameworks that incorporate both data and modelling studies (e.g., Trenberth and Otto-Bliesner, 2003; Rosenthal and Broccoli, 2004; McPhaden et al., 2006).

## 8.2 Limitations of the methods applied and assessment of model uncertainties

For simplicity we have assumed that our model is 'perfect', of course that is inaccurate, there are four potential sources of error: the input variables (temperature, salinity and their conversion into δ$^{18}$O$_{sw}$ and δ$^{18}$O$_{eq}$); the model's error with respect to

real world values (Roche et al., 2018); the statistical test's errors (associated Type I – in which attribution of significance is given to an insignificant random event, a false 'positive' – and Type II – in which a significant event is attributed to be insignificant, a false 'negative' - errors); and reducing the complexities of foraminiferal biology via parameterization. The input variables can have errors associated with both the absolute values of temperature and salinity used here, and the limitation of input values to a single value per month. Whilst it is possible to interpolate to a daily resolution, this is problematic for two reasons: (1) daily temperature records have much more high frequency oscillations than the data here and (2) the lifecycle of a single foraminifera is approximately monthly, therefore by using monthly data it provides an estimate of the average population signal. Conversion of salinity and temperature into $\delta^{18}O_{sw}$ and $\delta^{18}O_{eq}$ uses a quadratic approximation, one source of error is the unknown influence of carbonate ion concentration on both the Kim and O'Neil (1997) equation and the foraminiferal microenvironment (de Nooijer et al., 2008, 2009; Spero et al., 1997; Spero and DeNiro, 1987; Spero and Lea, 1996) which has implications due to the upwelling of cool, low $p$H, waters in the eastern Tropical Pacific (Cole and Tudhope, 2017; Raven et al., 2005). The spatial variability in salinity, particularly within regions underlying the intertropical convergence zone (ITCZ) and the moisture transport from the Caribbean into the eastern Pacific along the topographic low that represents Panama Isthmus, the resultant conversion of salinity to $\delta^{18}O_{sw}$ and then $\delta^{18}O_{eq}$ may contain further error. If such errors are independent of the absolute value of the variable, *i.e.* the error on cold temperature is the same and not larger than warm temperatures, then the error terms effectively cancel one another out. A point of note, is that the $\delta^{18}O$ to °C conversion of Kim and O'Neil (1997) is considered to be marginally larger at the cold end then at the warm end (0.2 ‰ per 1°C to 0.22 ‰ per 1°C) than that originally discerned (O'Neil et al., 1969).

The comparison of the pseudo-Mg/Ca temperature signal produced here ($T_c$) to a value corresponding to that reconstructed from measurements of Mg/Ca should be done with caution. Computation of pseudo-foraminiferal $\delta^{18}O$ in FAME is aided by the ability to compute an initial $\delta^{18}O$ equilibrium value for a given latitude-longitude grid-point and timestep. The weighting of $\delta^{18}O$ value used in FAME is an approximation of the foraminiferal shell, chambers being generally homogenous in $\delta^{18}O$ value, excluding either terminal features such as crust or gametogenic calcite which can lead to chamber heterogeneities (e.g., Wycech et al., 2018), although the latter can be approximated with an additional parameter (Roche et al., 2018). The same cannot be said for Mg/Ca, alongside heterogeneities in the shell which may be the result of diurnal processes, there are differences in both sample preparation and measurement techniques. Whilst the change in Mg/Ca with temperature has been validated (*e.g.*, Elderfield and Ganssen, 2000) the computation of a pseudo-proxy value for and from model parameters remains enigmatic. Construction of a matrix of equilibrium Mg/Ca would ideally be the most logical step in a second generation of the FAME model. Whilst simply solving the Mg/Ca palaeotemperature equation for an input of T and an output Mg/Ca is a first approximation, as stated previously several other parameters can alter this technique, this includes abiotic effects such as salinity (Allen et al., 2016; Gray et al., 2018; Groeneveld et al., 2008; Kısakürek et al., 2008) or carbonate ion concentration (Allen et al., 2016; Evans et al., 2018; Zeebe and Sanyal, 2002); biotic effects such as diurnal calcification (Eggins et al., 2003; Hori et al., 2018; Sadekov et al., 2008, 2009; Vetter et al., 2013); or additional factors such

as sediment (Fallet et al., 2009; Feldmeijer et al., 2013) or specimen (Barker et al., 2003; Greaves et al., 2005) 'cleaning' techniques. Given the role of Mg in inhibiting calcium carbonate formation, the manipulation of seawater similar to the modification of the cell's $pH$ (de Nooijer et al., 2008, 2009) may aid calcification and explain the formation of low-Mg by certain foraminifera (Zeebe and Sanyal, 2002). Scaling these processes up to a basin-wide model is beyond the remit of this

current paper.

Our modelling results also depend upon the species symbiotic nature and potential genotypes. For instance, mixotrophs, those organisms that utilise a mixture of sources for energy and carbon (planktonic foraminifera such as *G. ruber*; and/or *G. sacculifer*) can outcompete heterotrophic (or photoheterotrophic) organisms (planktonic foraminifera such as

*Neogloboquadrina pachyderma*; *Neogloboquadrina incompta*) especially in stratified-oligotrophic waters. Whilst FAME uses only the temperature component of FORAMCLIM (Roche et al., 2018) it is important to note that there are distinctions between the fundamental niche that FAME computes, *i.e.* the conditions that an organism can survive, and the realised niche, *i.e.* what an organism actually occupies given limiting factors within the environment. As FORAMCLIM and therefore FAME are based upon culture experiments, new observations highlight symbiotic or species associations (see Bird et al.,

2018, 2017). A species that hosts symbionts will likely have a restricted temperature that is associated with the temperature tolerance of their symbionts. Likewise, cryptic speciation may lead to foraminiferal genotypes exhibiting distinct environmental preferences (Bird et al., 2018, 2017; Darling et al., 2004, 2000, 1999; Huber et al., 1997; Morard et al., 2013; de Vargas et al., 1999, 2002). Incorporation of both a theoretical genotype abundance (Morard et al., 2013) and ecophysiological tolerances of different genotypes (Bird et al., 2018) within an ecophysiological model could further reduce

error within modelling of planktonic foraminiferal habitats, and thus reduce data-model comparison error. For instance, Morard *et al.* (2013) simulated the impact of genotypes upon palaeoceanographic reconstructions (in particular transfer functions) using a theoretical abundance, calculated with a best-fit gaussian response model, depending upon SST later using a similar approach (Morard et al., 2016) to deduce the impact upon $\delta^{18}O$.

**Conclusion**

Concentrating on the period spanning the instrumental record, we forward modelled the species-specific (*i.e.*, *G. ruber*; *G. sacculifer* and *N. dutertrei*) oxygen isotope values ($\delta^{18}O_c$) and pseudo-Temperature ($T_c$), computed from ocean reanalysis data using the temperature driven FAME module. The aim of this study was to determine whether the modelled values from different climate states are statistically different. If our assumptions are correct, including the reduction in Foraminiferal complexity and the choice of generic distribution (i.e., kernel) to the fit the data prior to performing an Anderson-Darling

test, our results suggest for large expanses of the Tropical Pacific the climate states do have different values. Whilst the results show that the values between El Niño states and Neutral climate states are statistically different for a large portion of the Tropical Pacific, the total variance is dominated by the interannual variance for much of the region. Overlaying our

computed foraminiferal distributions with the characteristics of the Pacific Ocean we infer that much of the region available for reconstructions corresponds to areas where several processes will alter the preservation of the foraminiferal signal. First, the inferred SAR for much of the region is critically low, and a simulation of bioturbation for different bioturbation depths and SAR typical for the Pacific indicates that discrete core depths can have a large temporal spread in single foraminifera, possibly precluding the extraction of ENSO-related climate variability. Second, a large proportion of the seafloor lies below the lysocline, the depth at which dissolution of foraminifera becomes apparent. These factors reduce the size of the area available for reconstructions considerably, thus arguably precluding the extraction of a temporally valid palaeoclimate signal using long-standing methods. It is our inference that only at exceptional ocean sediment core sites is it possible to determine the variability in ENSO based on planktonic foraminifer measurements, which makes it difficult to build a Pacific basin-wide understanding of past ENSO dynamics.

**Code and data availability**

The ocean reanalysis data used in this paper are available from the Universiteit Hamburg. An open source version of the FAME code is available from Roche et al. (2018). Statistical routines are available as part of the Statistical package of MATLAB; mapping tools (including the topographic colormap) are part of the Mapping Toolbox. The function to retrieve GEBCO bathymetry (data available at www.gebco.net) from netcdf format, gebconetcdf(FILE,Wlon,Elon,Slat,Nlat), is available from the MATLAB Central File Exchange (https://mathworks.com/matlabcentral/fileexchange/46669-gebconetcdf-file-wlon-elon-slat-nlat). The single foraminifera sediment accumulation simulator (SEAMUS) is published in Lougheed (2020), available at https://doi.org/10.5194/gmd-2019-155. A video of the $\delta^{18}O_{shell}$ output has been archived online (https://doi.org/10.5281/zenodo.2554843, Metcalfe et al., 2019).

**Author Contributions**

B.M. and D.M.R. designed the study. B.M. analysed the data. B.C.L. processed ocean SAR, depth data and ran the bioturbation model. B.M. drafted the manuscript with contributions from all authors.

**Competing Interests**

The authors declare no competing interests.

**Acknowledgements**

B.M. was supported by a Laboratoire d'excellence (LabEx) of the Institut Pierre-Simon Laplace (Labex L-IPSL), funded by the French Agence Nationale de la Recherche (grant no. ANR-10-LABX-0018). B.M. thanks both LSCE and the VU

University Amsterdam for guest status. D.M.R. is supported by the French agency Centre National de la Recherche Scientifique (CNRS) and the VU University Amsterdam. This is a contribution to the ACCLIMATE ERC project. The research leading to these results has received funding to C.W. from the European Research Council under the European Union's Seventh Framework Programme (FP7/2007-2013 Grant agreement n° 339108). B.C.L. acknowledges Swedish Research Council (Vetenskapsrådet – VR) grant 2018-04992, The Swedish National Infrastructure for Computing (SNIC) at the Uppsala Multidisciplinary Centre for Advanced Computational Science (UPPMAX) provided computer resources for running the SEAMUS model. We thank the Universiteit Hamburg for their online access server for ocean reanalysis data.

**Figure Captions**

**Figure 1 Anderson-Darling Results for Input datasets of Temperature and Equilibrium $\delta^{18}O$ ($\delta^{18}O_{eq}$). Results of the test in which input variables underwent the same statistical procedure (see section 2.0) as the modelled data for (A) temperature and (B) $\delta^{18}O_{eq}$ values. Here, model input data was extracted for a single depth of ~5 m without any growth weighting applied. Black regions are those grid points in which the null hypothesis ($H_0$), that the El Niño and Non- El Niño (Neutral) foraminifera populations (FP) are not statistically different ($FP_{El\ Niño} = FP_{Non-El\ Niño}$), cannot be rejected. Grey regions represent grid points where the $H_1$ hypothesis is accepted, therefore the distributions of the foraminiferal population for El Niño and Non- El Niño can be said to be unique ($FP_{El\ Niño} \neq FP_{Non-El\ Niño}$). The hatched regions represent areas were the $H_1$ hypothesis can be accepted, therefore the distributions of the foraminiferal population for El Niño and Non- El Niño can be said to be unique ($FP_{El\ Niño} \neq FP_{Non-El\ Niño}$), though the difference between the means of tested distribution are less than (A) 0.5°C or (B) 0.1 ‰. For a comparison with three different fixed depths (5; 149; and 235 m) without any growth weighting applied see Supplementary Figure 2.**

# Input Test: Temperature (°C)

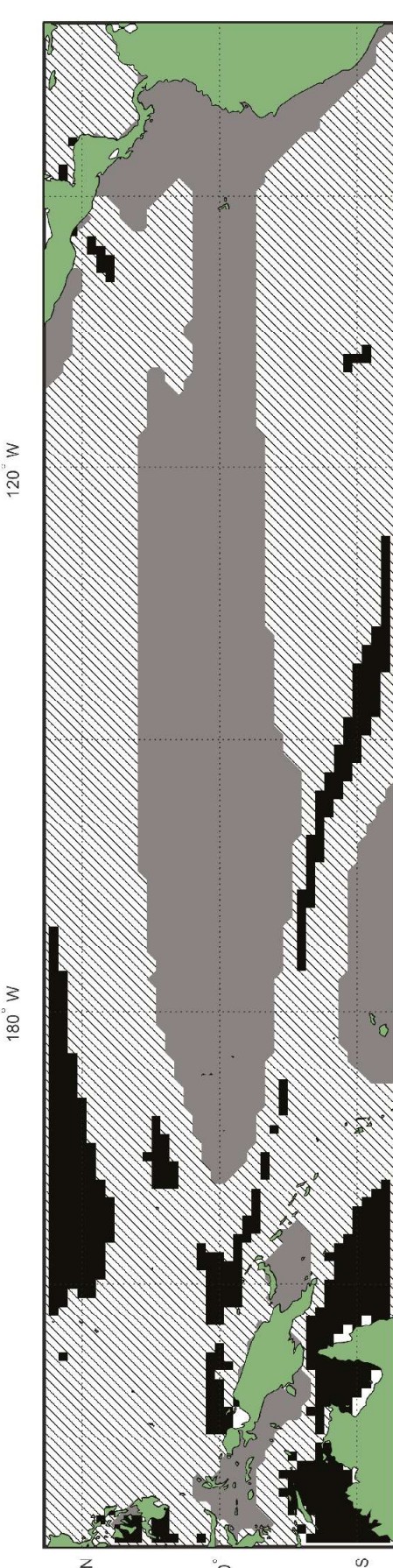

Ⓐ

# Input Test: Oxygen isotope equilibrium ($\delta^{18}O_{eq}$)

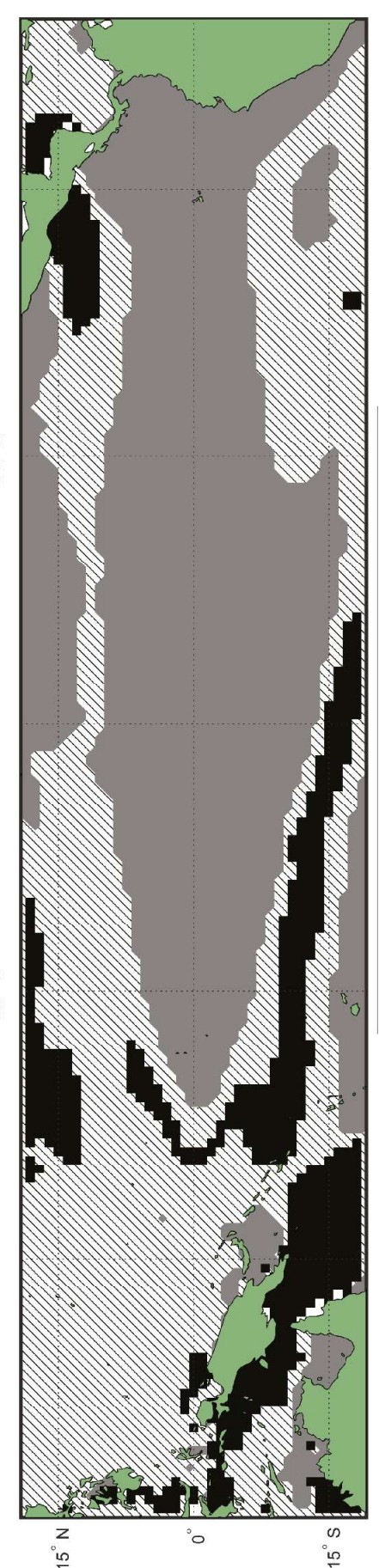

Ⓑ

Values are dissimilar = **possible** to discern El Nino values:

$$FP_{El\ Niño} \neq FP_{Non-El\ Niño}$$

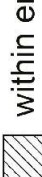 difference greater than error;   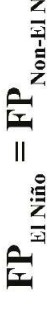 within error

Values are similar = **not possible** to discern El Nino values:

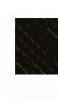

$$FP_{El\ Niño} = FP_{Non-El\ Niño}$$

**Figure 2 Anderson-Darling results plotted regionally in which species-specific results are overlain. Panels represent water depth locations where dissimilar and similar values for the two climate states for (a-b) FAME-$T_c$ modelled temperature (c-d) FAME-$\delta^{18}O_c$ modelled oxygen isotope values recorded in the calcite shells (Tc) occur. Each panel represents the Anderson-Darling test result, the results for *Globigerinoides sacculifer*, *Globigerinoides ruber* and *N. dutertrei* are overlaid with (A and C) cut-off depth of 60 m and (B and D) species-specific cut-off values. For all panels black areas reflect latitudinal and longitudinal grid points that failed to reject the null hypothesis ($H_0$) and therefore the foraminiferal population (FP) of the El Niño is similar to the Non-El Niño, and therefore the distribution between the neutral climate and El Niño cannot be said to be different ($FP_{El\ Niño} = FP_{Non-El\ Niño}$).**

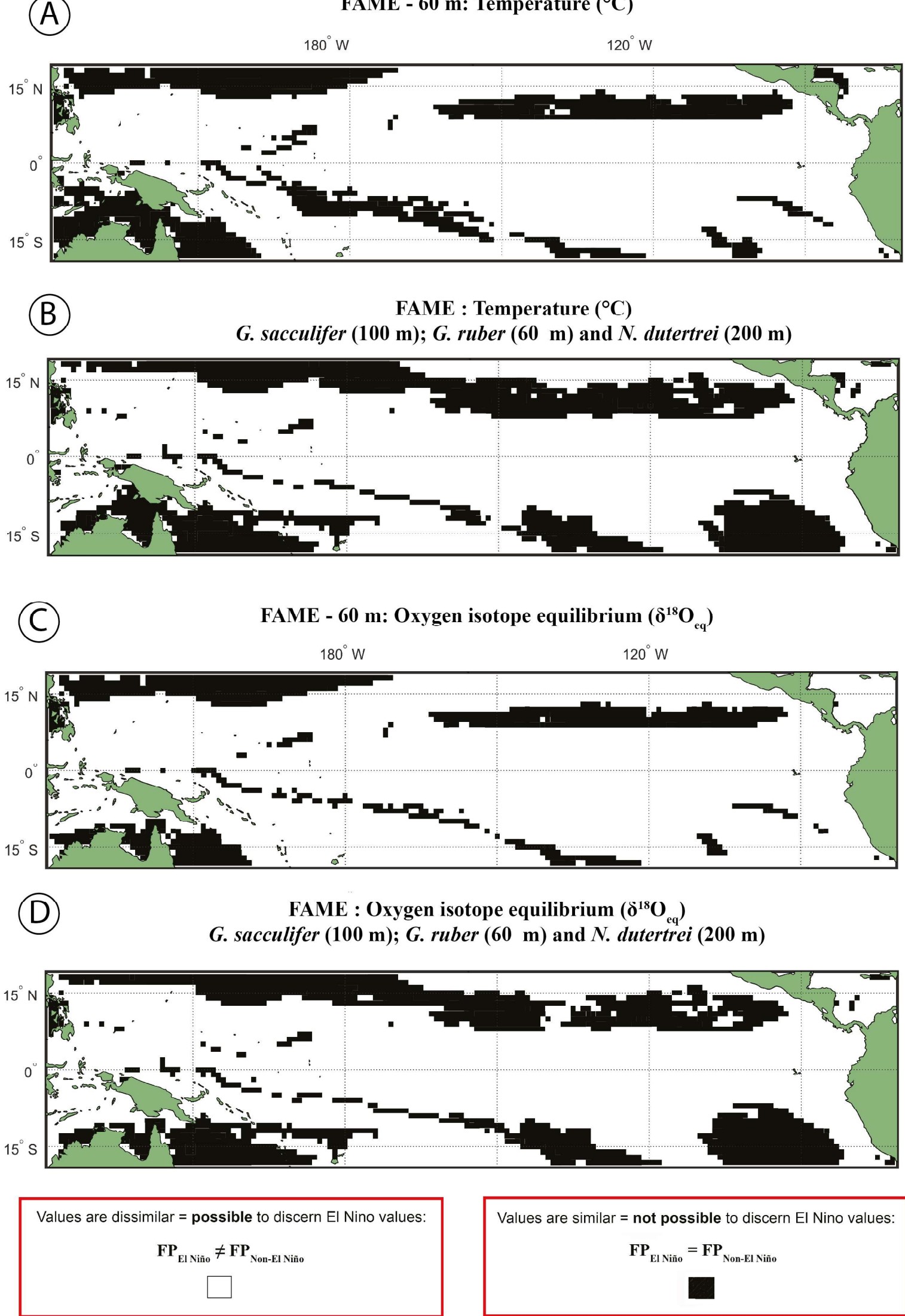

**FAME - 60 m: Temperature (°C)**

**FAME : Temperature (°C)**
*G. sacculifer* (100 m); *G. ruber* (60 m) and *N. dutertrei* (200 m)

**FAME - 60 m: Oxygen isotope equilibrium ($\delta^{18}O_{eq}$)**

**FAME : Oxygen isotope equilibrium ($\delta^{18}O_{eq}$)**
*G. sacculifer* (100 m); *G. ruber* (60 m) and *N. dutertrei* (200 m)

Values are dissimilar = **possible** to discern El Nino values:

$$FP_{El\ Niño} \neq FP_{Non\text{-}El\ Niño}$$

Values are similar = **not possible** to discern El Nino values:

$$FP_{El\ Niño} = FP_{Non\text{-}El\ Niño}$$

**Figure 3 Total variance and Interannual variance. (a) Total variance of *Globigerinoides sacculifer* $\delta^{18}O_c$, using FAME-$\delta^{18}O_{eq}$ for a cut-off value of 60 m. (b) The ratio of (a) and (c), where (c) is the Interannual variance of the timeseries of (a).**

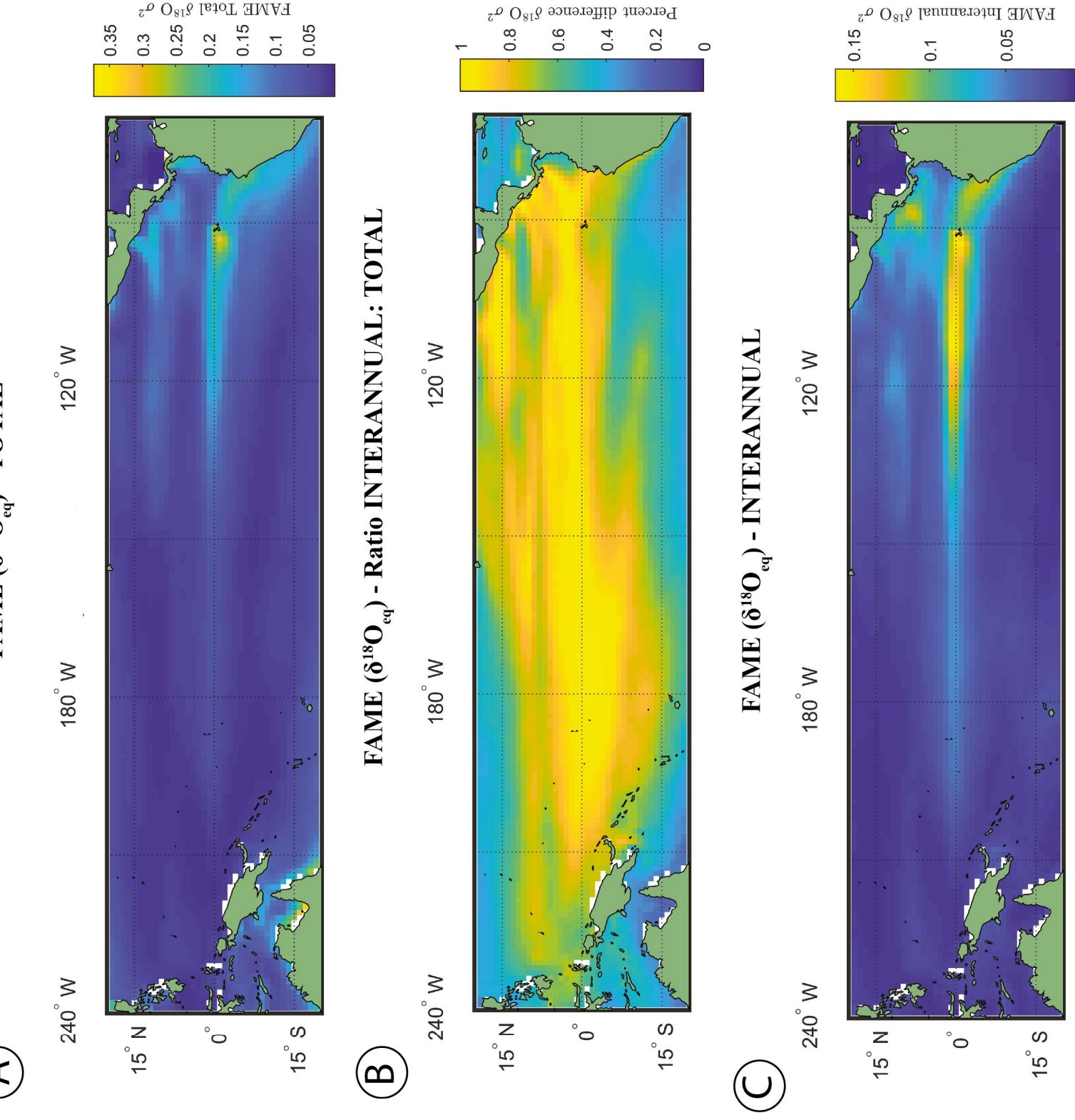

**Figure 4 The range in standard deviation of the Monte-Carlo experiments using FAME-$\delta^{18}O_c$ *G. sacculifer* with a depth cut-off of 60 m. In (a-f) we plot the range in standard deviation obtained by picking 60 months with replacement with 10,000 iterations, the experiments are as follows: (a) the same months were chosen for all grid-points for each iteration of the Monte-Carlo; (b) each grid-point has its own randomly selected months for each iteration of the Monte-Carlo; (c) the same as (b) but we weight the values by the total amount of growth per month; (d) the months selected for (c) were re-run but a random variability is added to each month (between -0.4 and 0.4 ‰); (e) the months selected for (b) were re-run but a random measurement error is added to each month (between -0.12 and 0.12 ‰); and (f) the months selected for (b) were re-run but the (d) random variability and (e) measurement error were combined.**

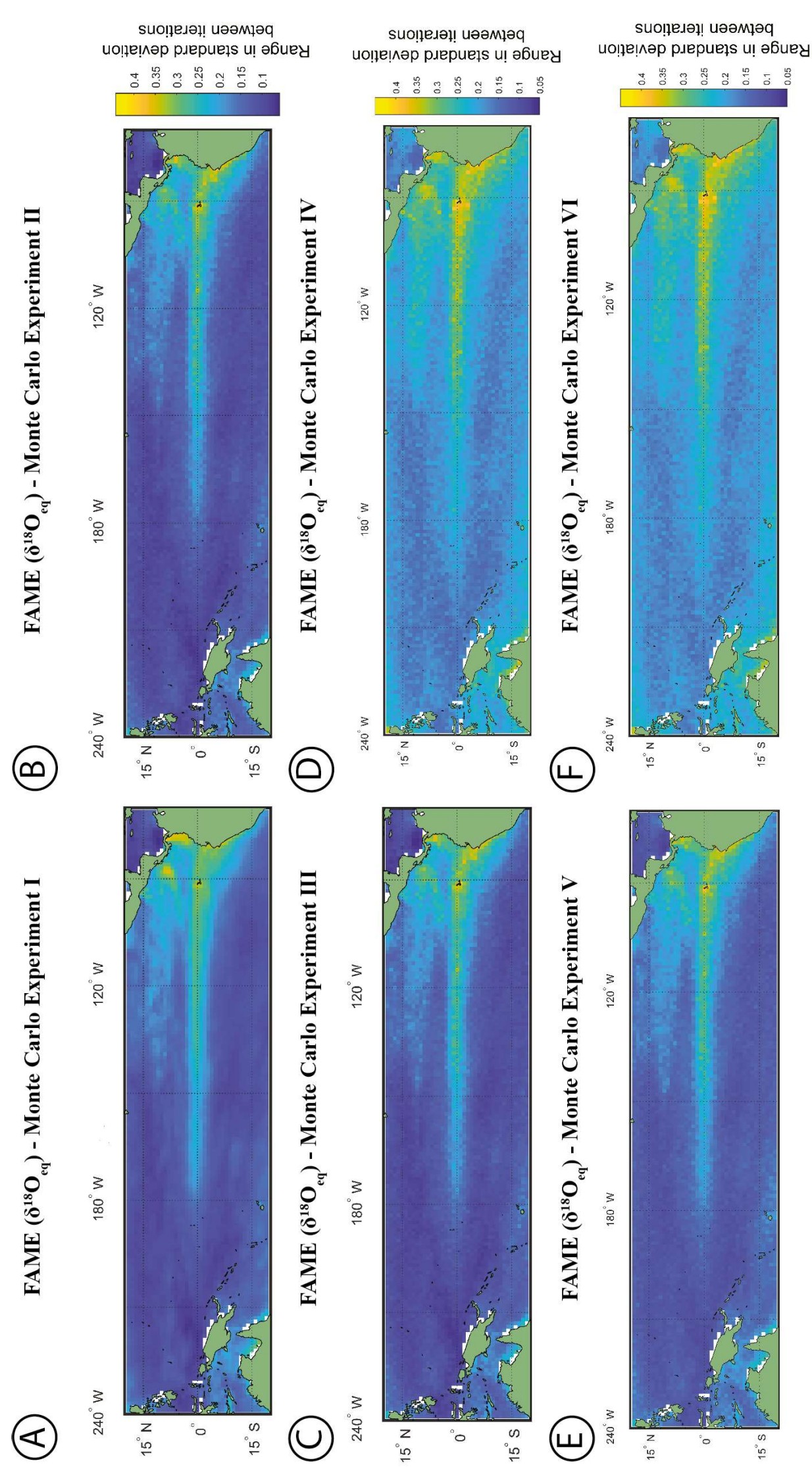

**Figure 5 (A) Map of the sedimentation rate and bathymetry of the Tropical Pacific. (A) Inferred sedimentation rate (Olson et 2016). White regions represent continental shelf. (B) GEBCO map of height relative to 0 m with location of seamounts plotted (white stars). (C) A binary colour map of the GEBCO data, yellow is values below cut-off depth value (3500 m below sea-level (bsl)) and purple above the cut-off depth value. See Supplementary Figure 8 for variation in cut-off values.**

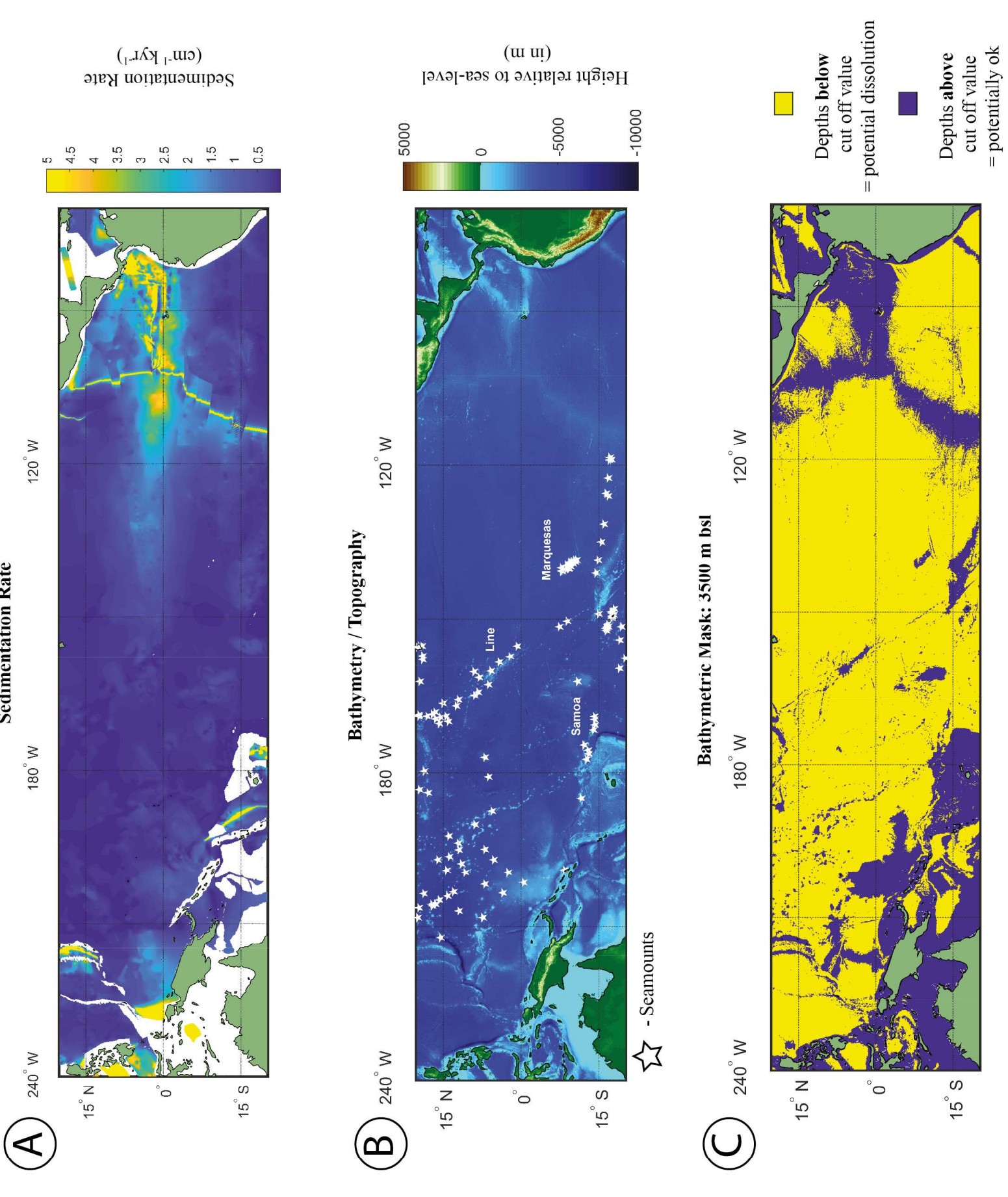

**Figure 6 Overlay between bathymetry and FAME results. The results of the FAME Anderson-Darling test for (A) temperature and (B) oxygen isotope values as input. Locations where the $H_1$ hypothesis can be accepted, i.e. the distributions can be said to be different ($FP_{El\ Niño} \neq FP_{Non\text{-}El\ Niño}$), are plotted as yellow where the depth is deeper than 3500 m bsl or purple where the depth is shallower than 3500 m bsl (see Figure 2). Purple locations are where our results suggest that the signal of ENSO has different values and the water depth allows for preservation.**

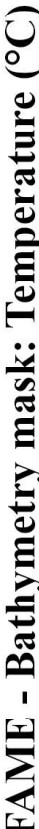

# FAME - Bathymetry mask: Temperature (°C)

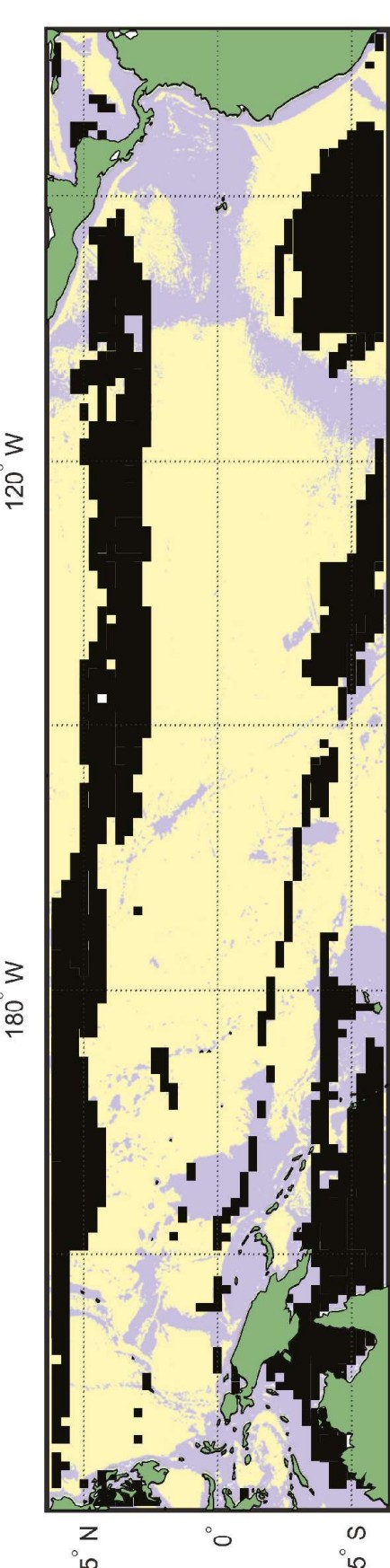

120° W

180° W

15° N

0°

15° S

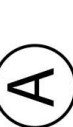

# FAME - Bathymetry mask: Oxygen isotope equilibrium ($\delta^{18}O_{eq}$)

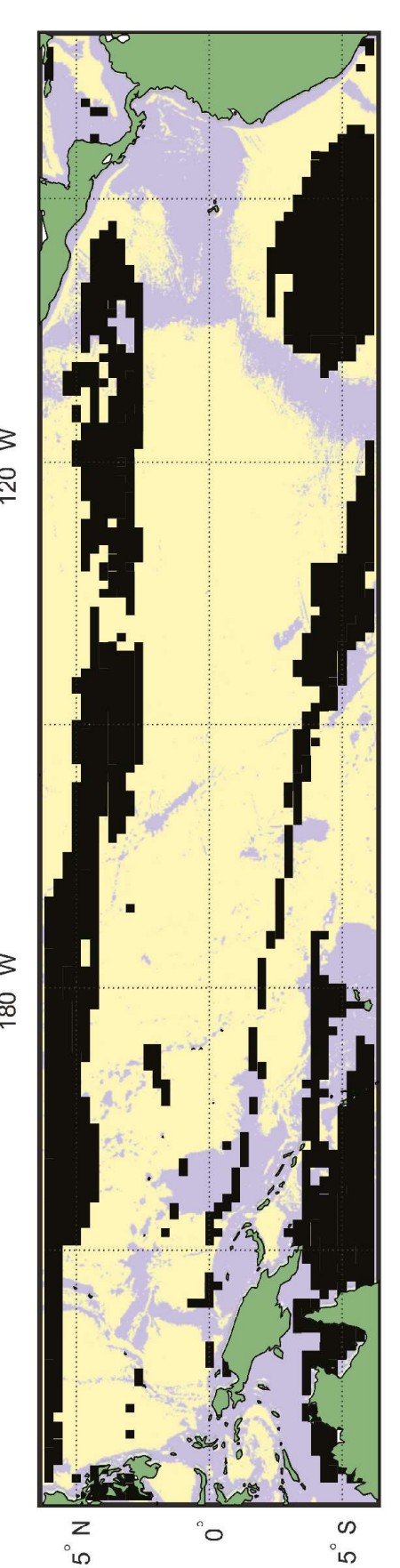

120° W

180° W

15° N

0°

15° S

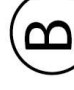

FAME & Depth = **possible** to discern El Nino values

Depth < 3500 m = above lysocline

FAME &/or Depth = not **possible** to discern El Nino values

Depth > 3500 m = below lysocline

Values similar

**Figure 7 Overlay between water depth and inferred SAR (Olson et al., 2016). Cut-off limits for bathymetry and SAR are 3500 m below sea-level and (A) ≥1 cm kyr$^{-1}$ and (B) ≥2 cm kyr$^{-1}$ respectively. The colours represent the following: Red / Pink: Continental shelf sediments that are (Red) shallower or (Pink) deeper than 3500 mbsl; Grey / White: grid point SAR is lower than SAR threshold and the seafloor depth is (grey) shallower or (white) deeper than 3500 mbsl; Light Yellow/Gold: Light yellow represents areas where the SAR is above the threshold but the water depth is deeper than 3500 mbsl in comparison Gold represents areas where the SAR is above the threshold and the water depth is deeper than 3500 mbsl. The ideal locations are therefore plotted as Gold.**

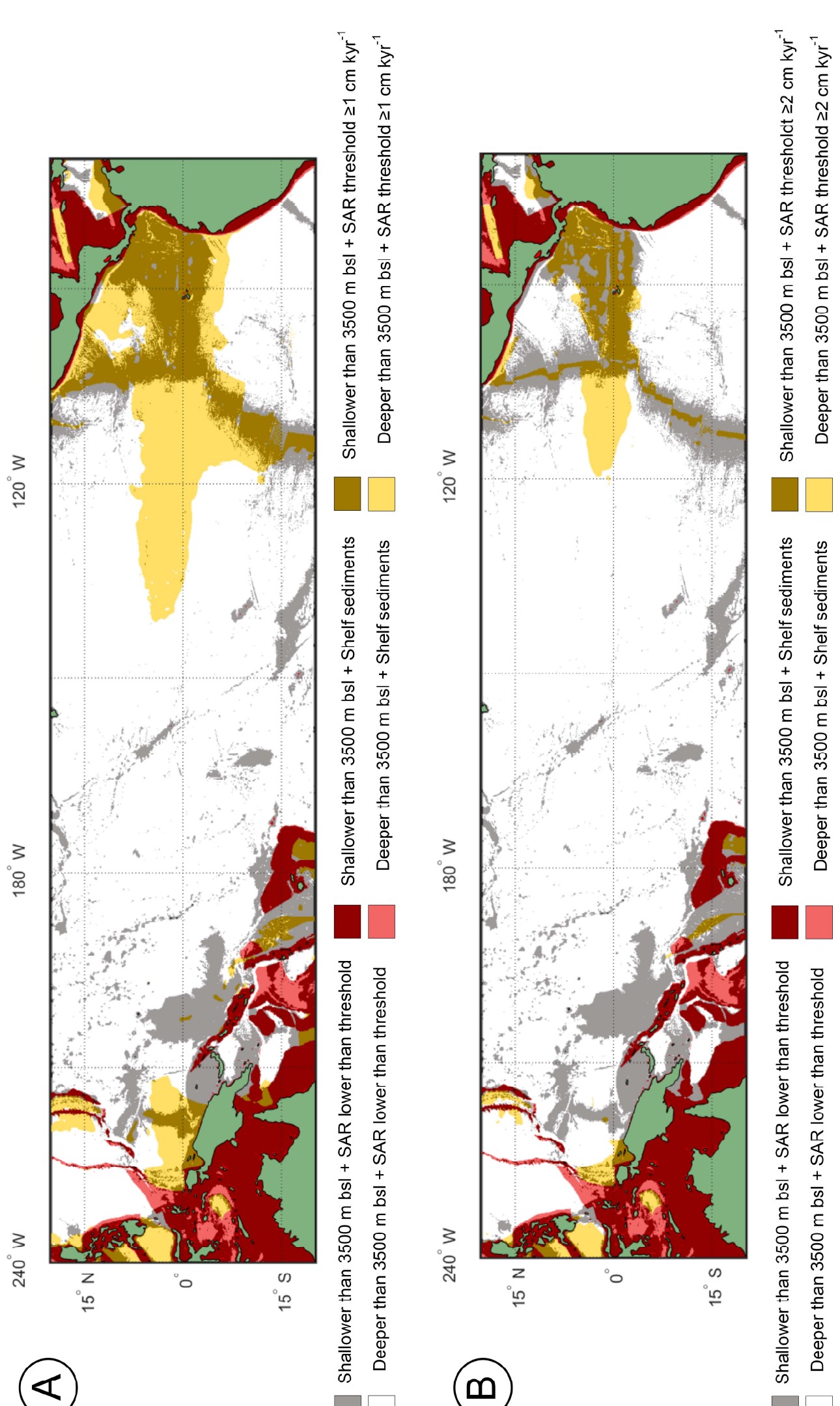

**Figure 8 Output of the bioturbation model SEAMUS. (A) The unbioturbated input signal, NGRIP (North Greenland Ice Core Project Members, 2004; Rasmussen et al., 2014; Seierstad et al., 2014), used in our simulation of bioturbation for different SAR with SEAMUS (Lougheed, 2019). Sediment mixed layer referred to here as bioturbation depth (BD) is fixed at (B, E , H, K) 5 cm, (C, F, I, L) 10 cm and (D, G, J, M) 15 cm for sedimentation accumulation rates (SAR) of (B-D) 1 cm kyr$^{-1}$; (E-G) 2 cm kyr$^{-1}$; (H-J) 5 cm kyr$^{-1}$ and (K-M) 10 cm kyr$^{-1}$. The output is plotted as the discrete 1 cm depth median age. In (B-M) grey values represent the unbioturbated input signal, NGRIP. Note, we retain the original units (V-SMOW) of the original timeseries used, no inference between Pacific climate and Greenland is intended by the use of NGRIP.**

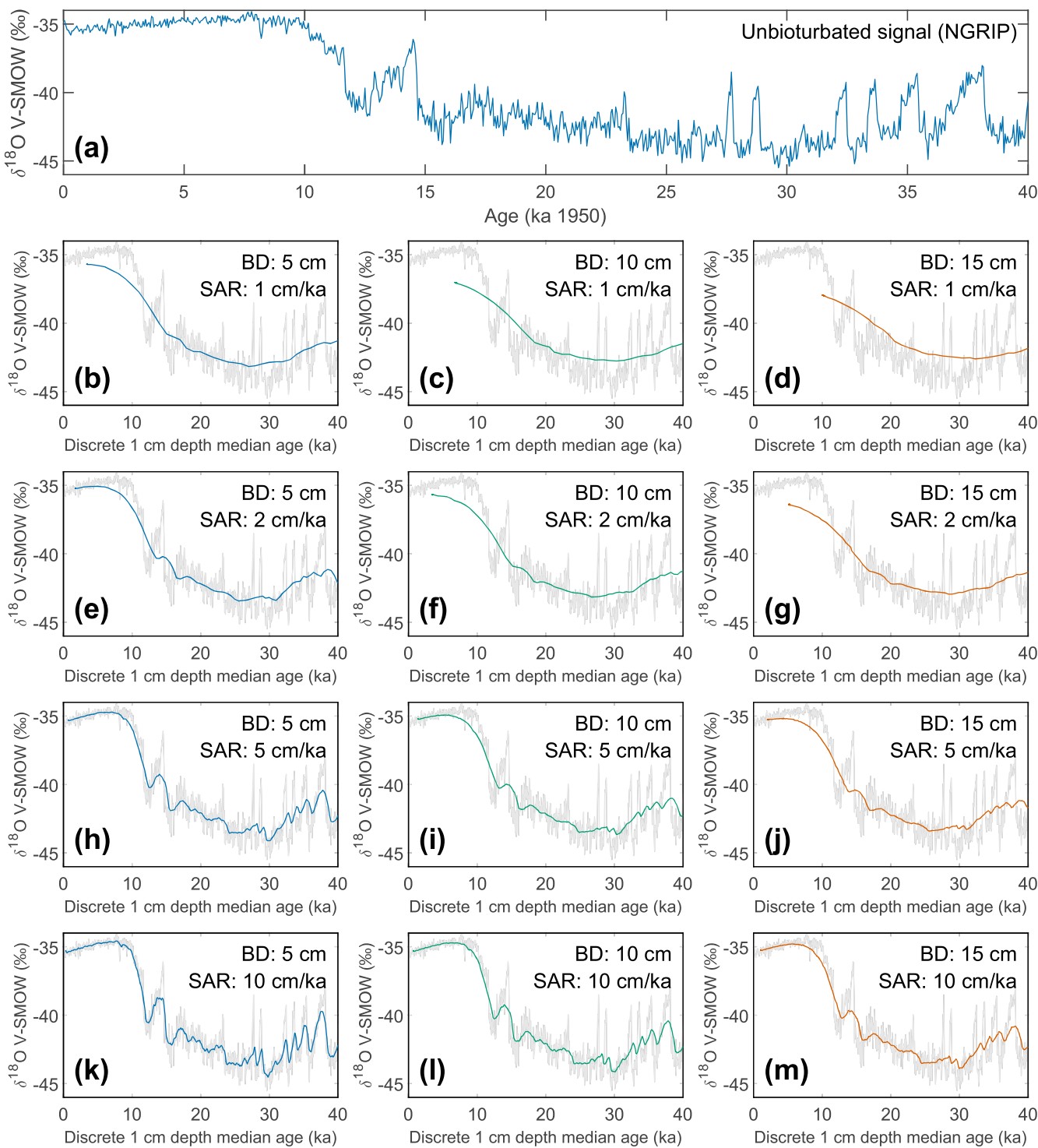

Example of bioturbation of a reference climate signal by typical Pacific SAR

**Figure 9 Histograms of simulated specimen age within the bioturbation depth. The simulated age distribution present within the sediment mixed layer, referred to here as bioturbation depth (BD). BD is fixed at (A, D, G, J) 5 cm, (B, E, H, K) 10 cm and (C, F, I, L) 15 cm for sedimentation accumulation rates (SAR) of (A-C) 1 cm kyr$^{-1}$; (D-F) 2 cm kyr$^{-1}$; (G-I) 5 cm kyr$^{-1}$ and (J-L) 10 cm kyr$^{-1}$. The output is plotted as the discrete 1 cm depth median age. Note the size of the BD varies, therefore the simulated age distribution comes from a varying 'core depth'.**

# Simulated single specimen age distribution for typical Pacific SAR

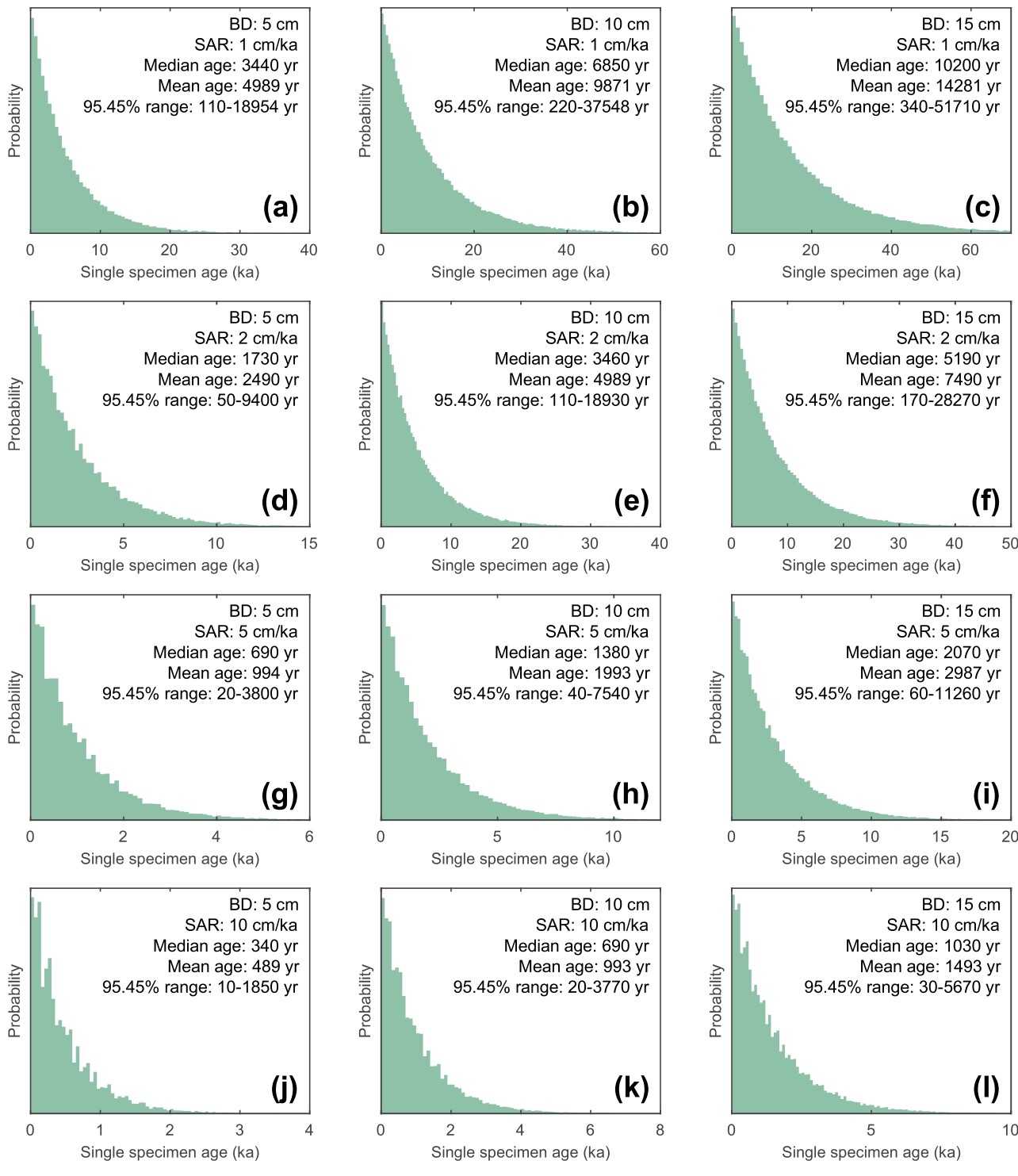

**Table Caption**

**Table 1. Data-model comparison.**

| FAME genus | FAME species | Model iteration cut-off depth (in m) | Total Standard deviation in ‰ Closest Grid Cell Value | Total Mean of 3*3 grid | Total Range of 3*3 grid | Interannual Standard deviation in ‰ Closest Grid Cell Value | Interannual Mean of 3*3 grid | Interannual Range of 3*3 grid | % difference Closest Grid Cell Value | % difference Mean of 3*3 grid | % difference Range of 3*3 grid | Data reference | Data | Data (age in kyr) | iCESM model Eulerian View (50 m); Standard deviation in ‰ | iCESM model Eulerian View (100 m); Standard deviation in ‰ | iCESM Lagrangian View; Standard deviation in ‰ |
|---|---|---|---|---|---|---|---|---|---|---|---|---|---|---|---|---|---|
| Globigerinodes | sacculifer | 60 | 0.32 | 0.33 | 0.03 | 0.29 | 0.29 | 0.05 | 0.90 | 0.89 | 0.10 | | | | 0.40 | 0.60 | 0.49 |
| Globigerinodes | sacculifer | 60 | 0.38 | 0.42 | 0.16 | 0.31 | 0.33 | 0.09 | 0.82 | 0.79 | 0.17 | | | | 0.53 | 0.75 | 0.35 |
| Globigerinodes | ruber | 60 | 0.26 | 0.27 | 0.01 | 0.23 | 0.23 | 0.03 | 0.88 | 0.88 | 0.13 | | | | 0.40 | 0.60 | 0.49 |
| Globigerinodes | ruber | 60 | 0.33 | 0.37 | 0.13 | 0.26 | 0.27 | 0.08 | 0.79 | 0.75 | 0.20 | B | 0.51 | 1.1 or 1.6 | 0.53 | 0.75 | 0.35 |
| Neogloboquadrina | dutertrei | 60 | 0.32 | 0.32 | 0.06 | 0.26 | 0.26 | 0.02 | 0.79 | 0.82 | 0.17 | A | 0.38 | 1.5 | 0.40 | 0.60 | 0.49 |
| Neogloboquadrina | dutertrei | 60 | 0.41 | 0.45 | 0.18 | 0.33 | 0.35 | 0.11 | 0.81 | 0.78 | 0.16 | C | 0.28 | 1.6 | 0.53 | 0.75 | 0.35 |
| Globigerinodes | sacculifer | 100 | 0.25 | 0.26 | 0.03 | 0.22 | 0.22 | 0.04 | 0.88 | 0.87 | 0.12 | | | | 0.40 | 0.60 | 0.49 |
| Globigerinodes | sacculifer | 100 | 0.33 | 0.36 | 0.14 | 0.28 | 0.29 | 0.08 | 0.84 | 0.81 | 0.16 | | | | 0.53 | 0.75 | 0.35 |
| Globigerinodes | ruber | 100 | 0.20 | 0.21 | 0.07 | 0.16 | 0.16 | 0.02 | 0.80 | 0.80 | 0.23 | | | | 0.40 | 0.60 | 0.49 |
| Globigerinodes | ruber | 100 | 0.27 | 0.31 | 0.11 | 0.22 | 0.22 | 0.08 | 0.79 | 0.73 | 0.23 | B | 0.51 | 1.1 or 1.6 | 0.53 | 0.75 | 0.35 |
| Neogloboquadrina | dutertrei | 100 | 0.29 | 0.29 | 0.05 | 0.23 | 0.23 | 0.02 | 0.79 | 0.82 | 0.17 | A | 0.38 | 1.5 | 0.40 | 0.60 | 0.49 |
| Neogloboquadrina | dutertrei | 100 | 0.40 | 0.43 | 0.15 | 0.33 | 0.34 | 0.09 | 0.83 | 0.81 | 0.11 | C | 0.28 | 1.6 | 0.53 | 0.75 | 0.35 |
| Globigerinodes | sacculifer | 200 | 0.21 | 0.22 | 0.04 | 0.17 | 0.18 | 0.02 | 0.83 | 0.81 | 0.20 | | | | 0.40 | 0.60 | 0.49 |
| Globigerinodes | sacculifer | 200 | 0.28 | 0.31 | 0.11 | 0.23 | 0.25 | 0.09 | 0.83 | 0.78 | 0.21 | | | | 0.53 | 0.75 | 0.35 |
| Globigerinodes | ruber | 200 | 0.20 | 0.20 | 0.09 | 0.16 | 0.16 | 0.03 | 0.78 | 0.79 | 0.24 | | | | 0.40 | 0.60 | 0.49 |
| Globigerinodes | ruber | 200 | 0.25 | 0.29 | 0.10 | 0.18 | 0.20 | 0.07 | 0.74 | 0.69 | 0.27 | B | 0.51 | 1.1 or 1.6 | 0.53 | 0.75 | 0.35 |
| Neogloboquadrina | dutertrei | 200 | 0.25 | 0.25 | 0.05 | 0.20 | 0.20 | 0.02 | 0.78 | 0.81 | 0.16 | A | 0.38 | 1.5 | 0.40 | 0.60 | 0.49 |
| Neogloboquadrina | dutertrei | 200 | 0.35 | 0.37 | 0.11 | 0.30 | 0.31 | 0.08 | 0.85 | 0.83 | 0.10 | C | 0.28 | 1.6 | 0.53 | 0.75 | 0.35 |
| Globigerinodes | sacculifer | 400 | 0.21 | 0.22 | 0.04 | 0.17 | 0.18 | 0.02 | 0.83 | 0.81 | 0.20 | | | | 0.40 | 0.60 | 0.49 |
| Globigerinodes | sacculifer | 400 | 0.28 | 0.31 | 0.11 | 0.23 | 0.24 | 0.09 | 0.83 | 0.78 | 0.21 | | | | 0.53 | 0.75 | 0.35 |
| Globigerinodes | ruber | 400 | 0.20 | 0.20 | 0.09 | 0.16 | 0.16 | 0.03 | 0.78 | 0.79 | 0.24 | | | | 0.40 | 0.60 | 0.49 |
| Globigerinodes | ruber | 400 | 0.25 | 0.29 | 0.10 | 0.18 | 0.20 | 0.07 | 0.74 | 0.69 | 0.27 | B | 0.51 | 1.1 or 1.6 | 0.53 | 0.75 | 0.35 |
| Neogloboquadrina | dutertrei | 400 | 0.24 | 0.23 | 0.05 | 0.18 | 0.19 | 0.02 | 0.77 | 0.80 | 0.17 | A | 0.38 | 1.5 | 0.40 | 0.60 | 0.49 |
| Neogloboquadrina | dutertrei | 400 | 0.34 | 0.36 | 0.11 | 0.29 | 0.30 | 0.07 | 0.85 | 0.83 | 0.10 | C | 0.28 | 1.6 | 0.53 | 0.75 | 0.35 |

Data from:

| | | | |
|---|---|---|---|
| A | Leduc et al., 2009; | B | Koutavas and Joanides, 2012; | C | Sadekov et al., 2013 |

Model (iCESM) values from supplement of Zhu et al., 2017 (converted from variance)

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
