# Peer review of "A proxy modelling approach to assess the potential of extracting ENSO signal from tropical Pacific planktonic foraminifera"

_Climate of the Past, 2019_

## Referee Comment (RC1) · Anonymous Referee #1 · 25 Mar 2019

SUMMARY

This manuscript by Metcalfe et al. tries to understand where in the tropical Pacific Ocean the analysis of distributions of individual foraminifera geochemistry results (d18Oc, T(MgCa)) could be used to find changes in the El Niño-Southern Oscillation. This type of analysis has been done before (Thirumalai, Ford, White), with a focus on the inverse problem of estimating ENSO change from individual foraminifera distributions. Here the novelty is the inclusion of a forward model of foraminifera growth rate. This model is used to estimate the biased sampling in depth and time that different foraminiferal species have, and how this contributes to the analysis of the ENSO signal. However, that part of the model is not really validated and it is unclear how much it adds to the analysis. Furthermore, the statistical analysis focuses on a forward prob-

lem rather than the inverse problem that is the real challenge for detecting changing ENSO from individual foraminiferal analysis. The forward problem is whether El Nino, neutral, and La Nina months have different distributions and requires that each individual d18Oc or T value be assigned beforehand to one of those three states. The inverse problem is to determine from comparison of two different d18Oc or T distributions (as would be measured in two sediment samples) whether any change in their distributions occurred and whether it can be ascribed to changes in the statistics of ENSO events (frequency, magnitude). Finally, there are also additional questions in the author's methodology that are opaque and need to be clarified.

As it stands, the focus on the forward problem and on statistical approaches not used for paleo-IF analysis make the manuscript in its present form not a good evaluation of the IF approach for ENSO reconstruction. The title is misleading and the abstract mis-states the conclusions from the study. For the reasons above and detailed below it is difficult to evaluate the utility and applicability of the manuscript to the questions the authors raise. With different analyses the authors could address the questions they pose. However, it could be very different from the manuscript in its current form and in my opinion would need to independently evaluated and reviewed.

GENERAL COMMENTS

1. Make sure that the abstract and conclusions follow from the analyses and are properly stated (see specific notes below).

2. Validate growth rate forward model

Validate the growth rate calculation through comparison with sedimentary relative abundances. This was done to some extent in the paper cited for the foraminifera model (Roche et al., 2018) but in that paper no clear assessment of the errors was presented. The model in Roche et al., 2018 is a simplification of earlier growth rate modeling of foraminifera. In Roche 2017 all parameters besides temperature are discarded. How well then does the model work? I think the authors should use the modeled growth

rate for the species they are targeting and calculate the relative abundance of those three species in a sediment sample. This can be compared to the measured relative abundance of those three species (summing to one) recalculated from their relative abundance amongst all species counted in coretop datasets. This should be shown as a scatterplot of observed vs. predicted on x- and y-axes rather than on a map as is shown in the supplement to Roche et al., 2018.

3. Validate the d18Oc predictions

Validate the d18Oc predictions from the growth rate and geochemistry model. This was done in Roche et al., 2017 but is also somewhat circular because the sedimentary d18O values were used to determine the depth of production. I admit I am not sure how to actually validate the approach except from an additional validation dataset not used for determining production depth.

4. Include the analysis of Tc for Mg/Ca reconstructions

Inexplicably the authors refuse to analyze the temperature distributions even though those are the data from the common Mg/Ca method of individual foraminifera analysis (Sadekov, Ford, White). The author's stated reason is due to "...the complexity in reconstructions of trace metal geochemistry...and the potential error associated with determining which carbonate phase is first used when foraminifera biomineralise...". While there are ongoing methodological and calibration efforts for this and other proxies (including d18Oc), to ignore such a widespread type of analysis seems very shortsighted. If the authors do not want to forward model the Mg/Ca proxy itself they can simply analyse the temperatures in their dataset. Either way this is something that should be included in the manuscript.

5. Remove maps of carbonate preservation/depth

It is fine for the authors to state the general problem in the text, but there are regions of shallow depth were carbonate is preserved that are not captured in the coarse DEM

used.

6. Remove map of sedimentation rate

Either quantitatively discuss the role of sedimentation rate and bioturbation or remove this map. The sedimentation rate threshold is intimately tied to the secular and non-ENSO variability and thus is a much more complicated analysis than the general discussion in the text. I think the discussion is a starting point but the author's miss that the important factors are really the magnitude of other, non-ENSO sources of variability at the timescale of a sediment sample (plus bioturbation) compared to the magnitude of the ENSO change signal and the non-ENSO variability.

7. Focus in the inverse problem

It is really the inverse problem of detecting a change in ENSO from a change in the distribution of foraminifera d18Oc or T that is the focus of IF ENSO reconstructions. The analysis in this paper basically asks the question: are the distributions from El Niño months different from neutral or La Niña months? This is a useful first step in the inverse problem but it doesn't really answer the question stated in the title about the validity of foraminifera-based ENSO reconstructions.

8. Apply statistical tests on parameters used on paleo-IF distributions

The author's use Anderson-Darling tests for differences in distributions. They should demonstrate how this might be useful for paleo-IF analysis. It would also be greatly to their advantage to test the approaches actually used for paleo-IF analysis (1-sigma, quantiles) to see how they perform in this framework. A welcome contribution would be demonstration that a new/different type analysis from those typically applied to paleo-IF distributions is better. As it stands, the focus on the forward problem and on statistical approaches not used for paleo-IF analysis make the manuscript in its present form not a good evaluation of the IF approach for ENSO reconstruction.

9. Definition of El Niño, neutral, and La Niña months

[Figure]

There is a large body of literature and accepted methods for defining El Niño, neutral, and La Niña periods. In the text the authors take a simplistic approach, but there is no reason for this. Why not actually use the societal and dynamically important definitions of these events including the requirement of a minimum consecutive number of months of anomalies and changing baseline for anomalies (to account for secular warming of the ocean)? This definition has a basis in theory as an El Nino (La Nina) event unfolds over a length of time and thus a single month anomaly may not be associated with the dynamics that are part of the coupled ENSO system.

10. Clearly separate the role that the growth model and (T,S) timeseries play in identifying ENSO change.

To what degree are the outcomes and conclusions of this paper depending upon the modeled growth rates versus the sea water properties (T,S,d18Ow)? Many prior workers have analyzed in different ways the reconstruction of ENSO from IF analysis. These approaches include summary statistics like the standard deviation (Thirumalai; Koutavas; Leduc; Sadekov; Rustic), as well as examination of changes in the quantiles of IF distributions (Ford; White). What is added here is the foraminifera growth rate weighting. What effect does this have? From the histograms in Roche et al. (2018) it appears that the growth-rate weighting does not have major consequences for the mean d18Oc value of a sediment sample. It may have consequences for the IF variability though. The authors could show a map that quantifies the growth-rate weighting effect with respect to the non-weighted results (ratio, difference).

11. Examine how ENSO amplitude vs. frequency change IF distributions

The authors raise an interesting point in their conclusion that has not been well addressed, namely how do changes in the statistics of ENSO (frequency, amplitude) affect IF distributions and reconstructions of ENSO variability. Evaluating these two different questions would be an important contribution to IF analysis of ENSO change. But, introducing the idea in the conclusions without a previous discussion in the manuscript

is not a good idea in my opinion.

SPECIFIC COMMENTS ON TEXT

– Abstract –

Page 1 line 15 "Our results show that it is possible to use d18Oc from foraminifera to disentangle the ENSO signal only in certain parts of the Pacific Ocean." This line in the abstract is sharply in contrast to the line in the conclusion at Page 12 line 21 "Overall, our results suggest that foraminiferal d18O for a large part of the Pacific Ocean can be used to reconstruct ENSO." Which is it?

Page 1 line 17 – "Furthermore, a large proportion of these areas coincide with sea-floor regions exhibiting a low sedimentation rate and/or water depth below the carbonate compensation depth, thus precluding the extraction of a temporally valid palaeoclimate signal using long-standing palaeoceanographic methods." The role of sedimentation rate in IF analysis is important but there is not any investigation of this effect in the present manuscript so it is not really a conclusion or finding. This statement should not be included in the paper in its present form.

Page 1 line 17 – "Furthermore, a large proportion of these areas coincide with sea-floor regions exhibiting a low sedimentation rate and/or water depth below the carbonate compensation depth, thus precluding the extraction of a temporally valid palaeoclimate signal using long-standing palaeoceanographic methods." The role of water depth and carbonate preservation is also important. But, there is not any investigation of the sedimentation rate effect in the present manuscript so it is not really a conclusion or finding. Furthermore, there are seamounts and other shallow sites not captured in the gridded dataset that can contain records for palaeoceanographic investigations. This statement should not be included in the paper in its present form.

– Main Text –

Page 3 line 23 – Here the authors introduce the 1-sigma d18Oc parameter than has

been used in some studies to look at changes in ENSO variance. But, they never really address whether this parameter is useful and can detect changes in ENSO. Thirumalai et al. (2013) took this question on already. More discussion of what has been done previously is needed. Also, why not test the actual way that IF analysis is used (e.g. 1-sigma, quantiles etc.) rather than a new method as introduced here (Anderson-Darling test)?

Page 4 line 1 – The new model for foraminifera growth only uses the temperature component of the previous model. Why? How different are the results?

Page 4 line 15 – Allowing symbiont-bearing foraminifera to possibly grow to 400 m simply based upon optimal temperatures seems not correct. They need to be in the photic zone.

–Methods–

Page 5 line 5 – The conversion of VSMOW to VPDB looks to be in error. The correct formula for this conversion is [d18O_VSMOW+1]/[d18O_VPDB+1] = 1.03091 where d18O does not include the 10ˆ3 term. Thus d18O_VPDB = d18O_VSMOW/1.03091 +(1/1.03091)-1 or d18O_VPDB = 0.97002*d18O_VSMOW - 0.02998. In d18O expressed with the 10ˆ3 term, the equation would read: d18O_VPDB = 0.97002*d18O_VSMOW - 29.98.

Page 5 line 10 – Why was growth rate arbitrarily constrained to these different depths? First, foraminifera with algal symbionts should be in the photic zone. Second, didn't the Roche et al., 2018 paper try to identify the depth-production relationship for the different species from the predicted d18Oc and measured MARGO d18Oc? Why not use those depths?

Page 6 line 1 – "…these for now can be ignored." Why can the other factors determining foraminifera growth be ignored? This cannot be a statement unless it is backed up. Or, the authors use only temperature but then go through an appropriate validation

process (more than what is shown in Roche et al., 2018) as suggested above.

Page 6 line 11 – Starting here, it is very unclear how and why the particular set of conditions for El Niño, La Niña, and neutral periods were chosen. What time series of sea surface temperatures were chosen for computing anomalies (in each grid square, Nino 3.4, Nino 3, Nino 4, etc.)? Were the anomalies based upon a 3-month running mean? Were the anomalies computed relative to a fixed period or, as is now the accepted approach, relative to 5-year interval means? Why not use the definition of El Nino etc. events that include the requirement for consecutive months of anomalies? This definition has a basis in theory as an El Nino (La Nina) event unfolds over a length of time and thus a single month anomaly may not be associated with the dynamics that are part of the coupled ENSO system.

Page 6 line 18 – Why and how was the pdf/cdf from the actual data fitted and smoothed with an Epanechnikov kernel? What impact did this fitting and smoothing (particularly the choice of bandwidth) have on the Anderson-Darling test and the results overall?

Page 6 line 24 – This paragraph is very unclear and the errors associated with binning prior to analysis of the pdf seem avoidable. For example, why not take the growth rate in each of the 696 months in each grid at each depth, and scale the growth rate to calculate an effective # of individuals such that they sum to 1000 across all months? Round those numbers to integers and then use the integer # of individuals for each month to replicate that actual months Tc or d18Oc value. The resulting ordered list of values can then be binned/smoothed etc. and represents a pseudo-distribution that one might find in a sediment sample?

–Results–

Page 7 line 3 – It says that the mean d18Oc for El Nino and neutral months are compared. How? Earlier and later it is stated that the A-D test is applied to compare distributions. What is meant by these lines?

Page 7 line 5 – "...ENSO events can potentially be detected by paleoceanographers and unmixed using, for example, a simple mixing algorithm with individual foraminiferal analysis..." This is not really practicable because it assumes complete stationarity in the El Nino, La Nina, and neutral distribution. This is unlikely as all are expected to change, and do in models and data (e.g. coral time series from middle Holocene show changed seasonal amplitude and ENSO cycles).

Page 7 line 7 –"In cases where FPEN and FPNEU do not exhibit significantly different means, then the chosen species and/or location represent a poor choice to study ENSO dynamics." This may not always be the case because the mean values could be similar but the distributions wildly different (such as long tails with different signs). Changing numbers of El Nino and neutral and La Nina events could that quite dramatically change the shape of the combined distribution that is ultimately preserved in sediments. And, it may be possible to find regions of such a distribution that can be used to diagnose changing ENSO.

Page 7 line 20 – Why is Anderson-Darling test done here but the mean values are discussed above? If the A-D test shows that the El Nino and Neutral distributions are different (at some statistical level) then that means alteration of those distributions (more/fewer, stronger/weaker events) would alter the summed distributions that one gets from a sediment sample. But, how would this actually be detected in the sediment sample? That the AD test demonstrates the El Nino, Neutral, and/or La Nina distributions are different is helpful but it does not get at whether ENSO change could actually be detected in a sediment sample.

Page 7 line 26 – Applying a 1-sigma value from modeled minus coretop comparisons to the AD test value does not seem appropriate. This value assesses the accuracy of the model in predicting the absolute value of the mean of a coretop sample. But it is not an appropriate estimate for the significance of the difference between two different IF values or the difference in the AD statistic.

Page 8 line 1 – This paragraph is rather confusing to understand. It sounds like the authors are comparing a depth-weighted reconstruction and non-depth weighted reconstructions at fixed depths (Fig. 3 vs. 4)?

Page 8 line 9 – Unclear what "on the low-end" means.

Page 8 line 16 – "...a large percentage of the tropical Pacific remains accessible to palaeoclimate studies." This is very much not the message in the Abstract and from the title of the paper. Those sections should reflect this finding.

Page 8 line 25 – "Indeed, one should view discrete sediment intervals, and the foraminifera contained within them, as representative of an integrated multi-decadal or even multi-centennial signal..." This is exactly how foraminifera paleo-IF studies have viewed them and should be stated up front (start of paragraph for instance).

Page 8 line 28 – "Therefore, in order to reliably extract short-term environmental information from foraminiferal-based proxies, the signal that one is testing or aiming to recover must exhibit a large enough amplitude in order to perturb the population by a significant degree from the background signal, otherwise it will be lost due to the smoothing effect of bioturbation..." This statement does not make sense to me. The background signal IS the signal, i.e. the seasonal cycle, ENSO etc. Changes in ENSO must be such that they alter that signal (the distribution of IF analyses), but bioturbation etc. should not erase the signal unless one is looking for short periods of change less than the time integrated into the sample.

Page 9 line 6 – "...a series of high magnitude, but low frequency El Niño events could be smoothed out of the downcore, discrete-depth record." They will not be smoothed out as the authors state. Those anomalous IF values may be rare, but will be present in the sediment sample and if measured can be used to examine changing ENSO.

Page 9 line 7 – The sediment accumulation rate needed to observe/reconstruct changes in ENSO is not fixed. It depends upon the magnitude and duration of secular

trends, and variability with respect to both the time integrated in a sediment sample and the magnitude of the ENSO signal and its change. This is a quite interesting but also complicated subject and arbitrarily cutting the sedimentation rate at 5 cm/ky is not justified.

Page 9 line 9 – The map of water depth is quite coarse and misses important locations that are above the CCD, accumulate carbonate (and foraminifera), and can be used for palaeoceanographic reconstructions. Thus, while the overall point is true, the map as shown is misleading.

–Discussion–

Page 9 line 20 – Why discount trace metal temperatures (Mg/Ca)? At least use the Tc analysis as a stand-in for Mg/Ca.

Page 9 line 23 – The focus of this paper is on IF analysis. Why are the Koutavas and Lynch- Stieglitz, 2003; Koutavas and Lynch-Stieglitz, 2003 etc. cited here? The whole discussion in this paragraph, lines 16-31 feels out of place.

Page 10 line 3 – The references to Cole and Tudhope, 2017; White et al., 2018 seem to be in error. These papers do not discuss lake core colour etc.

Page 10 line 3 – "If the number and magnitude of ENSO events were reduced, the relatively low downcore resolution of marine records may not accurately capture the dynamics of such lower amplitude ENSO events using existing methods." This statement is not justified by the author's analysis or a citation.

Page 10 line 5 – "The possibility of a marine sediment archive being able to reconstruct ENSO dynamics comes down to several fundamentals: the time-period captured by the sediment intervals (a combination of SAR and bioturbation), the frequency and intensity of ENSO events, as well as the foraminiferal abundance during ENSO and non-ENSO conditions." Also included is the magnitude of change in ENSO statistics and resulting foramifera Tc or d18Oc, sampling uncertainty on the IF distribution. See

also note above on the role of sedimentation rate.

Page 10 line 9 – "The results presented here imply that much of the Pacific Ocean is not suitable for reconstructing ENSO studies using palaeoceanography, yet several studies have exposed shifts within $\sigma$(d18Oc) of surface and thermocline dwelling foraminifera. One can, therefore, question what is being reconstructed in such studies." The results presented here don't really test whether individual foraminifera d18Oc (or Tc) studies can reconstruct ENSO. Furthermore, the water depth and sedimentation rate constraints are the reason for excluding much of the Pacific. This statement is therefore incorrect and the search for other explanations does not follow.

Page 10 line 19 – This second part of the paragraph is interesting and has been commented on before. But, at no point do the authors actually evaluate any of these effects or approaches so they can't really assess the different factors they raise here.

Page 10 line 28 – The discussion of model limitations does not ask what would seem to be the most important questions: Does the modeled growth rate actually reflect the real ocean (and the sampling bias for what is recorded in sediments)? Do the modeled growth-rate weighted d18O distributions match actual measured individual foraminifera d18O distributions (such as in Koutavas and Joanides or Rustic)? If no growth-rate weighting is applied are the results better or worse?

Page 11 line 14 – Why are the authors so dismissive of Mg/Ca analyses? The list of possible complications is important but it remains a fundamental observation that the Mg/Ca of foraminiferal calcite changes with growth temperature and has been validated in many different ways.

–Conclusion–

Page 12 line 17 – "Previous work. . ." The only citation here is to Zhu et al., 2017. There has been a lot of work comparing IFA different time slices (both d18Oc and Mc/Ca) that should be cited here (Koutavas et al., Leduc, Koutavas and Joanides, Sadekov et al,

Ford et al, Rustic et al, White et al). Furthermore, they have not all used 1-sigma d18Oc as the metric for detecting change.

Page 12 line 21 – "Overall, our results suggest that foraminiferal $\delta$18O for a large part of the Pacific Ocean can be used to reconstruct ENSO. . ." This contradicts what is said in the abstract and in some places in the text (but is similar to in other places in the text). Which is it?

Page 12 line 22 – What is meant by ". . .especially if an individual foraminiferal analysis. . . approach is used. . ." I thought the whole analysis in the paper was on whether individual foraminifera analysis can be used? Was there another method tested (for example the means analysis referred to at Page 7 line 3)?

Page 12 line 24 – "However, the sedimentation rate of ocean sediments in the region is notoriously slow (Olson et al., 2016) and much of the ocean floor is under the CCD. These factors reduce the size of the area available for reconstructions considerably (Lougheed et al., 2018), thus precluding the extraction of a temporally valid palaeoclimate signal using long-standing methods." This is generally true, but there are seamounts and other regions that may actually preserve carbonate. Furthermore, the sedimentation rate constraint is also somewhat arbitrary and depends upon secular trends and non-ENSO variability encompassed in a particular sample.

Page 12, line 27 – "We further highlight that the conclusions drawn from foraminiferal reconstructions should consider both the frequency and magnitude of El Niño events during the corresponding sediment time interval (with full error) to fully understand whether or not a strengthening or dampening occurred." While this is true, nowhere in the manuscript is this issue addressed. Inclusion as a conclusion to the paper is therefore not warranted.

Page 12 line 30 – "The use of ecophysiological models. . .are not limited to foraminifera and provide an important way to test whether proxies used for palaeoclimate reconstructions are suitable for the given research question." This is not really a conclusion

of the study. And, given the uncertainties and lack of rigorous testing of the foraminifera model in this study, this is a questionable statement overall.

–Figures–

Figure 3 – Why are there white and grey areas that mean the same thing?

Figure 4 – Are the temperature data growth weighted? What species? If not, why not analyze the Tc data in parallel to the d18Oc data to evaluate what advantage/disadvantage the two different signals have (e.g. from S).

Figure 5 – Why are the white and grey areas grouped together? What do they mean? Are these panels based upon growth-rate weighted values?

Figure 6 – Are these panels based upon growth-rate weighted values?

---

## Referee Comment (RC2) · Anonymous Referee #2 · 4 Apr 2019

In their manuscript Metcalfe et al. present a forward modeling approach through FAME to investigate the use of individual foraminifera analysis (IFA) for ENSO reconstruction. Based on the modeling results, they conclude that this proxy is only valid in part of the Pacific Ocean. However, these regions are often characterized by low sedimentation rate, therefore limiting the use of this proxy.

While the effort to incorporate forward models into paleoceanographic studies is commendable, I fail to see the practical application of this study. Inverse modeling would be impossible and the lack of comparison between the pseudo-proxy distributions and actual distributions of foraminifera prevents validation of the method.

Major comments

[Figure]

Inverse problem

The manuscript focuses on forward modeling of IFA analysis. Although definitely a valuable exercise for data-model comparison (assuming that the climate model can make use of the forward model), it doesn't solve the inverse problem. It would be almost impossible to evaluate the growth factor in the d18O record.

It's also not visually obvious what the difference between the output of a non-weighted model is vs FAME in Figure S1. Some statistics would help, or plotting the resulting kernel distributions on a separate panel,

Further, bioturbation is also likely to have a large impact on IFA, especially in areas of low sediment accumulation. Why not connect FAME to a bioturbation model and disentangle the influence of these factors?

Statistical analysis

Page 6, Line 25: Multiplying the bin counts will effectively skewed the results of a significance test. In practice, it would be impractical if not impossible to obtain 1000 samples in each bin. Similarly, page 7, line 4, how many foraminifera were artificial picked to produce these maps?

IFA model - data comparison

There are a number of recent studies with IFA results from the past ∼1000 years (some of them cited in the current manuscript). How do these distributions compare to the statistical ones?

Effect of SAR

Since a model of bioturbation was not implemented here, it's hard to examine the effect of bioturbation on the IFA. Furthermore, rapid accumulation rates should be possible around islands. The coarse map overlaid here fails to account for these. I would suggest adding to the text that in strategic locations (in the blue areas), sedimentation

rates may still be high enough.

Improper referencing

This is not the first study to use pseudo-proxy to examine whether IFA can be used for ENSO reconstruction. Thirumalai et al. present a model that can be more easily applied to a real application. First, reference this study (and others) at the beginning of the manuscript and second, why not extend their "picking" model to also evaluate the contributions of sample size?

Minor comments

Abstract: Should state that this is an IFA technique. Page 1, Line 23: specify that the interaction on interannual timescale is known as ENSO. On decadal, it's known as the PDO. Page 1, Line 27: SO is part of ENSO. Should rephrase as we have long instrumental records of the atmospheric variability but not the ocean. Page 3, line 3: Stott et al is not the only reconstruction in the Western Pacific, either use e.g., or as done previously cite multiple sources. Page 3, Line 30: Mg/Ca is not a simple function of temperature. There is a growing body of evidence that suggests that Mg/Ca is also sensitive to salinity and pH. In addition, the calcite saturation of the bottom waters on post-depositional preservation of the signal. Page 5, Line 3: Why not used species-specific equations? Page 5, Line 12: Not sure what is meant by "Which can compute eight foraminiferal species". Do you mean growth? Page 6, line 11: There is an abundant body of literature dealing with the definition of an ENSO event. Why not start there? Page 9, line 20-25: Most of these studies are based on pooled samples and were referencing to an ENSO-like signal rather than the interannual mode of variability that IFA is targeting.

---

## Referee Comment (RC3) · Anonymous Referee #3 · 11 Apr 2019

In this study, Metcalfe et al. aim to test whether the approach of using individual foraminifera analysis (IFA) can be used to assess ENSO variability. In order to accomplish this, they use the Foraminifera as Modeled Entities (FAME) model to calculate idealized foraminifera distributions across the tropical Pacific. These results are then combined with seafloor/ CCD depth and sedimentation rate to determine which regions of the Pacific Ocean are suitable targets for IFA approaches. Modeling of foraminifera populations in order to determine if ENSO change is detectable has been done before (e.g., Thirumalai 2013, White 2018), although these studies focus on the detection of ENSO from paleoclimate proxy records. This study's novel contribution is the inclusion of the FAME model and foraminiferal growth rates to the analysis of modeled response of biological calcite to tropical variability. However, the FAME portion of the model is

not validated against core-top data from the tropical Pacific, precluding assessment of its utility. The application of these results is likewise problematic, as it focuses on determining whether ENSO events (El Niño, La Niña) and neutral conditions have distinct distributions (forward modeling) rather than on how one could detect ENSO change (inverse modeling). Further, the discussion on sedimentation rate and CCD is broad-based and does not take in to consideration local changes in seafloor topography, changes in bottom-water oxygen availability that may alter bioturbation depths, and the variability characteristics of different regions with regard to the seasonal cycle, decadal-centennial variability, and ENSO change (e.g., Thirumalai 2013, Ford 2015, White 2018). Finally, there are aspects of the model that are unrealistic (e.g., a 400m depth for symbiont-bearing foraminifera; assuming sample sizes of 1000 for binning) or unrealized (e.g., how many individuals were selected for generating these estimates and a lack of model-data comparison) that present significant issues to the overall utility of this model for paleoceanographic reconstruction of ENSO from IFA. The title of the article does not represent the content or main goals of the study, and the conclusions stated in the abstract are different than those in the main paper. The questions the authors raise are valid and useful, but the results as stated do not support their conclusions. In fact, the stated conclusions of the article are, in several places, contradicted within the paper itself. These contradictions are not well-explained, and thus a clear summary of the findings is difficult to parse.

General Comments

The study here focuses on forward modeling using FAME for IFA. However, the authors fail to prove whether existing IFA-ENSO reconstructions are valid or provide the tools for evaluating proxy data (e.g., the "inverse problem", as mentioned in other reviews, whereby foraminifera records are analyzed to infer ENSO). Thus the application of these results to the paleodata world is limited. The more relevant application here is in targeting locations for performing IFA studies, but this is limited as well, as the sedimentological and bioturbation properties of regions across the Pacific are much

more variable captured here. The authors use their own definition of ENSO events, despite significant previous literature and established definitions that are commonly used. The use of single month anomalies does not adequately represent the actual ENSO phenomenon, which relies on ocean-atmosphere feedbacks expressed over a period of months, and thus their analysis of differences between El Niño, La Niña, and neutral conditions may be flawed and biased toward non-ENSO SST anomalies.

This study does not compare the results of their FAME analysis with existing IFA reconstructions of variability from the tropical Pacific. In the eastern Pacific, Rustic 2015 used $\delta$18O IFA on modern-era sediments to show close correspondence with calculated $\delta$18O from reanalysis data; in the central Pacific, White 2018 showed that the distributions of Mg/Ca-based SSTs from individual foraminifera in a 4ky coretop are statistically similar to modern reanalysis data.

Specific Comments

The authors focus on $\delta$18O proxies for IFA, and discount Mg/Ca reconstruction and the modeling efforts done with those (White 2018, Ford 2015). To discount Mg/Ca ratios as a paleoproxy without the kind of analysis provided for $\delta$18O seems premature . While changes in carbonate concentration, salinity, and preservation environment can indeed alter Mg/Ca ratios, significant study has been done and is underway to understand these roles. Species-specific calibrations and various corrections exist that are well quantified. Not using Mg/Ca for the Tc seems rather limited.

The number of foraminifera picked from a given sediment interval is an important component of IFA. Increasing bin counts to 1000 artificially (Page 6) does not represent the numbers typically used in such analyses; the numbers used for other analyses (Page 7) are not specified.

In the results, the first statistical test is to test whether the means of the FPen and FP-neu $\delta$18O distributions are different and use this to determine whether ENSO events can be detected. Comparison of the population means does not necessarily reflect

differences in the population distributions, and only provides a measure of mean conditions that may or may not be related to ENSO variability. The use of the Anderson-Darling test to assess differences in distribution is used later. It is unclear how these two different tests were related, and how the mean $\delta$18O FPen/neu was utilized. The author's use of the Anderson-Darling test to assess differences in distributions is novel, but results of this test are not compared to those that have been used to assess IFA results in previous studies (e.g., std dev (Thirumalai 2013, Koutavas and Joanides 2012, Rustic 2015) or Q-Q (White 2018, Ford 2015)). Is this more sensitive, less sensitive, or does it measure different aspects of the distribution change NOT captured in the other analyses? Without such comparison, the ability to assess the validity of IFA reconstruction (the purported goal of this paper) is limited.

The specifics of sedimentation rate and bioturbation vary greatly across the tropical Pacific and rely on multiple processes. The role that oxygen plays in bioturbation is important, especially as bottom-water oxygen levels vary across the tropical Pacific. Likewise, seafloor topography is highly variable, with ridges and sea mounts that are not apparent at the resolution used.

On P8: "Similarly, the individual characters of El Niño events, which are very short in duration, become lost in the bioturbated sediment record " The purpose of IFA is not to discern the properties of an individual event. Change in frequency or amplitude of events over a period of time can be statistically detected using various means to compare the distribution of integrated conditions over the period of sedimentation. Bioturbation serves, then, to extend that integrated time and the range of conditions experienced.

Bioturbation will also not remove anomalous values (page 9) – rather, such values may be present as part of a distribution representing more integrated time. Likewise, bioturbation has the effect of smoothing the signal, but the "signal" is a function of all sources of variability (ENSO, annual, decadal, centennial). The relative expression of these forms of variability along with the amount of time integrated by a sample are both

important in terms of the ability to capture ENSO signals.

On P.10, Cole and Tudhope (corals) and White et al (IFA) are cited in error when discussing lake colour intensity and precipitation-driven records.

Also on P10, the authors claim: "If the number and magnitude of ENSO events were reduced, the relatively low downcore resolution of marine records may not accurately capture the dynamics of such lower amplitude ENSO events using existing methods." – Which methods? Q-Q, std. dev, event counting, others? It's not entirely clear this is even referring to IFA reconstructions, as the records discussed previous are sedimentary, coral, and IFA (but noted as "precipitation driven", see above).

P.10: The possibility of a marine sediment archive being able to reconstruct ENSO dynamics comes down to several fundamentals: the time-period captured by the sediment intervals (a combination of SAR and bioturbation), the frequency and intensity of ENSO events, as well as the foraminiferal abundance during ENSO and non-ENSO conditions. This statement leaves out other key elements, including the relative expression of ENSO events, the seasonal cycle, and decade-and-longer variability. These elements are (arguably) more important for inverse modeling, where the ability to disentangle growth rates from other sources of variability is impossible, and thus the signatures of ENSO in such records need to be discerned.

A key point in the paper (P10) says "The results presented here imply that much of the Pacific Ocean is not suitable for reconstructing ENSO studies using paleoceanography, yet several studies have exposed shifts within std dev($\delta$18Oc) of surface and thermocline dwelling foraminifera. One can, therefore, question what is being reconstructed in such studies.". This study has, at this point, not tested whether the Std.dev of $\delta$18Oc from individual foraminifera have reconstructed ENSO (also, the wording of this sentence is odd).

The first paragraph of the discussion (p9) purports to be about paleoclimatological archives that "have been used to indirectly and directly study past ENSO". However, the

discussion is on mean-state reconstructions (Koutavas 2003, Dubois 2009). Koutavas 2003 is non-IFA mean-state reconstruction; likewise, the Dubois 2009 paper notes that "we prefer not to invoke any ENSO-like state for the glacial EEP based solely on our UK'37 SST." While it may be true that this result and Koutavas 2003 are at odds, this is not an issue of IFA or ENSO reconstruction, but rather aggregate analysis and mean -state reconstruction. Discussion of std.dev ENSO studies (modeled by Thirumalai, Koutavas 2006, Koutavas and Joanides 2012, Leduc 2009, Sadekov 2013, Rustic 2015) is not found, yet the following paragraph (see above) is largely about this approach. Further, significant discussion and analysis of IFA reconstructions of ENSO during the LGM is found in Ford 2015, which is not discussed here.

The main analysis uses an unrealistic mixed-layer depth of 400m for the models foraminifera. Symbiont -bearing forams (G. ruber and G. sacculifer) live in the photic zone, and thus modeling and analysis of these organisms should be constrained to these depths. The model results using the shallower depths and specific, photic zone depths (Figure 4, figure 5, Figure 6) show that much of the tropical Pacific is suitable for such analyses, provided adequate carbonate preservation. This is very much in contrast with the point made previously in the paper that much of the tropical Pacific is unsuitable. In these figures, confusingly, some figures show significant areas in white while others use gray for no discernable reason. The figures are also improperly labeled, according to the captions – in each figure, G. sacculifer is on the left, G. ruber is in the middle, and this is reversed in the caption. Which is which?

The conclusions are at odds with what is presented at various points in the paper. Specifically: "Overall, our results suggest that foraminiferal $\delta18O$ for a large part of the Pacific Ocean can be used to reconstruct ENSO, especially in an individual foraminifera Analysis approach is used, contrary to previous analysis (Thirumalai et al. 2013). This conclusion is contradicted in the abstract, and in various parts of the study (e.g., P10 – "the results presented here imply that much of the Pacific Ocean is not suitable for reconstruction ENSO studies with paleoceanography...") Which is it?

Again, Koutavas 2003 is cited here, but that is not an IFA study. In general, clearly noting which studies are IFA/ENSO and which are mean state / aggregate / non-IFA studies will clarify the discussion surrounding the use of IFA and IFA techniques to identify ENSO signals.

This study does not directly address the Thirumalai 2013 study, as presented. The role of seasonality does not appear to be well addressed in this study (a key factor of Thirumalai 2013), the questionable definition of ENSO events confounds direct comparison, and the lack of clarity on sampling rates and other facts precludes a direct comparison. If this was a goal in this analysis, the Thirumalai study should be discussed in detail at the beginning (and should be discussed, in any case, earlier when discussing approaches for quantifying the suitability of locations for ENSO reconstruction), and the differences between their approaches (e.g., forward vs. inverse modeling). Suitable criteria for comparison should be noted (e.g., std. dev. Vs A-D tests).

---

## Short Comment (SC1) · 30 Apr 2019

I have not been assigned to be a reviewer on this manuscript, yet a previously published paper authored by my colleagues and I entitled "Statistical constraints on El Niño Southern Oscillation reconstructions using individual foraminifera: A sensitivity analysis and me." (Thirumalai et al. 2013) is quite pertinent to this study. I wanted to point out some flawed rationale in this discussion paper especially considering the explicit lack of utilizing subsampling in their arguments concerning ENSO skill, seasonality, and individual foraminiferal reconstructions. I hope the authors are open to my comments and I would like to state that I am a big proponent of studies such as this one and that I support the FAMES approach.

- The authors' conception of reconstructing ENSO variability versus reconstructing mean state conditions in areas influenced by ENSO is critically flawed, especially with regards to commenting on and inferring changes in variability using individual foraminiferal analysis (IFA). Regardless of using Anderson-Darling statistical tests to assess whether subsampling occurs from their forward-modeled $\delta$18O and temperature histograms, their analyses completely discount that ENSO events are seasonally synchronized to the annual cycle, and much of the variance of subsampled IFA distributions across the Pacific Ocean represents this power. Foraminiferal reconstructions are sensitive to absolute temperatures and NOT to monthly anomalies of temperature. This has been clearly demonstrated previously (e.g., see Thirumalai et al. 2013) and also underpins that discussion of IFA is incomplete without discussing uncertainties in sampling (e.g., White et al. 2018). Thus, their arguments and results (as well as the abstract) need to be significantly revised with this in mind. If the authors want to comment on ENSO and foraminifera (as in their title), they MUST incorporate subsampling uncertainties and how this interacts with the seasonal cycle.

- By definition, the authors state that "FAME uses the associated temperature and $\delta$18Oeq at each grid cell to compute a time averaged $\delta$18Oc and Tc for a given species". In other words, the authors have shown in their analyses that ENSO events strongly alter the temperature and $\delta$18O in much of the tropical Pacific and that foraminiferal histograms (or foraminiferal distributions) are able to capture mean state conditions by sampling from these altered distributions (i.e., based on the utility of the Anderson-Darling test to account for histogram subsampling). Both of these aspects are well known. This, by no means, demonstrates a calculation of skill or validity of IFA-based ENSO reconstructions. Thus the title of the manuscript is inaccurate and misleading. For a demonstration of calculating ENSO skill in reconstructions, please read the literature: Carré et al. 2013, Emile-Geay and Tingly, 2017, Ford et al., 2014, Hereid et al. 2013, Khider et al. 2011, Tindall et al. 2017, Thirumalai et al. 2013, and so forth (only two of which are cited and not discussed). If this paper is proposed for revisions in this journal, I would strongly contend that the title of this paper should be

revised in addition to the recalculation or revision of their text wherein IFA-based ENSO skill is referred to (as opposed to mean state conditions that are influenced by ENSO.)

- The manuscript mischaracterizes Thirumalai et al. 2013 and does not refer to the advances contained therein appropriately. As one example, the authors write in their conclusions: "Overall, our results suggest that foraminiferal $\delta18O$ for a large part of the Pacific Ocean can be used to reconstruct ENSO, especially if an individual foraminiferal analysis (Lougheed et al., 2018; Wit et al., 2013) approach is used (Ford et al., 2015; Koutavas et al., 2006; Koutavas and Joanides, 2012; Koutavas and Lynch-Stieglitz, 2003; Sadekov et al., 2013; White et al., 2018), contrary to previous analysis (Thirumalai et al., 2013). " Firstly, considering that their analyses do not account for sampling uncertainty or ENSO skill in IFA-based reconstructions due to the lack of separation from the seasonal cycle as well as decadal and higher forms variability, their conclusion is not supported by their findings. Second, we demonstrate that, in fact, ENSO sensitivity is high (with minimal influences from seasonality) according to forward-modeled IFA (with uncertainty) in the subsurface Eastern equatorial Pacific and the surface-ocean in the central tropical Pacific Ocean (see Figs. 5-6 as well as Discussion). From the conclusions of Thirumalai et al. 2013: "Our results show that the IFA approach is insensitive to ENSO frequency changes (<20% probability) but nevertheless indicate that changes in ENSO amplitude or seasonal cycle amplitude (or a combination of both) can be detected depending on the ratio of interannual-to-annual variability at the location of the study."

References:

Carré, M., J. P. Sachs, A. J. Schauer, W. E. Rodríguez, and F. C. Ramos. 2013. Reconstructing El Niño-Southern Oscillation activity and ocean temperature seasonality from short-lived marine mollusk shells from Peru. Palaeogeography, Palaeoclimatology, Palaeoecology 371: 45-53.

Emile-Geay, J., and M. P. Tingley. 2015. Inferring climate variability from nonlinear

proxies: application to paleo-ENSO studies. Climate of the Past 11: 2763-2809.

Ford, H. L., A. C. Ravelo, and P. J. Polissar. 2015. Reduced El Niño–Southern Oscillation during the Last Glacial Maximum. Science 347: 255-258.

Hereid, K. A., T. M. Quinn, and Y. M. Okumura. 2013. Assessing spatial variability in El Niño-Southern Oscillation event detection skill using coral geochemistry. Paleoceanography 28: 14-23.

Khider, D., L. D. Stott, J. Emile-Geay, R. C. Thunell, and D. E. Hammond. 2011. Assessing El Niño Southern Oscillation variability during the past millennium. Paleoceanography 26: PA3222.

Tindall, J. C., A. M. Haywood, and K. Thirumalai. 2017. Modeling the stable water isotope expression of El Niño in the Pliocene: Implications for the interpretation of proxy data. Paleoceanography 19: 191-122.

Thirumalai, K., J. W. Partin, C. S. Jackson, and T. M. Quinn. 2013. Statistical constraints on El Niño Southern Oscillation reconstructions using individual foraminifera: A sensitivity analysis. Paleoceanography 28: 401-412.

White, S. M., A. C. Ravelo, and P. J. Polissar. 2018. Dampened El Niño in the Early and Mid-Holocene Due To Insolation-Forced Warming/Deepening of the Thermocline. Geophysical Research Letters 45: 316-326.

---

## Author Response (AR1)

**Track changes MS**

**Comments to the Author:**

Editor Decision: Reconsider after major revisions (03 Jun 2019) by Eric Wolff

*Your paper received a number of quite negative reviews. Nonetheless, I have decided to allow you to prepare a new version (major revisions) for consideration by reviewers.*

*I accept that there is value in a paper that forward models the proxy record to assess whether ENSO characteristics are even recorded in forams, as opposed to an inverse model that seeks to determine ENSO from the foram record. Nonetheless, you should recognise that several well-qualified reviewers felt that your study had not met the targets expressed in the title and abstract. Rather than (as your responses tended to) berate them, I would like to see you improve your presentation and explanations so that readers are not left disappointed.*

*This applies also to other aspects of the paper: as an example, like some reviewers I found the figures and the different shadings hard to understand; please improve either the figures or the captions, which is more constructive than implying that the reviewers have been deficient in not being able to follow them.*

*In your responses you have raised a number of issues that you state were not part of your paper - for example bioturbation, and alternative statitsical methods. While I do not expect you to change your model or methodology completely to meet these concerns, you clearly do need to add paragraphs to your paper to state what might be the effect if these issues were covered, and why their lack does not affect your main findings.*

*I look forward to receiving a substantially improved paper for further review.*

**Response to Editor**

Dear Prof. Eric Wolff,

Thank you for considering our manuscript for revisions. Please find attached our revised MS with track changes. As the 'track changes' document is heavily altered it may obscure the content, therefore we outline here the major changes in brief. We also include some rationale as to why we made the changes.

- We have changed the Title.
- We have altered the text of the abstract and conclusions to better reflect the fact that whilst 'foraminifera in the water' may 'sense' ENSO, the sedimentology of the Pacific reduces the useful area.
- We have edited the methodology and streamlined the explanation, as this is where the text may have led the reviewers to misinterpret our intentions.
- We have calculated the standard deviation/variance of each model run and each species, we have also calculated the interannual std. dev. vs. the total std. dev. We include a figure of *G. sacculifer* at 60 m and a supplementary table 1 in which we compare our results with three previous papers data and one model (iCESM)
- Differences with our model and data may reflect: Smaller standard deviation for larger n; the fact our dataset is in the "Anthropocene" whilst the data is from ~1000 years ago; there are reversals or evidence for bioturbation in the cores; subsample picking variation; and or the fact that our model may have limitations in terms of food, light, etc. (as stated in the discussion and methodology).

- We have run 4 picking experiments (picking 60 months, 10000 times at each grid cell: 170*40*60*10000). The four picking experiments highlight different ways of subsampling the model dataset.
- Unfortunately, it is not possible as the reviewers suggested to use QQ-plots as it would require 6800 plots; and interpolation of the skewed climate state data. We have included an explanation, as per your request, as to why we have not done this.
- We have moved the test of the input data from the supplement into the main text (figure 4) – this is not FAME instead we perform the Anderson-Darling on fixed depths. We have expanded and altered the original text to highlight that we had in fact tested the input signal (the reviewer had requested to see what the difference is between FAME and the input)
- Water depth map – as suggested by the reviewers, we have plotted seamount locations from the only scientifically verified (as opposed to text-mining Wikipedia entries for seamounts) available database of seamounts in the Pacific. An estimated 50,000 seamounts exist in the Pacific, but unfortunately this database contains only 291 values (and these include duplicate locations pertaining to be different parts of the same seamount system). Complementary to this, we have also added a new figure, to highlight that the GEBCO dataset is not a low resolution dataset, allowing us to show where the different water depth cutoffs for CCD alter or expand the available area suitable for sediment core retrieval.
- We have plotted the FAME overlay (figure 7) only with the water depth (previously it was with the SAR plus water depth). This does not alter our results. Instead we now contrast the water depth with different variations of SAR (figure 9) and highlight the potential altered signal through bioturbation of each scenario (further expanded upon in figure 10). This is intended to better explain our results.
- Sedimentation rate map - We have expanded the figures to include the inferred sediment accumulation rate, the oxygen saturation of the bottom water (unfortunately the 2018 pre-release of WOA as yet does not include this oxygen parameter in netcdf format although we will happily revise this if it becomes available) in response to a reviewers comment that oxygen will alter the bioturbation depth, and bioturbation model output. The main text was expanded to reflect those changes.
- We have run a bioturbation model, SEAMUS (model code and outline has been submitted by co-author Bryan Lougheed to Geoscientific Model Discussions and is online:  Lougheed, B. C.: SEAMUS (v1.0): a $\Delta14C$-enabled, single-specimen sediment accumulation simulator, Geosci. Model Dev. Discuss., https://doi.org/10.5194/gmd-2019-155, in discussion, 2019.), using the Uppsala Supercomputer and added corresponding new figures and text.  As outlined in our responses to the reviewers, a 60-year ocean reanalysis dataset is too short for a bioturbation model (due to thousands of years of model spin-up time being necessary to simulate a sediment archive). We have, therefore, used a Greenland ice core record as an input dataset at 10-year intervals for several thousand years, simply to understand the effects of bioturbation. The use of this dataset is in no way meant to infer some connection between ENSO and ice cores. It was used because it represented a temporally high-resolution climate input dataset with which to demonstrate the effects of bioturbation. The equivalent marine record, the benthic stack LR04, is already bioturbated, seeing as it is a stack of sediment cores with an average SAR of ~2.5 cm ka$^{-1}$
- We have, as suggested by the reviewer, included a discussion point of the potential amplitude changes, a point we had raised in the conclusions regarding future work.
- Figures - As outlined in our response to reviewer 3, we agree that some figures could have been clearer (As we had stated in the reply: "*We will correct this – the label on the figure is correct (left G. sacculifer; mid G. ruber; right N. dutertrei), we will correct the figure captions. The reviewer is correct that some are white and some grey when we originally made the figure everything was white, black and hashed. Unfortunately, these hashes draw the eye (and can hide some small locations) away from the white only locations we decided to make it grey to highlight this (The top panel of Figure 5 'G. sacculifer 60 m' is missing the grey which is our mistake but demonstrates the drawing of the eye). The N. dutertrei dataset does not have the hashing so we decided to make it white only.*"). We have corrected the error in the caption and one panel of figure 5. I hope the addition of an explanatory legend in our figures can solve this problem.
- As explained to the reviewer the rationale for having white vs. black OR grey and hashed vs. black is threefold: white background with a hashed section can draw the eye; the two schemes are meant to be distinct if we used the grey (means are greater than the error) for N. dutertrei one could infer that those results are comparable with G. ruber and G. sacculifer grey regions (though we do not know that); and the binary grayscale approach was used for its contrast.

- The color scheme white, grey, black, hashed can be changed with only a minor amount of work – however, if our explanatory legends are still not clear we would appreciate input from the reviewers in terms of what colour scheme to use. Likewise, the color schemes of 7 and 8 (i.e., yellow/blue-purple) can be altered as well, however we felt that this nicely picks out the bathymetric highs. In addition, Figure 9C; 9E and 9G color scheme can also be altered, however the red grey and yellow color scheme works well with dark and light contrast. This allows for 6 combinations to be plotted on one figure.

*On behalf of all-co-authors*

Brett Metcalfe          Bryan Lougheed          Claire Waelbroeck          Didier Roche

**Using a foraminiferal ecology model to test if tropical Pacific planktonic foraminifera are suitable recorders of ENSO**

[revised manuscript text omitted]

For the specific case of foraminifera populations in the water, it particularly arises from the species-specific ecological niche. The mapping of proxy value to climate value can therefore be skewed, a major factor governing the spatiotemporal distribution of a given planktonic foraminiferal species is the presence of an ideal water temperature. Proxies of past ENSO and Pacific SST (Ford et al., 2015; Koutavas et al., 2006; Koutavas and Joanides, 2012; Koutavas and Lynch-Stieglitz, 2003; Leduc et al., 2009; Sadekov et al., 2013; White et al., 2018) are based upon the biomineralisation of the calcite, or a polymorph such as verite (Jacob et al., 2017),  shells of foraminifera (Emiliani, 1955; Evans et al., 2018; Zeebe and Wolf-Gladrow, 2001). In general, there are three major types of foraminifera-based palaeoceanographic proxies:

**(1)** those associated with the faunal composition and their abundance within deep-sea sediments that utilises either a qualitative approach (Phleger et al., 1953; Schott, 1952); a weighted average (Berger and Gardner, 1975; Jones, 1964; Lynts and Judd, 1971); a selected species approach (e.g. coiling direction, or warm-water species presence; Ericson et al., 1964; Ericson and Wollin, 1968; Hutson, 1980b; Parker, 1958; Peeters et al., 2004; Ruddiman, 1971; Schott, 1966); a regression analysis (Hecht, 1973; Imbrie and Kipp, 1971; Williams and Johnson, 1975); or, a transfer function (CLIMAP Project Members, 1976; McIntyre et al., 1976; Williams, 1976; Williams and Johnson, 1975) that compares the down-core records with a dataset of 'modern' values and their associated water column parameters (Hutson, 1977, 1978).

**(2)** those associated with the stable oxygen isotope composition of a whole shell analysed either individually ( Ganssen et al., 2011; Koutavas et al., 2006; Koutavas and Joanides, 2012; Leduc et al., 2009) or pooled (Garidel-Thoron et al., 2007; Koutavas et al., 2002; Stott et al., 2002, 2004), herein $\delta^{18}O_c$ (c = calcite), which can  be used to reconstruct SST

and past oxygen isotope values in seawater $\delta^{18}O_{sw}$ (sw = seawater) when paired with a proxy that can either reconstruct temperature or salinity;

and (3) those associated with the trace metal geochemistry (e.g., Ford et al., 2015; Sadekov et al., 2013; Stott et al., 2002, 2004; White et al., 2018)(Stott et al., 2002, 2004), more specifically the natural logarithm of the relative concentration of Mg and Ca (ln(Mg/Ca), of the shell, based upon the temperature dependent (Elderfield and Ganssen, 2000; Nürnberg et al., 1996) incorporation and substitution of a Mg cation into the calcite lattice (Branson et al., 2013, 2016).

However, tThe interpretation of these proxiesproxies, however, is not straightforward, for example, calibration of foraminiferal assemblage based transfer functions with surface temperatures as opposed to a deeper temperature signal may skew the reconstructed temperature (Telford et al., 2013); $\delta^{18}O_c$ is further can be affected by species-specific size effects (Feldmeijer et al., 2015; Metcalfe et al., 2015; Pracht et al., 2018), disequilibria or vital effects, which clouds the accurate reconstruction of past SST and $\delta^{18}O_{sw}$. There is no simple bijective function between $\delta^{18}O_c$ and the oceanic variables $\delta^{18}O_{sw}$ and temperature used in its calculation, with variability in $\delta^{18}O_{sw}$ limiting the use of $\delta^{18}O_c$ as a pure temperature proxy. 
[revised manuscript text omitted]

The difference between the constant of Hut (1987) and the dynamic value (Brand et al., 2014) is minor.

**2.2 Foraminifera as modelled entities (FAME)**

For each latitude and longitude grid, a monthly growth rate was calculated using the ORA S4 temperature and a Michaelis-Menton kinetics to predict growth calculated from culture experiments, described in Lombard et al. (2009). Growth rate was artificially constrained to arbitrary values of the upper 60; 100; 200 and 400 m for all species to reflect the symbiotic nature of various foraminiferal species. Foraminifera as modelled entities has been developed as a tool for translating, a climatic input (typically a reanalysis dataset or climate model output) into a (simulated-) climatic signal, a signal that aims to approximate the depth integrated growth of foraminifera (e.g., Pracht et al., 2019; Wilke et al., 2006; Steindhardt et al., 2015). Data-model comparison studies suffer from an inability to directly compare like with like so that there are differences in (i) the units used *i.e.*, most proxies reconstructing temperature do not give values of temperature in degrees °C or K but in their own proxy units (e.g., per mil ‰; mmol/mol; species abundance or ratio) necessitating a conversion; and (ii) there is a

reduction in scales, *i.e.*, models give a wealth of information (multiple depth layers and high resolution time slices) in the time-depth domain. A number of models and modelling studies exist to determine the foraminiferal responses to present (Fraile et al., 2008, 2009; Kageyama et al., 2013; Kretschmer et al., 2017; Lombard et al., 2009, 2011; Roy et al., 2015; Waterson et al., 2016; Žarić et al., 2005, 2006), past (Fraile et al., 2009; Kretschmer et al., 2016) and future (Roy et al., 2015) climate

5    scenarios, FAME uses the associated temperature and $\delta^{18}O_{eq}$ at each grid cell to compute a time averaged $\delta^{18}O_c$ and $T_c$ for a given species. FAME was produced as an attempt to reduce the error associated with data-model comparisons by (i) generating simulated-proxy time-series from model runs that can be compared with age-depth values down core; and (ii) to reduce the model information for a given time-slice into a manageable and relevant value using an integration that would make sense on a biological point of view (Roche et al., 2018).

10    The FAME model utilises the temperature-growth rate equations of Lombard *et al.* (2009) to simulate temperature-derived growth rate (Kageyama et al., 2013; Lombard et al., 2009, 2011), this growth rate is then used as a weighing to produce a growth rate-weighted proxy value (Roche et al., 2018). The original Lombard et al. (2009, 2011) equations are based upon a synthesis of culture studies, pooled together irrespective of experimental design or rationale, therefore they can be considered to conceptually represent the fundamental niche of a given foraminiferal species, *i.e.* the range in environment

15    that the species can survive. The basic structure of FAME is based upon temperature based Michaelis-Menton kinetics to predict growth rate, described in Lombard et al. (2009), without using the parameters (e.g., light, respiration, food) associated with FORAMCLIM (Lombard et al., 2011). The absence of known values or proxy values for the full set of parameters associated with FORAMCLIM has led us to an Occam's Razor favoured approach in model parameterisation for FAME (Roche et al., 2018). Although other processes may also impact species such as mixed layer depth and nutrients these

20    variables for now can be set aside, as temperature provides the dominant signal, it is worth noting that in all probability some variance will arise from these processes and deviation between observed and expected values should consider this.

Using the MARGO core top $\delta^{18}O_c$ database (MARGO Project Members*, 2009), Roche *et al.* (2018) validated and computed the optimum depth habitat (the depth habitat that exhibits the strongest correlation when comparing FAME $\delta^{18}O_c$ and MARGO $\delta^{18}O_c$) for each species in the MARGO database (MARGO Project Members*, 2009). Whilst, both models,

25    FAME and FORAMCLIM, can compute the growth rate of eight foraminiferal species from culture studies (Kageyama et al., 2013; Lombard et al., 2009, 2011; Roche et al., 2017), the limited number of species available for a global core top comparison led to a reduction in the number of species modelled (Roche et al., 2018). Here the output of FAME is further restricted to three species that have been the main focus of foraminifera-based studies that have been used to infer ENSO variability, namely the upper ocean dwelling *Globigerinoides sacculifer* and *Globigerinoides ruber*, as well as the

30    thermocline dwelling *Neogloboquadrina dutertrei* (Ford et al., 2015; Koutavas et al., 2006; Koutavas and Joanides, 2012; Koutavas and Lynch-Stieglitz, 2003; Leduc et al., 2009; Sadekov et al., 2013). The MARGO database does not include *N. dutertrei,* meaning that we concentrate our efforts mainly on *G. ruber* and *G. sacculifer.*

In this study, ORA S4 temperature was used as the input variable, with the growth rate computations artificially constrained to arbitrary values of the upper 60; 100 and 200 m to reflect the presence of photosymbiotic algae in the various

foraminiferal species. By identifying the optimum depth habitat, Roche *et al.* (2018) established the realised niche, *i.e.* the range in environment that the species can be found, for these species for the late Holocene. Unlike some foraminiferal models, FAME does not include limiting factors such as competition, respiration or predation variables as no reliable proxy exists for such parameterisation in the geological record, therefore aspects such as interspecific competition that may limit the niche width of a species are not computed. As these depth constraints (60 m; 100 m; and 200 m) may induce some variability we opted to include an extreme value of 400 m that grossly exaggerates the potential depth window. It is important to note however that as the computation of FAME is based on growth occurring within a temperature window it does not necessarily mean that for a given grid point modelled foraminifera will grow down to 400 m (or whichever cut-off value is used), only that the model in theory can do so (depending if optimal temperature conditions are met) to capture the total theoretical niche width. As the optimised depths computed from the MARGO dataset of Roche *et al.* (2018) are shallower, and upper ocean water is more prone to temperature variability, our approach likely dampens both the modelled $\delta^{18}O_c$ and $T_c$. The FAME is based upon FORAMCLIM a Michaelis-Menton kinetics to predict growth calculated from culture experiments, described in Lombard et al. (2009).which can compute eight foraminiferal species (Kageyama et al., 2013; Lombard et al., 2009, 2011; Roche et al., 2017), however comparison with a core top database has been limited to five foraminiferal species (Roche et al., 2017). Here the output of FAME is further restricted to three species that have been the main focus of foraminifera-based studies that have been used to infer ENSO variability, namely the upper ocean dwelling *Globigerinoides sacculifer* and *Globigerinoides ruber*, as well as the thermocline dwelling *Neogloboquadrina dutertrei* (Ford et al., 2015; Koutavas et al., 2006; Koutavas and Joanides, 2012; Koutavas and Lynch-Stieglitz, 2003; Leduc et al., 2009; Sadekov et al., 2013). Growth rate was artificially constrained to arbitrary values of the upper 60; 100; 200 and 400 m for all species to reflect the symbiotic nature of various foraminiferal species.

2.3 Monthly growth rate (GR) weighted $\delta^{18}O$ and $T_C$

The FAME growthmodelled growth rate output was used to compute the monthly depth-weighted oxygen isotope distribution ($\sum_{z=0}^{zb(k)} \delta^{18}O_c$, where $z$ is the depth) for each species, using the aforementioned computed $\delta^{18}O_{eq}$ for a given latitudinal and longitudinal grid point (Figures 2 and 3). This was repeated four times, during which the lower depth limit, where $zb(k)$ is either of the growth rate computation was set to 60; 100; 200 andor 400 m. No correction for species specific disequilibria, such as vital effect, was applied to the data.

At each grid point the total growth rate ($\sum_{t=1}^{nt} \sum_{z=0}^{zb(k)} GR$, where $nt$ is the number of time steps) was calculated and each individual monthly growth rate ($\sum_{z=0}^{zb(k)} GR$) normalised to this value. A weighted histogram was constructed from the corresponding oxygen isotope value. (Figure S1) which will sum to 1 (or 100 %). The rationale for constructing a rflux/GR weighted histogram is related to the probability of an oxygen isotope value occurring, essentially for an unweighted distribution this is $p(\delta^{18}O_c) = \frac{x}{12}$, with each month contributing $x$ and therefore, having a maximum value for $C_i$ of $z$ (al., 1996). However, in reality the monthly contribution, if remineralisation of the settling flux and benthic seafloor processes (Lougheed et al., 2018) are for now largely ignored, is modulated by various hydrographic processes and related to the

temperature sensitivity (Jonkers and Kučera, 2017; Mix, 1987). Although other processes may also impact species, such as mixed-layer-depth and nutrients these for now can be ignored. Therefore, both the resultant $\delta^{18}O_c$ and flux are sufficiently perturbed/altered by T°C that for a weighted distribution each monthly contribution is $\frac{\sum_{i=1}^{n} f_i \cdot m_i}{\sum_{i=1}^{n} f_i}$. The same procedure was executed for the $T_c$, where the computed $\delta^{18}O_{eq}$ is replaced with the ORA-S4 temperature profile for a given latitudinal and longitudinal grid point.

**2.4 Statistical analysis**

The tropical Pacific Ocean is divided into four Niño regions based on historical ship tracks, from east to west: Niño 1 and 2 (0° to -10°S, 90°W to 80°W), Niño 3 (5°N to -5°S, 150°W to 90°W), Niño 3.4 (5°N to -5°S, 170°W to 120°W) and Niño 4 (5°N to -5°S, 160°E to 150°W). Pan-Pacific meteorological agencies differ in their definition (An and Bong, 2016, 2018) of an El Niño, with each country's definition reflecting socio-economic factors. Therefore, for simplicity we use One index for ENSO, the Oceanic Niño Index (ONI), based upon the Niño 3.4 region (because of the region's importance for interactions between ocean and atmosphere) which is a 3-month running mean of SST anomalies in ERSST.v5 (Huang et al., 2017). However, Pan-Pacific meteorological agencies differ in their definition (An and Bong, 2016, 2018) of an El Niño, with each country's definition reflecting socio-economic factors, therefore, for simplicity we We utilise a threshold of $\chi \geq +0.5$°C (where $\chi$ is the value of ONI) as a proxy for El Niño, $-0.5$°C $\leq \chi \geq +0.5$°C for neutral climate conditions and $-0.5$°C $\leq \chi$ for a La Niña in the Oceanic Niño Index. Many meteorological agencies consider that five consecutive months of $\chi \geq +0.5$°C must occur for the classification of an El Niño event. However, here it is the only difference is that we considered that any single month falling within our threshold values as representative of El Niño, neutral or La Niña conditions (grey bars in Figure 1). This simplification reflects the lifecycle of planktonic foraminifera (~4 weeks) seeing that the population at time step $t$ does not record what happened at $t-1$ or what will happen at $t+1$. As we are producing the mean population growth weighted $\delta^{18}O_c$ values, 'almost' El Niño or 'almost' La Niña would be indistinguishable from the build-up and subsequent climb-down of actual El Niño and La Niña events. Therefore, these 'almost' El Niño or 'almost' La Niña are placed within their respective climatological pools as El Niño or La Niña.

Each time-step for the entirety of the Pacific was classified as one of three climate states (El Niño; Neutral; and La Niña), where after By using this threshold, three weighted histograms for the resultant each $\delta^{18}O_c$ and $T_c$ at each timestep produced by FAME and their resultant distributions (El Niño; Neutral; and La Niña) were computed for every month and for every latitude and longitude grid-point for the 1958-2015 period for each grid-point were binned into their respective categories. Monthly output representing different climate conditions were compared against each another for $\delta^{18}O_c$ and $T_c$ separately. An Epanechnikov-kernel distribution was first fitted to the binned monthly output of a single climate state, the bandwidth varies between grid-points to provide for an optimal kernel distribution. The use of an Epanechnikov-kernel distribution to fit the data, as opposed to other types of distribution, represents a trade-off between keeping as many parameters constant whilst mimicking the underlying dataset for a large number of grid points. The conversion of the data from dataset to distribution

may induce some small error induced by: rounding to whole integers; the use of a $\delta^{18}O_{\text{mid-point}}$ which gives an error associated with the bin size ($\pm 0.05$ ‰) that is symmetrical close to the distributions measures of central tendency but asymmetrical at the sides; and finally, the associated rounding error at the bin edges within a histogram ($\pm 0.005$ ‰). Subsequently any two desired distributions  can be compared for (dis)similarity using an Anderson-Darling test (1954). Here, all values, i.e., the population, associated with a climatological state are compared with the other populations representing the different climatological state, the results plotted here are Neutral climate state vs. El Niño climate state. , e.g., the grey/white area in Figure's 3 to 6.

~~Using the repeat matrix function (MatLab function: *repmat*), a matrix of $\delta^{18}O_c$ was produced using each bin's mid-point ($\delta^{18}O_{\text{mid-point}}$) there is a threefold error combined with this methodology which may account for minor variation between discrete runs of the model: first the counts values were rounded to whole integers so an exact number of cells could be added to a matrix; secondly the $\delta^{18}O_{\text{mid-point}}$ was used which gives an error associated with the bin size ($\pm 0.05$ ‰) that is symmetrical close to the distributions measures of central tendency but asymmetrical at the sides; and finally, the associated rounding error at the bin edges within a histogram ($\pm 0.005$ ‰).~~ 2.5 Test of input data (Temperature and calculated $\delta^{18}O_{eq}$)

Foraminifera as modelled entities produces a modulated response that seeks to replicate how foraminifera modify the climate signal, several studies have approximated the foraminiferal signal in a different way (e.g., Thirmulai et al., 2013; Zhu et al., 2017a). In order to understand how FAME has altered the signal, and the degree to which the conclusions drawn depend upon the modelled growth rates, the input datasets of the sea water properties (Temperature and calculated $\delta^{18}O_{eq}$), underwent a similar statistical test (Figure 4). Unlike FAME, which integrates over several depth levels using the computed growth rate, the test of the input datasets was with fixed depths without any growth rate weighting. These fixed depths are 5, 149 and 235 m, giving a Eulerian view (Zhu et al., 2017a) in which to observe the implications of FAME's dynamic depth habitat. As per the FAME output, each timestep value was placed into its climate state and an Anderson-Darling test performed to compare the (dis)similarity of on the resultant distributions.

**2.6 Alternative statistical tests**

In order to compare our results with previously published studies using planktonic foraminifera we employed a series of simple statistical tests, mimicking those applied to sediment archives by the palaeoclimate community. A chief parameter that has been employed in previous ENSO proxy work using  foraminiferal analysis (more specifically, individual foraminiferal analysis; IFA) is the measure of individual foraminifera downcore standard deviation ($\sigma(\delta^{18}O_c)$) . Increased

$\sigma(\delta^{18}O_c)$ is considered to correlate to increased variation in SST and, in turn, increased ENSO incidence and/or magnitude (Leduc et al., 2009; Zhu et al., 2017a) or even increased inter-annual variance (Thirmulai et al., 2013)in temperature. The variance ($\sigma^2(\delta^{18}O_c)$) of the timeseries were computed both as the total variance and as the interannual variance, the latter is computed as outlined in Zhu et al. (2017a). For the interannual variance, the mean monthly climatology is subtracted from

5   the dataset, producing monthly anomalies and a linear trend removed (using the detrend function of MatLab 2019a) – the resultant data was left unfiltered (*i.e.*, Zhu et al., 2017a used a 1-2-1 filter). Four 'picking' experiments were performed, as FAME computes the average value for a given time step and given the single foraminiferal isotope variance for an equivalent time step (e.g., weeks; Steinhardt et al., 2015) it is more than likely that this computation reduces the real spread in values. Therefore, rather than use the terminology specimen we prefer to use months. Given the complexity in

10  reconstructions of trace metal geochemistry (Elderfield and Ganssen, 2000; Nürnberg et al., 1996): the potential error associated with determining which carbonate phase is first used when foraminifera biomineralise (Jacob et al., 2017); growth-band integration; secondary factors (e.g., salinity, carbonate ion) the focus of the picking here has been on the $\delta^{18}O_c$. Irrespective of which experiment, 60 months were drawn, with replacement, and the number of Monte Carlo iterations is set at 10,000. We assume that the 'picker' is taxonomically well-trained and/or has a procedure in which species can be checked

15  taxonomically post-analysis (e.g. photographing all specimens prior to analysis, Pracht et al. (2019)) and therefore do not include an error that deals with incorrect identification. Although we note that parameterisation of misidentification would be difficult, as it requires understanding of the variability in both standard deviation and absolute values for species co-occurring downcore (Feldmeijer et al., 2015; Metcalfe et al., 2015; 2019). For each run of experiment's (i) to (iii) the drawn months were saved in order to perform (iv):

    (i) In Picking Experiment-I (Figure 3D), the months drawn for each iteration of the Monte Carlo were selected and each grid-point was sampled (i.e., there are 10000·60 selected months). This assumes that the same months are selected at grid point A as point B.

25    (ii) In Picking Experiment-II (Figure 3E), at each grid-point a Monte Carlo was run (*i.e.*, there are 170·40·10000·60 selected months). This assumes that different months could be selected between grid point A and point B.

    (iii) In Picking Experiment-III (Figure 3F), at each grid-point a Monte Carlo was run using the growth rate

30    weighting for each month (*i.e.*, there are 170·40·10000·60 selected months), this assumes that in periods of higher growth there will be a higher flux of the species and therefore a greater chance of selecting that month. The rationale being that not only are different months selected between grid point A and point B, but if A and B differ climatologically there may be an over subscription of ecologically beneficial habitats in one core location compared to the other.

(iv) In Picking Experiment-IV (Figures 3G to 3I), the experiment of (ii) was re-run but with the addition of two sources of error: The first error is based upon FAME producing the average value for a given time slice, therefore short-term variability in temperature and/or the spread in the population (*i.e.*, variance in depth of an individual; variance in chamber growth per individual), as evidenced by single foraminiferal analysis of sediment trap samples (*e.g.*, Steinhardt et al., 2015), is potentially lost. Therefore, for each picked month between -0.40 and 0.40 ‰ is added to the picked month value (in intervals of 0.02 ‰), this is approximately ±2° C (*i.e.*, ~4° C). The second error is the analytical error that an individual measurement will have. Machine measurement error is assumed to lie between -0.12 and 0.12 ‰ (in intervals of 0.005 ‰ – the 3rd decimal place is an exaggeration of machine capabilities although it will have repercussions for rounding) the 1σ of within run (as opposed to long-term average) of international stable isotope standards. The intervals of both errors (0.02 ‰ and 0.005 ‰) were chosen to give a similar number (n = 41 and 49) of potential randomly selected error for each picked month. Each picked month has their own randomly selected error for both of these errors, i.e., each value is the sum of the month picked and their own error. The values for within month variability (Figures 3G ) and machine error (Figure 3H) are calculated separately and then combined (Figure 3I), as they may have a corresponding or conflicting signs, either 'cancelling' out each other or amplifying the difference.

An associated statistical methodology is the graphical summary (as opposed to a numerical summary via a test value) of plotting the quantiles of two probability or the quantiles of sample probability distribution against a theoretical distribution distributions also referred to as a Quantile-Quantile, or Q-Q plot (*e.g.*, Ford et al., 2015; White et al., 2018). A complimentary (*i.e.*, used in association with, not as replacement, Filliben 1975) test metric, the Probability plot correlation coefficient (Filliben, 1975) can be used as a numerical summation of this approach, which bases its rationale on near linearity between the two tested distributions. This graphical technique is not used here for the following reasons, (i) the climatic categories (*i.e.*, El Nino, Neutral, La Nina) imposed upon the data give uneven sized sample distributions requiring an interpolated quantile estimate; and (ii) the large graphical computation required (170·40).

**2.7 An approximation of sedimentary archives: Water depth & Sedimentation Rate**

The resolution of the ocean reanalysis data for the time period 1958-2015 would essentially be analogous with a sediment core representing 50 yr⁻¹ cm⁻¹ (or 20 cm⁻¹ kyr⁻¹). Based on our analysis, such a hypothetical core with a rapid sediment accumulation rate (SAR) could allow for the possible disentanglement of El Niño related signals from the climatic signal using IFA, but only in a best-case scenario involving minimal bioturbation, which is unlikely in the case of oxygenated waters. Indeed, one should view discreteDiscrete sediment intervals retrieved from systematically bioturbated deep-sea sediment cores contain, and the foraminifera with ages spanning many centuriescontained within them, as representative of an integrated multi-decadal or even multi-centennial signal (Lougheed et al., 2018; Peng et al., 1979). This is in contrast to,

as opposed to other proxies such as corals (Cole and Tudhope, 2017), speleotherms (Chen et al., 2016) and molluscs (Butler et al., 2013; Milano et al., 2017), or varves in which distinct layers correspond to discrete time intervals where distinct time-specific banding is present (true 'time-series' proxies). The ambient signal following translation into a foraminiferal signal within the water is therefore further modulated by several post mortem processes, which include: the latitudinal-longitudinal shift in position of sinking foraminifera - the so-called 'funnel affect' (van Sebille et al., 2015; Deuser et al., 1981); dissolution of calcium carbonate either in the water (Schiebel et al., 2007), at the seafloor, or due to pore fluids; and bioturbation. As mentioned, mixing by bioturbation, results in an apparent smoothing of the downcore, discrete-depth multi-specimen signal (Hutson, 1980a; Löwemark, 2007; Löwemark et al., 2005, 2008; Löwemark and Grootes, 2004; Cole and Tudhope, 2017; Mix, 1987), thus leading to the possibility of interpreting single outlying foraminifera values within a specific depth as representing an 'extreme' climate, when they may in fact represent climate from a different time or epoch. This is especially apparent in $\delta^{18}O_c$ where there is a difference temporally of $\delta^{18}O_{sw}$ (e.g., the ice volume effect in glacial and interglacial cycles ~1.25 ‰) meaning that the same temperature can have radically different $\delta^{18}O_c$ values, a consequence of this is that a series of high magnitude, but low frequency El Niño events could be disturbed in a discrete-depth record. Therefore, in order to reliably extract short-term environmental information from foraminiferal-based proxies, the signal that one is testing or aiming to recover must have a large enough magnitude, be largely unaffected by dissolution (*i.e.*, above the lysocline) so as not to adversely affect the population and the sedimentation rate must be high enough to give sufficient temporal coverage and rule out upwards bioturbation of single foraminifera from significantly different climate periods.

In our first step in consideration of post-mortem signal alteration we focus on dissolution. The lysocline, the depth at which dissolution first becomes apparent (Berger, 1968; 1970), and the Calcite (or Calcium Carbonate) Compensation Depth (CCD; Bramlette, 1961) vary between the different ocean basins; the Atlantic Ocean in which deep water forms has a relatively deep CCD as a by-product of 'young' well ventilated bottom waters whereas the Pacific Ocean the final section of the thermohaline circulation conveyor belt, has a shallower CCD. In order to highlight the potential for dissolution, the bathymetry of the Pacific was extracted from the General Bathymetric Chart of the Oceans GEBCO 2014 30 arc-second grid (version 20150318, www.gebco.net) between -20°S to 20°N and 120°E to -70°W (Figure 8). Depths of 3500 m below sea-level (bsl), 4000 m bsl and 4500 m bsl are used here as cut-off values, these depths represent multiple possible depths under which there is the potential for noticeable dissolution (i.e., lysocline) or be dissolved (i.e., CCD). Whilst our intention here is a generalised view to be used as an approximate guide, it is important to note that the Pacific Ocean has the largest proportion globally of >1 km tall seamounts that are smaller than <100 km (Wessel, 1997). Which may have important, relatively shallow-water sedimentary sequences, which may also be of sufficient sediment accumulation rate, therefore we supplement the GEBCO bathymetric data with the locations of seamounts. However, whilst exhibit a large enough amplitude in order to perturb the population by a significant degree from the background signal, there are an estimated 50,000 seamounts in the Pacific that are taller than a km (Menard, 1964; Wessel and Lyons, 1997), only 12,000 have been documented on charts (Batiza, 1982), and approximately 291 have been dated (Koppers et al., 2003; Clouard and Bonneville,

2005; Hillier, 2007). It is these 291, <1% of the estimated seamounts, we have overlain onto the bathymetric data (Figure 8b), although this number is further reduced as we only plot between 20°S and 20°N.

The second step when considering post mortem signal alteration is the sediment accumulation rate (SAR). We first plot the time-averaged deep-sea SAR (Olson et al., 2016), adapted by Lougheed *et al.* (2018) for the Tropical Pacific (Figure 9). New

5 geochronological tools, such as dual $^{14}$C-$\delta^{18}$O measurements on single foraminifera (Lougheed et al., 2018), show that low sedimentation rate cores can have large variances in age between individual foraminifera present within a discrete 1 cm depth interval (Berger and Heath, 1968; Lougheed et al., 2018). In order to model bioturbation, a number of papers have used a diffusion style approach that reduces the parameters down to sediment mixing intensity and sediment mixing depth (herein referred to as bioturbation depth, BD), although this may be an artificial division purely driven by mathematical need

10 rather than biological constraints (Boudreau, 1998). The bioturbation depth has been shown to have a global average of 9.8 cm (1σ: ± 4.5 cm) that is independent of both water depth and sedimentation rate (Boudreau, 1998), though likely controlled as a result of the energy efficiency of foraging, *e.g.* deeper burrows may cost more energy to produce than can be offset in extracted food resources, and potential decay in labile food resources with sediment depth. It is not possible to carry out a transient bioturbation model upon the temperature and salinity ocean reanalysis data that we used for FAME, as it only

15 covers half a century of data, whereas thousands of years of input data are required to force a transient bioturbation model. To investigate how much temporal signal is integrated into discrete-depth intervals for typical tropical Pacific SAR, we, therefore, utilised the single foraminifera sediment accumulation simulator (SEAMUS, Lougheed, 2019) to bioturbated, as the input climate signal (Figure's 9 to 11), the 0-40,000 year $\delta^{18}O_{sw}$ of NGRIP (North Greenland Ice Core Project Members, 2004; Rasmussen et al., 2014; Seierstad et al., 2014). The ice core time series is an ideal input for a bioturbation simulator, as

20 it represents a highly temporally resolved climate input signal. SEAMUS simulates foraminifera in 10-year timesteps. The use of the NGRIP timeseries here is purely as an input parameter to investigate the effect of bioturbation upon a given climate signal -, it is important to stress that by using NGRIP as an input signal for SEAMUS we are neither implying that tropical Pacific cores should have signal similar to NGRIP or inferring some kind of causal relationship. As we seek to investigate the effect of bioturbation, no attempt has been made to modulate the input signal's absolute values to mimic

25 expected $\delta^{18}O_c$ values and why each plot of the synthetic down core time series retains the use of V-SMOW, despite carbonates being required to be V-PDB (Coplen 1995). Keeping all things constant, and varying a single parameter between experiments with SEAMUS, the sediment accumulation rate (SAR) was varied to fixed values of either 1, 2, 5 or 10 cm kyr$^{-1}$ (representative of typical Pacific SAR) and a bioturbation depth (BD) of either 5, 10 or 15cm based upon the global estimate and it's error bounds (Boudreau, 1998). For each experiment, the selected values of SAR and BD were kept constant for the

30 entire SEAMIS model run (i.e., the intensity and magnitude of bioturbation was not varied). In reality, SAR and BD may vary temporally depending on local conditions (*e.g.*, food, oxygen). Finally, the FAME results for the three species are overlaid with a water depth mask of above and below 3500 m below sea-level (mbsl), to also show seafloor areas under the CCD depth, where carbonate material is not preserved (Berger, 1967, 1970b). A comparison between water depth and time-averaged deep-sea SAR (Olson et al., 2016), adapted by Lougheed *et al.* (2018) is shown in Figure's 7 and 9.

otherwise it will be lost due to the smoothing effect of bioturbation (Hutson, 1980a; Löwemark, 2007; Löwemark et al., 2005, 2008; Löwemark and Grootes, 2004) upon the downcore, discrete-depth signal (Cole and Tudhope, 2017; Mix, 1987). Similarly, the individual characters of El Niño events, which are very short in duration, become lost in the bioturbated sediment record. New geochronological tools, such as dual $^{14}$C-$\delta^{18}$O measurements on single foraminifera (Lougheed et al., 2018), show that low sedimentation rate cores can have large variances in age between individual foraminifera present within a discrete 1 cm depth interval (Berger and Heath, 1968; Lougheed et al., 2018). In the case of a high SAR core (20 cm$^{-1}$kyr$^{-1}$), assuming a sediment mixing layer of 10 cm, benthic seafloor processes would result in a minimum 700-year 1σ confidence interval for a 1 cm discrete depth age (Lougheed et al., 2018). A consequence of bioturbation in sediment cores is that a series of high magnitude, but low frequency El Niño events could be smoothed out of the downcore, discrete-depth record. To investigate which areas of the sea floor can potentially preserve a foraminiferal downcore signal, we overlay our results with a map of time-averaged deep-sea SAR (Olson et al., 2016), adapted by Lougheed *et al.* (2018) to also show seafloor areas under the CCD depth, where carbonate material is not preserved (Berger, 1967, 1970b).

**3.0 Results**

The results of the forward model (Figure 2 and 3) are compared with their input (Figure 4) in order to identify regions in which the values are statistically distinct (Figure's 5-7). These results are then shown against the water depth (Figure's 7 to 9) and the SAR (Figure's 9-11) for the region. The results utilise Foraminifera as Modelled Entities (FAME; Figures 2, 3, 5, 6, and 7); the original Ocean Reanalysis data with computed $\delta^{18}$O$_{eq}$ (Figure 4); the General Bathymetric Chart of the Oceans (GEBCO; Figures 7 to 9); and the single foraminifera sediment accumulation simulator (SEAMUS; Figures 9 to 11).

**3.1 FAME Output: Variance**

We compute growth-weighted $\delta^{18}$O$_c$ (Figure 35 and 75) and temperature (Figure 6 and 7) distributions for each 1° by 1° grid cell in the fifty-eight year simulation using FAME (Roche et al., 2018), constraining the calculation to the Tropical Pacific Ocean (between -20°S to 20°N and 120°E to -70°W). Our model produces 696 individual monthly maps for all three species (Figure 2). While two of the three species (*G. ruber* and *G. sacculifer*) have similar ecologies, they show differences in their resultant $\delta^{18}$O$_c$ for the same ocean conditions (Figure 2). A comparison of our computed variance with measured data (Supplementary Table 1) is made, we compare both the value of the nearest grid-cell and because of the size of the grid and drift of foraminifera (van Sebille et al., 2015) an average of a 3 by 3 grid in which the nearest grid-cell to the core location is in the center. A comparison is made with both the iCESM model output and the core's that match this output (Zhu et al., 2017a). For the Late Holocene sample (~1.5 ka) MD02-2529 (08°12.33'N 84°07.32'W; 1619 m) in which *N. dutertrei* individual foraminifera were analysed from >250 μm (Leduc et al., 2009) giving a calculated standard deviation of measured foraminifera of 0.38 ‰. Whereas the full ~60 year time series (n = 696) of FAME presented here, gives a standard deviation

for all species, at depth cut off 60 m between 0.26 and 0.32 ‰; at depth cut off 100 m between 0.20 and 0.29 ‰; at depth cut off 200 m between 0.20 and 0.25 ‰; and at depth cut off 400 m between 0.20 and 0.24 ‰ (see Supplementary Table 1). Although these values vary if the average of the surrounding grid cells is used (see Supplementary Table 1). In comparison the iCESM results have the following standard deviation values, for a Eulerian (fixed) depth of 50 m: 0.4 ‰; Eulerian 100 m: 0.6 ‰; and Lagrangian value of 0.49 ‰. There are three samples (Koutavas and Joanides, 2012; Sadekov et al., 2013) located south of core site MD02-2529, these are the Late Holocene (~1.6 ka) samples of V21-30 (01°13'S 89°41'W; 617 m) and (~1.1 ka) V21-29 (01°03'S 89°21'W; 712 m) in which *G. ruber* was measured individually (Sadekov et al., 2013). For these two sites the measured standard deviation is 0.507 ‰ and 0.510 ‰ for V21-30 and V21-29 respectively (Koutavas and Joanides, 2012). The third core site at a similar location is (~1.6ka) CD38-17P (01°36'04 S 90°25'32W; 2580 m) was not analysed individually, instead replicates of pooled samples of 2 or 3 shells of *N. dutertrei* (Sadekov et al., 2013) were made these measured values give a standard deviation of 0.28 ‰. The full ~60 year time series (n = 696) of FAME presented here, gives a standard deviation for all species, at depth cut off 60 m between 0.33 and 0.41 ‰; at depth cut off 100 m between 0.27 and 0.40 ‰; at depth cut off 200 m between 0.25 and 0.35 ‰; and at depth cut off 400 m between 0.25 and 0.34 ‰ (see Supplementary Table 1). Although these values vary if the average of the surrounding grid cells is used (see Supplementary Table 1). In comparison the iCESM results have the following standard deviation values, for a Eulerian (fixed) depth of 50 m: 0.53 ‰; Eulerian 100 m: 0.75 ‰; and Lagrangian value of 0.35 ‰.

The study of ENSO has focused on whether the variability is entirely in response to ENSO or whether it is dominated by interannual variability (Xie, 1994, 1995; Wang et al 1994, 2010), here the interannual (Figure 3C), and total variance (Figure 3A) was computed and a ratio between the two calculated (Figure 3B; see Supplementary Table 1). Like the same analysis of interannual and total variance computed for iCESM and SODA reanalysis (Carton et al., 2000), outlined in Zhu et al. (2017a), there is also high ratio of interannual to total variance in our computed FAME dataset (Figure 3B). Although there are regions in the Eastern Equatorial Pacific wherein this ratio reduces. Despite this reduction, the ratio between total and interannual variance is still above > 0.5.

The Monte-Carlo experiments (Figure 3D-I) highlight the variation in picking a subset of the months, here 60, from the full timeseries. The FAME-$\delta^{18}O_{eq}$ *G. sacculifer* with a depth cut-off of 60 m is plotted here, the values for each grid point is the range in standard deviation (*i.e.*, the maximum standard deviation minus the minimum standard deviation) between iterations of the Monte-Carlo (n= 10,000). The range in standard deviations between iterations is plotted instead of the mean of the standard deviations. With increasing $n$ the mean converges toward the sample mean, however as the point of the Monte-Carlo is to generate plausible 'samples' it is more important to take into account the range in possible values which would help to establish the potential variability of subsampling. For the most part, regions with high total variance (Figure 3A) also have a larger range in standard deviations between the iterations 'picked'. It is interesting to note that by changing from the same months picked for each grid-point (Monte-Carlo I: Figure 3D) to varying the months picked between grid-points (Monte-Carlo II: Figure 3E or Monte-Carlo III: 3F) the range goes from 'smooth' to a more noisy dataset. Whilst the values plotted here are not the absolute values (as they are the range in standard deviation for a given grid point for the entire

10,000 iterations), it can be seen that some of the inter-core comparisons could in essence relate to differences in picking, *i.e.* different 'months' picked between grid-points may exacerbate or accentuate differences. Likewise, adding random variability, between -0.4 and 0.4 ‰ (Figure 3G and 3I), may also reduce the differences between areas of high Total variance and low Total variance. Though the values associated with machine error (-0.12 to 0.12 ‰) appear to do little to affect the range (Figure 3H and 3I). Whilst again the values plotted are not the absolute values, variability added in an attempt to mimic biological variation of a given time slice, increases the range of possible standard deviations in regions with low Total variance (Figure 3G and 3I), therefore understanding the biological variability on shorter timescale which here may be over exaggerated may be crucial (e.g., Steinhardt et al., 2015; Mikis et al., 2019).

**3.2 FAME Output: Anderson-Darling test**

Using a basin-wide statistical test, we examine whether the mean $\delta^{18}O_c$ values of a given El Niño foraminifera population ($FP_{EN}$) and a given non-El Niño ('Neutral conditions') foraminifera population ($FP_{NEU}$) can be expected to be significantly different at any given specific location. Where $FP_{EN}$ and $FP_{NEU}$ exhibit significantly different distributions mean $\delta^{18}O_c$ values, ENSO events can potentially be detected by paleoceanographers and unmixed using, for example, a simple mixing algorithm with individual foraminiferal analysis (Wit et al., 2013). In cases where $FP_{EN}$ and $FP_{NEU}$ do not exhibit significantly different means values, then the chosen species and/or location represent a poor choice to study ENSO dynamics. Each simulation time step was placed into a climate states: identification of timesteps that represent El Niño (EN), Neutral (NEU), and La Niña climate conditions was done using the Oceanic Nino Index (ONI) derivative (Huang et al., 2017) (Figure 1). Comparison, for each species, FAME's predicted growth-weighted $\delta^{18}O_c$ and $T_c$ distributions associated with each climate event was done using an Anderson-Darling (AD) test. This statistical test can be used to determine whether or not two distributions can be said to come from the same population. The results of this test are presented in the following way, in which there are four criteria: areas where  the population distributions of the two climate states are found to be statistically similar have black grid cells in all panels referring to the Anderson-Darling test results (Figure's 4-7); the results in which the areas where the populations distributions of two climate states are found to be statistically distinct are shown with two distinct colour schemes depending on whether a computable error can be included (Grey and Hashed) or not (White). For FAME-$\delta^{18}O_c$ the results where the populations are dissimilar are either plot as grey and hashed for *G. ruber* and *G. sacculifer* or white for *N. dutertrei* (Figure 5). This is because for these two species (*G. ruber* and *G. sacculifer*) we have the possibility to determine how robust these results are. We use the 1σ values of the observed (FAME) minus expected (MARGO), as computed by Roche et al. (2018) with the MARGO core top $\delta^{18}O_c$ database, as the potential error associated with the FAME model. Regions in which the difference between the two populations are larger than the potential error are associated with grey, whereas those less than the potential error as hashed regions (Figure 5A and B), these errors should be seen as a guide rather than a rejection of a site. Because the MARGO database does not contain *N. dutertrei* we have given the panels concerning this species a separate colour scheme, black represents grid-cells for which the two populations cannot

be said to be statistically different, white grid-cells are those in which the two populations can be said to be statistically different (Figure 5C). As we do not have a similar way to calculate the error for $T_c$, FAME-$T_c$ results are shown (in Figure 6) with this same binary pattern (*i.e.*, white grid-cells are those in which the two populations can be said to be statistically different and black are those in which the two populations can be said to be similar). To reduce the complexity, the overlay

5    of the species Anderson-Darling results (Figure 7) also uses the binary colour scheme (white or black).

[revised manuscript text omitted]
) is for temperature (Figure 4A-C) 0.5 °C and for $\delta^{18}O_{eq}$ (Figure 4D-F) 0.10 ‰, these errors should be seen as a guide rather than a rejection of a site. The results of thisBy using a fixed depth, non-FAME, these results show that the shallowest depths produce populations that are significantly different both in terms of

30 their mean values and their distributions. In the upper panel of Figure 4, the canonical El Niño 3.4 region is clearly visible at 5 m depth. Whilst differences exist between Anderson-Darling results for the temperature input data (Figure 4) and the FAME AD results ($\delta^{18}O$ (Figure 5) and $T_c$ (Figure 6)Figure 3), for instance close to the Panama isthmus, there are significant similarities between the plots.

-These plots also show that our FAME data (Figure 5-73), in which we allow foraminiferal growth down deeper than the depths in Roche et al. (2018), are a conservative estimate and thus are on the low-end (Figure 4), to account for potential discrepancies with depth habitats. In the original paper on depth habitats based upon temperatures derived from $\delta^{18}O_c$, Emiliani (1954) cautioned that the depth habitats obtained would represent a weighted average of the total population, and while foraminiferal depth habitats are likely to vary spatiotemporally, the average depth habitat is skewed toward the dominant signal (Mix, 1987).

To further test the model-driven results and to assess if they are still consistent when the depth limitation (400 m) is shoaled (to 60, 100, 200 m), the analysis was rerun with the aforementioned range of depths (Figure 5 and 6). Whilst it is possible to discern differences between the depths, it is important to note that a large percentage of the tropical Pacific remains accessible to palaeoclimate studies. A shallower depth limitation in the model increases the area for the 'warm' species, suggesting that the influence of a reduced variability in temperature or $\delta^{18}O_{eq}$ with a deeper depth limit causes the differences between $FP_{EN}$ and $FP_{NEL}$ to be reduced.

**3.42 Water depth and SAR**

Our analysis uses reanalysis data for the time period 1958-2015, a hypothetical core that had a comparable resolution would essentially be analogous with a sediment core with a rapid sediment accumulation rate (SAR), representing 50 yr cm$^{-1}$ (or 20 cm kyr$^{-1}$). Based on our analysis, such a hypothetical core could allow for the possible disentanglement of El Niño related signals from the climatic signal, but only in a best-case scenario involving minimal bioturbation, which is unlikely in the case of oxygenated waters. Extracting the oxygen saturation ($SO_2$) state, of the Pacific Ocean bottom waters from the Annual Climatology WOA13 give values that are predominantly >40 % (Figure 9B). Oxygen saturation is the concentration of Oxygen in a medium against the maximum that can be dissolved in the same medium. Whilst annual variability may exist, it is unlikely that bioturbation would be prevented by low oxygen. Therefore, using a cut off value that has been considered sufficiently high enough to outpace bioturbation (*e.g.*, Koutavas and Lynch-Stieglitz, 2003) of 5 cm kyr$^{-1}$ (Figure 9A) it can be demonstrated that much of the Pacific has an inferred lower sedimentation rate (< 5 cm kyr$^{-1}$; (Figure 9C) than this cut off value. To test the influence of bioturbation, the bioturbation simulator SEAMUS was run using the NGRIP time series. The results of SEAMUS highlight the potential single foraminifera depth displacement that low sedimentation rates can result in (Figure 9. Following the current available geochronological method (i.e., age-depth method) such specimens that are displaced in depth are assigned the average age of the depth that they were displaced to, which could result in erroneous interpretations of climate variability when analysis such as IFA is applied (Lougheed et al., 2018). The results of SEAMUS are plotted both as time series of the bioturbated NGRIP signal (Figure 10) and as histograms of the probability of finding a particularly pseudo-foraminifera with a given age in the bioturbation depth (Figure 11). As the bioturbation depth varies between 5, 10 and 15 cm for the different simulations of SAR, the histogram in each panel (in Figure 11) represent different thicknesses of sediment, i.e., for Figure 11 panels a, d, g, and j histograms represent data with a BD thickness of 5 cm. Likewise, the timeseries is plotted with the discrete 1 cm depth median age; the median age of the bioturbation depth (Figure

11) is the reason why each timeseries does not 'start' at 0 age (Keigwin and Guilderson, 2009). The variance within a single depth in a core largely represents the integrated time signal for that depth, as opposed to the variance of a climatic signal for an inferred (or measured) average age for the depths in question. The proxy variance will be based both upon a non-uniform distribution in temporal frequency of specimens, i.e., older specimens are few compared to younger specimens. A large

5 proportion of the specimens in the BD come from years that are 'proximal' this may give undue confidence that the probability of picking a specimen from these years is higher, however the long-tail of the distribution means that there is an equally high chance of picking a specimen that has come from several thousand years earlier than the discrete-depth's median age. If we consider for the moment this as picking specimens from a box, there is a high chance of picking from a single box that represents the age you want however there is an equally high chance of picking from numerous boxes with

10 varying age. If the spread in the climatic variable is uniform throughout this time then it can be possible to reproduce a similar signal, although this would not by definition represent the actual spread in the actual climatic variable for a given time, however the spread in the climate variable is unlikely to be constant. With a varying spread in the climatic signal bioturbation can introduce the possibility of spurious interpretations, but it is of course more obvious where the measured distributions over-exaggerate the climate signal (e.g., Wit et al., 2013). Furthermore, if we consider that researchers do not

15 pick as randomly as they profess, there is both a size and preservation bias to specimens selected, and size is not constant down-core (e.g., Metcalfe et al., 2015) we can further introduce bias within the dataset. The SEAMUS output that corresponds with our chosen SAR cut-off value of 5 cm kyr $^{-1}$ (Figure 10 and 11), the lower limit of our mask (Figure 9), is shown in panels in Figure 10 and 11, much of the region for which FAME is calculated upon has inferred sedimentation rates lower than this cut-off value (Figure 9C to 9H).

20 An additional factor in the post mortem preservation of the oceanographic signal in foraminiferal shells is whether the shells can be preserved. The GEBCO bathymetry data is binned into 250 m wide bins, and the data normalised to 1.0. As the data contains both bathymetric and topographic (below and above 0 m), the grey area in each histogram represent > 0 m (Figure 8). Whilst, there are differences depending on the cut-off value (Figures 8C to 8E) it is important to note that much of the canonical El Niño 3.4 region (Wang et al., 2017) used in oceanography (Figure 1) is also excluded from these suitable areas.

25 Overlaying the water depth and the SAR with the Anderson-Darling results (Figure 7) highlights that ~~The resolution of the ocean reanalysis data for the time period 1958-2015 would essentially be analogous with a sediment core representing 50 yr$^{-1}$ cm$^{-1}$ (or 20 cm$^{-1}$kyr$^{-1}$). Based on our analysis, such a hypothetical core with a rapid sediment accumulation rate (SAR) could allow for the possible disentanglement of El Niño related signals from the climatic signal using IFA, but only in a best-case scenario involving minimal bioturbation, which is unlikely in the case of oxygenated waters. Indeed, one should view~~

30 ~~discrete sediment intervals, and the foraminifera contained within them, as representative of an integrated multi-decadal or even multi-centennial signal (Lougheed et al., 2018; Peng et al., 1979), as opposed to other proxies such as corals (Cole and Tudhope, 2017), speleotherms (Chen et al., 2016), or varves in which distinct layers correspond to discrete time intervals (true 'time-series' proxies). Therefore, in order to reliably extract short-term environmental information from foraminiferal-based proxies, the signal that one is testing or aiming to recover must exhibit a large enough amplitude in order to perturb the~~

population by a significant degree from the background signal, otherwise it will be lost due to the smoothing effect of bioturbation (Hutson, 1980a; Löwemark, 2007; Löwemark et al., 2005, 2008; Löwemark and Grootes, 2004) upon the downcore, discrete-depth signal (Cole and Tudhope, 2017; Mix, 1987). Similarly, the individual characters of El Niño events, which are very short in duration, become lost in the bioturbated sediment record. New geochronological tools, such as dual $^{14}$C-$\delta^{18}$O measurements on single foraminifera (Lougheed et al., 2018), show that low sedimentation rate cores can have large variances in age between individual foraminifera present within a discrete 1 cm depth interval (Berger and Heath, 1968; Lougheed et al., 2018). In the case of a high SAR core (20 cm⁻¹kyr⁻¹), assuming a sediment mixing layer of 10 cm, benthic seafloor processes would result in a minimum 700-year 1σ confidence interval for a 1 cm discrete depth age (Lougheed et al., 2018). A consequence of bioturbation in sediment cores is that a series of high magnitude, but low frequency El Niño events could be smoothed out of the downcore, discrete-depth record. To investigate which areas of the sea floor can potentially preserve a foraminiferal downcore signal, we overlay our results with a map of time-averaged deep-sea SAR (Olson et al., 2016), adapted by Lougheed *et al.* (2018) to also show seafloor areas under the CCD depth, where carbonate material is not preserved (Berger, 1967, 1970b). Oof the total area where $FP_{EN}$ is significantly different from $FP_{NEU}$ (*i.e.* those areas where planktonic foraminiferal flux is suitable for reconstructing past ENSO dynamics), only a small proportion corresponds to areas where the sea floor is both above the CCD (< 3500 mbsl) and SAR is at least 5 cm/ka (Figure 3D 9). It is important to note that at certain locations, near islands or seamounts, the SAR and water depth may be high enough to allow for a signal to be preserved (Figure 8B). Also of note is that fact that much of the canonical El Niño 3.4 region (Wang et al., 2017) used in oceanography (Figure 1) is also excluded from these suitable areas in Figure 3D.

**4.0 Discussion**

**4.1 From Life to Sedimentary Assemblages**

Whilst we are principally interested in understanding whether living foraminifera can theoretically reconstruct ENSO, comparison with data requires an additional analysis, data-model comparisons are subjective, nominally supposing that the data is the value to be achieved by the model. However, if the foraminifera modulate the original climate signal, then preservation selectively filters which specimens are conserved whereas bioturbation acts to reorder, transposing the order in which they are recovered from the depth domain. Once the sediment is recovered, the researcher acts as a final filter, which is in essence a random picking. Although technically most researchers will pick whole shells so alongside size selectivity (*e.g.*, Metcalfe et al., 2015) there is also preservation bias associated with picking of foraminifera (*e.g.* Koutavas and Lynch-Stieglitz, 2003). Whilst the presence of depths in the ocean whereupon calcite is absent from sediments was described in the earliest work (e.g., Murray and Renaud, 1891; Sverdrup, 1942), overlaying maps of measured surface sediment carbonate percentage with water depth in the Pacific Ocean led Bramlette (1961) to coin the term 'compensation depth' (Wise, 1978). This work highlighted the 'narrow' depths (4-5000 m) in the Central Pacific of the CCD. Conceptually Berger (1971) placed

three levels in the Pacific ocean that were descriptive of the aspects (e.g., chemical, palaeontological and sedimentological) of the calcite budget; the saturation depth, demarking supersaturated from undersaturated; the lyscoline, the depth at which dissolution becomes noticeable (Berger 1968, 1971); and compensation depth (Bramlette, 1961), in which supply is compensated through dissolution. The aspects of the lysocline was estimated by the faunal assemblages of Parker and Berger (1971, figure's 14 and 15 of that publication), for much of the equatorial Pacific the lysocline is estimated at ~3800 m. As the lysocline is where dissolution becomes apparent, ergo it is a sample already visibly degraded, we therefore set the limit of the water depth mask shallower. In fact, in regions of high fertility, such as the Eastern Equatorial Pacific, the lysocline was estimated to be present at ~2800 m (Thunell et al., 1981) or ~3000 m (Berger, 1971; Parker and Berger, 1971) for instance core V21-28 close to the Galapagos Islands (01°05'N, 87°17'W) has a shallower dissolution affect even at 2714 m (Luz, 1973). A comparison between the hydrographic and sedimentary lyscoline, using a mooring in the Panama Basin showed that the sedimentary lyscocline is a product of where the hydrographic lyscocline meets the seafloor (Thunell et al., 1981), this could lead to dissolution within the water (e.g., Schiebel et al., 2007). In the EEP region the shallower lyscoline has an equally shallow CCD, at ~3600 m, here the high fertility is considered responsible for its shoaling, lowering the pH through increased $CO_2$ (Berger et al., 1976). The correspondence between lyscoline depth and CCD depth does not hold true for the entirety of the Pacific, plotting a N-S cross-section from 50°N to 50°S Berger (1971) noted that in the Central Equatorial Pacific, the high fertility region generates a larger zone of dissolution resistant facies even with a shoaled lysocline. If we factor in the sedimentation rate of the Pacific, which has been estimated to be considerably lower than 1 cm (Blackman and Somayajulu, 1966; Berger, 1969; Menard, 1964), then dissolution may become further exacerbated. The longer a shell remains at the sediment-water interface the greater the prospects for it to be dissolved become, therefore low SAR increases the chance of dissolution (Bramlette, 1961). For instance in 15 equatorial Pacific cores, below 4000 m, the average SAR was presented (Hays et al., 1969; here calculated) at 0.96 cm kyr[-1] (1σ ± 0.43 cm kyr[-1]). Although there are regions and/or core locations in which the SAR is higher, for instance eight EEP cores shallower than the lyscoline depth (Thunell et al., 1981) of ~2800 m were presented by Koutavas and Lynch-Stieglitz (2003) which have an average SAR, calculated, at 7.20 cm kyr[-1] (1σ ± 2.82 cm kyr[-1]). The average age for these same core's 0 cm core depth is 2184 years (1σ ± 1521 yrs), whilst it cannot be assumed that there has been no loss during recovery (i.e., core top is not sediment-water interface), a non-zero core top age is expected for both bioturbation (Keigwin and Guilderson, 2009) and dissolution. Alongside, the potential for dissolution there is the also the mixing of ocean sediments by the benthos (Bramlette and Bradley, 1942). For instance, Arrhenius (1961) noted that ash beds present in cores of the EEP (Worzel, 1959; Ewing et al., 1959) had a 2-3 cm layer above and below what should have originally been a sharp boundary in which they estimated that ~50% of the material originated from the other side of the boundary. If one assumes 1 cm kyr[-1] sedimentation rates, then the range in age of the obvious 6 cm mixed sediments is minimally ~6000 years per cm, comparison with an analogous SEAMUS simulation (bioturbation depth 5 cm; SAR 1cm) highlights the considerable spread in age, placing the 95.45% range between 110 and 18954 years (Figure 11). Much of this temporal variability will be hidden, especially when proxy values correspond with the expected values, and more obvious when the values are larger than expected (*e.g.*, Wit et al.,

2013). Owing to the lack of absolute variability during the Holocene the apparent confirmation of similarity between proxy values and modern distributions of the '*to be reconstructed*' variable is not a confirmation of proxy reliability. Especially in the tropics wherein seasonal variability is limited. The effects of both bioturbation and dissolution are further amplified when combined with finite sampling strategies. Therefore, 
[revised manuscript text omitted]

event should be recorded, our Anderson-Darling test for instance highlights that there are locations that cannot discern the difference between El Niño and other climate states whilst for the same time period there are locations where the different climate states can be differentiated. Whilst our analysis is a statistical treatment of the data, each species, and different types of phyto- or zooplankton preserved in ocean sediments, are likely to record the same set of environmental conditions differently (Mix, 2006). This is, in brief, the rationale for the development of FAME, the same climate signal seen through the view of species-specific proxies will give a fractured view constrained by each species particular ecophysiological constraints (Mix, 1987; Roche et al., 2018). A dynamic depth habitat in which the environmental signal becomes a weighted average of the water column can further confound the original signal (Wilke et al., 2006). What can be seen as contradictory reconstructions can therefore be viewed as the prevailing or dominant conditions at a given location at the time when environmental conditions overlap ecological constraints for a given species.

especially considering that tTerrestrial records suggest the number of El Niño events per century in the early Holocene (8-6 ka BP) was minimal (Moy et al., 2002), with between 0 and 10 events occurring per century. This dampened ENSO is observed within lake core colour intensity and records driven primarily by precipitation - although like other proxies this can also be interpreted differently, *i.e.* as a large changes in the hydrological cycle shifting precipitation away regionally (Trenberth and Otto-Bliesner, 2003) (Cole and Tudhope, 2017; White et al., 2018). If we assume for now that the number and magnitude of ENSO events wasere reduced, the relatively low downcore resolution of marine records may not accurately capture the dynamics of such lower amplitude ENSO events using existing methods. The sensitivity and probability of detecting a change in IFA with changes in frequency and amplitude, has been dealt with before (Thirmulai et al., 2013), although without considering bioturbation. The possibility of a marine sediment archive being able to reconstruct ENSO dynamics comes down to several fundamentals: the time-period captured by the sediment intervals (a combination of SAR and bioturbation), the frequency and intensity of ENSO events, as well as the foraminiferal abundance during ENSO and non-ENSO conditions.

The synthesis of pseudo-timeseries to discern the potential distribution for different scenarios, whilst a necessary approximation, is nonetheless one that is free of cause and causality. Modulating a timeseries for events with enhanced or weakened amplitude or fewer or greater number of events assumes in essence that there is limited feedback both regionally (between two sites) and internally within the timeseries (i.e., a process that operates on a higher level). Reconstructions of the past can benefit from inclusion within conceptual frameworks that incorporate both data and modelling studies (e.g., Trenberth and Otto-Bliesner, 2003; Rosenthal and Broccoli, 2004; McPhaden et al., 2006). The use of coupled ocean-atmosphere models (e.g., Clement et al., 1999; Zebiak and Cane, 1987); isotope enabled Earth system models (e.g., iCESM; Zhu et al., 2017); or multi-model ensembles with prescribed boundary conditions can be used for the generation of timeseries in which the physics of atmospheric and oceanic circulation are constrained and feedbacks between sites can occur. The perceived failure of several climate models to resolve ENSO adequately, resulting in variable ENSO frequency and amplitude between models, could therefore be used to determine the proxy signal from model derived timeseries at different frequencies and intensities of ENSO. Albeit a timeseries of variable ENSO that is grounded in ocean-atmosphere coupling.

Such analysis could also provide information on a secondary assumption, in which time slices from the same core inherently assume that where a particular oceanographic feature exists now is also where it may have existed before. This gives a somewhat binary view, the feature either occurs or does not occur, and if it occurs then it has either enhanced or weakened. Yet this can (though not always) preclude a scenario in which the feature has shifted. Analysis of the El Nino patterns suggests that there are two types of El Nino that are spatially delineated: the dateline Central Pacific El Nino and the Eastern Pacific El Nino. The expansion, contraction or shift of certain large scale oceanographic features (e.g., Polar Front, Upwelling) during periods of warmer than average (e.g., the last interglacial) or colder than average temperatures (e.g. the LGM) can complicate the comparison of two down core samples, i.e., a static core continuously recording a particular climate event as opposed to a shifting oceanographic regime 
[revised manuscript text omitted]
 the integrating the δ¹⁸O$_{eq}$ values using a growth-rate based weighting (FAME; Roche et al., 2018), a computed integrated signal based upon the growth rate and the δ¹⁸O$_{eq}$ values. Values are in per mil (‰ V-PDB).**

[Figure]

[Figure]

**Figure 3.** Total variance and Interannual variance and the range in standard deviation of the Monte-Carlo experiments. (a) Total variance of *Globigerinoides sacculifer* $\delta^{18}O_c$, using FAME-$\delta^{18}O_{c0}$ for a cut-off value of 60 m. (b) The ratio of (a) and (c), where (c) is the Interannual variance of the timeseries of (a). In (d-i) we plot the range in standard deviation obtained by picking 60 months with replacement with 10,000 iterations, the experiments are as follows: (d) the same months were chosen for all grid-points for each iteration of the Monte-Carlo; (e) each grid-point has its own randomly selected months for each iteration of the Monte-Carlo; (f) the same as (e) but we weight the values by the total amount of growth per month; (g) the months selected for (e) were re-run but a random variability is added to each month; (h) the months selected for (e) were re-run but a random measurement error is added to each month; and (i) the months selected for (e) were re-run but the (g) random variability and (h) measurement error were added. Note the scale change between (d-f) and (g-i).

hypothesis ($H_0$) and therefore the foraminiferal population (FP) of the El Niño is similar to the Non-El Niño, and therefore the distribution between the neutral climate and El Nino cannot be said to be different ($FP_{El\ Niño} = FP_{Non-El\ Niño}$). White/grey areas reflect regions in which the $H_1$ hypothesis is accepted, therefore the distributions can be said to be unique ($FP_{El\ Niño} \neq FP_{Non-El\ Niño}$). Hatching represents the FAME species specific values of 0.32 ‰ and 0.20 ‰. The final, bottom, panel represents a synthesis of the upper panels areas in which reconstructions are possible, represented by both the blue and white area. The black area represents those areas where the null hypothesis for all three species cannot be rejected. Blue represents sedimentological limitations that preclude the extraction of a foraminiferal signal from the sediment: deeper than 3500 mbsl (below CCD) and time averaged sediment accumulation rate (SAR) of less than <5 cm/ka (Lougheed et al., 2018; Olson et al., 2016) (see Figure S7 for the different sedimentological components).

[Figure]

**Temperature at 5 m**

**Temperature at 149 m**

**Temperature at 235 m**

$FP_{El\ Niño} \neq FP_{Non\text{-}El\ Niño}$     $(FP_{El\ Niño} \neq FP_{Non\text{-}El\ Niño})$ within error     $FP_{El\ Niño} = FP_{Non\text{-}El\ Niño}$

[Figure]

**Figure 4.** Anderson-Darling Results for Input datasets of Temperature and Equilibrium δ¹⁸O (δ¹⁸O$_{eq}$). Results of the test in which input variables underwent the same statistical procedure (see section 2.0) as the modelled data for (A-C) temperature and (D-F) δ¹⁸O$_{eq}$ values. Here, model input data was extracted for three fixed depths ([A & D] 5 m; [B & E] 149 m; [C & F] and 235 m) without any growth weighting applied. Black regions are those grid points in which the null hypothesis (H$_0$), that the El Niño and Non- El Niño populations are not statistically different (FP$_{El Niño}$ = FP$_{Non-El Niño}$), cannot be rejected. Gray regions  **represent** grid pointss **where the H$_1$ hypothesis is accepted, therefore the distributions of the foraminiferal population (FP) for El Niño and Non- El Niño can be said to be unique (FP$_{El Niño}$ ≠ FP$_{Non-El Niño}$)**. The hatched regions represent areas were the H$_1$ hypothesis can be accepted, therefore the distributions of the foraminiferal population (FP) for El Niño and Non- El Niño can be said to be unique (FP$_{El Niño}$ ≠ FP$_{Non-El Niño}$), though the difference between the means of tested distribution are less than (A-C)  **0.5°C** or (D-F) 0.1 ‰**. Each panel represents a single depth (5, 15**049 **and 23**550 **m)** .

[Figure]

**Figure 5.** Anderson-Darling Results for modelled FAME- (δ¹⁸Oeqc): Panels representing  locations of
where dissimilar and similar values of FAME modelled species δ¹⁸O occur between climate states, for (columns) particular species
and (rows) particular model depth cut-off limits. Each panel represents the Anderson-Darling test result, which are plotted with
([A] *Globigerinoides sacculifer* and [B] *Globigerinoides ruber*) and without ([C] *N. dutertrei*) model derived error. For all panels
black areas reflect latitudinal and longitudinal grid points that failed to reject the null hypothesis (H₀) and therefore the
foraminiferal population (FP) of the El Niño is similar to the Non-El Niño (FP_El Niño = FP_Non-El Niño). The results in which the H₁
hypothesis is accepted, in which the, therefore the distributions can be said to be unique (FP_El Niño ≠ FP_Non-El Niño), are plotted as
either: (A – *G. sacculifer*, B – *G. ruber*) grey and hatched or (C – *N. dutertrei*) solely as white regions. For species with calculatable
error, grey regions represent values where the difference between the two means of the population is greater than species-specific

standard deviation of the FAME model and hatched regions represent those in which the means are less than this standard deviation (Roche et al., 2018). For species without a calculatable error, the regions are plotted in white. The rows represent the model runs with a depth cut-off limit at: (A-C) 60 m; (D) 100 m; (E) 200 m; and (F) 400 m.

5 ~~that can reconstruct El Niño events for (Left Panel) *G. ruber*; (Middle panel) *G. sacculifer*; and (Right panel) *N. dutertrei*. Panels represent the Anderson-Darling test results for the species *G. ruber*; *G. sacculifer*; and *N. dutertrei* where the black areas reflect latitudinal and longitudinal grid points that failed to reject the null hypothesis (H0) and therefore the foraminiferal population (FP) of the El Niño is similar to the Non-El Niño, and therefore the distribution between the neutral climate and El Nino cannot be said to be different (FP El Niño = FP Non-El Niño). White/grey areas reflect regions in which the H1 hypothesis is accepted, therefore the distributions can be said to be unique (FP El Niño ≠ FP Non-El Niño). Hatching represents the FAME species specific values of 0.32 ‰ and 10 0.20 ‰. Each row represents a particular depth limit for the model, from top to bottom: 60 m; 100 m; 200 m and 400 m.~~

[Figure]

**Figure 6. Anderson-Darling Results for modelled FAME-T$_c$: Panels representing locations of where dissimilar and similar values of FAME modelled temperature recorded in the calcite shells (Tc) occur between climate states, for (columns) particular species and (rows) particular model depth cut-off limits. Each panel represents the Anderson-Darling test result, which are plotted with ([A]** *Globigerinoides sacculifer* **and [B]** *Globigerinoides ruber***) and without ([C]** *N. dutertrei***) model derived error. For all panels black areas reflect latitudinal and longitudinal grid points that failed to reject the null hypothesis (H$_0$) and therefore the**

foraminiferal population (FP) of the El Niño is similar to the Non-El Niño, and therefore the distribution between the neutral climate and El Nino cannot be said to be different (FP$_{El\ Niño}$ = FP$_{Non-El\ Niño}$). The results in which the H$_1$ hypothesis is accepted, in which the distributions can be said to be different (FP$_{El\ Niño}$ ≠ FP$_{Non-El\ Niño}$), are plotted as white regions. The rows represent the model runs with a depth cut-off limit at: (A-C) 60 m; (D) 100 m; (E) 200 m; and (F) 400 m.

[Figure]

Figure 7. Combined A-D plots. As figure 5 and figure 6, in that panels represent locations of where dissimilar and similar values for the two climate states for (a-d) FAME-δ$^{18}$O$_{eq}$ modelled oxygen isotope values or (e-h) FAME-T$_c$ modelled temperature recorded in the calcite shells (Tc) occur. Each panel represents the Anderson-Darling test result, the results for *Globigerinoides sacculifer, Globigerinoides ruber* and *N. dutertrei* are overlaid. For all panels black areas reflect latitudinal and longitudinal grid points that failed to reject the null hypothesis (H$_0$) and therefore the foraminiferal population (FP) of the El Niño is similar to the Non-El Niño, and therefore the distribution between the neutral climate and El Nino cannot be said to be different (FP$_{El\ Niño}$ = FP$_{Non-El\ Niño}$). The results in which the H$_1$ hypothesis is accepted, in which the distributions can be said to be different (FP$_{El\ Niño}$ ≠ FP$_{Non-El\ Niño}$), are plotted as yellow where the depth is deeper than 3500 m bsl or purple where the depth is shallower than 3500 m bsl (see Figure 8). Purple locations are where our results suggest that the signal of ENSO has different values and the water depth allows for preservation – although this purple region will be smaller when inferred SAR is taken into account (see Figure 9). The rows represent the model runs with a depth cut-off limit at: (A and E) 60 m; (B and F) 100 m; (C and G) 200 m; and (D and H) where a combination of depths were utilised (Pracht et al., 2019).

[Figure]

**Figure 8.** Bathymetric map of the Tropical Pacific Ocean highlighting the areas above and below the Lysocline and/or Calcite compensation depth (CCD). (A) GEBCO map of height relative to 0 m; (B) same as (A) with location of seamounts plotted (white stars); (C-E) binary colour map of GEBCO data, yellow is values below cut-off depth value ([C] 3500 m below sea-level (bsl); [D] 4000 m bsl; and [E] 4500 m bsl respectively) and purple above the cut-off depth value. The histograms represent the normalised frequency of grid cell height in bins of 250 m wide, yellow is values below cut off value ([C] 3500 m below sea-level (bsl); [D] 4000 m bsl; and [E] 4500 m bsl respectively), purple above cut off value. The grey bins in each histogram are those above 0 m.

[Figure]

**Figure 9. Map of the sedimentation rate and oxygen saturation for the Tropical Pacific. (A) Inferred sedimentation rate (Olsen et 2016). White regions represent continental shelf. (B) Oxygen saturation of the bottom grid layer of World Ocean Atlas 2013 (data from: https://www.nodc.noaa.gov/cgi-bin/OC5/woa13/woa13oxnu.pl ). (C, E, G) Overlay between water depth and inferred SAR, Red / Pink: Continental shelf sediments that are (Red) shallower or (Pink) deeper than 3500 mbsl; Gray / White: SAR lower than SAR threshold and the seafloor depth is (grey) shallower or (white) deeper than 3500 mbsl; Light Yellow/Gold: Light yellow represents areas where the SAR is above the threshold but the water depth is deeper than 3500 mbsl in comparison Gold represents areas where the SAR is above the threshold and the water depth is deeper than 3500 mbsl. The ideal locations are therefore plotted as Gold. Cut-off limits for SAR are (C) >1 cm kyr$^{-1}$; (E) >2 cm kyr$^{-1}$ and (G) >5 cm kyr$^{-1}$, (D, F, H) bioturbation simulations for the minimum threshold for each SAR (see Figure 10 and Figure 11 for the output of SEAMUS). Each plot gives the**

**input values of NGRIP (grey) and for each SAR three analysis were performed with different bioturbation depths (BD) of: (Blue) 5 cm; (Green) 10 cm; and (Orange) 15 cm.**

[revised manuscript text omitted]

Font: (Default) +Headings (Times New Roman), Not Highlight

| Page 17: [2] Formatted | Brett Metcalfe | 21/07/2019 14:49:00 |

Not Highlight

| Page 17: [3] Formatted | Brett Metcalfe | 21/07/2019 14:49:00 |

Font: (Default) +Headings (Times New Roman), Not Highlight

| Page 17: [4] Formatted | Brett Metcalfe | 21/07/2019 14:49:00 |

Not Highlight

| Page 17: [5] Formatted | Brett Metcalfe | 21/07/2019 14:49:00 |

Font: (Default) +Headings (Times New Roman), Not Highlight

| Page 17: [6] Formatted | Brett Metcalfe | 21/07/2019 14:49:00 |

Not Highlight

| Page 17: [7] Formatted | Brett Metcalfe | 21/07/2019 14:49:00 |

Font: (Default) +Headings (Times New Roman), Not Highlight

| Page 17: [8] Formatted | Brett Metcalfe | 21/07/2019 14:49:00 |

Not Highlight

| Page 17: [9] Formatted | Brett Metcalfe | 21/07/2019 14:49:00 |

Font: (Default) +Headings (Times New Roman), Not Highlight

| Page 17: [10] Formatted | Brett Metcalfe | 21/07/2019 14:49:00 |

Not Highlight

| Page 17: [11] Formatted | Brett Metcalfe | 19/07/2019 18:07:00 |

Font: Italic, Not Highlight

| Page 17: [12] Formatted | Brett Metcalfe | 19/07/2019 18:07:00 |

Not Highlight

| Page 17: [13] Formatted | Brett Metcalfe | 19/07/2019 18:08:00 |

Not Highlight

| Page 17: [14] Formatted | Brett Metcalfe | 19/07/2019 18:08:00 |

Font: Italic, Not Highlight

| Page 17: [15] Formatted | Brett Metcalfe | 19/07/2019 18:08:00 |

Not Highlight

| Page 17: [16] Formatted | Brett Metcalfe | 21/07/2019 14:49:00 |

Not Highlight

| Page 17: [17] Formatted | Brett Metcalfe | 21/07/2019 14:49:00 |

Not Highlight

| Page 17: [18] Formatted | Brett Metcalfe | 19/07/2019 14:48:00 |

Highlight

| Page 17: [19] Formatted | Brett Metcalfe | 19/07/2019 18:08:00 |

Not Highlight

| Page 17: [20] Formatted | Brett Metcalfe | 09/07/2019 14:04:00 |

Highlight

| Page 17: [21] Formatted | Brett Metcalfe | 09/07/2019 14:04:00 |

Highlight

| Page 17: [22] Formatted | Brett Metcalfe | 10/07/2019 11:23:00 |

Highlight

| Page 17: [23] Formatted | Brett Metcalfe | 20/07/2019 19:09:00 |

Highlight

| Page 17: [24] Formatted | Brett Metcalfe | 21/07/2019 10:41:00 |

Superscript

| Page 17: [25] Formatted | Brett Metcalfe | 21/07/2019 10:41:00 |

Subscript

| Page 17: [26] Formatted | Brett Metcalfe | 21/07/2019 10:43:00 |

Font: Italic

| Page 17: [27] Formatted | Brett Metcalfe | 20/07/2019 19:09:00 |

Font: Italic

| Page 17: [28] Formatted | Brett Metcalfe | 21/07/2019 16:17:00 |

Font: Italic

| Page 17: [29] Formatted | Brett Metcalfe | 21/07/2019 15:06:00 |

Highlight

| Page 17: [30] Formatted | Brett Metcalfe | 10/07/2019 12:25:00 |

Highlight

| Page 17: [31] Formatted | Brett Metcalfe | 10/07/2019 12:25:00 |

Highlight

| Page 17: [32] Formatted | Brett Metcalfe | 10/07/2019 12:25:00 |

Highlight

**Comments to the Author:**

*Editor Decision: Reject (27 May 2019) by Helen McGregor*

*Comments to the Author:*

*Dear Dr Metcalfe*

*Thank you for your manuscript titled 'On the validity of foraminifera-based ENSO reconstructions' (manuscript #cp-2019-9) submitted for publication in Climate of the Past. The manuscript has received three referee reviews and one short comment during the Discussion phase.*

*After careful consideration of your Author Responses I regret to inform you that we are unable to accept the manuscript for publication at this time. The three referee reports and short comment are supportive of the FAMES approach, however they bring up similar issues on this specific manuscript, and fundamentally, there appears to be a mismatch between the intention of the study as read by the referees (and myself) in the submitted manuscript, and the intention of your study as stated in the Author Responses. The proposals for revision in the Author Response do not to my mind reconcile this key issue, nor adequately address the other major issues, and therefore I cannot accept the manuscript.*

*The detailed referee reports are made available for your reference and I encourage you to engage with these comments constructively in your future studies.*

*I wish to clarify for future reference that Climate of the Past referees are not required to cross-reference each other's reports on the Discussions forum and are free to develop their reviews independently.*

*Again, thank you for your interest in the journal and I regret that I cannot be more positive on this occasion.*

*Yours sincerely,*

*Helen McGregor*

Having received reviews from all three anonymous referees we would like to make some general comments before posting a line by line response to referees #2 and #3.

We note that all three referees have submitted very similar reviews. We invested quite some effort in responding to the points raised by the first referee as quickly as possible, so that referees #2 and #3 would be able to consult our response before submitting their own personal reviews. It seems, however, that these two referees submitted very similar reviews as referee #1, which would suggest that referees #2 and #3 have not taken our response to referee #1 into account. It is unfortunate that the full potential of the Climate of the Past Discussions forum format was not exploited. It would indeed have been much more profitable to discuss the comments of referee #2 & #3 if they had been written in the light of our initial response. We also note that even though we strongly disagree in our initial response with the first reviewer comment, at time of writing, there has been no further comments from the reviewers on the discussion page on this first response although the discussion period is not ended. In the interest of stimulating such a discussion, we will briefly synthesise and address the three main criticisms raised by all three reviewers here and our arguments against them. There are several other, minor criticisms/corrections raised by the referees which we find very valid, for which we thank the reviewers and which we will gladly address. The main criticisms raised by the three reviews can be distilled into the following four points:

*(1) The study is a forward model and, therefore, does not inverse model ENSO recorded by sediments.*

This is correct. Our study uses 60 years of observed monthly climate data to forward model planktonic foraminifera populations accounting for habitat water depths and growth season for the entire Pacific Ocean, allowing us to investigate whether or not ENSO dynamics are recorded by foraminifera populations *in the water.* We think that this investigation is in itself a fundamentally interesting subject for research/publication, since it is the first step/prerequisite to investigate whether it is possible to distinguish ENSO from non-ENSO conditions from the planktonic foraminifer fossil record. This question has not been investigated before and is, therefore, new and has the potential for interesting conclusions.

Our model approach is indeed not an inverse model. It is also not a sediment model. The manuscript that we have submitted is a forward model of planktonic foraminifera populations in the water. Clearly, the reading of the referees' comments shows that they would have preferred if we had submitted a completely different manuscript; as they have reviewed our manuscript as if it were yet another study on the inverse problem of detecting a change in ENSO from a change in the distribution of sediment archive planktonic foraminifer $\delta^{18}O$. We are disappointed and strongly disagree with being reviewed in the view of a different manuscript that could have been written, and not evaluated based on the actual manuscript submitted and the science contained therein. We feel this is against the basic principles of a standard review process and strongly object to this treatment of our work.

*(2) The FAME output in this study has not been tested against core tops / sediment records, etc.*

This is correct. In the present manuscript, we have indeed used a 60 years monthly record of climate input to run our FAME foraminifera population model. Core tops and sediment records in the Pacific integrate foraminifera populations from centuries/millennia of time, and are therefore totally unsuitable to compare to our model output in the context of evaluating the impact of ENSO. The best thing that we could think of to compare our FAME output to, in the interest of this particular study, would be data from monthly plankton tows spanning multiple decades, but we couldn't find any. There is thus simply no data we know of that can be used meaningfully in the context of our current work on ENSO. We note, however, that the FAME model itself has been validated against

core tops in Roche et al. (2018), including core tops from the Pacific Ocean. The claim that FAME has not been validated in the Pacific Ocean is therefore unfounded.

*(3) The statistics are not applicable to palaeo records*

This is correct. We did indeed not select a statistical test on the basis of it being applicable to palaeo records. We chose the statistical test (Anderson Darling test) that was most suitable for our purpose and hypothesis: to examine the (dis)similarity of different high-resolution probability distributions produced by our FAME output driven by 60 years of monthly observational data. This approach allows us to use the full potential of our model output to directly test our hypothesis. The standard deviation (as suggested by all three reviewers), is unsuitable for comparing the dissimilarity of distributions, as it can return spurious results: for example, it is possible that two distributions of differing shape could have the same standard deviation, whereas an A-D test would detect a difference. The research question and its associated hypotheses are the central tenant upon which we based our experimental design. We note that we are not developing a statistical toolbox for palaeorecords and readers should of course always assess, on a case by case basis, statistics that are appropriate for their own studies.

*(4) The authors have not run a bioturbation model.*

This is correct. Since we are not modelling sediments in this manuscript, we did not carry out a transient bioturbation model. It would not be suitable to carry out a transient bioturbation model upon 60 years' worth of input data, seeing as bioturbation models require, due to the interplay between sediment accumulation rate (SAR) and bioturbation depth, many millennia of data/spin up time in order to produce valid output.

We were, however, curious as to whether the parts of the Pacific Ocean where FAME predicts that foraminifera in the water can record ENSO dynamics coincide with high SAR sediments. Therefore, as an extra, we included a rough map of regions of the Pacific Ocean where SAR < 5 cm/ka. The reviewers asked why 5 cm/ka was used as a cut-off point. We intentionally used a very generous cut-off point of 5 cm/ka and assumed that the reader would understand that 5 cm/ka is a very slow SAR that would severely hamper the detection of average ENSO dynamics for specific time periods (i.e. less than one millennia). It is not, however, necessary to carry out a transient bioturbation model to understand the time periods integrated by a 5 cm/ka SAR. We can simply carry out a calculation following established understanding of the influence of bioturbation upon the age dispersal of single foraminifera (Berger and Heath, 1968; Berger and Johnson, 1978; Berger and Killingley, 1982), the same understanding that is included in transient bioturbation models themselves (e.g. Trauth, 2013; Dolman and Laepple, 2018; Lougheed et al., 2018). In such a case, assuming the common bioturbation depth of 10 cm (Peng et al., 1979; Trauth et al., 1997; Boudreau, 1998), we can calculate that the 1 sigma age value of foraminifera contained within a single cm of a 5 cm/ka core is:

10 [cm] / (5 [cm/ka] /1000) = 2000 years (from which follows that 2 sigma = 4000 years).

Our map may of course miss limited parts of the Pacific Ocean with SAR higher than 5 cm/ka. We will mention this and/or highlight such places in the final version of the manuscript.

Kind Regards,

Brett Metcalfe

Bryan Lougheed

Claire Waelbroeck

Didier Roche

**B. Metcalfe on behalf of the co-authors, Response to Reviewer 1**

*[reviewer comments as quoted blocks]*

We appreciate the effort the reviewer has taken in reviewing our paper and thank them for their interest in our work. The rapid publishing of a review on the same day as the reviewer has been nominated for review, especially given the level of detail contained within the review, must have involved great effort, and we thank the reviewer for making time for us. The reviewer raises some important points, but we believe that the reviewer may have misunderstood the goals of our paper, or that we have may not have explained them clearly enough. We would, therefore, like to take this opportunity to clear up these matters.

> *["Page 12 line 22 – What is meant by "…especially if an individual foraminiferal analysis… approach is used…" I thought the whole analysis in the paper was on whether individual foraminifera analysis can be used? Was there another method tested (for example the means analysis referred to at Page 7 line 3)? ; Page 9 line 23 – The focus of this paper is on IF analysis. Why are the Koutavas and Lynch- Stieglitz, 2003; Koutavas and Lynch-Stieglitz, 2003 etc. cited here? The whole discussion in this paragraph, lines 16-31 feels out of place."].*

We should make clear that we are principally not undertaking an evaluation of sediment-based individual foraminifera analysis, but rather an evaluation of the foraminifera populations in the water, before they have been incorporated into the sediment. We will endeavour to make this clearer in the manuscript, as we may not have done so (: we do not explicitly say 'we are modelling individual foraminifera distributions' as the reviewer assumes we are, however we also do not explicitly say 'we are not modelling individual foraminifera distributions' therefore, a clarifying sentence we will be added to a revised manuscript). Although, clearly, the question of whether foraminifera populations in the water are themselves able to record ENSO or not is important for sediment-based reconstructions, so we have included some minor discussion of sediment dynamics. Furthermore, the FAME methodology we apply does not simulate individual foraminifera, rather, it produces what the likely $\delta^{18}O_c$, $T_c$ value for a time-step using a function that 'weights' water depths by foraminiferal growth. In other words, FAME is producing mean population values for a given time slice.

The reviewer has suggested that we carry out analysis that is outside the scope and purposes of our manuscript, for example:

> *"Furthermore, the statistical analysis focuses on a forward problem rather than the inverse problem that is the real challenge for detecting changing ENSO from individual foraminiferal analysis".*

We should make clearer, and will be glad to do so in the final version, that the purpose of our study is to determine whether or not the foraminiferal population produced under El Nino conditions are statistically different from the Neutral and La Nina conditions during the period of the observed climate record. Such a difference is an important prerequisite for any analysis: Can we detect the change we are searching for? In order to address this question, we choose a forward model as the most suitable tool. In the following reply to the reviewer, we will expand upon why the inverse method may not be suitable for our application/research question; discuss the use of the statistical methods in our paper which will hopefully address the reviewers concerns about palaeo-applications; discuss the reviewer's comments regarding validation; and answer specific questions.

**Inverse Problem**

> *[Focus in the inverse problem. It is really the inverse problem of detecting a change in ENSO from a change in the distribution of foraminifera d18Oc or T that is the focus of IF ENSO reconstructions. The analysis in this paper basically asks the question: are the distributions from El Niño months different from neutral or La Niña months? This is a useful first step in the inverse problem but it doesn't really answer the question stated in the title about the validity of foraminifera-based ENSO reconstructions.]*

While interesting, we believe that the inverse approach is not suitable to our particular research question. We will make this clearer in the manuscript. The inverse problem, as its very name

suggests, flips the question: "we have this data what variables must have occurred to produce them". One is inverting the scientific method to explain causal factors from observations rather than explaining observations with causal factors. The reviewer points out several papers that have done this approach, but there is a lack of large-scale analysis beyond single cores or forcing a climate model with these boundary conditions to explain inter basin variability. So, why did we not use an inverse problem, well, firstly, it has been done before, as the reviewer suggests:

> ("This type of analysis has been done before (Thirumalai, Ford, White), with a focus on the inverse problem of estimating ENSO change from individual foraminifera distributions. Here the novelty is the inclusion of a forward model of foraminifera growth rate.")

And, secondly, it would not address the central question we are asking. Using the inverse problem would change our fundamental question from '*determining whether the foraminiferal population produced under El Nino conditions are statistically different from the Neutral and La Nina conditions*' to an entirely different question, namely, '*with this dataset what magnitude and frequency of ENSO would have to have occurred to produce these observations*'. The reviewer seems to partly realise that these are not the same question:

> ("The forward problem is whether El Nino, neutral, and La Nina months have different distributions and requires that each individual d18Oc or T value be assigned beforehand to one of those three states.")

Crucially and in contrast to the inverse methodology, our research question does not exclude the possibility of there being no detectable change, while the reviewer's proposed question forces the ENSO parameters to contort into those that generate a particular dataset. In conclusion (of this point), the inverse problem approach would not give us an answer to the hypothesis and/or research question that we have chosen: **with a chosen set of input parameters (temperature, salinity) using an ecological model what would the theoretical observations of $T_C$ or $\delta^{18}O$ be? And would the populations of El Nino, Neutral and La Nina climate be similar or different?**

In addition, for the purposes of carrying out the inverse problem, there is a lack of sediment-based data to do a large, basin-wide inverse-analysis. As we have already shown in our study (namely in the SAR and water depth/CCD plots), the seafloor of the Pacific is not conducive to providing samples with which to perform an inverse analysis on a basin-wide scale, and there is also a sampling bias, as the reviewer correctly alludes to: ("Page 10 line 28 – The discussion of model limitations does not ask what would seem to be the most important questions: Does the modeled growth rate actually reflect the real ocean (and the sampling bias for what is recorded in sediments)?").

A further problem is that we would need to vary temperature AND salinity to realistically produce an inverse model, and not just temperature, as is the case when producing an accurate $\delta^{18}O_C$.

> [ Furthermore, the statistical analysis focuses on a forward problem rather than the inverse problem that is the real challenge for detecting changing ENSO from individual foraminiferal analysis. The forward problem is whether El Nino, neutral, and La Nina months have different distributions and requires that each individual d18Oc or T value be assigned beforehand to one of those three states. The inverse problem is to determine from comparison of two different d18Oc or T distributions (as would be measured in two sediment samples) whether any change in their distributions occurred and whether it can be ascribed to changes in the statistics of ENSO events (frequency, magnitude).]

We would like to point out that neither in our response to the reviewer, nor in the paper, are we being critical of inverse modelling. We believe that it can be an appropriate and valuable technique, but it is not the suitable technique for answering our central research question. There is a fundamental difference in what the reviewer would like us to produce and what we have done (*"With different analyses the authors could address the questions they pose. However, it could be very different from the manuscript in its current form and in my opinion would need to independently evaluated and reviewed"),* namely that our paper sets out to use FAME to produce distributions using an input of temperature and salinity. The

reviewer would like us to produce temperature and salinity from the distributions. However, how we should create these distributions without the input temperature parameter required for FAME is not clear. Indeed, as should be clear from the reading of the FAME methodology already published (Roche et al., 2018), there is no simple bijective relationship between the $\delta^{18}O_c$ and the oceanic variables (T, $\delta^{18}O_{sw}$).

**Statistics**

*[Apply statistical tests on parameters used on paleo-IF distributions . The author's use Anderson-Darling tests for differences in distributions. They should demonstrate how this might be useful for paleo-IF analysis. It would also be greatly to their advantage to test the approaches actually used for paleo-IF analysis (1-sigma, quantiles) to see how they perform in this framework. A welcome contribution would be demonstration that a new/different type analysis from those typically applied to paleo-IF distributions is better. As it stands, the focus on the forward problem and on statistical approaches not used for paleo-IF analysis make the manuscript in its present form not a good evaluation of the IF approach for ENSO reconstruction.*

*[Page 3 line 23 – Here the authors introduce the 1-sigma d18Oc parameter than has been used in some studies to look at changes in ENSO variance. But, they never really address whether this parameter is useful and can detect changes in ENSO. Thirumalai et al. (2013) took this question on already. More discussion of what has been done previously is needed. Also, why not test the actual way that IF analysis is used (e.g. 1sigma, quantiles etc.) rather than a new method as introduced here (Anderson-Darling test)?]*

To reiterate, the reason we use the long-established Anderson-Darling (AD) (1954; doi:10.2307/2281537) approach is because it is the most suitable method for the computer modelling study that we are carrying out, for the following reasons: The FAME model, coupled to the observational climate data that we have inputted, can produce high-resolution probability density functions (PDF) associated with El Nino, La Nina and neutral conditions. An Anderson-Darling test allows us to directly test if these PDFs are significantly different from one another or not. Obviously, an Anderson-Darling test may be more difficult to apply to foraminifera sampled from natural archives, where workers are limited in data resolution by the number of foraminifera that can be picked for analysis, by bioturbation of the natural archive, etc. Subsequently, in those cases, it might indeed make sense to use simpler statistics such as standard deviation and quantiles. Since we are not analysing natural archives, but rather data produced by a model for which we control what is generated, it makes more sense, in our case, to use a more powerful method such as the Anderson-Darling test.

Nonetheless, we appreciate that workers analysing natural archives are accustomed to using more straightforward statistical analyses, and would also like to see the 1 sigma and quantile intervals, so we will additionally report those for comparison in a revised version of the manuscript. Of course, these statistical parameters may not answer our research question and will not impact the answer to our research question (as AD is the appropriate test). These tests may however have flaws, given that the standard deviation is not the best descriptor of non -normal data and outside of the realms of statistics its usage assumes that the data will significantly impact the standard deviation or that standard deviation can be used as a measure of ENSO. If for instance one follows the reasoning of Mix (1987), a species may actually calcify solely in the anomaly regions (the la nina or el nino), and such species may not have a standard deviation that due to ENSO. This argument can be used for quantile-quantile, if the species does not calcify for the full year, an assumption of such an approach, it will mean that the data does not reflect the full year but a subset.

**Validation of the forward model**

*[Here the novelty is the inclusion of a forward model of foraminifera growth rate. This model is used to estimate the biased sampling in depth and time that different foraminiferal species have, and how this contributes to the analysis of the ENSO signal.*

The model is indeed a simplification of the earlier FORAMCLIM model, because light, food etc. are not easily parameterised in models or validated with proxies. However, to claim that Roche et al. has "no clear assessment of the errors [was] presented" is not correct in our opinion. For the question relating on how the model works, the reviewer is referred to the initial publication where all the equations are described in detail; the code is itself also made available in the supplementary online material so that the reviewer can even try the model itself. While the reviewer's comments on Roche et al. (2018) relate to another publication, in the interest of discussion let us focus on how the reviewer would validate the model.

> *"I think the authors should use the modeled growth rate for the species they are targeting and calculate the relative abundance of those three species in a sediment sample. This can be compared to the measured relative abundance of those three species (summing to one) recalculated from their relative abundance amongst all species counted in coretop datasets. This should be shown as a scatterplot of observed vs. predicted on x- and y-axes rather than on a map as is shown in the supplement to Roche et al., 2018."*

Unfortunately, this would not work due to the closed sum problem. Relative abundance between species is based upon a closed sum calculation, i.e. not the true abundance as a fraction of the total foraminifera flux. Therefore, variation caused by other species not being considered/modelled has the potential to alter the relative abundance of the species being considered. In other words, if you take the relative abundance of *G. ruber*, *G. sacculifer* and *N. dutertrei* and sum them to 1 that would not get rid of the closed sum problem, because it fails to consider other species which are not being modelled. In fact, you would not only magnify the counting error, but you would also be basing your data on a small percentage of the total foraminifera flux. Additionally, as clearly stated in Roche et al., 2018, the FAME model is constructed so as to produce a $\delta^{18}O_c$ value where the given species of foraminifera is assumed to be able to grow: "It should be clearly understood that this approach is not able to and does not attempt to determine the relative abundances of the different species. Instead FAME provides a simplified approach to compute the $\delta^{18}Oc$ of a generic population of foraminifers if environmental conditions permit its growth. From a model–data perspective, this approach enables one to compute the calcite $\delta^{18}O$ for a given species, were it to exist in the sedimentary record ". A further useful reference in this instance is the actual formulation of the model in equation 8, page 3590 of Roche et al. (2018).

> *Page 10 line 28 – The discussion of model limitations does not ask what would seem to be the most important questions: Does the modeled growth rate actually reflect the real ocean (and the sampling bias for what is recorded in sediments)?*

As the reviewer states in this question, there is a sampling bias within sediments hence the selection of a forward model and why in this instance it is more logical than an inverse model.

> *Do the modeled growth-rate weighted d18O distributions match actual measured individual foraminifera d18O distributions (such as in Koutavas and Joanides or Rustic)? If no growth-rate weighting is applied are the results better or worse?*

As discussed earlier, FAME is not attempting to produce the measured IF distributions from natural archives.

> *Clearly separate the role that the growth model and (T,S) timeseries play in identifying ENSO change.*

*To what degree are the outcomes and conclusions of this paper depending upon the modeled growth rates versus the sea water properties (T,S,d18Ow)? Many prior workers have analyzed in different ways the reconstruction of ENSO from IF analysis. These approaches include summary statistics like the standard deviation (Thirumalai; Koutavas; Leduc; Sadekov; Rustic), as well as examination of changes in the quantiles of IF distributions (Ford; White). What is added here is the foraminifera growth rate weighting. What effect does this have? From the histograms in Roche et al. (2018) it appears that the growth-rate weighting does not have major consequences for the mean d18Oc value of a sediment sample. It may have consequences for the IF variability though. The authors could show a map that quantifies the growth-rate weighting effect with respect to the non-weighted results (ratio, difference).*

Thank you for the comment. Figure 4 and supplementary figure S3 are already aiming at this, and we will endeavour to make this clearer. We will expand upon the section that is already included in the paper: **"The model-driven results were assessed with the underlying observational dataset**, to check **how the dataset alters with FAME** the input data (temperature and δ18Oeq) under**went** statistical test**ing** (Figure 4 and Figure S3). Instead of a variable depth, we opted for fixed depths at 5, 149 and 235 m, giving a Eulerian view (Zhu et al., 2017a) in which to observe the implications of a dynamic depth habitat. By using a fixed depth, these results show that the shallowest depths produce populations that are significantly different both in terms of their mean values and their PDF. In the upper panel of Figure 4, the canonical El Niño 3.4 region is clearly visible at 5 m depth. Whilst differences exist between the temperature (Figure 4) and the FAME Anderson Darling results (Figure 3), for instance close to the Panama isthmus, there are significant similarities between the plots. These plots also show that our FAME data (Figure 3), in which we allow foraminiferal growth down deeper than the depths in Roche et al. (2018), are a conservative estimate and thus are on the low-end (Figure 4), to account for potential discrepancies with depth habitats."

We do not fully understand the following comment of the reviewer:

*"Page 8 line 1 – This paragraph is rather confusing to understand. It sounds like the authors are comparing a depth-weighted reconstruction and non-depth weighted reconstructions at fixed depths (Fig. 3 vs. 4)?".*

In the way FAME is set up, the weighting for depth is based upon growth, without FAME we would be unable to integrate the required weighting function. Hence why it necessitated fixing the depth for the analysis of the input temperature and $\delta^{18}O_{eq}$ values.

**Validation of $\delta^{18}O_C$**

*["Validate the $\delta^{18}O_c$ predictions from the growth rate and geochemistry model. This was done in Roche et al., 2017 but is also somewhat circular because the sedimentary $\delta^{18}O$ values were used to determine the depth of production. I admit I am not sure how to actually validate the approach except from an additional validation dataset not used for determining production depth.*

*Page 5 line 10 – Why was growth rate arbitrarily constrained to these different depths? First, foraminifera with algal symbionts should be in the photic zone. Second, didn't the Roche et al., 2018 paper try to identify the depth-production relationship for the different species from the predicted $\delta^{18}O_c$ and measured MARGO $\delta^{18}O_c$? Why not use those depths?"]*

Thank you for your comment. It is important to stress that there is no geochemistry model in our approach. The optimisation procedure of Roche et al. (2018), gives the maximum allowed growth depth of each species. However, as the reviewer discussed previously, we should test how this influences the resultant distributions we generate, therefore we constrained the model to four depths including to a depth known to be below the photic zone. This is what is stated in page 4 line 7-19; page 5 line 10. We will rephrase this with clarity in mind.

**Dismissive of Mg/Ca-Temperature?**

*[Include the analysis of Tc for Mg/Ca reconstructions. Inexplicably the authors refuse to analyze the temperature distributions even though those are the data from the common Mg/Ca method of individual foraminifera analysis (Sadekov, Ford, White). The author's stated reason is due to "...the complexity in reconstructions of trace metal geochemistry...and the potential error associated with determining which carbonate phase is first used when foraminifera biomineralise...". While there are ongoing methodological and calibration efforts for this and other proxies (including d18Oc), to ignore such a widespread type of analysis seems very shortsighted. If the authors do not want to forward model the Mg/Ca proxy itself they can simply analyse the temperatures in their dataset. Either way this is something that should be included in the manuscript.*

*Page 9 line 20 – Why discount trace metal temperatures (Mg/Ca)? Include the analysis of Tc for Mg/Ca reconstructions.*

*Page 11 line 14 – Why are the authors so dismissive of Mg/Ca analyses? The list of possible complications is important but it remains a fundamental observation that the Mg/Ca of foraminiferal calcite changes with growth temperature and has been validated in many different ways.]*

The reviewer is alluding to Figure 6, the Tc or calcite/recorded temperature, which is essentially our pseudo-Mg/Ca* produced with FAME (* = It is a weighting of temperature rather than $\delta^{18}O_{eq}$). This is discussed in the dataset, we also ran the temperature (Figure 4) of the dataset by itself (without the foraminiferal growth rates). It is true that we don't go into too much detail and we will expand our section discussing the FAME produced temperature (Tc).

However, it is not as the reviewer states as us being 'so dismissive of Mg/Ca analyses' (we are sorry if we created such an impression). A great many researchers are dedicating their time to this valuable geochemical analysis. However, given that the species-specific conversion from temperature to Mg/Ca is not as straight-forward as $\delta^{18}O_c$ and $\delta^{18}O_{eq}$, it would therefore require more parameters, which we are not confident in modelling at this stage. A pseudo Mg/Ca would also need to be validated (yet techniques are not standardised nor cross calibrated to a sufficient degree, with users using laser ablation; pooled specimens and/or whole shell) and the problems associated with dissolution, cleaning for analysis, are not easily parameterised in a model. We do welcome discussions on the computation of pseudo Mg/Ca and consider it something that could be included in the future, possibly in a second generation of the FAME model.

**Removal of maps**

*[Remove maps of carbonate preservation/depth. It is fine for the authors to state the general problem in the text, but there are regions of shallow depth were carbonate is preserved that are not captured in the coarse DEM used; Remove map of sedimentation rate. Either quantitatively discuss the role of sedimentation rate and bioturbation or remove this map. The sedimentation rate threshold is intimately tied to the secular and nonENSO variability and thus is a much more complicated analysis than the general discussion in the text. I think the discussion is a starting point but the author's miss that the important factors are really the magnitude of other, non-ENSO sources of variability at the timescale of a sediment sample (plus bioturbation) compared to the magnitude of the ENSO change signal and the non-ENSO variability.]*

As stated in the paper: - "The resolution of the ocean reanalysis data for the time period 1958-2015 would essentially be analogous with a sediment core representing 50 yr-1 cm-1 (or 20 cm-1 kyr-1). Based on our analysis, such a hypothetical core with a rapid sediment accumulation rate (SAR) could allow for the possible disentanglement of El Niño related signals from the climatic signal using IFA, but only in a best-case scenario involving minimal/no bioturbation, which is unlikely in the case of oxygenated sediments". This is an important caveat to communicate to the reader. We believe that removing the maps would remove this valuable piece of information.

**Definition of ENSO components**

*["Definition of El Niño, neutral, and La Niña months there is a large body of literature and accepted methods for defining El Niño, neutral, and La Niña periods. In the text the authors take a simplistic approach, but there is no reason for this. Why not actually use the societal*

*and dynamically important definitions of these events including the requirement of a minimum consecutive number of months of anomalies and changing baseline for anomalies (to account for secular warming of the ocean)? This definition has a basis in theory as an El Nino (La Nina) event unfolds over a length of time and thus a single month anomaly may not be associated with the dynamics that are part of the coupled ENSO system."]*

What we said: "The tropical Pacific Ocean is divided into four Niño regions based on historical ship tracks, from east to west: Niño 1 and 2 (0° to -10°S, 90°W to 80°W), Niño 3 (5°N to -5°S, 150°W to 90°W), Niño 3.4 (5°N to -5°S, 170°W to 120°W) and Niño 4 (5°N to -5°S, 160°E to 150°W). One index for ENSO, the Oceanic Niño Index (ONI), based upon the Niño 3.4 region (because of the region's importance for interactions between ocean and atmosphere) is a 3-month running mean of SST anomalies in ERSST.v5 (Huang et al., 2017). However, Pan-Pacific meteorological agencies differ in their definition (An and Bong, 2016, 2018) of an El Niño, with each country's definition reflecting socio-economic factors, therefore, for simplicity we utilise a threshold of $\chi \geq +0.5°C$ as a proxy for El Niño, $-0.5°C \leq \chi \geq +0.5°C$ for neutral climate conditions and $-0.5°C \leq \chi$ for a La Niña in the Oceanic Niño Index. Many meteorological agencies consider that five consecutive months of $\chi \geq +0.5°C$ must occur for the classification of an El Niño event. However, here it is considered that any single month falling within our threshold values as representative of El Niño, neutral or La Niña conditions (grey bars in Figure 1). By using this threshold, three weighted histograms for each $\delta18Oc$ and Tc and their resultant distributions (El Niño; Neutral; and La Niña) were computed for every month and for every latitude and longitude grid-point for the 1958-2015 period."

Why did we do this? Because if a foraminifer lives for 30 days then how appropriate would *"including the requirement of a minimum consecutive number of months of anomalies and changing baseline for anomalies (to account for secular warming of the ocean)"* be? Because sediments can't resolve annual/sub-annual resolution (like corals or molluscs), therefore the periods where the threshold passes 0.5 would in the sediment be mixed with with El Nino or La Nina (as in they would have potentially similar values as an El Nino, and as time cannot be resolved they would be considered as El Nino). In the sediment the minimum consecutive months is not used to define an El Nino, as it is impossible, an arbitrary value or any kind of quantile- or sigma distribution is.

**Next paper.**

*["Examine how ENSO amplitude vs. frequency change IF distributions. The authors raise an interesting point in their conclusion that has not been well addressed, namely how do changes in the statistics of ENSO (frequency, amplitude) affect IF distributions and reconstructions of ENSO variability. Evaluating these two different questions would be an important contribution to IF analysis of ENSO change. But, introducing the idea in the conclusions without a previous discussion in the manuscript is not a good idea in my opinion."]*

We are not attempting, at this stage, to specifically reproduce single foraminifera analysis. How ENSO amplitude and frequency impact foraminiferal distributions is a separate paper we are working upon, because it actually cannot be dealt within as a simple discussion topic (and would require a specific and different dataset from the current paper's dataset), it is something that we thought about as we worked on this manuscript. Therefore, we suggested this approach in our conclusions / perspectives as something that could be worked on in the future, which is something that we believe is normal in scientific manuscripts. We will attempt to make clearer that we are referring to possible future work.

**Specific comments**

*Page 1 line 17 – "Furthermore, a large proportion of these areas coincide with sea-floor regions exhibiting a low sedimentation rate and/or water depth below the carbonate compensation depth, thus precluding the extraction of a temporally valid palaeoclimate signal using long-standing palaeoceanographic methods." The role of sedimentation rate in IF analysis is important but there is not any investigation of this effect in the present manuscript so it is not really a conclusion or finding. This statement should not be included*

*in the paper in its present form; Page 1 line 17 – "Furthermore, a large proportion of these areas coincide with sea-floor regions exhibiting a low sedimentation rate and/or water depth below the carbonate compensation depth, thus precluding the extraction of a temporally valid palaeoclimate signal using long-standing palaeoceanographic methods." The role of water depth and carbonate preservation is also important. But, there is not any investigation of the sedimentation rate effect in the present manuscript so it is not really a conclusion or finding. Furthermore, there are seamounts and other shallow sites not captured in the gridded dataset that can contain records for palaeoceanographic investigations. This statement should not be included in the paper in its present form*

We disagree, we believe that the inclusion of SAR and water depth (CCD) adds important context to our paper. Those are the two main factors that allow for the carbonate signal to be preserved in sufficient temporal resolution. We can consider adding the location of sea mounts to the map, thank you for this idea.

.

*Page 4 line 1 – The new model for foraminifera growth only uses the temperature component of the previous model. Why? How different are the results?*

The 'why' have been dealt within in Roche et al. (2018). The 'how different' is comparing apples and oranges, FAME requires temperature as an input whereas FORAMCLIM needs temperature, light, and organic carbon (food). Light and food are not included in many datasets, nor are they parameterised or have proxies. A validation step of the two different models using the same observational input data is thus not simply attainable. The input data here does not have either of these additional variables (we considered for example using a long term chlorophyll record from satellite data, but such datasets ignore the deep chlorophyll maximum).

*Page 4 line 15 – Allowing symbiont-bearing foraminifera to possibly grow to 400 m simply based upon optimal temperatures seems not correct. They need to be in the photic zone.*

Four different depths (60; 100; 200 and 400 m) have been used in the model, the use of the shallow and deeper depths likely don't capture one or more of the species actual ecologies, however that is why we ran it with different depths to understand how chosen depth alters (or doesn't alter) the results.

What we said: - "Consequently, we allow all the species of foraminifera to grow down to ~ 400 m (depending if optimal temperature conditions are met) to capture the total theoretical niche width. As the optimised depths of Roche et al. (2018) are shallower, and upper ocean water is more prone to temperature variability, our approach likely dampens both the modelled $\delta 18 O_c$ and $T_c$. Therefore, the sensitivity of the model was tested by applying the same procedure but with the limitation of the depth set to 60; 100 and 200 m."

**–Methods–**

*Page 5 line 5 – The conversion of VSMOW to VPDB looks to be in error. The correct formula for this conversion is [d18O_VSMOW+1]/[d18O_VPDB+1] = 1.03091 where d18O does not include the 10^3 term. Thus d18O_VPDB = d18O_VSMOW/1.03091 +(1/1.03091)-1 or d18O_VPDB = 0.97002*d18O_VSMOW 0.02998. In d18O expressed with the 10^3 term, the equation would read: d18O_VPDB = 0.97002*d18O_VSMOW - 29.98.*

The reviewer is referring to the fractionation difference between water and carbonate for which the given expressions are indeed correct. However, what is referred to in the manuscript is the conversion between two scales with measured $\delta^{18}O_{sw}$ ; those can be converted from V-SMOW to V-PDB by – 0.27 ‰ (please see Figure 1 in Hut, 1987 for the original reference).

*Page 6 line 1 – "...these for now can be ignored." Why can the other factors determining foraminifera growth be ignored? This cannot be a statement unless it is backed up. Or, the authors use only temperature but then go through an appropriate validation process (more than what is shown in Roche et al., 2018) as suggested above.*

We agree with the reviewer that our initial sentence was somewhat ill-formulated. What we meant here is that the major driver of foraminiferal growth is temperature and hence taking it (temperature) into account will provide the first order signal, as was discussed already in Roche et al. (2018). A revised version of the manuscript will include a modified statement as follow: "these variables for now can be set aside as temperature provides the dominant signal, it is worth noting that in all probability some variance will arise from these processes and deviation between observed and expected values should consider this." Regarding the validation process we refer the reviewer to the discussion already given above.

> *Page 6 line 11 – Starting here, it is very unclear how and why the particular set of conditions for El Niño, La Niña, and neutral periods were chosen. What time series of sea surface temperatures were chosen for computing anomalies (in each grid square, Nino 3.4, Nino 3, Nino 4, etc.)? Were the anomalies based upon a 3-month running mean? Were the anomalies computed relative to a fixed period or, as is now the accepted approach, relative to 5-year interval means? Why not use the definition of El Nino etc. events that include the requirement for consecutive months of anomalies? This definition has a basis in theory as an El Nino (La Nina) event unfolds over a length of time and thus a single month anomaly may not be associated with the dynamics that are part of the coupled ENSO system.*

Pg. 6, Line's 8 to 10. The ONI dataset was used, this is based upon the 3 month smoothed anomaly in the El Nino 3.4 region, we decided to set thresholds including anything above 0.5 and below -0.5 in their respective EL Nino and La Nina bins because unlike, e.g. corals, a palaeoceanographer using sediment core foraminifera cannot discern a specific year. An 'almost El Nino' anomaly won't be discernible from a full El Nino period in the fossil record, because unlike coral records we cannot determine what the previous 3 months were like.

> *Page 6 line 18 – Why and how was the pdf/cdf from the actual data fitted and smoothed with an Epanechnikov kernel? What impact did this fitting and smoothing (particularly the choice of bandwidth) have on the Anderson-Darling test and the results overall?*

The data was fitted using a fit distribution procedure in MatLab because the statistical function requires a distribution to test. We chose to use the kernel distribution because it mimics the underlying dataset well and we were testing a large number of grid points, therefore we decided to keep numerous parameters constant (for instance we could have decided to change the distribution using a find the best fit distribution but this would have made intercomparison problematic), however to allow our fitted distribution to better mimic the underlying distribution we allowed the programme to vary the bandwidth between grid points for an optimal kernel distribution.

> *Page 6 line 24 – This paragraph is very unclear and the errors associated with binning prior to analysis of the pdf seem avoidable. For example, why not take the growth rate in each of the 696 months in each grid at each depth, and scale the growth rate to calculate an effective # of individuals such that they sum to 1000 across all months? Round those numbers to integers and then use the integer # of individuals for each month to replicate that actual months Tc or d18Oc value. The resulting ordered list of values can then be binned/smoothed etc. and represents a pseudo-distribution that one might find in a sediment sample?*

The reviewer is correct that it would solve the minor binning error – but it wouldn't solve the rounding error, if you round these numbers….

What we wrote:- "As the weighted distributions are effectively probability distributions, in order to fit a distribution, we multiplied the bin counts by 1000, effectively converting probability into a hypothesised distribution. Using the repeat matrix function (MatLab function: repmat), a matrix of $\delta 18Oc$ was produced using each bin's mid-point ($\delta 18Omid$-point) there is a threefold error combined with this methodology which may account for minor variation between discrete runs of the model: first the counts values were rounded to whole integers so an exact number of cells could be added to a matrix; secondly the $\delta 18Omid$-point was used which gives an error associated with

the bin size (±0.05 ‰) that is symmetrical close to the distributions measures of central tendency but asymmetrical at the sides; and finally, the associated rounding error at the bin edges within a histogram (±0.005 ‰)."

**–Results–**

*Page 7 line 3 – It says that the mean d18Oc for El Nino and neutral months are compared. How? Earlier and later it is stated that the A-D test is applied to compare distributions. What is meant by these lines?*

We will reword this sentence for clarity.

*Page 7 line 5 – "...ENSO events can potentially be detected by paleoceanographers and unmixed using, for example, a simple mixing algorithm with individual foraminiferal analysis..." This is not really practicable because it assumes complete stationarity in the El Nino, La Nina, and neutral distribution. This is unlikely as all are expected to change, and do in models and data (e.g. coral time series from middle Holocene show changed seasonal amplitude and ENSO cycles).*

Here we are discussing an unmixing analysis for a single time slice, if enough foraminifera are measured then it can be possible to disentangle mathematically the various components that go into a single distribution. However, this is only possible if the values of El Nino, La Nina etc. have a different absolute $\delta^{18}O$ value, our point here. This is true regardless of an unmixing analysis or and holds true for any proxy.

*Page 7 line 7 –"In cases where FPEN and FPNEU do not exhibit significantly different means, then the chosen species and/or location represent a poor choice to study ENSO dynamics." This may not always be the case because the mean values could be similar but the distributions wildly different (such as long tails with different signs). Changing numbers of El Nino and neutral and La Nina events could that quite dramatically change the shape of the combined distribution that is ultimately preserved in sediments. And, it may be possible to find regions of such a distribution that can be used to diagnose changing ENSO.*

*Page 7 line 20 – Why is Anderson-Darling test done here but the mean values are discussed above? If the A-D test shows that the El Nino and Neutral distributions are different (at some statistical level) then that means alteration of those distributions (more/fewer, stronger/weaker events) would alter the summed distributions that one gets from a sediment sample. But, how would this actually be detected in the sediment sample? That the AD test demonstrates the El Nino, Neutral, and/or La Nina distributions are different is helpful but it does not get at whether ENSO change could actually be detected in a sediment sample.*

True, that is why we tested the distributions as well. Here we are discussing the fact that similar values would be impossible to unravel – we will make this clear. This *("Changing numbers of El Nino and neutral and La Nina events could that quite dramatically change the shape of the combined distribution that is ultimately preserved in sediments. And, it may be possible to find regions of such a distribution that can be used to diagnose changing ENSO")* is why we chose to use a statistical test that looks into the distribution.

*Page 7 line 26 – Applying a 1-sigma value from modeled minus coretop comparisons to the AD test value does not seem appropriate. This value assesses the accuracy of the model in predicting the absolute value of the mean of a coretop sample. But it is not an appropriate estimate for the significance of the difference between two different IF values or the difference in the AD statistic.*

We disagree, if the model has some measurable error, it is appropriate to advertise the fact to readers that at some locations the distributions whilst significantly different with one test fall within the model 'error'. Hence the use of hashing.

*Page 8 line 9 – Unclear what "on the low-end" means.*

We will clarify this.

*Page 8 line 16 – "...a large percentage of the tropical Pacific remains accessible to palaeoclimate studies." This is very much not the message in the Abstract and from the title of the paper. Those sections should reflect this finding.*

First the title is "On the validity of foraminifera-based ENSO reconstructions" is ambiguous as to whether they are or are not valid. Second the abstract is referring to the entire discussion, the calculated distributions and the SAR/depth. We will clarify this statement.

*Page 8 line 25 – "Indeed, one should view discrete sediment intervals, and the foraminifera contained within them, as representative of an integrated multi-decadal or even multi-centennial signal..." This is exactly how foraminifera paleo-IF studies have viewed them and should be stated up front (start of paragraph for instance).*

Yes, we will make this clearer. We note that we cite our individual foram paper (Lougheed et al, 2018) using single shell $^{14}$C to show that a single cm can be not just multi-centennial, but multi-millennial.

*Page 8 line 28 – "Therefore, in order to reliably extract short-term environmental information from foraminiferal-based proxies, the signal that one is testing or aiming to recover must exhibit a large enough amplitude in order to perturb the population by a significant degree from the background signal, otherwise it will be lost due to the smoothing effect of bioturbation..." This statement does not make sense to me. The background signal IS the signal, i.e. the seasonal cycle, ENSO etc. Changes in ENSO must be such that they alter that signal (the distribution of IF analyses), but bioturbation etc. should not erase the signal unless one is looking for short periods of change less than the time integrated into the sample.*

The reviewer has stated what we are stating in that sentence, we will rephrase for clarity. However, ENSO is not the background signal otherwise it would not be detectable through a temperature anomaly. It is a short term climatic event when one considers that a single cm in ocean sediments can reflect hundred to thousands of years.

*Page 9 line 6 – "...a series of high magnitude, but low frequency El Niño events could be smoothed out of the downcore, discrete-depth record." They will not be smoothed out as the authors state. Those anomalous IF values may be rare, but will be present in the sediment sample and if measured can be used to examine changing ENSO.*

Thank you for pointing this sentence out. We will clarify this sentence to explain in better detail what we mean. Firstly, the absolute magnitude of events would obviously be smoothed out if one were to be apply discrete, multi-specimen sample downcore analysis (i.e. not single foram analysis), as the reviewer is obviously aware of. Were single foram analysis to be applied, the single foram values corresponding to high-magnitude ENSO events would indeed still be present in the sediment record, as the reviewer correctly points out. However, single foraminifera from multiple ENSO events, and non-ENSO climate, would all be mixed into the same discrete interval, meaning that a time-series of ENSO is essentially not possible to produce, and therefore: (1) the frequency of ENSO events becomes difficult to detect, and (2) that one is forced to make *a priori* assumptions regarding the behaviour of background climate and ENSO climate in past times in order to differentiate between ENSO and non-ENSO single foraminifera in the palaeo record.

*Page 9 line 7 – The sediment accumulation rate needed to observe/reconstruct changes in ENSO is not fixed. It depends upon the magnitude and duration of secular trends, and variability with respect to both the time integrated in a sediment sample and the magnitude of the ENSO signal and its change. This is a quite interesting but also complicated subject and arbitrarily cutting the sedimentation rate at 5 cm/ky is not justified.*

We agree with the reviewer it is not fixed, the '*sediment accumulation rate needed to observe/reconstruct changes*' ideally would reflect the percentage of foraminifera within the sediment growing during ENSO events and the magnitude of the events not just the number of events. We furthermore note that, to avoid using a SAR cut off that could be considered arbitrary, we intentionally used a very generous cut-off of 5 cm/ka. Were we to set the cut-off to be higher, following the more traditional

lower cut-off of 10 cm/ka (Bard et al, Shackleton et al), then the areas of the Pacific basin that could be considered suitable would be even more limited.

> *Page 9 line 9 – The map of water depth is quite coarse and misses important locations that are above the CCD, accumulate carbonate (and foraminifera), and can be used for palaeoceanographic reconstructions. Thus, while the overall point is true, the map as shown is misleading.*

We used the latest GEBCO, but we would be more than happy to include higher resolution data.. However, we are obviously not saying a seamount would not be useable. We can consider adding sea mounts to the map.

> *Page 10 line 3 – The references to Cole and Tudhope, 2017; White et al., 2018 seem to be in error. These papers do not discuss lake core colour etc.*

We will rephrase this sentence. Here we referring to the interpretation, for instance figure 19.3 of Cole and Tudhope (2017)

> *Page 10 line 3 – "If the number and magnitude of ENSO events were reduced, the relatively low downcore resolution of marine records may not accurately capture the dynamics of such lower amplitude ENSO events using existing methods." This statement is not justified by the author's analysis or a citation.*

We will add a citation(s).

> *Page 10 line 5 – "The possibility of a marine sediment archive being able to reconstruct ENSO dynamics comes down to several fundamentals: the time-period captured by the sediment intervals (a combination of SAR and bioturbation), the frequency and intensity of ENSO events, as well as the foraminiferal abundance during ENSO and non-ENSO conditions." Also included is the magnitude of change in ENSO statistics and resulting foramifera Tc or d18Oc, sampling uncertainty on the IF distribution. See also note above on the role of sedimentation rate.*

At the reviewer's suggestion we will add in 'sampling-bias' into the sentence

> *Page 10 line 9 – "The results presented here imply that much of the Pacific Ocean is not suitable for reconstructing ENSO studies using palaeoceanography, yet several studies have exposed shifts within σ(d18Oc) of surface and thermocline dwelling foraminifera. One can, therefore, question what is being reconstructed in such studies." The results presented here don't really test whether individual foraminifera d18Oc (or Tc) studies can reconstruct ENSO. Furthermore, the water depth and sedimentation rate constraints are the reason for excluding much of the Pacific. This statement is therefore incorrect and the search for other explanations does not follow.*

This sentence may have led the reviewer to misinterpret our results as sediment-based individual foraminiferal analysis centric, we will rephrase this sentence for clarity and suitability.

> *Page 10 line 19 – This second part of the paragraph is interesting and has been commented on before. But, at no point do the authors actually evaluate any of these effects or approaches so they can't really assess the different factors they raise here.*

Here we are discussing other's findings, for instance, Zhu computed the variance and found that some of the signals detected could be a by-product of the annual cycle.

**–Conclusion–**

> *Page 12 line 17 – "Previous work..." The only citation here is to Zhu et al., 2017. There has been a lot of work comparing IFA different time slices (both d18Oc and Mc/Ca) that should be cited here (Koutavas et al., Leduc, Koutavas and Joanides, Sadekov et al,Ford et al, Rustic et al, White et al). Furthermore, they have not all used 1-sigma d18Oc as the metric for detecting change.*

The reviewer is right, "*they have not all used 1-sigma d18Oc as the metric for detecting change*", that is why they are not cited. The reviewer would be justified in suggesting these references here, had we not repeatedly cited them throughout our paper. Whilst we will attempt to make this clearer for the reader, it is worth noting that a few sentences later, we directly refer to the papers the reviewers cites (see comment below:) .

We will add clarity to this statement, however we would like to note that this quote neglects the second part, the first word overall here being 'generally speaking' eludes to the fact that it's a sentence that has a follow up: "Overall, our results suggest that foraminiferal $δ_{18}O$ for a large part of the Pacific Ocean can be used to reconstruct ENSO, **especially if an individual foraminiferal analysis (Lougheed et al., 2018; Wit et al., 2013) approach is used** (Ford et al., 2015; Koutavas et al., 2006; Koutavas and Joanides, 2012; Koutavas and Lynch-Stieglitz, 2003; Sadekov et al., 2013; White et al., 2018), contrary to previous analysis (Thirumalai et al., 2013). **However**, the sedimentation rate of ocean sediments in the region is notoriously slow (Olson et al., 2016) and much of the ocean floor is under the CCD. **These factors reduce the size of the area available for reconstructions considerably (Lougheed et al., 2018), thus precluding the extraction of a temporally valid palaeoclimate signal using long-standing methods.**"

As the reviewer states our statement is 'generally true', therefore the difference is our interpretation vs. the reviewer's interpretation, we politely disagree.

Conclusions do not have to include the main focus of the study but can include information that present the findings in a different light (as in what it means to the readers) or what can be next done. We don't agree with the reviewer's suggestion that it is an untested model, but also doubting that Ecophysiological models are not limited to foraminifera – this is a factual statement and they can be of use.

**–Figures–**

Where we have used FAME they are growth rate weighted values – this is explained in the methodology and Roche et al. (2018). Figure 4 and supplementary figure 3 use the input data therefore they are non growth rate weighted.

*Figure 3 – Why are there white and grey areas that mean the same thing?*

As the key shows they represent where the populations are statistically different, the hashing draws the eye too much so for those panels with hashes we make it grey. As one of our species does not have these hashes it remains white.

*Figure 4 – Are the temperature data growth weighted? What species? If not, why not analyze the Tc data in parallel to the d18Oc data to evaluate what advantage/disadvantage the two different signals have (e.g. from S).*

What the caption says: - "Figure 4. Results of an Anderson-Darling test between El Nino and Neutral climate conditions based upon the Temperature input data: Fixed depth." This is the temperature input data

*Figure 5 – Why are the white and grey areas grouped together? What do they mean? Are these panels based upon growth-rate weighted values?*

As the key shows they represent where the populations are statistically different, the hashing draws the eye too much so for those panels with hashes we make it grey. As one of our species does not have these hashes it remains white.

*Figure 6 – Are these panels based upon growth-rate weighted values?*

Where we have used FAME they are growth rate weighted values

**B. Metcalfe on behalf of the co-authors, Response to Reviewer 2**

*[reviewer comments as red text in blocks]*

We thank Reviewer 2 for their time in submitting this review of our work, before we address the main and minor comments we like to (re)address a point raised by Reviewer 1 and expand upon application of our study.

> In their manuscript Metcalfe et al. present a forward modeling approach through FAME to investigate the use of individual foraminifera analysis (IFA) for ENSO reconstruction. Based on the modeling results, they conclude that this proxy is only valid in part of the Pacific Ocean. However, these regions are often characterized by low sedimentation rate, therefore limiting the use of this proxy.

The reviewer correctly identifies that we have carried out a forward model of foraminifera $\delta^{18}$O. Specifically, we used the FAME model, driven by observed climate data, to predict foraminifera population $\delta^{18}$O accounting for their habitat water depths and growth season, to test whether foraminifera populations in the water can record the ENSO signal. We are not modelling IFA in sediment cores. We carry out a minor discussion of sediment core IFA, because, clearly, whether or not ENSO dynamics can be recorded by foraminifera populations in the water itself has consequences for sediment core records (which obviously source their foraminifiera tests from the water column).

> While the effort to incorporate forward models into paleoceanographic studies is commendable, I fail to see the practical application of this study.

The reviewer might prefer if our study was more similar to an inverse model approach, however, the manuscript that we submitted is a forward model. We regret if we have not elucidated enough in our paper the practical applications of our forward model. We see several applications:

**Research question / proxy validation** – Pg. 12 ln 30-32 "The use of ecophysiological models (Kageyama et al., 2013; Lombard et al., 2009, 2011) are not limited to foraminifera and provide an important way to test whether proxies used for palaeoclimate reconstructions are suitable for the given research question." One application can be validating the research question, i.e., whether foraminifera in the water will in the first place (even before they become fossilised, disturbed, winnowed away) record the signal we are looking (e.g., similar to Leduc et al., 2009; Ford et al., 2015). A pre-screening using the available information (temperature, salinity, etc.) and our understanding of certain processes can be used to focus our research, determine species selection. Which leads on to…

**Site selection** – In an ideal world the location of (palaeo-) data would be at least one datapoint every 1x1 degree latitude longitude (or a higher resolution) so that a direct comparison between model and data could be obtained. However, this is neither practical nor feasible in the real world where access to time and funding (etc.) is limited. The time and effort (from personal experience) that goes into sampling, washing, picking, measuring (and dependent upon pooled or individual, replicates or no replicates, etc) foraminifera is a lot. Forward models can therefore be used to select sites not on geographic proximity to existing published records but based on their strategic location, which can provide critical information about certain climatological processes for data-model comparison. Naturally, a bioturbation module would further elucidate such an approach. There are three models/analytical tools that exist that deal with picking and signal modulation: Sedproxy; FIRM or INFAUNAL. Therefore, we focussed upon the construction of a signal, which leads on to….

**Sandbox modelling -** Pg. 12 ln 26-27: "We further highlight that the conclusions drawn from foraminiferal reconstructions should consider both the frequency and magnitude of El Niño events during the corresponding sediment time interval (with full error) to fully understand whether or not a strengthening or dampening" occurred. FAME is intended for climate models where boundary conditions can be varied.

> Inverse modeling would be impossible and the lack of comparison between the pseudo-proxy distributions and actual distributions of foraminifera prevents validation of the method.
>
> ### Major comments
>
> ### Inverse problem
>
> The manuscript focuses on forward modeling of IFA analysis. Although definitely a valuable exercise for data-model comparison (assuming that the climate model can make use of the forward model), it doesn't solve the inverse problem. It would be almost impossible to evaluate the growth factor in the d18O record.

Our commentary regarding the inverse problem is available with respect to Reviewer 1, we will surmise our key points here: (i) It is not suitable/applicable to our research question; (ii) a lack of a large scale dataset to perform a basin-wide analysis; and (iii) lack of a bijective relationship between $\delta^{18}O_c$ and the oceanic variables (T, $\delta^{18}O_{sw}$).

"(assuming that the climate model can make use of the forward model)" - FAME is built with climate models in mind. Data-model comparison studies suffer from an 'apples and oranges' problem, of which there are two key problems: (i) the conversion of units i.e., most proxies reconstructing temperature do not give values of temperature in degrees C or K but in their own proxy units (per mil, mmol/mol, etc) requiring a conversion, and (ii) a reduction in scales, i.e., models give a wealth of information (multiple layers) in the time-depth domain. FAME was produced (Roche et al., 2018) to (i) to generate pseudo-proxy time-series from model runs that can be compared with age-depth values down cores. Naturally, including a bioturbation, or mixing, module into down-core work is prudent for core datasets (please see comment below; and to (ii) reduce the information for a given time-slice into a manageable value using an integration that would make sense on a biological point of view (integrating various depths with equal weightings might seem logical, but foraminifera for instance grow at different rates depending on their temperature) and).

It's also not visually obvious what the difference between the output of a non-weighted model is vs FAME in Figure S1. Some statistics would help, or plotting the resulting kernel distributions on a separate panel,

We will elaborate further on these plots to make them more understandable, Figure S1 are produced solely by FAME, the difference is one is weighted for a larger proportion of growth per month and the other bins the total number of months, these plots were picked at random from the dataset.

Further, bioturbation is also likely to have a large impact on IFA, especially in areas of low sediment accumulation. Why not connect FAME to a bioturbation model and disentangle the influence of these factors?

There are three points we would like to address with this comment, the first is the reviewers admission that IFA (and by association pooled) distributions can be impacted by bioturbation. The second point is that it asks the follow-up question to our own research question, i.e. "Are the populations between different climatological end members significantly different, **and can this difference be resolved in the sediment record**", this second question would need different input parameters, as discussed next. The third point we would like to address is that our input dataset is ~60 years, in the grand scheme of things a relatively small contribution to the sediment. We choose not to link the dataset to a bioturbation module because the time series is not long enough, a 1cm kyr$^{-1}$ SAR with a 10 cm mixed layer depth would need several thousand years of input data (please see comment further below). Furthermore, the application of bioturbation to a monthly time-series can be done in several ways, for instance should one could bioturbate the 'settling flux', i.e. the monthly signal, as soon as it is encoded or at the end of the time-series.

**Statistical analysis**

Page 6, Line 25: Multiplying the bin counts will effectively skewed the results of a significance test. In practice, it would be impractical if not impossible to obtain 1000 samples in each bin.

We will reword this section as whilst we state 'total bin counts' we add a clarification: - "As the weighted distributions are effectively probability distributions, in order to fit a distribution, we multiplied the total bin counts by 1000 **(i.e. so that the total sum is 1000).**

Similarly, page 7, line 4, how many foraminifera were artificial picked to produce these maps?

We did not artificially pick foraminifera (here we are referring to population rather than a simulated sample). We are testing the water populations that would be El Nino or La Nina or Neutral. As we are not modelling IFA in sediment cores, we therefore also do not include a picking routine.

**IFA model - data comparison**

There are a number of recent studies with IFA results from the past 1000 years (some of them cited in the current manuscript). How do these distributions compare to the statistical ones?

We will address this in a revised form of the MS.

**Effect of SAR**

Since a model of bioturbation was not implemented here, it's hard to examine the effect of bioturbation on the IFA.

We have indeed not specifically modelled the effect of bioturbation upon our data, because it is simply not possible to carry out a transient bioturbation model run upon 60 years data. Therefore, to avoid appearing biased against sediment archives in our SAR map, we intentionally used a very generously low SAR cutoff of 5 cm/ka. However, it is not necessary to run a transient bioturbation model to demonstrate the limitations of a SAR that is less than 5 cm/ka. We can carry out simple calculation following established understanding of the influence of bioturbation upon the age dispersal of single foraminifera (Berger and Heath, 1968; Berger and Johnson, 1978; Berger and Killingley, 1982), the same understanding that is included in transient bioturbation models themselves (e.g. Trauth, 2013; Dolman and Laepple, 2018; Lougheed et al., 2018). In such a case, assuming a bioturbation depth of 10 cm (Peng et al., 1997; Trauth et al., 1997; Boudreau, 1998), we can calculate that the 1 sigma age value of foraminifera contained within a single cm of a 5 cm/ka core is 10 / (5/1000) = 2000 years (from which follows that 2 sigma = 4000 years). – We will include a version of this paragraph in a revised manuscript

Furthermore, rapid accumulation rates should be possible around islands. The coarse map overlaid here fails to account for these. I would suggest adding to the text that in strategic locations (in the blue areas), sedimentation rates may still be high enough.

We agree seamounts and islands are places where the depth is shallower than our prescribed carbonate compensation cut-off depth and could also potentially have higher SAR, we will add a clarifying statement to the figures and the main-text.

**Improper referencing**

This is not the first study to use pseudo-proxy to examine whether IFA can be used for ENSO reconstruction. Thirumalai et al. present a model that can be more easily applied to a real application. First, reference this study (and others) at the beginning of the manuscript and second, why not extend their "picking" model to also evaluate the contributions of sample size?

We reiterate that our forward model seeks to model the d18O of the full foraminifera populations in the water. We are not modelling sediment core IFA.

**Minor comments**

Abstract: Should state that this is an IFA technique.

See answer to your previous point.

Page 1, Line 23: specify that the interaction on interannual timescale is known as ENSO. On decadal, it's known as the PDO.

We will rephrase to:

"Predictions of short-term, abrupt changes in regional climate are imperative for improving the spatiotemporal precision and accuracy when forecasting future climate. Coupled ocean-atmosphere interactions (wind circulation and sea surface temperature) in the tropical Pacific, collectively known as the El Niño-Southern Oscillation (ENSO) **on interannual timescales and the Pacific Decadal Oscillation on decadal timescales**, represent global climate's largest source (Wang et al., 2017) of inter-annual climate variability (Figure 1)."

Page 1, Line 27: SO is part of ENSO. Should rephrase as we have long instru- mental records of the atmospheric variability but not the ocean.

We will rephrase to:

"The instrumental record of the past century provides important information (*i.e.* the Southern Oscillation Index; SOI), however, detailed **oceanographic** observations **of the components** of ENSO (**both the El Nino and Southern Oscillation**), such as the Tropical Oceans Global Atmosphere (TOGA; 1985-1994) experiment only provide information from the latter half of the twentieth century (Wang et al., 2017)."

Page 3, line 3: Stott et al is not the only reconstruction in the Western Pacific, either use e.g., or as done previously cite multiple sources.

We will include multiple sources and an e.g., so it will read as follows: "and (3) those associated with the trace metal geochemistry (**e.g., Ford et al., 2015; Sadekov et al., 2013;** Stott et al., 2002, 2004; **White et al., 2018**), more specifically the natural logarithm of the relative concentration of Mg and Ca (ln(Mg/Ca), of the shell, based upon the temperature dependent (Elderfield and Ganssen, 2000; Nürnberg et al., 1996)incorporation and substitution of a Mg cation into the calcite lattice (Branson et al., 2013, 2016).".

However, we would like to clarify that the rationale for the citation here is not referring exclusively to the Western Pacific pool, but rather citing papers (including earlier papers) around the topic of 3 proxy types as noted on Pg. 2 ln23:

"Proxies of past ENSO and Pacific SST (Ford et al., 2015; Koutavas et al., 2006; Koutavas and Joanides, 2012; Koutavas and Lynch-Stieglitz, 2003; Leduc et al., 2009; Sadekov et al., 2013; White et al., 2018) are based upon the biomineralisation of the calcite, or a polymorph such as verite (Jacob et al., 2017), and shells of foraminifera (Emiliani, 1955; Evans et al., 2018; Zeebe and Wolf-Gladrow, 2001). There are three major types of foraminifera-based palaeoceanographic proxies: (1)…. (2)… and (3)…"

Page 3, Line 30: Mg/Ca is not a simple function of temperature. There is a growing body of evidence that suggests that Mg/Ca is also sensitive to salinity and pH. In addition, the calcite saturation of the bottom waters on post-depositional preservation of the signal.

We agree that is why on Pg. 11 Ln 16 – 23 we state : "several other parameters can alter this technique, this includes abiotic effects such as salinity (Allen et al., 2016; Gray et al., 2018; Groeneveld et al., 2008; Kısakürek et al., 2008) or carbonate ion concentration (Allen et al., 2016; Evans et al., 2018; Zeebe and Sanyal, 2002); biotic effects such as diurnal calcification (Eggins et al., 2003; Hori et al., 2018; Sadekov et al., 2008, 2009; Vetter et al., 2013); or additional factors such as sediment (Fallet et al., 2009; Feldmeijer et al., 2013) or specimen (Barker et al., 2003; Greaves et al., 2005) 'cleaning' techniques. Given the role of Mg in inhibiting calcium carbonate formation, the manipulation of seawater similar to the modification of the cell's $p$H (de Nooijer et al., 2008, 2009) may aid calcification and explain the formation of low-Mg by certain foraminifera (Zeebe and Sanyal, 2002)"

As such we are producing a pseudo proxy of what the proxy aims to test, i.e. temperature, we will add some text with regards to these later lines, where the reviewer has pointed it out, as follows:
"Here, we use the recently developed *Foraminifera as Modelled Entities* (FAME) model (Roche et al., 2018) to take into account potential modulation of $\delta_{18}O_c$ and the Temperature recorded in the calcite, herein $T_c$, by foraminifera growth. $T_c$ can thus be considered as an estimate of the proxy Mg/Ca (**albeit one uninfluenced by secondary factors**)."

Page 5, Line 3: Why not used species- specific equations?

A first reason is that the use of equilibrium as opposed to species-specific equations places all foraminiferal species against an equal benchmark. More importantly, most species-specific equations have been produced in culture (though some others in tows and pump samples) so that they contain variability in both growth and environmental (the same data that produced the growth functions in FORAMCLIM/FAME) conditions.

As an equilibrium oxygen isotope matrix (time-depth) is used to produce the FAME weighted distributions, said equilibrium can be switched out for 'equilibrium species values', however, such a terminology does not clearly outline the influence of time (growth season) and depth on the different species' oxygen isotopes.

Page 5, Line 12: Not sure what is meant by "Which can compute eight foraminiferal species". Do you mean growth?

The sentence will be altered as follows:
"FAME is based upon FORAMCLIM which can compute **the growth of** eight foraminiferal species (Kageyama et al., 2013; Lombard et al., 2009, 2011; Roche et al., 2017), however comparison with a core top database has been limited to five foraminiferal species (Roche et al., 2017)."

Page 6, line 11: There is an abundant body of literature dealing with the definition of an ENSO event. Why not start there?

We are not attempting to challenge or redefine ENSO or discount ONI or the BJ-Index (etc.) derived event chronologies, instead our simplification is based upon two things: (1) the simplicity of the input variables and (2) whether foraminifera (~4 weeks life) would 'sense' the event (as stated in our previous comment to reviewer 1).  To clarify we will alter the paragraph as follows:
Pg. 6 Ln 10 – 15: "However, Pan-Pacific meteorological agencies differ in their definition (An and Bong, 2016, 2018) of an El Niño, with each country's definition reflecting socio-economic factors, therefore, for simplicity we utilise a threshold of $\chi \geq +0.5°C$ **(where $\chi$ is the value of ONI)** as a proxy for El Niño, -$0.5°C \leq \chi \geq +0.5°C$ for neutral climate conditions and $-0.5°C \leq \chi$ for a La Niña in the Oceanic Niño Index. Many meteorological agencies consider that five consecutive months of $\chi \geq +0.5°C$ must occur for the classification of an El Niño event. However, here it is considered that any single month falling within our threshold values as representative of El Niño, neutral or La Niña conditions (grey bars in Figure 1)**. This simplification reflects the lifecycle of planktonic foraminifera (~4 weeks) as the population at time step t knows not what happened at t-1 or will happen at t+1. As we are producing the mean population growth weighted $\delta^{18}O$ values, 'almost' El Niño or 'almost' La Niña would be indistinguishable from the build-up and subsequent climb-down of actual El Niño and La Niña events. Therefore, these 'almost' El Niño or 'almost' La Niña are placed within their respective climatological pools as El Niño or La Niña.**"

Page 9, line 20-25: Most of these studies are based on pooled samples and were referencing to an ENSO-like signal rather than the interannual mode of variability that IFA is targeting.

We state that these datasets are from pooled and individual (Pg. 9 ln 18-20 "Several authors have focussed on individual foraminifera analysis (IFA) or pooled foraminiferal analysis in the Pacific region, either for trace metal or stable isotope geochemistry.") and later we point out that these are the authors inference (Pg. 9 ln 22-23 "The resultant data of such studies have been used to infer a relatively"). We agree that depending upon the method authors can be referencing 'ENSO-like' and/or 'interannual mode of variability', we make reference to this point at Pg 10 ln 9 -26 (e.g., "Our own analysis using our FAME $\delta_{18}O_c$ and $T_c$ output mimics foraminiferal sedimentary archives, pooling several decades worth of data in which the resolution is coarse enough to obscure and prevent individual El Niño events being visible but allowing for some kind of long-term mean state of ENSO activity to be reconstructed").

As we already take into consideration the reviewers point, we can assume that this comment merely reflects the opinion we are performing IFA.

**B. Metcalfe on behalf of the co-authors, Response to Reviewer 3**

*[reviewer comments as red text in blocks]*

We thank reviewer 3 for their time in reviewing our paper. However, it is unfortunate that the reviewer did not take the time to read our response to reviewer 1 as many of the same concerns have been addressed there and we have responded to those questions in detail. It is also unfortunate because it would have, hopefully, aided the reviewer in realising that this paper is about testing whether the given foraminiferal populations are statistically different so that they could potentially unravel different climatic states. Our approach is not about IFA research; IFA research is referred to because it provides an excellent sample dataset (and we highlight that it is one of the ways to understand a climate history more thoroughly). It is however not the only dataset and hence we make reference to other studies as well. We also find it disappointing that once more we are having to discuss inverse and forward modelling, as well as the statistical tests used; all aspects that where already explained in our previous response.

> *In this study, Metcalfe et al. aim to test whether the approach of using individual foraminifera analysis (IFA) can be used to assess ENSO variability. In order to accom- plish this, they use the Foraminifera as Modeled Entities (FAME) model to calculate idealized foraminifera distributions across the tropical Pacific. These results are then combined with seafloor/ CCD depth and sedimentation rate to determine which regions of the Pacific Ocean are suitable targets for IFA approaches. Modeling of foraminifera populations in order to determine if ENSO change is detectable has been done before (e.g., Thirumalai 2013, White 2018), although these studies focus on the detection of ENSO from paleoclimate proxy records. This study's novel contribution is the inclusion of the FAME model and foraminiferal growth rates to the analysis of modeled response of biological calcite to tropical variability.*

We once more state we are not testing IFA or have a model which tests IFA approaches. IFA is a suitable dataset to compare results with because it gives a lot more information than pooled analysis.

> *However, the FAME portion of the model is not validated against core-top data from the tropical Pacific, precluding assessment of its utility.*

The FAME model is validated against the whole MARGO dataset which includes the tropical Pacific. Please read Roche et. al. (2018).

> *The application of these results is likewise problematic, as it focuses on determining whether ENSO events (El Niño, La Niña) and neutral conditions have distinct distributions (forward modeling) rather than on how one could detect ENSO change (inverse modeling).*

There are a number of applications of this method, which we have outlined in response to reviewer 2. However, we consider that the reviewer comment is not an argument against what we have done – the first basic principle of understanding a proxy is 'can we detect', not 'how could we detect', as the how implies we know we can. Our manuscript is devoted to answer the question "can we detect".

> *Further, the discussion on sedimentation rate and CCD is broad-based and does not take in to consideration local changes in seafloor topography, changes in bottom-water oxygen availability that may alter bioturbation depths, and the variability characteristics of different regions with regard to the seasonal cycle, decadal-centennial variability, and ENSO change (e.g., Thirumalai 2013, Ford 2015, White 2018).*

It is true we have used broad based and conservative estimators (5 cm$^{-1}$ kyr$^{-1}$) as the bioturbation mixed layer is known to vary from 1 to 35 cm depending upon the various aspects the reviewer states. However, we fail to understand how "and the variability characteristics [?] of different regions with regard to the seasonal cycle, decadal-centennial variability and ENSO change" would somehow relate to SAR and dissolution depth.

> *Finally, there are aspects of the model that are unrealistic (e.g., a 400m depth for symbiont-bearing foraminifera; assuming sample sizes of 1000 for binning) or unrealized (e.g., how many individuals were selected for generating these estimates and a lack of model-data comparison) that present significant issues to the overall utility of this model for paleoceanographic reconstruction of ENSO from IFA.*

In the paper we state we use more than one depth, we first apply a CUT-OFF value of 400 m then progressively shallower (hence the reason for multiple panels) – we know this CUT-OFF value is deeper than symbiotic species (e.g., Pracht et al., 2019, Biogeosciences) hence the use of shallower depths (the reviewer is arguing against a simple test of our model here). It should be pointed out, whilst the model can in principle run down to 400 m it will only register a value if the temperature is applicable for that species (i.e., temperatures outside of the temperature window will not give growth as highlighted by the equations in Roche et al., 2018). Second, we multiply the TOTAL BIN COUNTS by 1000 to convert it into a simple distribution to test the distributions of the various climate - this is not the picking / assumed sample size. Were we to test sample picking we would need (and would have stated) to have done a larger test (although the computation required would be enormous [samples in group * replications * resampling] *[lat * lon], i.e., an individual foraminifera picking would be [1*40*10,000]*[40*120] = ~192 million computations).

*The title of the article does not represent the content or main goals of the study, and the conclusions stated in the abstract are different than those in the main paper.*

We disagree. If we break down the title:

**Validity – is the quality or state of something being valid (valid - being logically correct or well-grounded/justifiable).** We are testing whether the distributions of different climate events are statistically different (this is a fundamental test)

**Foraminifera-based** – we use a foraminiferal model. An alternative could include the word "populations" here, as in "foraminiferal populations".

**ENSO reconstructions -** The reviewer, we assume, is arguing that as we say ENSO reconstructions, and judging from their previous point ("as it focuses on determining whether ENSO events (El Niño, La Niña) and neutral conditions have distinct distributions (forward modeling) rather than on how one could detect ENSO change (inverse modeling).") that we haven't focused on the 'how one could detect', yet we are testing whether the foraminiferal distributions of different climate events are statistically different, hence we have carried out a test on a more fundamental level (i.e. foraminifera in the water, before they are incorporated in the sediment archive).

We will rephrase the abstract and conclusions for clarity.

*The questions the authors raise are valid and useful, but the results as stated do not support their conclusions. In fact, the stated conclusions of the article are, in several places, contradicted within the paper itself. These contradictions are not well-explained, and thus a clear summary of the findings is difficult to parse.*

We will rephrase those sections the reviewer elaborates on in the general comments.

**General Comments**

*The study here focuses on forward modeling using FAME for IFA. However, the authors fail to prove whether existing IFA-ENSO reconstructions are valid or provide the tools for evaluating proxy data (e.g., the "inverse problem", as mentioned in other reviews, whereby foraminifera records are analyzed to infer ENSO). Thus the application of these results to the paleodata world is limited.*

We are not forward modelling for IFA in sediment records, we are forward modelling foraminifera populations in the water.

The Paleodata world exists to answer questions regarding our understanding of past climate, therefore understanding if our proxies work for the period covering the observed climate record represents a fundamental test. The use of foraminiferal records to infer ENSO starts off with the prerequisite that what you are recording is ENSO, so our current research asked whether the values of the different climatic states for two proxies (calcite $\delta^{18}O$ and temperature of calcification) are statistically different. The application of these results downcore or the provision of tools for evaluating palaeoproxy data were not the stated aim of our particular research question.

*The more relevant application here is in targeting locations for performing IFA studies, but this is limited as well, as the sedimentological and bioturbation properties of regions across the Pacific are much more variable captured here.*

We included a rough SAR/CCD map for the benefit of the reader. However, if a reader or potential researcher in palaeoceanography has access to much better data regarding SAR/sedimentological properties, our research would still be of value because they can compare their chosen location with the FAME results that we have generated. The fact that the FAME results and FAME-SAR-DEPTH results are plotted separately allows users to pick and choose whichever plot they find necessary.

*The authors use their own definition of ENSO events, despite significant previous literature and established definitions that are commonly used. The use of single month anomalies does not adequately represent the actual ENSO phenomenon, which relies on ocean-atmosphere feedbacks expressed over a period of months, and thus their analysis of differences between El Niño, La Niña, and neutral conditions may be flawed and biased toward non-ENSO SST anomalies.*

We extensively discussed the definition of ENSO events in the paper and our answer to reviewer 1. In the latter we explain the rationale for our choice of definition (that foraminifera life cycle is shorter than a month and therefore several populations would exist that could conceivably have the same isotopic value as a true event).

*This study does not compare the results of their FAME analysis with existing IFA reconstructions of variability from the tropical Pacific. In the eastern Pacific, Rustic 2015 used δ18O IFA on modern-era sediments to show close correspondence with calculated δ18O from reanalysis data; in the central Pacific, White 2018 showed that the distributions of Mg/Ca-based SSTs from individual foraminifera in a 4ky coretop are statistically similar to modern reanalysis data.*

As per our comment to a similar question by reviewer 2, we will address this in a revised form of the MS.

**Specific** Comments

*The authors focus on δ18O proxies for IFA, and discount Mg/Ca reconstruction and the modeling efforts done with those (White 2018, Ford 2015). To discount Mg/Ca ratios as a paleoproxy without the kind of analysis provided for δ18O seems premature. While changes in carbonate concentration, salinity, and preservation environment can indeed alter Mg/Ca ratios, significant study has been done and is underway to understand these roles. Species-specific calibrations and various corrections exist that are well quantified. Not using Mg/Ca for the Tc seems rather limited.*

This point was discussed in our answer to reviewer 1. We use integrated temperature as a pseudo proxy for Mg/Ca – we did not attempt (though it would be interesting to perform such an analysis in another dedicated paper) to convert the input temperature into proxy values, i.e., a pseudo equilibrium Mg/Ca, like how the oxygen isotope values are calculated. Most ocean reanalysis and model datasets do not include the full variables required, although some models do (e.g., Grey and Evans 2019, Paleoocean. Paleoclim.).

The section of the paper that deals with and discusses Mg/Ca, is not "to discount" the proxy but was written to preempt comment(s) regarding our paper about why we did not attempt to model the full variables of this proxy (as we state in the paper why it would be beyond the remit of the paper).

Nor have we, as the reviewer states, 'discounted' the modelling efforts done with Mg/Ca (e.g. White et al., 2018; Ford et al., 2015) – if the reviewer would like to clarify this point, we would gratefully alter the text. However, this would seem to be the reviewer's own projection on to our paper and not something we categorically stated. We will try to make this clearer.

*The number of foraminifera picked from a given sediment interval is an important component of IFA. Increasing bin counts to 1000 artificially (Page 6) does not represent the numbers typically used in such analyses; the numbers used for other analyses (Page 7) are not specified.*

They are not specified because we didn't used the number of foraminifera picked, we artificially convert the distribution into testable values by multiplication. However, as we have stated throughout we are testing the distribution of the *population* not a *sample*.

*In the results, the first statistical test is to test whether the means of the FPen and FP-neu δ18O distributions are different and use this to determine whether ENSO events can be detected. Comparison of the population means does not necessarily reflect differences in the population distributions, and only provides a measure of mean conditions that may or may not be related to ENSO variability. The use of the Anderson-Darling test to assess differences in distribution is used later. It is unclear how these two different tests were related, and how the mean δ18O FPen/neu was utilized.*

As per a similar comment as reviewer 1, we will reword this sentence for clarity. We agree it does not necessarily reflect differences hence why we used the AD test.

*The author's use of the Anderson-Darling test to assess differences in distributions is novel, but results of this test are not compared to those that have been used to assess IFA results in previous studies (e.g., std dev (Thirumalai 2013, Koutavas and Joanides 2012, Rustic 2015) or Q-Q (White 2018, Ford 2015)). Is this more sensitive, less sensitive, or does it measure different aspects of the distribution change NOT captured in the other analyses? Without such comparison, the ability to assess the validity of IFA reconstruction (the purported goal of this paper) is limited.*

Different statistical techniques plot, test or validate different aspects of a sample or dataset being used. There seems to be some confusion by all three reviewers as to why the Anderson-Darling is being used. As we have already stated this before (in our reply to reviewer 1), a statistical test should be chosen by a study's author that can be used to test the research question devised by that author. We seek to investigate the (dis)similarity of distributions, therefore the Anderson-Darling test is the suitable test.

*The specifics of sedimentation rate and bioturbation vary greatly across the tropical Pacific and rely on multiple processes. The role that oxygen plays in bioturbation is important, especially as bottom-water oxygen levels vary across the tropical Pacific. Likewise, seafloor topography is highly variable, with ridges and sea mounts that are not apparent at the resolution used.*

*On P8: "Similarly, the individual characters of El Niño events, which are very short in duration, become lost in the bioturbated sediment record " The purpose of IFA is not to discern the properties of an individual event. Change in frequency or amplitude of events over a period of time can be statistically detected using various means to compare the distribution of integrated conditions over the period of sedimentation. Bioturbation serves, then, to extend that integrated time and the range of conditions experienced.*

Bioturbation causes, in the case of low SAR sediment record, thousands of years of time to become mixed into a single interval of sediment core. Hence, this may serve to mix values associated with the long-term climate signal and the ENSO signal.

With respect to the comment regarding IFA, we understand (e.g. Ganssen et al., 2011) that IFA is not a way to deduce a particular event (e.g. monsoon) but a way to characterize the samples. However, our sentence on page 8 is not saying that individual events are to be reconstructed, but that the characteristics of single event get muddled up in time. Now this would not be a problem (fundamentally) if $\delta^{18}O$ did not have a $\delta^{18}O_{sw}$ component. But $\delta^{18}O$ does have a $\delta^{18}O_{sw}$ component, thus, shells that are anomalous in one time period (e.g. LGM) may - with a change in the ice volume effect on the $\delta^{18}O_{sw}$ - have a value that is similar to 'background signal' in another time period (e.g. Holocene). Hence, the use of the word 'lost'.

*Bioturbation will also not remove anomalous values (page 9) – rather, such values may be present as part of a distribution representing more integrated time. Likewise, bioturbation has the effect of smoothing the signal, but the "signal" is a function of all sources of variability (ENSO, annual, decadal, centennial). The relative expression of these forms of variability along with the amount of time integrated by a sample are both important in terms of the ability to capture ENSO signals.*

Anomalous values are only anomalous in relation to the rest of the dataset. As our answer to the comment above explains, if a samples anomalous values are moved into a sample with similar values then they will no longer be anomalous. We will clarify what we meant by smoothing the signal in a revised form of the MS.

*On P.10, Cole and Tudhope (corals) and White et al (IFA) are cited in error when discussing lake colour intensity and precipitation-driven records.*

We were referring to some of the analysis within those papers (as per our comment to a similar point of reviewer 2 we will alter this sentence).

*Also on P10, the authors claim: "If the number and magnitude of ENSO events were reduced, the relatively low downcore resolution of marine records may not accurately capture the dynamics of such lower amplitude ENSO events using existing methods."*
*– Which methods? Q-Q, std. dev, event counting, others? It's not entirely clear this is even referring to IFA reconstructions, as the records discussed previous are sedimentary, coral, and IFA (but noted as "precipitation driven", see above).*

As our comment to reviewer 1 very clearly explained that we are not modelling IFA it is a shame that reviewer 3 did not have the time to read our replies. Here, "methods" is generally referring to proxies – we will elaborate in the revised version to reduce the confusion.

*P.10 line 5: **"The possibility of a marine sediment archive being able to reconstruct ENSO dynamics comes down to several fundamentals: the time-period captured by the sediment intervals (a combination of SAR and bioturbation), the frequency and intensity of ENSO events, as well as the foraminiferal abundance during ENSO and non-ENSO conditions."** This statement leaves out other key elements, including the relative expression of ENSO events, the seasonal cycle, and decade-and-longer variability. These elements are (arguably) more important for inverse modeling, where the ability to disentangle growth rates from other sources of variability is impossible, and thus the signatures of ENSO in such records need to be discerned.*

[Bold is to clarify what part of the reviewers comment is a quote] – The reviewer suggests we have left out key elements. We agree that we missed out '*seasonal cycle, and decade-and-longer variability*' which we will add. But the '*relative expression of ENSO events*' is already included within the reviewer's chosen quote (underlined).

*A key point in the paper (P10) says "The results presented here imply that much of the Pacific Ocean is not suitable for reconstructing ENSO studies using paleoceanography, yet several studies have exposed shifts within std dev(δ18Oc) of surface and thermocline dwelling foraminifera. One can, therefore, question what is being reconstructed in such studies.". This study has, at this point, not tested whether the Std.dev of δ18Oc from individual foraminifera have reconstructed ENSO (also, the wording of this sentence is odd).*

If the two populations are statistically similar (as in La Nina and El Nino have statistically indistinguishable distributions) then it is logical to question what the measure of dispersion (std dev) of the measured sample is linked to (i.e. it is seasonality, or species depth habitat change). We will rephrase in the revised version of the MS to improve readability.

*The first paragraph of the discussion (p9) purports to be about paleoclimatological archives that "have been used to indirectly and directly study past ENSO". However, the discussion is on mean-state reconstructions (Koutavas 2003, Dubois 2009). Koutavas 2003 is non-IFA mean-state reconstruction; likewise, the Dubois 2009 paper notes that "we prefer not to invoke any ENSO-like state for the glacial EEP based solely on our UK'37 SST." While it may be true that this result and Koutavas 2003 are at*

*odds, this is not an issue of IFA or ENSO reconstruction, but rather aggregate analysis and mean -state reconstruction. Discussion of std.dev ENSO studies (modeled by Thirumalai, Koutavas 2006, Koutavas and Joanides 2012, Leduc 2009, Sadekov 2013, Rustic 2015) is not found, yet the following paragraph (see above) is largely about this approach. Further, significant discussion and analysis of IFA reconstructions of ENSO during the LGM is found in Ford 2015, which is not discussed here.*

[remaining 66,900 characters of this post omitted]

---

## Referee Report (RR1)

**Review of**

**Using a foraminiferal ecology model to test if tropical Pacific planktonic foraminifera are suitable recorders of ENSO, version 4**

by Metcalfe et al.

**Recommendation**: *Reject*

Summary

The manuscript uses a number of forward models to assess the possibility of recovering distinct ENSO states from individual foraminiferal analyses (IFAs). While the goal is excellent and the general approach sound, the presentation is so abstruse that it removes all credibility from the paper's claims. After what looks like 4 trials, it is unclear that another round of revisions could fix these fatals flaws.

**1    General Comments**

As a prelude, I note that the paper has received prior reviews, which I refrained from reading to avoid biasing my judgment. I apologize in advance if the following comments are redundant, or if they contradict the recommendations of previous referees – I know firsthand that one cannot please every referee.

**1.1    Statistics**

As said above, the general approach is sound, though the tediousness of the exposition leads the authors to belabor obvious points at the expense of critical explanations.

My biggest question mark is on the kernel density estimation (section 2.4). The authors use an Epanechikov kernel with the following justification *"The use of an Epanechnikov kernel distribution to fit the data, as opposed to other types of distribution, represents a trade-off between keeping as many parameters constant whilst mimicking the underlying dataset for a large number of grid points."*. I fail to see how this justifies an Epanechikov kernel, as opposed to any other kernel. While the choice of kernel is typically unimportant, the fact that it is so awkwardly justified raises a red flag. If your results depend sensitively on the choice of kernel, you are in deep, deep trouble. It would be important to include (as a supplement) an analysis with a different kernel choice (e.g. Gaussian).

The authors also mention a variable bandwidth, which is fine, but do not explain how it is chosen (e.g. Silvermann criterion). Given that the entire premise of the paper is to compare distributions, this is a crucial detail that needs to be better explained, possibly with a sensitivity analysis.

On the broader point of reporting the results of the Anderson-Darling test, the authors rely exclusively on whether the p-values are above or below 5%. As emphasized by *Wasserstein and Lazar* [2016], the American Statistical Association explicitly warns against relying exclusively on p-values, and recommends additional metrics like effect sizes and confidence intervals. I think mapping effect sizes, possibly stippled to indicate whether the p-values are above or below 5%, would be better practice.

Finally, and though I would be the last reader to request that the paper get any lengthier, I am surprised that the authors did not focus on the most obvious Achille's heel of IFAs, as practiced, for instance, by *White et al.* [2018]. Contrary to the author's claim, there is nothing inherently wrong with using quantile-quantile plots to compare distributions – indeed it can be a fine idea. The one issue with QQ plots as applied in studies like *White et al.* [2018] is that it is a handful of extreme values that determine the slope, making the results extremely brittle to outliers. To my mind, this is the most urgent statistical issue to address about the way IFAs are currently presented.

**1.2 Structure**

The paper follows the classic structure of Introduction/Methods/Results/Discussion. The only issue here is that, because they consider 3 distinct questions, the methods are varied and lengthy, and by the time the reader gets to the end of Section 3, they have largely forgotten the relevant methods. It would seem more natural to me to structure one section per question, with relevant methods introduced where needed. This would look like: 1 introduction 2) distinguishing variance statistics 3) distinguishing distributions, 4) sensitivity to input parameters, 5) impacts of dissolution and bioturbation 6) discussion. One thing is for sure: the present structure is extremely indigestible, and squanders any goodwill that the reader might still have after reading that pompous introduction.

**1.3 Grammar**

I have reviewed dozens of papers over my career, but this one takes the prize for the most abstruse writing coming from native English speakers. A few times I had to look up whether some of the quirks might be differences between British and American English, but I could find no justification in any grammar book for spelling figures "figure's" (P14L30), for starting sentences by "Whilst" followed by a comma, for writing Proustian run-on sentences, or for being generally so incoherent that, after reviewing the paper on my iPad, the number one suggestion from my autocomplete is "incoherent grammar" (see annotated manuscript). I am surprised that experienced scientists like Didier Roche or Claire Waelbroeck let the paper be submitted once, let alone four times, with such flaws.

**1.4 Figures**

The figures are a piece of work. First, this is the first time I have seen figures so large that they make the text pages of the PDF look like microfilm. To add insult to injury, they are all of different sizes, making the document's navigation extremely tedious. The substance is no better than the style, unfortunately, as (apart from Fig 10), they are all so poorly designed that I would tell my students to redo them. It seems like the authors cannot decide what point to make, so they bombard the reader with lots of similar, overloaded figures. It is imperative to focus the design around the key points, and put the other figures in an appendix/supplement.

**2 Line by line Comments**

see annotated manuscript.

In summary, this paper is not appropriate for publication in present form, and it is unclear if, after this many trials, it can ever be brought up to that standard.

**References**

Wasserstein, R. L., and N. A. Lazar (2016), The ASA Statement on p-Values: Context, Process, and Purpose, *The American Statistician*, *70*(2), 129–133, doi:10.1080/00031305.2016.1154108.

White, S. M., A. C. Ravelo, and P. J. Polissar (2018), Dampened El Niño in the Early and Mid-Holocene Due To Insolation-Forced Warming/Deepening of the Thermocline, *Geophysical Research Letters*, *45*(1), 316–326, doi:10.1002/2017GL075433.

**Using a foraminiferal ecology model to test if tropical Pacific planktonic foraminifera are suitable recorders of ENSO**

[revised manuscript text omitted]

**1.2 Foraminiferal Proxies**

Such an approach, however, requires an abundance of reliable spatiotemporal proxy data from the entire Pacific Ocean. Moreover, such proxy reconstructions are subject to several unknowns, uncertainties and biases. For the specific case of foraminifera populations in the water, it particularly arises from the species-specific ecological niche. The mapping of proxy value to climate value can therefore be skewed, a major factor governing the spatiotemporal distribution of a given planktonic foraminiferal species is the presence of an ideal water temperature. Proxies of past ENSO and Pacific SST (Ford et al., 2015; Koutavas et al., 2006; Koutavas and Joanides, 2012; Koutavas and Lynch-Stieglitz, 2003; Leduc et al., 2009; Sadekov et al., 2013; White et al., 2018) are based upon the biomineralisation of the calcite, or a polymorph such as verite (Jacob et al., 2017), shells of foraminifera (Emiliani, 1955; Evans et al., 2018; Zeebe and Wolf-Gladrow, 2001). In general, there are three major types of foraminifera-based palaeoceanographic proxies:

**(1)** those associated with the faunal composition and their abundance within deep-sea sediments that utilises either a qualitative approach (Phleger et al., 1953; Schott, 1952); a weighted average (Berger and Gardner, 1975; Jones, 1964; Lynts and Judd, 1971); a selected species approach (e.g. coiling direction, or warm-water species presence; Ericson et al., 1964; Ericson and Wollin, 1968; Hutson, 1980b; Parker, 1958; Peeters et al., 2004; Ruddiman, 1971; Schott, 1966); a regression analysis (Hecht, 1973; Imbrie and Kipp, 1971; Williams and Johnson, 1975); or, a transfer function (CLIMAP Project Members, 1976; McIntyre et al., 1976; Williams, 1976; Williams and Johnson, 1975) that compares the down-core records with a dataset of 'modern' values and their associated water column parameters (Hutson, 1977, 1978);

**(2)** those associated with the stable oxygen isotope composition of a whole shell analysed either individually (Ganssen et al., 2011; Koutavas et al., 2006; Koutavas and Joanides, 2012; Leduc et al., 2009) or pooled (Garidel-Thoron et al., 2007; Koutavas et al., 2002; Stott et al., 2002, 2004), herein $\delta^{18}O_c$ (c = calcite), which can be used to reconstruct SST and past oxygen isotope values in seawater $\delta^{18}O_{sw}$ (sw = seawater) when paired with a proxy that can either reconstruct temperature or salinity;

**(3)** those associated with trace metal geochemistry (e.g., Ford et al., 2015; Sadekov et al., 2013; Stott et al., 2002, 2004; White et al., 2018), more specifically the natural logarithm of the relative concentration of Mg and Ca (ln(Mg/Ca), of the shell, based upon the temperature dependent (Elderfield and Ganssen, 2000; Nürnberg et al., 1996) incorporation and substitution of a Mg cation into the calcite lattice (Branson et al., 2013, 2016).

The interpretation of these proxies, however, is not straightforward, for example, calibration of foraminiferal assemblage based transfer functions with surface temperatures as opposed to a deeper temperature signal may in fact skew the reconstructed temperature (Telford et al., 2013); $\delta^{18}O_c$ can be affected by species-specific size effects (Feldmeijer et al., 2015; Metcalfe et al., 2015; Pracht et al., 2018), disequilibria or vital effects, which clouds the accurate reconstruction of past SST and $\delta^{18}O_{sw}$. There is also no simple bijective function between $\delta^{18}O_c$ and the oceanic variables $\delta^{18}O_{sw}$ and temperature used in its calculation, with variability in $\delta^{18}O_{sw}$ limiting the use of $\delta^{18}O_c$ as a pure temperature proxy. 
[revised manuscript text omitted]

The difference between the constant of Hut (1987) and the dynamic value (Brand et al., 2014) is minor.

**2.2 Foraminifera as modelled entities (FAME)**

Foraminifera as modelled entities has been developed as a tool for translating, a climatic input (typically a reanalysis dataset or climate model output) into a (simulated-) climatic signal, a signal that aims to approximate the depth integrated growth of foraminifera (e.g., Pracht et al., 2019; Wilke et al., 2006; Steindhardt et al., 2015). Data-model comparison studies suffer from an inability to directly compare like with like so that there are differences in (i) the units used *i.e.*, most proxies reconstructing temperature do not give values of temperature in degrees °C or K but in their own proxy units (e.g., per mil ‰; mmol/mol; species abundance or ratio) necessitating a conversion; and (ii) there is a reduction in scales, *i.e.*, models give a wealth of information (multiple depth layers and high resolution time slices) in the time-depth domain. A number of models and modelling studies exist to determine the foraminiferal responses to present (Fraile et al., 2008, 2009; Kageyama et al., 2013; Kretschmer et al., 2017; Lombard et al., 2009, 2011; Roy et al., 2015; Waterson et al., 2016; Žarić et al., 2005, 2006), past (Fraile et al., 2009; Kretschmer et al., 2016) and future (Roy et al., 2015) climate scenarios, FAME uses the associated temperature and $\delta^{18}O_{eq}$ at each grid cell to compute a time averaged $\delta^{18}O_c$ and $T_c$ for a given species. FAME was produced as an attempt to reduce the error associated with data-model comparisons by (i) generating simulated-proxy time-series from model runs that can be compared with age-depth values down core; and (ii) to reduce the model information for a given time-slice into a manageable and relevant value using an integration that would make sense on a biological point of view (Roche et al., 2018).

[revised manuscript text omitted]

Each time-step for the entirety of the Pacific was classified as one of three climate states (El Niño; Neutral; and La Niña), where after the resultant $\delta^{18}O_c$ and $T_c$ at each timestep produced by FAME for each grid-point were binned into their respective categories. An Epanechnikov-kernel distribution was first fitted to the binned monthly output of a single climate state, the bandwidth varies between grid-points to provide for an optimal kernel distribution. The use of an Epanechnikov-kernel distribution to fit the data, as opposed to other types of distribution, represents a trade-off between keeping as many parameters constant whilst mimicking the underlying dataset for a large number of grid points. The conversion of the data from dataset to distribution may induce some small error induced by: rounding to whole integers; the use of a $\delta^{18}O_{mid-point}$ which gives an error associated with the bin size ($\pm0.05$ ‰) that is symmetrical close to the distributions measures of central tendency but asymmetrical at the sides; and finally, the associated rounding error at the bin edges within a histogram ($\pm0.005$ ‰). Subsequently any two desired distributions can be compared for (dis)similarity using an Anderson-Darling test (1954). Here, all values, i.e., the population, associated with a climatological state are compared with the other populations representing the different climatological state, the results plotted here are Neutral climate state vs. El Niño climate state.

**2.5 Test of input data (Temperature and calculated $\delta^{18}O_{eq}$)**

Foraminifera as modelled entities produces a modulated response that seeks to replicate how foraminifera modify the climate signal, several studies have approximated the foraminiferal signal in a different way (e.g., Thirmulai et al., 2013; Zhu et al., 2017a). In order to understand how FAME has altered the signal, and the degree to which the conclusions drawn depend upon the modelled growth rates, the input datasets of the sea water properties (Temperature and calculated $\delta^{18}O_{eq}$), underwent a similar statistical test (Figure 4). Unlike FAME, which integrates over several depth levels using the computed growth rate, the test of the input datasets was with fixed depths without any growth rate weighting. These fixed depths are 5, 149 and 235 m, giving a Eulerian view (Zhu et al., 2017a) in which to observe the implications of FAME's dynamic depth habitat. As per the FAME output, each timestep value was placed into its climate state and an Anderson-Darling test performed to compare the (dis)similarity of on the resultant distributions.

**2.6 Alternative statistical tests**

In order to compare our results with previously published studies using planktonic foraminifera we employed a series of simple statistical tests, mimicking those applied to sediment archives by the palaeoclimate community. A chief parameter that has been employed in previous ENSO proxy work using foraminiferal analysis (more specifically, individual foraminiferal analysis; IFA) is the measure of individual foraminifera downcore standard deviation ($\sigma(\delta^{18}O_c)$). Increased $\sigma(\delta^{18}O_c)$ is considered to correlate to increased variation in SST and, in turn, increased ENSO incidence and/or magnitude (Leduc et al., 2009; Zhu et al., 2017a) or increased interannual variance (Thirmulai et al., 2013). The variance ($\sigma^2(\delta^{18}O_c)$) of the timeseries were computed both as the total variance and as the interannual variance, the latter is computed as outlined in Zhu et al. (2017a). For the interannual variance, the mean monthly climatology is subtracted from the dataset, producing monthly anomalies and a linear trend removed (using the detrend function of MatLab 2019a) – the resultant data was left unfiltered (*i.e.*, Zhu et al., 2017a used a 1-2-1 filter). Four 'picking' experiments were performed, as FAME computes the average value for a given time step and given the single foraminiferal isotope variance for an equivalent time step (e.g., weeks: Steinhardt et al., 2015) it is more than likely that this computation reduces the real spread in values. Therefore, rather than use the terminology specimen we prefer to use months. Given the complexity in reconstructions of trace metal geochemistry (Elderfield and Ganssen, 2000; Nürnberg et al., 1996): the potential error associated with determining which carbonate phase is first used when foraminifera biomineralise (Jacob et al., 2017); growth-band integration; secondary factors (e.g., salinity, carbonate ion) the focus of the picking here has been on the $\delta^{18}O_c$. Irrespective of which experiment, 60 months were drawn, with replacement, and the number of Monte Carlo iterations is set at 10,000. We assume that the 'picker' is taxonomically well-trained and/or has a procedure in which species can be checked taxonomically post-analysis (e.g. photographing all specimens prior to analysis, Pracht et al. (2019)) and therefore do not include an error that deals with incorrect identification. Although we note that parameterisation of misidentification would be difficult, as it requires understanding of the variability in both standard deviation and absolute values for species co-occurring downcore (Feldmeijer et al., 2015; Metcalfe et al., 2015; 2019). For each run of experiment's (i) to (iii) the drawn months were saved in order to perform (iv):

(i) In Picking Experiment-I (Figure 3D), the months drawn for each iteration of the Monte Carlo were selected and each grid-point was sampled (i.e., there are $10000 \cdot 60$ selected months). This assumes that the same months are selected at grid point A as point B.

(ii) In Picking Experiment-II (Figure 3E), at each grid-point a Monte Carlo was run (*i.e.*, there are $170 \cdot 40 \cdot 10000 \cdot 60$ selected months). This assumes that different months could be selected between grid point A and point B.

(iii) In Picking Experiment-III (Figure 3F), at each grid-point a Monte Carlo was run using the growth rate weighting for each month (*i.e.*, there are $170 \cdot 40 \cdot 10000 \cdot 60$ selected months), this assumes that in periods of higher growth there will be a higher flux of the species and therefore a greater chance of selecting that month. The rationale being that not only are different months selected between grid point A and point B, but if A and B differ climatologically there may be an over subscription of ecologically beneficial habitats in one core location compared to the other.

(iv) In Picking Experiment-IV (Figures 3G to 3I), the experiment of (ii) was re-run but with the addition of two sources of error: The first error is based upon FAME producing the average value for a given time slice, therefore short-term variability in temperature and/or the spread in the population (*i.e.*, variance in depth of an individual; variance in chamber growth per individual), as evidenced by single foraminiferal analysis of sediment trap samples (*e.g.*, Steinhardt et al., 2015), is potentially lost. Therefore, for each picked month between -0.40 and 0.40 ‰ is added to the picked month value (in intervals of 0.02 ‰), this is approximately

±2° C (*i.e.*, ~4° C). The second error is the analytical error that an individual measurement will have. Machine measurement error is assumed to lie between -0.12 and 0.12 ‰ (in intervals of 0.005 ‰ – the 3rd decimal place is an exaggeration of machine capabilities although it will have repercussions for rounding) the 1σ of within run (as opposed to long-term average) of international stable isotope standards. The intervals of both errors (0.02 ‰ and 0.005 ‰) were chosen to give a similar number (n = 41 and 49) of potential randomly selected error for each picked month. Each picked month has their own randomly selected error for both of these errors, *i.e.*, each value is the sum of the month picked and their own error. The values for within month variability (Figures 3G) and machine error (Figure 3H) are calculated separately and then combined (Figure 3I), as they may have a corresponding or conflicting signs, either 'cancelling' out each other or amplifying the difference.

An associated statistical methodology is the graphical summary (as opposed to a numerical summary via a test value) of plotting the quantiles of two probability or the quantiles of sample probability distribution against a theoretical distribution distributions also referred to as a Quantile-Quantile, or Q-Q plot (*e.g.*, Ford et al., 2015; White et al., 2018). A complimentary (*i.e.*, used in association with, not as replacement, Filliben 1975) test metric, the Probability plot correlation coefficient (Filliben, 1975) can be used as a numerical summation of this approach, which bases its rationale on near linearity between the two tested distributions. This graphical technique is not used here for the following reasons, (i) the climatic categories (*i.e.*, El Nino, Neutral, La Nina) imposed upon the data give uneven sized sample distributions requiring an interpolated quantile estimate; and (ii) the large graphical computation required (170·40).

**2.7 An approximation of sedimentary archives: Water depth & Sedimentation Rate**

Discrete sediment intervals retrieved from systematically bioturbated deep-sea sediment cores contain foraminifera with ages spanning many centuries (Lougheed et al., 2018; Peng et al., 1979).  The ambient signal following translation into a foraminiferal signal within the water is therefore further modulated by several post mortem processes, which include: the lat long shift in position of sinking foraminifera - the so-called 'funnel affect' (van Sebille et al., 2015; Deuser et al., 1981); dissolution of calcium carbonate either in the water (Schiebel et al., 2007), at the seafloor, or due to pore fluids; and bioturbation. As mentioned, mixing by bioturbation, results in an apparent smoothing of the downcore, discrete-depth multi-specimen signal (Hutson, 1980a; Löwemark, 2007; Löwemark et al., 2005, 2008; Löwemark and Grootes, 2004; Cole and Tudhope, 2017; Mix, 1987), thus leading to the possibility of interpreting single outlying foraminifera values within a specific depth as representing an 'extreme' climate, when they may in fact represent climate from a different time or epoch. This is especially apparent in $\delta^{18}O_c$ where there is a difference temporally of $\delta^{18}O_{sw}$ (e.g., the ice volume effect in glacial and interglacial cycles ~1.25 ‰) meaning that the same temperature can have radically different $\delta^{18}O_c$ values, a consequence of this is that a series of high magnitude, but low frequency El Niño events could be disturbed in a discrete-depth record.

Therefore, in order to reliably extract short-term environmental information from foraminiferal-based proxies, the signal that one is testing or aiming to recover must have a large enough magnitude, be largely unaffected by dissolution (*i.e.*, above the lysocline) so as not to adversely affect the population and the sedimentation rate must be high enough to give sufficient temporal coverage and rule out upwards bioturbation of single foraminifera from significantly different climate periods.

In our first step in consideration of post-mortem signal alteration we focus on dissolution. The lysocline, the depth at which dissolution first becomes apparent (Berger, 1968; 1970), and the Calcite (or Calcium Carbonate) Compensation Depth (CCD; Bramlette, 1961) vary between the different ocean basins; the Atlantic Ocean in which deep water forms has a relatively deep CCD as a by-product of 'young' well ventilated bottom waters whereas the Pacific Ocean the final section of the thermohaline circulation conveyor belt, has a shallower CCD. In order to highlight the potential for dissolution, the bathymetry of the Pacific was extracted from the General Bathymetric Chart of the Oceans GEBCO 2014 30 arc-second grid (version 20150318, www.gebco.net) between -20°S to 20°N and 120°E to -70°W (Figure 8). Depths of 3500 m below sea-level (bsl), 4000 m bsl and 4500 m bsl are used here as cut-off values, these depths represent multiple possible depths under which there is the potential for noticeable dissolution (i.e., lysocline) or be dissolved (i.e., CCD). Whilst our intention here is a generalised view to be used as an approximate guide, it is important to note that the Pacific Ocean has the largest proportion globally of >1 km tall seamounts that are smaller than <100 km (Wessel, 1997). Which may have important, relatively shallow-water sedimentary sequences, which may also be of sufficient sediment accumulation rate therefore we supplement the GEBCO bathymetric data with the locations of seamounts. However, whilst there are an estimated 50,000 seamounts in the Pacific that are taller than a km (Menard, 1964; Wessel and Lyons, 1997), only 12,000 have been documented on charts (Batiza, 1982), and approximately 291 have been dated (Koppers et al., 2003; Clouard and Bonneville,

2005; Hillier, 2007). It is these 291, <1% of the estimated seamounts, we have overlain onto the bathymetric data (Figure 8b), although this number is further reduced as we only plot between 20°S and 20°N.

The second step when considering post-mortem signal alteration is the sediment accumulation rate (SAR). We first plot the time-averaged deep-sea SAR (Olson et al., 2016), adapted by Lougheed *et al.* (2018) for the Tropical Pacific (Figure 9). New geochronological tools, such as dual $^{14}C$-$\delta^{18}O$ measurements on single foraminifera (Lougheed et al., 2018), show that low sedimentation rate cores can have large variances in age between individual foraminifera present within a discrete 1 cm depth interval (Berger and Heath, 1968; Lougheed et al., 2018). In order to model bioturbation, a number of papers have used a diffusion style approach that reduces the parameters down to sediment mixing intensity and sediment mixing depth (herein referred to as bioturbation depth, BD), although this may be an artificial division purely driven by mathematical need rather than biological constraints (Boudreau, 1998). The bioturbation depth has been shown to have a global average of 9.8

cm (1σ: ± 4.5 cm) that is independent of both water depth and sedimentation rate (Boudreau, 1998), though likely controlled as a result of the energy efficiency of foraging, *e.g.* deeper burrows may cost more energy to produce than can be offset in extracted food resources, and potential decay in labile food resources with sediment depth. It is not possible to carry out a transient bioturbation model upon the temperature and salinity ocean reanalysis data that we used for FAME, as it only covers half a century of data, whereas thousands of years of input data are required to force a transient bioturbation model.

To investigate how much temporal signal is integrated into discrete-depth intervals for typical tropical Pacific SAR, we, therefore, utilised the single foraminifera sediment accumulation simulator (SEAMUS, Lougheed, 2019) to bioturbate, as the input climate signal (Figure's 9 to 11), 0-40,000 year $\delta^{18}O_w$ of NGRIP (North Greenland Ice Core Project Members, 2004; Rasmussen et al., 2014; Seierstad et al., 2014). The ice core time series is an ideal input for a bioturbation simulator, as it represents a highly temporally resolved climate input signal. SEAMUS simulates foraminifera in 10-year timesteps. The use of the NGRIP timeseries here is purely as an input parameter to investigate the effect of bioturbation upon a given climate signal - it is important to stress that by using NGRIP as an input signal for SEAMUS we are neither implying that tropical Pacific cores should have signal similar to NGRIP or inferring some kind of causal relationship. As we seek to investigate the effect of bioturbation, no attempt has been made to modulate the input signal's absolute values to mimic expected $\delta^{18}O_c$

values and this is why each plot of the synthetic down core time series retains the use of V-SMOW, despite carbonates being required to be V-PDB (Coplen 1995). Keeping all things constant, and varying a single parameter between experiments with SEAMUS, the sediment accumulation rate (SAR) was varied to fixed values of either 1, 2, 5 or 10 cm kyr$^{-1}$ (representative of typical Pacific SAR) and a bioturbation depth (BD) of either 5, 10 or 15cm based upon the global estimate and it's error bounds (Boudreau, 1998). For each experiment, the selected values of SAR and BD were kept constant for the entire

SEAMIS model run (i.e., the intensity and magnitude of bioturbation was not varied). In reality, SAR and BD may vary temporally depending on local conditions (*e.g.*, food, oxygen). Finally, the FAME results for the three species are overlaid with a water depth mask that highlights whether grid points are above or below 3500 m below sea-level (mbsl), to also show seafloor areas under the CCD depth, where carbonate material is not preserved (Berger, 1967, 1970b). A comparison between water depth and time-averaged deep-sea SAR (Olson et al., 2016), adapted by Lougheed *et al.* (2018) is shown in

Figure's 7 and 9.

**3. Results**

The results of the forward model (Figure 2 and 3) are compared with the input values (Figure 4) in order to identify regions in which the values are statistically distinct for different climate states (Figure's 5-7). These results are then shown against the water depth (Figure's 7 to 9) and the SAR (Figure's 9-11) for the region. The results utilise Foraminifera as Modelled

Entities (FAME; Figures 2, 3, 5, 6, and 7); the original Ocean Reanalysis data with computed $\delta^{18}O_{eq}$ (Figure 4); the General Bathymetric Chart of the Oceans (GEBCO; Figures 7 to 9); and the single foraminifera sediment accumulation simulator (SEAMUS; Figures 9 to 11).

**3.1 FAME Output: Variance**

We compute growth-weighted $\delta^{18}O_c$ (Figure 5 and 7) and temperature (Figure 6 and 7) distributions for each grid cell in the fifty-eight year simulation using FAME (Roche et al., 2018), constraining the calculation to the Tropical Pacific Ocean (between -20°S to 20°N and 120°E to -70°W). Our model produces 696 individual monthly maps for all three species (Figure 2). While two of the three species (*G. ruber* and *G. sacculifer*) have similar ecologies, they show differences in their resultant $\delta^{18}O_c$ for the same ocean conditions (Figure 2). A comparison of our computed variance with measured data (Supplementary Table 1) is made, we compare both the value of the nearest grid-cell and because of the size of the grid and drift of foraminifera (van Sebille et al., 2015) an average of a 3 by 3 grid in which the nearest grid-cell to the core location is in the center. A comparison is made with both the iCESM model output and the core's that match this output (Zhu et al., 2017a). For the Late Holocene sample (~1.5 ka) MD02-2529 (08°12.33'N 84°07.32'W; 1619 m) in which *N. dutertrei* individual foraminifera were analysed from >250 µm (Leduc et al., 2009) giving a calculated standard deviation of measured foraminifera of 0.38 ‰. Whereas, the full ~60 year time series (n = 696) of FAME presented here, gives a standard deviation for all species, at depth cut off 60 m between 0.26 and 0.32 ‰; at depth cut off 100 m between 0.20 and 0.29 ‰; at depth cut off 200 m between 0.20 and 0.25 ‰; and at depth cut off 400 m between 0.20 and 0.24 ‰ (see Table 1). Although these values vary if the average of the surrounding grid cells is used (see Table 1). In comparison the iCESM results have the following standard deviation values, for a Eulerian (fixed) depth of 50 m: 0.4 ‰; Eulerian 100 m: 0.6 ‰; and Lagrangian value of 0.49 ‰. There are three samples (Koutavas and Joanides, 2012; Sadekov et al., 2013) located south of core site MD02-2529, these are the Late Holocene (~1.6 ka) samples of V21-30 (01°13'S 89°41'W; 617 m) and (~1.1 ka) V21-29

(01°03'S 89°21'W; 712 m) in which *G. ruber* was measured individually (Sadekov et al., 2013). For these two sites the measured standard deviation is 0.507 ‰ and 0.510 ‰ for V21-30 and V21-29 respectively (Koutavas and Joanides, 2012). The third core site at a similar location is (~1.6ka) CD38-17P (01°36'04 S 90°25'32W; 2580 m) was not analysed individually, instead replicates of pooled samples of 2 or 3 shells of *N. dutertrei* (Sadekov et al., 2013) were made these measured values give a standard deviation of 0.28 ‰. The full ~60 year time series (n = 696) of FAME presented here, gives a standard deviation for all species, at depth cut off 60 m between 0.33 and 0.41 ‰; at depth cut off 100 m between 0.27 and 0.40 ‰; at depth cut off 200 m between 0.25 and 0.35 ‰; and at depth cut off 400 m between 0.25 and 0.34 ‰ (see Table 1). Although these values vary if the average of the surrounding grid cells is used (see Table 1). In comparison the iCESM results have the following standard deviation values, for a Eulerian (fixed) depth of 50 m: 0.53 ‰; Eulerian 100 m: 0.75 ‰; and Lagrangian value of 0.35 ‰.

The study of ENSO has focused on whether the variability is entirely in response to ENSO or whether it is dominated by interannual variability (Xie, 1994, 1995; Wang et al 1994, 2010), here the interannual (Figure 3C)  and total variance (Figure 3A) was computed and a ratio between the two calculated (Figure 3B; see Supplementary Table 1). Like the same analysis of interannual and total variance computed for iCESM and SODA reanalysis (Carton et al., 2000), outlined in Zhu et al. (2017a), there is also high ratio of interannual to total variance in our computed FAME dataset (Figure 3B). Although there are regions in the Eastern Equatorial Pacific wherein this ratio reduces. Despite this reduction, the ratio between total and interannual variance is still above > 0.5.

[revised manuscript text omitted]

For FAME-$\delta^{18}O_c$ the results where the populations are dissimilar are either plot as grey and hashed for *G. ruber* and *G. sacculifer* or white for *N. dutertrei* (Figure 5). This is because for these two species (*G. ruber* and *G. sacculifer*) we have the possibility to determine how robust these results are. We use the $1\sigma$ values of the observed (FAME) minus expected (MARGO), as computed by Roche et al. (2018) with the MARGO core top $\delta^{18}O_c$ database, as the potential error associated with the FAME model. Regions in which the difference between the two populations are larger than the potential error are associated with grey, whereas those less than the potential error as hashed regions (Figure 5A and B), these errors should be seen as a guide rather than a rejection of a site. Because the MARGO database does not contain *N. dutertrei* we have given the panels concerning this species a separate colour scheme, black represents grid-cells for which the two populations cannot be said to be statistically different, white grid-cells are those in which the two populations can be said to be statistically different (Figure 5C). As we do not have a similar way to calculate the error for $T_c$, FAME-$T_c$ results are shown (in Figure 6) with this same binary pattern (*i.e.*, white grid-cells are those in which the two populations can be said to be statistically different and black are those in which the two populations can be said to be similar). To reduce the complexity, the overlay of the species Anderson-Darling results (Figure 7) also uses the binary colour scheme (white or black).

Our results show that much of the Pacific Ocean can be considered to have statistically different population between $FP_{EN}$

and $FP_{NEU}$ for both $\delta^{18}O$ (Figure 5) and $T_c$ (Figure 6). We consider that the likely cause for such a remarkable result is due to FAME computing a weighted average and, therefore, the lack of a signal found exclusively within the regions demarked in Figure 1 as El Niño regions could represent how the temperature signal is integrated via an extension of the growth rate; growing season and depth habitat of distinct foraminiferal populations. Taking into account the FAME-$\delta^{18}O_c$ error for *G. ruber* and *G. sacculifer*, we have computed regions in which the difference in oxygen isotopes between the two populations ($\Delta\delta^{18}O_c$) compared with the AD-test is smaller than the aforementioned error (Hatching in Figure 5), *i.e.* where the mean difference between $FP_{EN}$ and $FP_{NEU}$ is within the error. The hatched regions in Figure 5 considerably reduce the areal extent of significant difference between $FP_{EN}$ and $FP_{NEU}$, with the remaining regions aligning with the El Niño 3.4 region (Figure 1). It is important to note that this error relates to the model and in reality, the difference between the climate states could be larger or smaller. No such test was performed on the *N. dutertrei* dataset, because of its absence from the MARGO dataset.

To further test the model-driven results and to assess if they are still consistent when the depth limitation is varied, the analysis was rerun with depths of 100, 200 and an extreme value of 400 m (Figure 5-7). Whilst it is possible to discern differences between the depths, it is important to note that a large percentage of the tropical Pacific remains accessible to palaeoclimate studies. A shallower depth limitation in the model increases the area for the 'warm' species, suggesting that the influence of a reduced variability in temperature or $\delta^{18}O_{eq}$ with a deeper depth limit causes the differences between $FP_{EN}$

and $FP_{NEU}$ to be reduced. Overlaying the results of the Anderson-Darling test for all three species (Figure 7) per depth for 60, 100 and 200 m highlights the areas where multi-species comparisons could be made. To account for potential differences in depth habitat we make a combination of shallower depth for *G. ruber* and deeper depths for *G. sacculifer* and *N. dutertrei* (Pracht et al., 2019) in the final panels (Figures 7D and 7H).

**3.3 Test of input parameters (fixed depth: temperature and $\delta^{18}O_{eq}$)**

The model-driven results were assessed with the underlying input dataset (temperature and $\delta^{18}O_{eq}$), these underwent the same statistical test (Figure 4), although with fixed depths of 5 m, 149 m and 235 m (see section 2.5). The results for each grid point are presented as either black, grey or hashed. Areas where the population distributions of the two climate states are found to be statistically similar have black grid cells. Regions in which the difference between the two populations are larger than the potential error are associated with grey, whereas those less than the potential error as hashed regions. The threshold error (i.e., the difference between the means of each distribution) is for temperature (Figure 4A-C) 0.5 °C and for $\delta^{18}O_{eq}$ (Figure 4D-F) 0.10 ‰, these errors should be seen as a guide rather than a rejection of a site. The results of this fixed depth, non-FAME, test show that the shallowest depths produce populations that are significantly different both in terms of their mean values and their distributions. In the upper panel of Figure 4, the canonical El Niño 3.4 region is clearly visible at 5 m depth. Whilst differences exist between Anderson-Darling results for the input data (Figure 4) and the FAME $\delta^{18}O$ (Figure 5) and $T_c$ (Figure 6), for instance close to the Panama isthmus, there are significant similarities between the plots. These plots also show that our FAME data (Figure 5-7), in which we allow foraminiferal growth down deeper than the depths in Roche et al. (2018), are a conservative estimate and thus are on the low-end (Figure 4), to account for potential discrepancies with depth habitats. In the original paper on depth habitats based upon temperatures derived from $\delta^{18}O_c$, Emiliani (1954) cautioned that the depth habitats obtained would represent a weighted average of the total population, and while foraminiferal depth habitats are likely to vary spatiotemporally, the average depth habitat is skewed toward the dominant signal (Mix, 1987).

**3.4 Water depth and SAR**

Our analysis uses reanalysis data for the time period 1958-2015, a hypothetical core that had a comparable resolution would essentially be analogous with a sediment core with a rapid sediment accumulation rate (SAR), representing 50 yr cm$^{-1}$ (or 20 cm kyr$^{-1}$). Based on our analysis, such a hypothetical core could allow for the possible disentanglement of El Niño related signals from the climatic signal, but only in a best-case scenario involving minimal bioturbation, which is unlikely in the case of oxygenated waters. Extracting the oxygen saturation (**SO₂**) state, of the Pacific Ocean bottom waters from the Annual Climatology WOA13 give values that are predominantly >40 % (Figure 9B). Oxygen saturation is the concentration of Oxygen in a medium against the maximum that can be dissolved in the same medium. Whilst annual variability may exist, it is unlikely that bioturbation would be prevented by low oxygen. Therefore, using a cut off value that has been considered sufficiently high enough to outpace bioturbation (*e.g.*, Koutavas and Lynch-Stieglitz, 2003) of 5 cm kyr$^{-1}$ (Figure 9A) it can be demonstrated that much of the Pacific has an inferred lower sedimentation rate (< 5 cm kyr$^{-1}$; Figure 9C) than this cut off value. To test the influence of bioturbation, the bioturbation simulator SEAMUS was run using the NGRIP time series. The results of SEAMUS highlight the potential single foraminifera depth displacement that low sedimentation rates can result in (Figure 9). Following the current available geochronological method (i.e., age-depth method) such specimens that are displaced in depth are assigned the average age of the depth that they were displaced to, which could result in erroneous interpretations of climate variability when analysis such as IFA is applied (Lougheed et al., 2018). The results of SEAMUS are plotted both as time series of the bioturbated 'NGRIP' signal (Figure 10) and as histograms of the probability of finding a particularly pseudo-foraminifera with a given age in the bioturbation depth (Figure 11). As the bioturbation depth varies between 5, 10 and 15 cm for the different simulations of SAR, the histogram in each panel (in Figure 11) represent different thicknesses of sediment, i.e., for Figure 11 panels a, d, g, and j histograms represent data with a BD thickness of 5 cm. Likewise, the timeseries is plotted with the discrete 1 cm depth median age; the median age of the bioturbation depth (Figure 11) is the reason why each timeseries does not 'start' at 0 age (Keigwin and Guilderson, 2009).

The variance within a single depth in a core largely represents the integrated time signal for that depth (Figure 11), as opposed to the variance of a climatic signal for an inferred (or measured) average age for the depths in question. The proxy variance will be based both upon a non-uniform distribution in temporal frequency of specimens, i.e., older specimens are few compared to younger specimens. A large proportion of the specimens in the BD come from years that are 'proximal' (i.e., close to the youngest age) this may give undue confidence that the probability of picking a specimen from these years is higher, however the long-tail of the distribution means that there is an equally high chance of picking a specimen that has come from several thousand years earlier than the discrete-depth's median age. If we consider for the moment this as picking specimens from a box, there is a high chance of picking from a single box that represents the age you want however there is an equally high chance of picking from numerous boxes with varying age. If the spread in the climatic variable is uniform throughout this time then it can be possible to reproduce a similar signal, although this would not by definition represent the actual spread in the actual climatic variable for a given time, however the spread in the climate variable is unlikely to be constant. With a varying spread in the climatic signal bioturbation can introduce the possibility of spurious interpretations, but it is of course more obvious where the measured distributions over-exaggerate the climate signal (e.g., Wit et al., 2013). Furthermore, if we consider that researchers do not pick as randomly as they profess, there is both a size and preservation bias to specimens selected, and size is not constant down-core (e.g., Metcalfe et al., 2015) we can further introduce bias within the dataset. The SEAMUS output that corresponds with our chosen SAR cut-off value of 5 cm kyr [-1] (Figure 10 and

11), the lower limit of our mask (Figure 9), is shown in panels in Figures 10H to 10J and Figures 11G to 11I. It is important to note however, that much of the region for which FAME is calculated upon has inferred sedimentation rates lower than this cut-off value (Figure 9C to 9H).

An additional factor in the post-mortem preservation of the oceanographic signal in foraminiferal shells is whether the shells can be preserved. The GEBCO bathymetry data is binned into 250 m wide bins, and the data normalised to 1.0. As the data contains both bathymetric and topographic (below and above 0 m), the grey area in each histogram represent > 0 m (Figure 8). Whilst, there are differences depending on the cut-off value (Figures 8C to 8E) much of the canonical El Niño 3.4 region (Wang et al., 2017) used in oceanography (Figure 1) is also excluded from these suitable areas. Overlaying the water depth and the SAR with the Anderson-Darling results (Figure 7) highlights that of the total area where $FP_{EN}$ is significantly different from $FP_{NEU}$ (i.e. those areas where planktonic foraminiferal flux is suitable for reconstructing past ENSO

dynamics), only a small proportion corresponds to areas where the sea floor is both above the CCD (< 3500 mbsl) and SAR is at least 5 cm/ka (Figure 9). However, at certain locations, near islands or seamounts, the SAR and water depth may be high enough to allow for a signal to be preserved (Figure 8B) that may not be represented here.

**4. Discussion**

**4.1 From Life to Sedimentary Assemblages**

Whilst we are principally interested in understanding whether living foraminifera can theoretically reconstruct ENSO, comparison with data requires an additional analysis. This is because data-model comparisons are subjective, nominally supposing that the data is the value to be achieved by the model. However, if the foraminifera modulate the original climate signal, then preservation selectively filters which specimens are conserved whereas bioturbation acts to reorder, transposing the order in which they are recovered from the depth domain. Once the sediment is recovered, the researcher acts as a final filter, which is in essence a random picking. Although technically most researchers will pick whole shells so alongside size selectivity (*e.g.*, Metcalfe et al., 2015) there is also preservation bias associated with picking of foraminifera (*e.g.* Koutavas and Lynch-Stieglitz, 2003). Whilst the presence of depths in the ocean whereupon calcite is absent from sediments was described in the earliest work (e.g., Murray and Renaud, 1891; Sverdrup, 1942), overlaying maps of measured surface sediment carbonate percentage with water depth in the Pacific Ocean led Bramlette (1961) to coin the term 'compensation depth' (Wise, 1978). This work highlighted the 'narrow' depths (4-5000 m) in the Central Pacific of the CCD. Conceptually Berger (1971) placed three levels in the Pacific ocean that were descriptive of the aspects (e.g., chemical, palaeontological and sedimentological) of the calcite budget; the saturation depth, demarking supersaturated from undersaturated; the lyscoline, the depth at which dissolution becomes noticeable (Berger 1968, 1971); and compensation depth (Bramlette, 1961), in which supply is compensated through dissolution. The aspects of the lysocline was estimated by the faunal assemblages of Parker and Berger (1971, figure's 14 and 15 of that publication), for much of the equatorial Pacific the lysocline is estimated at ~3800 m. As the lysocline is where dissolution becomes apparent, ergo it is a sample already visibly degraded, we therefore set the limit of the water depth mask shallower, at 3500 m bsl. In fact, in regions of high fertility, such as the Eastern Equatorial Pacific, the lysocline was estimated to be present at ~2800 m (Thunell et al., 1981) or ~3000 m (Berger, 1971; Parker and Berger, 1971). For instance, core V21-28 close to the Galapagos Islands (01°05'N, 87°17'W) has a shallower dissolution affect than either of these two values despite being collected from a water depth of 2714 m (Luz, 1973). A comparison between the hydrographic and sedimentary lyscoline, using a mooring in the Panama Basin showed that the sedimentary lyscocline is a product of where the hydrographic lyscocline meets the seafloor (Thunell et al., 1981), therefore, this could lead to dissolution within the water of the settling flux (e.g., Schiebel et al., 2007). In the EEP region the shallower lyscoline is accompanied by an equally shallower CCD (located at ~3600 m) for which the highly fertile is considered responsible for its shoaling, lowering the pH through increased $CO_2$ (Berger et al., 1976). The correspondence between lyscoline depth and CCD depth does not hold true for the entirety of the Pacific, plotting a N-S cross-section from

50°N to 50°S Berger (1971) noted that in the Central Equatorial Pacific, the high fertility region generates a larger zone of dissolution resistant facies even with a shoaled lysocline. If we factor in the sedimentation rate of the Pacific, which has been estimated to be considerably lower than 1 cm (Blackman and Somayajulu, 1966; Berger, 1969; Menard, 1964), then dissolution may become further exacerbated. The longer a shell remains at the sediment-water interface the greater the prospects for it to be dissolved become, therefore low SAR increases the chance of dissolution (Bramlette, 1961). For instance, in 15 equatorial Pacific cores, below 4000 m, the average SAR was presented (Hays et al., 1969; here calculated) at 0.96 cm kyr$^{-1}$ (1σ ± 0.43 cm kyr$^{-1}$). Although there are regions and/or core locations in which the SAR is higher, for instance eight EEP cores shallower than the lysocline depth (Thunell et al., 1981) of ~2800 m were presented by Koutavas and Lynch-Stieglitz (2003) which have an average SAR, calculated at 7.20 cm kyr$^{-1}$ (1σ ± 2.82 cm kyr$^{-1}$). The average age for these same core's 0 cm core depth is 2184 years (1σ ± 1521 yrs), whilst it cannot be assumed that there has been no loss during recovery (i.e., core top is not sediment-water interface), a non-zero core top age is expected for both bioturbation (Keigwin and Guilderson, 2009) and dissolution. Alongside, the potential for dissolution there is the also the mixing of ocean sediments by the benthos (Bramlette and Bradley, 1942). For instance, Arrhenius (1961) noted that ash beds present in cores of the EEP (Worzel, 1959; Ewing et al., 1959) had a 2-3 cm layer above and below what should have originally been a sharp boundary in which they estimated that ~50% of the material originated from the other side of the boundary. If one assumes 1 cm kyr$^{-1}$ sedimentation rates, then the range in age of the obviously 6 cm mixed sediments is minimally ~6000 years per cm, comparison with an analogous SEAMUS simulation (bioturbation depth 5 cm; SAR 1cm) highlights the considerable spread in age, placing the 95.45% range between 110 and 18954 years (Figure 11). Much of this temporal variability (either through bioturbation or dissolution) will be hidden, especially when proxy values correspond with the expected values, and more obvious when the values are larger than expected (*e.g.*, Wit et al., 2013). Owing to the lack of absolute variability during the Holocene the apparent confirmation of similarity between proxy values and modern distributions of the '*to be reconstructed*' variable is not a confirmation of proxy reliability. Especially in the tropics wherein seasonal variability is limited. The effects of both bioturbation and dissolution are further amplified when combined with finite sampling strategies. Therefore, the results of the sedimentological features, presented here, imply that much of the

Pacific Ocean is not suitable for preserving (Figures 7-9) the ENSO signal, despite the possibility of the species of foraminifera having unique values for different climate states (Figures 4-7). ENSO studies using palaeoceanography have exposed shifts, one can, therefore, question what is being reconstructed in such studies.

**4.2 Palaeoceanographic Implications**

**4.2.1 Pacific climate reconstructions**

[revised manuscript text omitted]

ENSO conditions; as well as what the proxy is recording. There is also the presumption that a particular climate event should be recorded, our Anderson-Darling test for instance highlights that there are locations that cannot discern the difference between El Niño and other climate states whilst for the same time period there are locations where the different climate states can be differentiated. Whilst our analysis is a statistical treatment of the data, each species, and different types of phyto- or zooplankton preserved in ocean sediments, are likely to record the same set of environmental conditions differently (Mix, 2006). This is, in brief, the rationale for the development of FAME, the same climate signal seen through the view of species-specific proxies will give a fractured view constrained by each species particular ecophysiological constraints (Mix, 1987; Roche et al., 2018). A dynamic depth habitat in which the environmental signal becomes a weighted average of the water column can further confound the original signal (Wilke et al., 2006). What can be seen as contradictory reconstructions can therefore be viewed as the prevailing or dominant conditions at a given location at the time when environmental conditions overlap ecological constraints for a given species.

Terrestrial records suggest the number of El Niño events per century in the early Holocene (8-6 ka BP) was minimal (Moy et al., 2002), with between 0 and 10 events occurring per century. This dampened ENSO is observed within lake core colour intensity and records driven primarily by precipitation - although like other proxies this can also be interpreted differently, *i.e.* as a large change in the hydrological cycle shifting precipitation away regionally (Trenberth and Otto-Bliesner, 2003). If we assume for now that the number and magnitude of ENSO events was reduced, the relatively low downcore resolution of marine records may not accurately capture the dynamics of such lower amplitude ENSO events using existing methods. The sensitivity and probability of detecting a change in IFA with changes in frequency and amplitude, has been dealt with before (Thirmulai et al., 2013), although without considering bioturbation. The synthesis of pseudo-timeseries to discern the potential distribution for different scenarios, whilst a necessary approximation, is nonetheless one that is free of cause and causality. Modulating a timeseries for events with enhanced or weakened amplitude or fewer or greater number of events assumes in essence that there is limited feedback both regionally (between two sites) and internally within the timeseries (i.e., a process that operates on a higher level). Reconstructions of the past can benefit from inclusion within conceptual frameworks that incorporate both data and modelling studies (e.g., Trenberth and Otto-Bliesner, 2003; Rosenthal and Broccoli, 2004; McPhaden et al., 2006). The use of coupled ocean-atmosphere models (e.g., Clement et al., 1999; Zebiak and Cane, 1987); isotope enabled Earth system models (e.g., iCESM; Zhu et al., 2017); or multi-model ensembles with prescribed boundary conditions can be used for the generation of timeseries in which the physics of atmospheric and oceanic circulation are constrained and feedbacks between sites can occur. The perceived failure of several climate models to resolve ENSO adequately, resulting in variable ENSO frequency and amplitude between models, could therefore be used to determine the proxy signal from model derived timeseries at different frequencies and intensities of ENSO. Albeit a timeseries of variable ENSO that is grounded in ocean-atmosphere coupling. Such analysis could also provide information on a secondary assumption, in which time slices from the same core inherently assume that where a particular oceanographic feature exists now is also where it may have existed before. This gives a somewhat binary view, the feature either occurs or does not occur, and if it occurs then it has either enhanced or weakened. Yet this can (though not always) preclude a scenario in which the feature has shifted. Analysis of the El Nino patterns suggests that there are two types of El Nino that are spatially delineated: the dateline Central Pacific El Nino and the Eastern Pacific El Nino. The expansion, contraction or shift of certain large scale oceanographic features (e.g., Polar Front, Upwelling) during periods of warmer than average (e.g., the last interglacial) or colder than average temperatures (e.g. the LGM) can complicate the comparison of two down core samples, i.e., a static core continuously recording a particular climate event as opposed to a shifting oceanographic regime 
[revised manuscript text omitted]
 the integrating the $\delta^{18}O_{eq}$ values using a growth-rate based weighting (FAME; Roche et al., 2018). Values are in per mil (‰ V-PDB).**

**Figure 4. Anderson-Darling Results for Input datasets of Temperature and Equilibrium $\delta^{18}O$ ($\delta^{18}O_{eq}$). Results of the test in which input variables underwent the same statistical procedure (see section 2.0) as the modelled data for (A-C) temperature and (D-F) $\delta^{18}O_{eq}$ values. Here, model input data was extracted for three fixed depths ([A & D] 5 m; [B & E] 149 m; [C & F] and 235 m) without any growth weighting applied. Black regions are those grid points in which the null hypothesis ($H_0$), that the El Niño and Non- El Niño populations are not statistically different ($FP_{El\ Niño} = FP_{Non-El\ Niño}$), cannot be rejected. Gray regions represent grid points where the $H_1$ hypothesis is accepted, therefore the distributions of the foraminiferal population (FP) for El Niño and Non-El Niño can be said to be unique ($FP_{El\ Niño} \neq FP_{Non-El\ Niño}$). The hatched regions represent areas were the $H_1$ hypothesis can be accepted, therefore the distributions of the foraminiferal population (FP) for El Niño and Non- El Niño can be said to be unique ($FP_{El\ Niño} \neq FP_{Non-El\ Niño}$), though the difference between the means of tested distribution are less than (A-C) 0.5°C or (D-F) 0.1 ‰. Each panel represents a single depth (5, 149 and 235 m).**

**Figure 5. Anderson-Darling Results for modelled FAME-$\delta^{18}O_{eq}$: Panels representing locations of where dissimilar and similar values of FAME modelled species $\delta^{18}O$ occur between climate states, for (columns) particular species and (rows) particular model depth cut-off limits. Each panel represents the Anderson-Darling test result, which are plotted with ([A]** *Globigerinoides sacculifer* **and [B]** *Globigerinoides ruber*) **and without ([C]** *N. dutertrei*) **model derived error. For all panels black areas reflect latitudinal and longitudinal grid points that failed to reject the null hypothesis ($H_0$) and therefore the foraminiferal population (FP) of the El Niño is similar to the Non-El Niño (FP$_{El\ Niño}$ = FP$_{Non-El\ Niño}$). The results in which the $H_1$ hypothesis is accepted, in which the, therefore the distributions can be said to be different (FP$_{El\ Niño}$ ≠ FP$_{Non-El\ Niño}$), are plotted as either: (A –** *G. sacculifer*, **B –** *G. ruber*) **grey and hatched or (C –** *N. dutertrei*) **solely as white regions. For species with calculatable error, grey regions represent values where the difference between the two means of the population is greater than species-specific standard deviation of the FAME model and hatched regions represent those in which the means are less than this standard deviation (Roche et al., 2018). For species without a calculatable error, the regions are plotted in white. The rows represent the model runs with a depth cut-off limit at: (A-C) 60 m; (D) 100 m; (E) 200 m; and (F) 400 m.**

.

**Figure 6. Anderson-Darling Results for modelled FAME-$T_c$:** Panels representing locations of where dissimilar and similar values of FAME modelled temperature recorded in the calcite shells (Tc) occur between climate states, for (columns) particular species and (rows) particular model depth cut-off limits. Each panel represents the Anderson-Darling test result, which are plotted with ([A] *Globigerinoides sacculifer* and [B] *Globigerinoides ruber*) and without ([C] *N. dutertrei*) model derived error. For all panels black areas reflect latitudinal and longitudinal grid points that failed to reject the null hypothesis ($H_0$) and therefore the foraminiferal population (FP) of the El Niño is similar to the Non-El Niño, and therefore the distribution between the neutral climate and El Nino cannot be said to be different ($FP_{El Niño} = FP_{Non-El Niño}$). The results in which the $H_1$ hypothesis is accepted, in which the distributions can be said to be different ($FP_{El Niño} \neq FP_{Non-El Niño}$), are plotted as white regions. The rows represent the model runs with a depth cut-off limit at: (A-C) 60 m; (D) 100 m; (E) 200 m; and (F) 400 m.

**Figure 7. Combined A-D plots. As figure 5 and figure 6, in that panels represent locations of where dissimilar and similar values for the two climate states for (a-d) FAME-$\delta^{18}O_{eq}$ modelled oxygen isotope values or (e-h) FAME-$T_c$ modelled temperature recorded in the calcite shells (Tc) occur. Each panel represents the Anderson-Darling test result, the results for *Globigerinoides sacculifer*, *Globigerinoides ruber* and *N. dutertrei* are overlaid. For all panels black areas reflect latitudinal and longitudinal grid points that failed to reject the null hypothesis ($H_0$) and therefore the foraminiferal population (FP) of the El Niño is similar to the Non-El Niño, and therefore the distribution between the neutral climate and El Nino cannot be said to be different ($FP_{El\ Niño}$ = $FP_{Non-El\ Niño}$). The results in which the $H_1$ hypothesis is accepted, in which the distributions can be said to be different ($FP_{El\ Niño} \neq FP_{Non-El\ Niño}$), are plotted as yellow where the depth is deeper than 3500 m bsl or purple where the depth is shallower than 3500 m bsl (see Figure 8). Purple locations are where our results suggest that the signal of ENSO has different values and the water depth allows for preservation – although this purple region will be smaller when inferred SAR is taken into account (see Figure 9). The rows represent the model runs with a depth cut-off limit at: (A and E) 60 m; (B and F) 100 m; (C and G) 200 m; and (D and H) where a combination of depths were utilised (Pracht et al., 2019).**

**Figure 8. Bathymetric map of the Tropical Pacific Ocean highlighting the areas above and below the Lysocline and/or Calcite compensation depth (CCD). (A) GEBCO map of height relative to 0 m; (B) same as (A) with location of seamounts plotted (white stars); (C-E) binary colour map of GEBCO data, yellow is values below cut-off depth value ([C] 3500 m below sea-level (bsl); [D] 4000 m bsl; and [E] 4500 m bsl respectively) and purple above the cut-off depth value. The histograms represent the normalised frequency of grid cell height in bins of 250 m wide, yellow is values below cut off value ([C] 3500 m below sea-level (bsl); [D] 4000 m bsl; and [E] 4500 m bsl respectively), purple above cut off value. The grey bins in each histogram are those above 0 m.**

**Figure 9. Map of the sedimentation rate and oxygen saturation for the Tropical Pacific. (A)** Inferred sedimentation rate (Olsen et 2016). White regions represent continental shelf. **(B)** Oxygen saturation of the bottom grid layer of World Ocean Atlas 2013 (data from: https://www.nodc.noaa.gov/cgi-bin/OC5/woa13/woa13oxnu.pl ). **(C, E, G)** Overlay between water depth and inferred SAR, Red / Pink: Continental shelf sediments that are (Red) shallower or (Pink) deeper than 3500 mbsl; Gray / White: grid point SAR is lower than SAR threshold and the seafloor depth is (grey) shallower or (white) deeper than 3500 mbsl; Light Yellow/Gold: Light yellow represents areas where the SAR is above the threshold but the water depth is deeper than 3500 mbsl in comparison Gold represents areas where the SAR is above the threshold and the water depth is deeper than 3500 mbsl. The ideal locations are therefore plotted as Gold. Cut-off limits for SAR are **(C)** $\geq$1 cm kyr$^{-1}$; **(E)** $\geq$2 cm kyr$^{-1}$ and **(G)** $\geq$5 cm kyr$^{-1}$, **(D, F, H)** alongside the maps the bioturbation simulations for the minimum SAR threshold is plotted (see Figure 10 and Figure 11 for the output of SEAMUS). Each plot gives the input values of NGRIP (grey) and for each SAR three analysis were performed with different bioturbation depths (BD) these are (Blue) 5 cm; (Green) 10 cm; and (Orange) 15 cm.

**Figure 10.** Output of the bioturbation model SEAMUS. (A) The unbioturbated input signal, NGRIP (North Greenland Ice Core Project Members, 2004; Rasmussen et al., 2014; Seierstad et al., 2014), used in our simulation of bioturbation for different SAR with SEAMUS (Lougheed, 2019). Sediment mixed layer referred to here as bioturbation depth (BD) is fixed at (B, E , H, K) 5 cm, (C, F, I, L) 10 cm and (D, G, J, M) 15 cm for sedimentation accumulation rates (SAR) of (B-D) 1 cm kyr$^{-1}$; (E-G) 2 cm kyr$^{-1}$; (H-J) 5 cm kyr$^{-1}$ and (K-M) 10 cm kyr$^{-1}$. The output is plotted as the discrete 1 cm depth median age. In (B-M) grey values represent the unbioturbated input signal, NGRIP. Note, we retain the original units (V-SMOW) of the original timeseries used, no inference between Pacific climate and Greenland is intended by the use of NGRIP (see section 2.7).

**Figure 11. Histograms of simulated specimen age within the bioturbation depth. The simulated age distribution present within the sediment mixed layer, referred to here as bioturbation depth (BD). BD is fixed at (A, D, G, J) 5 cm, (B, E, H, K) 10 cm and (C, F, I, L) 15 cm for sedimentation accumulation rates (SAR) of (A-C) 1 cm kyr[-1]; (D-F) 2 cm kyr[-1]; (G-I) 5 cm kyr[-1] and (J-L) 10 cm kyr[-1]. The output is plotted as the discrete 1 cm depth median age. Note the size of the BD varies, therefore the simulated age distribution comes from a varying 'core depth'.**

[revised manuscript text omitted]

---

## Author Response (AR2)

**Private correspondence to the Editor: NOT FOR PUBLIC PEER REVIEW FILE**

In my response to reviews (for the eventual public peer review) I have ignored the tone of comments that are within the appended pdf and the review in general of Reviewer 5 since they are not constructive and generally quite abrasive. For example,

Pg. 2 *the present structure is extremely indigestible, and squanders any goodwill that the reader might still have after reading that pompous introduction;*

Pg. 10 *for heaven's sake, everyone knows this.;*

Pg. 21 *Without treatment yes, but what idiot does it that way? I don't know a single climate scientist dumb enough to compare models and data literally;*

Pg. 26 *dear lord, again?;*

Pg. 27 *one more for the road; etc. etc.*

Pg. 2 *I am surprised that experienced scientists like Didier Roche or Claire Waelbroeck let the paper be submitted once, let alone four times, with such flaws.*

I bring them up here, though, because they also aren't professional, and I'm surprised that the reviewer has felt that this is appropriate. Especially, the following:

Pg. 2 *I am surprised that experienced scientists like Didier Roche or Claire Waelbroeck let the paper be submitted once, let alone four times, with such flaws*

which crosses the line into what can be considered an ad hominem attack. I am a quite an experienced scientist (started PhD in 2009, defended in 2013), and yet I found that these comments somewhat discouraged even me. However, I am encouraged by my co-authors to press on. I do worry, however, about how such a review would be received by a more easily discouraged / less experienced author, perhaps making their first submission, who may think that comments like these are typical of either the publication process, Climate of the Past, Copernicus or the EGU. Frustration at the author is not an excuse, we all read papers, theses, student submissions, et cetera where we would like the author to have done things differently, but it is our job as reviewers to tone our criticism in a way that is constructive and treats the work sent to us with a modicum of respect (even if we disagree about the science or the conclusions). Whilst I greatly value the open-discussion aspect of the journal, , can I suggest that future reviews for Copernicus from this reviewer be checked?

**Editor comments**

Comments to the Author: I have obtained two further reviews of your paper, both from referees who were not involved in the first round of reviewing. You will be relieved to know that both reviewers, in contrast to the previous rounds, do see merit in your approach - this means we have passed one barrier to publication. Referee 4 does not go into great detail after that, considering that your paper has already received a lot of comments. However referee 5 is very unhappy with the style of the paper and its readability, to the extent that they recommend rejection.

In view of the fact that we have votes in favour of the principle of the paper, and that I generally agree that this does seem like a useful approach, I have decided to let you proceed with a further revision. However I also agree with referee 5 that the paper is extremely tough going. As well as the review, referee 5 provided an annotated pdf with further comments and I am hoping Copernicus will load this for you to see soon. However the paper definitely needs yet another rewrite to make it easier to follow.

*The figures are called in a bizarre order (as far as I could see it went Fig 1, then 8 and 9, then in practice 5 and 7).

* The intro and methods are not too bad, but once we reach section 3, the paper needs to be structured in a logical and linear fashion so that the reader is guided from figure to figure and told what the conclusions from each one are. this is simply not the case at present and I could barely follow what you were doing.

*Additionally the density and number of figures is daunting. I strongly recommend that you strip down to a smaller number of essential panels and figures and put the rest into the supplement.

I realise this paper has already been through numerous iterations, but I'm afraid it is essential to make further improvements if it is to be accepted. I hope you will be able to achieve this.

**Reply to Editor**

Thank you for considering the value of our manuscript and sending it out for further review.

As suggested by yourself and Anonymous Reviewer 5 we have altered the layout / structure, in doing so we have also reduced the amount of text. We have reduced the number of panels in the figures because we have split the paper into five 'experiments' or 'mini papers', but the number of figures is still 9 (but with less panels per figure).

**Reply to Reviewer 4: M. Kucera**

Considering the original manuscript, the reviews and the response to the comments, I believe much of the criticism of the manuscript has been misdirected. The reviewers seem to wish that the manuscript would have dealt with a technique to reconstruct ENSO from fossil foraminifera. Anything else they consider not useful. I fear that this view is too simple and I concur with the statement that the authors have made in defense of their study:

"We seek to understand whether or not foraminifera populations in the water are intrinsically capable of recording ENSO dynamics, and at which locations. We consider this to be the most fundamental consideration for foraminifera-based ENSO studies….. Assessment of [sedimentary records] essentially becomes a moot point if the events to be reconstructed are not recorded by foraminifera in the water [in the first place]."

I cannot see anything in the statement that should not be true and contrary to the referees, I see much merit in the concept of preceding complex paleoceanographical interpretations by feasibility assessments of this kind. For this reason, I believe the revised manuscript can be considered for publication.

**We thank the reviewer for the time in reviewing our manuscript.**

Realizing that this is a contentious issue, I offer some additional suggestions:

- The title still does not reflect the essence of the approach. Foraminifera do not record ENSO, they record seawater properties that change in response to ENSO. I recommend amending the title towards something like: "Proxy modelling approach to assess the potential of extracting past ENSO signal from planktonic foraminifera proxies ..." or "...that ENSO-related patterns are recorded..."

**We thank the reviewer for suggesting alternative titles and have gone with their first suggestion.**

- Neither the abstract nor the conclusion acknowledge sufficiently the key assumption of the approach: that the FAME model is able to robustly predict the seasonality and the vertical dimension of foraminifera habitat. The author mention this issue clearly (e.g., in section 4.2), but what is missing is a qualifier in the statement of the result in the abstract and conclusion "provided the assumptions of the model are correct, our results indicate that….".

**We have added into the text qualifiers that the reviewer seeks. We agree that the model relies upon a set of assumptions, which do not necessarily reduce the usefulness of the model.**

- This is not to say that I believe that the question of the validity of the model discredits the papers. I do have my doubts about the robustness of a niche model that is based exclusively on temperature, but I also note that is not really critical for the value of this study. If formulated correctly then all it does is showing how, assuming it is valid, an explicit forward model can be used a-priori to test if it is plausible that the given recording system can record an oceanographic signal to allow robust reconstructions. This, I believe, is the way forward in paleoceanography and it should be emphasized more in the manuscript.

Line 19 (commented text) should read vaterite, not verite

**Section removed from paper.**

**Reply to Reviewer 5.**

**Structure: Rearranged format / Length**

We thank the reviewer for their suggestion regarding the format and have taken this onboard, modifying the text accordingly. Each section is now presented as a self-contained mini-study. We also agree with the reviewer that by combining multiple questions our text has become unwieldy, something we ourselves alluded to in the first round of review. We appreciate the reviewer for giving our paper the consideration to read it afresh ("*As a prelude, I note that the paper has received prior reviews, which I refrained from reading to avoid biasing my judgment.*"). As they themselves may have guessed (" *I apologize in advance if the following comments are redundant, or if they contradict the recommendations of previous referees – I know firsthand that one cannot please every referee*") a number of their comments were in fact referring to the previous reviewers suggestions and modifications which lengthened the manuscript. We apologize for the over emphasis of certain points which may have become tedious to the reader ("*though the tediousness of the exposition leads the authors to belabor obvious points at the expense of critical explanations*" ; "Pg. 14 Needlessly tedious. Cut to the chase!"; " Pg. 9 So why mention it at all?")

**Grammar and Figures**

We have altered the text and figures accordingly.

**Choice of distribution fit**

> Pg.11 *If your results depend on the choice of kernel, you are in deep, deep trouble* ;
> Pg. 11 *While the choice of kernel is typically unimportant, the fact that it is so awkwardly justified raises a red flag.*"

A statistical test that includes a fitted distribution will of course add an assumption to any results regarding the fit of the distribution. There are multiple kernel distributions, whilst it is unimportant which kernel (as we stated to the previous reviewers), in hindsight we felt it was wise to include the full title so that other researchers can (if they wish) replicate our work. Therefore, we are not justifying an Epanechikov kernel over other kernels ('*fail to see how this justifies an Epanechikov kernel, as opposed to any other kernel*') but over other types of distributions (beta, gamma, gaussian, etc.). Our justification for seeking a generic fit was added at the behest of the previous round of reviews, the aim of the paper is not to test which distribution best fits ocean parameters and therefore a generic fit was sought.

Matlab's fit distribution "`fitdist` uses a normal kernel smoothing function and chooses an optimal bandwidth for estimating normal densities, unless you specify otherwise" (Matlab website).

**QQ-plots**

> "*Finally, and though I would be the last reader to request that the paper get any lengthier, I am surprised that the authors did not focus on the most obvious Achille's heel of IFAs, as practiced, for instance, by White et al. [2018]. Contrary to the author's claim, there is nothing inherently wrong with using quantile-quantile plots to compare distributions – indeed it can be a fine idea. The one issue with QQ plots as applied in studies like White et al. [2018] is that it is a handful of extreme values that determine the slope, making the results extremely brittle to outliers. To my mind, this is the most urgent statistical issue to address about the way IFAs are currently presented.*"

Our study seeks to test the presence of water conditions that would be favourable for recording of ENSO by foraminifera populations in the water. We also briefly investigate if such parts of the ocean water coincide with areas of high sediment accumulation rate deep-sea archives or not. As the reviewer points out, our manuscript already covers

a lot of ground. We are of course aware of a number of other challenges associated with sediment based IFA reconstructions, such as the interpretive limitations of QQ plots when combined with limited sample sizes. Whilst we (generally) agree with the reviewer, after the previous rounds of review we felt that the manuscript was already long enough.

We are actually working on a manuscript that focuses on the recording of ENSO in the sediment domain on millennial timescales (i.e., regarding QQ plots and other aspects), but it would be much too much to fit into this current manuscript, which already covers substantial ground (and focuses on decadal timescales)

**Rounds of review: Fourth submission?**

Finally, the reviewer suggests that this is the fourth submission, as opposed to the second round of review, this wouldn't matter if it wasn't in part their rationale for rejection:

> "*I am surprised that experienced scientists like Didier Roche or Claire Waelbroeck let the paper be submitted once, **let alone four times**, with such flaws*"

> "*After what looks **like 4 trials**, it is unclear that another round of revisions could fix these fatals flaws.*"

> "*In summary, this paper is not appropriate for publication in present form, and it is unclear if, **after this many trials**, it can ever be brought up to that standard.*"

**References suggested by the reviewer:**

Whilst the reviewer suggested the following references, we have altered the text, so that some do not fit into the revised manuscript. Or they are redundant.

D. Khider L. D. Stott, J. Emile-Geay, R. Thunell, D. E. Hammond, 2011. Assessing El Niño Southern Oscillation variability during the past millennium. Palaeoceanography and Palaeoclimatology, https://doi.org/10.1029/2011PA002139

D. Khider G. Huerta C. Jackson L. D. Stott J. Emile-Geay, 2015. A Bayesian, multivariate calibration for Globigerinoides ruber Mg/Ca. $G^3$, https://agupubs.onlinelibrary.wiley.com/doi/full/10.1002/2015GC005844

S. Dee J. Emile-Geay M. N. Evans A. Allam E. J. Steig D.M. Thompson, 2015. PRYSM: An open-source framework for PRoxY System Modeling, with applications to oxygen-isotope systems. Journal in Advances in Modelling Earth Systems, https://agupubs.onlinelibrary.wiley.com/doi/full/10.1002/2015MS000447

M. Comboul and J. Emile-Geay, 2015. Paleoclimate Sampling as a Sensor Placement Problem. AMS, https://doi.org/10.1175/JCLI-D-14-00802.1

Evans, M. N., S. E. Tolwinski- Ward, D. M. Thompson, and K. J. Anchukaitis (2013), Applications of proxy system modeling in high resolution paleoclimatology, Quaternary Science Reviews doi:10.1016/j.quascirev.2013.05.024

**Comments from Reviewer's annotated PDF file:**

**NOTE: Some comments by the anonymous reviewer appear to be 'off the cuff' remarks and therefore we have left them without reply.**

Reviewer comments are in red, *the text they are referring to in italic* **and our reply in bold.**

Pg. 2 Proustian, and neither complete nor readable

**This section was removed to streamline the text.**

Pg. 2 What do you mean?

Sentence referred to: *Yet, the simulation of past ENSO using climate models has been fraught with difficulties due to the associated feedbacks of ENSO upon model boundary conditions (e.g., SST, pCO2) (Ford et al., 2015).* – **ENSO as a source of the largest climate variability is an intrinsic component of the climate system, makes it a complex feature to model.**

Pg. 3 Surely these two studies do not have a monopoly on model-data comparison? Why single them out?

**This section was removed to streamline the text.**

Pg. 3: Also, Khider et al 2011: https://agupubs.onlinelibrary.wiley.com/doi/full/10.1029/2011PA002139

**This section was removed to streamline the text.**

Pg. 3 Period

**This section was removed to streamline the text.**

Pg. 3: "Bijective" is mathematically pedantic. "One-to-one mapping" will be just as accurate, and much easier to understand.
**The reviewer could have just said simplify to one-to-one mapping. This section was removed to streamline the text.**

Pg. 3 Add Khider et al. 2015  https://agupubs.onlinelibrary.wiley.com/doi/full/10.1002/2015GC005844

**This section was removed to streamline the text.**

Pg. 4: This is actually called a proxy system model, and the canonical reference is Evans et al 2013: Evans, M. N., S. E. Tolwinski- Ward, D. M. Thompson, and K. J. Anchukaitis (2013), Applications of proxy system modeling in high res- olution paleoclimatology, Quaternary Science Reviews, 76 (0), 16�28, doi:10.1016/j.quascirev.2013.05.024.

Sentence referred to: *Recent attempts at circumnavigating proxy related problems have employed isotope-enabled models (Caley et al., 2014; Roche et al., 2014; Zhu et al., 2017a), proxy models (Dolman and Laepple, 2018; Jonkers and Kučera, 2017; Roche et al., 2018) or uncertainty analysis (Thirumalai et al., 2013; Fraass and Lowery, 2017; Dolman and Laepple, 2018) to predict both the potential δ18Oc values in foraminifera and/or the probability of detection of a climatic event.*
**Altered**

Pg. 4: There's already a section 1.2

**Altered**

Pg. 4: Do you need to cite 5 papers for "sedimentation rate"? That seems to be a basic notion that does not need much defining.

**Altered. The interaction between sedimentation rate and bioturbation and their influence upon the temporal resolution of sediment archives is an often misunderstood principle.**

Pg. 5: Too many methods for too many questions. I suggest organizing by question, so readers don't have to constantly flip between 2 and 3.

**Altered**

Pg. 5: More precisely?

Sentence referred to: *The difference between the constant of Hut (1987) and the dynamic value (Brand et al., 2014) is minor.*

**Altered to streamline the text**

Pg. 5: Ecological, no? Otherwise the sentence makes no sense.

Sentence referred to: *Foraminifera as modelled entities has been developed as a tool for translating, a climatic input (typically a reanalysis dataset or climate model output) into a (simulated-) **climatic** signal, a signal that aims to approximate the depth integrated growth of foraminifera (e.g., Pracht et al., 2019; Wilke et al., 2006; Steindhardt et al., 2015).*

**Altered**

Pg. 5: Cite Dee et al 2015

Sentence referred to: *Data-model comparison studies suffer from an inability to directly compare like with like so that there are differences in (i) the units used i.e., most proxies reconstructing temperature do not give values of temperature in degrees °C or K but in their own proxy units (e.g., per mil ‰; mmol/mol; species abundance or ratio) necessitating a conversion; and (ii) there is a reduction in scales, i.e., models give a wealth of information (multiple*

*depth layers and high resolution time slices) in the time-depth domain.* - **This is basic information so does not need a citation defining it.**

Pg. 6: Does a poor job of explaining how FAME differs from competitors. Rephrase to better motivate this model.

Sentence referred to: *A number of models and modelling studies exist to determine the foraminiferal responses to present (Fraile et al., 2008, 2009; Kageyama et al., 2013; Kretschmer et al., 2017; Lombard et al., 2009, 2011; Roy et al., 2015; Waterson et al., 2016; Žarić et al., 2005, 2006) , past (Fraile et al., 2009; Kretschmer et al., 2016) and future (Roy et al., 2015) climate scenarios, FAME uses the associated temperature and δ18Oeq at each grid cell to compute a time averaged δ18Oc and Tc for a given species. FAME was produced as an attempt to reduce the error associated with data-model comparisons by (i) generating simulated-proxy time-series from model runs that can be compared with age-depth values down core; and (ii) to reduce the model information for a given timeslice into a manageable and relevant value using an integration that would make sense on a biological point of view (Roche et al., 2018).*
**We have added text throughout to better motivate this model**

Pg. 6. From
**Altered**

Pg. 6. Weight
**Altered**

Pg. 6. Comma
**Section altered**

Pg. 6: So why mention it at all?

Sentence referred to: *The MARGO database does not include N. dutertrei, meaning that we concentrate our efforts mainly on G. ruber and G. sacculifer.* – **Because it is a species that has been used for this type of study, therefore whilst we focus on the other two FAME has the option for more species than the MARGO database.**

Pg. 7: this seems to contradict what was just said above. Please clarify

Sentence referred to: *This was repeated four times, during which the lower depth limit of the growth rate computation was set to 60; 100; 200 and 400 m.* – **Redundant sentence removed**

Pg. 7: for heaven's sake, everyone knows this.

Sentence referred to: *The tropical Pacific Ocean is divided into four Niño regions based on historical ship tracks, from east to west: Niño 1 and 2 (0° to -10°S, 90°W to 80°W), Niño 3 (5°N to -5°S, 150°W to 90°W), Niño 3.4 (5°N to -5°S, 170°W to 120°W) and Niño 4 (5°N to -5°S, 160°E to 150°W).* – **Removed**

Pg. 8: If your results depend on the choice of kernel, you are in deep, deep trouble.

**-see comment at start of reply**

**Pg. 8** Prodigiously unclear

Sentence referred to: *Here, all values, i.e., the population, associated with a climatological state are compared with the other populations representing the different climatological state, the results plotted here are Neutral climate state vs. El Niño climate state.* – **Altered to 'For each test, comparison is made between all the values of one climatological state and all the values of another climatological state.'**

Pg. 8 You're not testing input data here, you're comparing to two other studies, one of which uses an isotope enabled GCM. but one would never guess from the text.

**The reviewer is referring to the following section:**

Sentence referred to: *2.5 Test of input data (Temperature and calculated δ18Oeq)*

*Foraminifera as modelled entities produces a modulated response that seeks to replicate how foraminifera modify the climate signal, several studies have approximated the foraminiferal signal in a different way (e.g., Thirmulai et al., 2013; Zhu et al., 2017a). In order to understand how FAME has altered the signal, and the degree to which the conclusions drawn depend upon the modelled growth rates, the input datasets of the sea water properties (Temperature and calculated δ18Oeq), underwent a similar statistical test (Figure 4). Unlike FAME, which integrates over several depth levels using the computed growth rate, the test of the input datasets was with fixed depths without any growth rate weighting. These fixed depths are 5, 149 and 235 m, giving a Eulerian view (Zhu et al., 2017a) in which to observe the implications of FAME's dynamic depth habitat. As per the FAME output, each timestep value was placed into its climate state and an Anderson-Darling test performed to compare the (dis)similarity of on the resultant distributions.*

**This section is about testing our input data and not comparing to two other studies (which we do later on in the paper). As stated in the section header we are testing the input dataset used for FAME (i.e.,** *In order to understand how FAME has altered the signal, and the degree to which the conclusions drawn depend upon the modelled growth rates, the input datasets of the sea water properties (Temperature and calculated δ18Oeq), underwent a similar statistical test (Figure 4)***) using the same statistical tests we use later in the paper for FAME.**

Pg. 8: Linear or not? the software is unimportant here - it's the method that counts

Sentence referred to: monthly anomalies and a **linear** trend removed (using the detrend function of MatLab 2019a) – the resultant data was left unfiltered (*i.e.*, Zhu et al., 2017a used a 1-2-1 filter) – **Yes linear, as already stated. We are referring to the detrend function of Matlab, so it is important to refer to which software we are using.**

Pg. 9: most cryptic
Sentence referred to underlined, however a larger section is required for our reply: *Four 'picking' experiments were performed, as FAME computes the average value for a given time step and given the single foraminiferal isotope variance for an equivalent time step (e.g., weeks: Steinhardt et al., 2015) it is more than likely that this computation reduces the real spread in values. Therefore, rather than use the terminology specimen we prefer to use months.*

Pg. 9 Incoherent grammar
*Although we note that parameterisation of misidentification would be difficult, as it requires understanding of the variability in both standard deviation and absolute values for species co-occurring downcore (Feldmeijer et al., 2015; Metcalfe et al., 2015; 2019).* - **Altered**

Pg. 9 Incoherent grammar
Sentence referred to: *Therefore, for each picked month between -0.40 and 0.40 ‰ is added to the picked month value (in intervals of 0.02 ‰), this is approximately ±2° C (i.e., ~4° C).* - **Altered**

Pg. 10 Enormous!

Sentence referred to: *Therefore, for each picked month between -0.40 and 0.40 ‰ is added to the picked month value (in intervals of 0.02 ‰), this is approximately ±2° C (i.e., ~4° C).* - **As we outline in our rationale this is equivalent to the range found in trap data, this could reflect the variation in absolute depth habitat experienced by individual foraminifera, slight variation in growth or other aspects of the individual (size, food, metabolism, etc.).**

Pg. 10 Incoherent
Sentence referred to: *Each picked month has their own randomly selected error for both of these errors, i.e., each value is the sum of the month picked and their own error.* - **Altered**

Pg. 10 But you're already using KDE, which is basically a interpolation over quantiles! So why interpolate on top of that?; This not a large computation compared to what is described above
**Section Removed**

Pg. 10 Focus on what you have, not the rest [strikethrough]
This is in contrast to other proxies such as corals (Cole and Tudhope, 2017), speleotherms (Chen et al., 2016) and molluscs (Butler et al., 2013; Milano et al., 2017), where distinct time-specific banding is present (true 'time-series' proxies). – **As suggested section removed**

Pg. 10 Incoherent Grammer
Sentence referred to underlined: *This is especially apparent in δ18Oc where there is a difference temporally of δ18Osw (e.g., the ice volume effect in glacial and interglacial cycles ~1.25 ‰) meaning that the same temperature can have radically different δ18Oc values, a consequence of this is that a series of high magnitude, but low frequency El Niño events could be disturbed in a discrete-depth record.*

Pg. 11 Needlessly tedious. Cut to the chase!
Section referred to: *Whilst our intention here is a generalised view to be used as an approximate guide, it is important to note that the Pacific Ocean has the largest proportion globally of >1 km tall seamounts that are smaller than <100 km (Wessel, 1997). Which may have important, relatively shallow-water sedimentary sequences, which may also be of sufficient sediment accumulation rate, therefore we supplement the GEBCO bathymetric data with the locations of seamounts. However, whilst there are an estimated 50,000 seamounts in the Pacific that are taller than a km (Menard, 1964; Wessel and Lyons, 1997), only 12,000 have been documented on charts (Batiza, 1982), and approximately 291 have been dated (Koppers et al., 2003; Clouard and Bonneville, 2005; Hillier, 2007). It is these 291, <1% of the estimated seamounts, we have overlain onto the bathymetric data (Figure 8b), although this number is further reduced as we only plot between 20°S and 20°N.*
**Previous reviewers requested the addition of seamounts, yet there is no easily accessible database of such a feature. We have condensed this section (leaving the references behind) as we agree.**

Pg. 11 Can you detail differences with TURBO2, with which I am more familiar?
**The reviewer/reader is invited to read a recently published paper (Lougheed, 2020) in GMD detailing SEAMUS, where the similarities and differences with TURBO2 are outlined in detail.**

Pg. 12 Incoherent grammer
Sentence referred to: To investigate how much temporal signal is integrated into discrete-depth intervals for typical tropical Pacific SAR, we, therefore, utilised the single foraminifera sediment accumulation simulator (SEAMUS, Lougheed, 2019) to bioturbate, as the input climate signal (Figure's 9 to 11)…

Pg. 12 Strange choice. Why not use simulated data, like SST from TraCE-21k? I understand the idea of using NGRIP as prototypical of a high-res paleoclimate record, but putting it in the middle of the Pacific is rather incongruous.

We wanted to use an oxygen isotope record, as there are global and regional changes which we discussed in the original manuscript. In the original MS and comments made in reply to the previous (first) round of reviewers we discussed that bioturbating a foraminifera between time intervals with different global d18Osw may alter the interpretation. As we state in this section this is the only record that is long enough and whose data is available.

Regarding the other point(s) as we had expected the reader might think this, which is why we had added a section. We refer the reviewer to this section already in the text: *The use of the NGRIP timeseries here is purely as an input parameter to investigate the effect of bioturbation upon a given climate signal - it is important to stress that by using NGRIP as an input signal for SEAMUS we are neither implying that tropical Pacific cores should have signal similar to NGRIP or inferring some kind of causal relationship.* **We are not putting an ice core in the middle of the Pacific. We are using it, as the reviewer themselves points out, as a high-resolution palaeoclimate record that can be bioturbated using SAR and BD values that are similar to the Pacific Ocean.**

Pg. 13 Incoherent grammar
**Altered**

Pg. 14 Yes, could be
*i.e.10 different 'months' picked between grid-points may exacerbate or accentuate differences.*

Pg. 14 At what level?
Pg. 14 Again, do this here, not the methods
**Don't understand what these comments are referring to**

Pg. 14 Oxymoronic
Sentence referred to underlined, however a larger section is required for our reply: *Therefore, understanding the biological variability on shorter timescale (e.g., Steinhardt et al., 2015; Mikis et al., 2019) which, maybe here over exaggerated, may be crucial for understanding discrepancies between cores.* – **Not an example of an oxymoron, if the reviewer is referring to over exaggerated.**

Pg. 15 Better to report the p-values and effect sizes, rather than whether p $<$ or $>$ 0.05
**We don't refer to p< or > here but to the test metric (not the p value)**

Pg. 15 Can you summarize what this shows?
**Section altered**

Pg. 16 Statistically?
**We agree that using the wording significant implies statistical correlation and have altered this section**

Pg. 16 What you are really testing here are the effects of dissolution and bioturbation
**Section 7 in the revised version now refers to this as 'Approximation of sedimentary archives'**

Pg. 17 And what? ("both" implies a second clause)
**Altered**

Pg. 17 That's what needs to be done, but figures are so messy i can't get to it.
**We have altered the figures**

Pg. 18 And that is where people would sample

Sentence referred to: *However, at certain locations, near islands or seamounts, the SAR and water depth may be high enough to allow for a signal to be preserved (Figure 8B) that may not be represented here.* – **We agree, but close to seamounts there may also be enhanced bioturbation as these underwater obstacles alter the circulation leading to resuspension or upwelling of nutrients. Likewise, the stability of the sediment on the seamount and potential for resuspension of older material may alter the preserved stratigraphy (through slumping, or winnowing).**

Pg. 18: Without treatment yes, but what idiot does it that way? I don't know a single climate scientist dumb enough to compare models and data literally.

Sentence referred to underlined, however a larger section is required for our reply: *Whilst we are principally interested in understanding whether living foraminifera can theoretically reconstruct ENSO, comparison with data requires an additional analysis. This is because data-model comparisons are subjective, nominally supposing that the data is the value to be achieved by the model. However, if the foraminifera modulate the original climate signal, then preservation selectively filters which specimens are conserved whereas bioturbation acts to reorder, transposing the order in which they are recovered from the depth domain. Once the sediment is recovered, the researcher acts as a final filter, which is in essence a random picking.* – **We are elaborating on all the factors that make the data potentially biased or erroneous, one can use a proxy system model to generate pseudo proxy time series so that a comparison between models and data can be made. The attempt to fit pseudo-data produced from model output to 'garbage data' is our point, rather than** literally **comparing data and models. In other words, we agree with the comment of the reviewer.**

Pg. 18 This review is off topic

**We have altered the text**

Pg. 19 Assuming you are right

Sentence referred to: *ENSO studies using palaeoceanography have exposed shifts, one can, therefore, question what is being reconstructed in such studies.*

Pg. 19 of what?
**Text altered**

Pg. 20 A point made originally by Thirumalai et al 2013

Sentence referred to: *our own analysis using the ratio of total to interannual variance also suggests that much of the variance in the simulated foraminiferal signal is dominated by interannual variance.*

Pg. 20 off topic
*4.2.2 The use of models in reconstructions*
**The previous reviewers wanted justification for such a model, we have altered this section in light of this round of review**

Pg. 20 Carbonate preservation
**This sentence is referring to the carbonate ion effect and not carbonate preservation**

Pg. 21 Agreed, but this is FAME, not this paper

Sentence referred to: *A dynamic depth habitat in which the environmental signal becomes a weighted average of the water column can further confound the original signal (Wilke et al., 2006).* **– I am not entirely sure what point the reviewer is making here. Wilke et al. show that the depth habitat of foraminifera can be approximated by a weighted average of the various calcifying depths rather than a specific water depth (the signal most researchers want to reproduce).**

Pg. 21 Gibberish. Makes zero sense
Sentence referred to**:** *The synthesis of pseudo-timeseries to discern the potential distribution for different scenarios, whilst a necessary approximation, is nonetheless one that is free of cause and causality. Modulating a timeseries for events with enhanced or weakened amplitude or fewer or greater number of events assumes in essence that there is limited feedback both regionally (between two sites) and internally within the timeseries (i.e., a process that operates on a higher level).*- **We have altered this section of text**

Pg. 21 Those won't have feedback btwn sites. That is not what climate models do
Sentence referred to**:** *or multi-model ensembles with prescribed boundary conditions can be used for the generation of timeseries in which the physics of atmospheric and oceanic circulation are constrained and feedbacks between sites can occur.*
**We have altered this section of text**

**Pg. 21** Need to back up with refs. This is a serious claim
Sentence referred to**:** *The perceived failure of several climate models to resolve ENSO adequately, resulting in variable ENSO frequency and amplitude between models, could therefore be used to determine the proxy signal from model derived timeseries at different frequencies and intensities of ENSO.*
**We have altered this section of text**

Pg. 22 How can something be "somewhat binary "? Are we in fuzzy logic territory?
Sentence referred to**:** *This gives a somewhat binary view, the feature either occurs or does not occur, and if it occurs then it has either enhanced or weakened.* **– Colloquialism that does not refer to fuzzy logic. Binary thinking of researchers to either something occurs or does not occur. Have altered**

Pg. 22: Unclear and verbose
Sentence referred to**:** *Yet this can (though not always) preclude a scenario in which the feature has shifted.*

Pg. 22 You are discussing optimal sampling design. This has been done before, though not for forams: https://journals.ametsoc.org/doi/10.1175/JCLI-D-14-00802.1
Sentence referred to**:** Climate models could therefore also be used to determine applicable core locations for comparison of proxy values with '*like with like*' oceanographic features (similar to the analysis of Evans et al. (1998) for predicting coral sites), without necessarily the cost of a time-slice project (e.g., CLIMAP, MARGO). **–I assume the point of this comment is that we should refer to Comboul and Emile-Geay, 2014? As we know it has not been done before hence why we refer to a paper on corals.**

**Pg. 22** Number already taken. Get your numbering straight!
Referring to**:** *4.2 Limitations of the methods applied and assessment of model uncertainties*
**Altered**

Pg. 22  Isn't LeGrande and Schmidt only for surface values, not subsurface?

Sentence referred to**:** *The spatial variability in salinity, particularly within regions underlying the intertropical convergence zone (ITCZ) and the moisture transport from the Caribbean into the eastern Pacific along the topographic low that represents Panama Isthmus, the resultant conversion of salinity to δ18Osw and then*

*δ18Oeq may contain further error.*
**There are some parameters for subsurface in LeGrande and Schmidt.**

Pg. 23: dear lord, again?

Referring to**: Whilst** the change in Mg/Ca with temperature has been validated (*e.g.*, Elderfield and Ganssen, 2000), the computation of a pseudo-proxy value for and from model parameters remains enigmatic.

Pg. 23: That is one enigmatic sentence

Sentence referred to**:** Whilst the change in Mg/Ca with temperature has been validated (*e.g.*, Elderfield and Ganssen, 2000), the computation of a pseudo-proxy value for and from model parameters remains enigmatic.

Pg. 24: The conclusions agree with my intuition, but the paper is too messy to back them up.

Review of

**Using a foraminiferal ecology model to test if tropical Pacific planktonic foraminifera are suitable recorders of ENSO, version 4**
by Metcalfe et al.

**Recommendation**: *Reject*

Summary

The manuscript uses a number of forward models to assess the possibility of recovering distinct ENSO states from individual foraminiferal analyses (IFAs). While the goal is excellent and the general approach sound, the presentation is so abstruse that it removes all credibility from the paper's claims. After what looks like 4 trials, it is unclear that another round of revisions could fix these fatals flaws.

**1 General Comments**

As a prelude, I note that the paper has received prior reviews, which I refrained from reading to avoid biasing my judgment. I apologize in advance if the following comments are redundant, or if they contradict the recommendations of previous referees – I know firsthand that one cannot please every referee.

**1.1 Statistics**

As said above, the general approach is sound, though the tediousness of the exposition leads the authors to belabor obvious points at the expense of critical explanations.

My biggest question mark is on the kernel density estimation (section 2.4). The authors use an Epanechikov kernel with the following justification *"The use of an Epanechnikov kernel distribution to fit the data, as opposed to other types of distribution, represents a trade-off between keeping as many parameters constant whilst mimicking the underlying dataset for a large number of grid points."*. I fail to see how this justifies an Epanechikov kernel, as opposed to any other kernel. While the choice of kernel is typically unimportant, the fact that it is so awkwardly justified raises a red flag. If your results depend sensitively on the choice of kernel, you are in deep, deep trouble. It would be important to include (as a supplement) an analysis with a different kernel choice (e.g. Gaussian).

The authors also mention a variable bandwidth, which is fine, but do not explain how it is chosen (e.g. Silvermann criterion). Given that the entire premise of the paper is to compare distributions, this is a crucial detail that needs to be better explained, possibly with a sensitivity analysis.

On the broader point of reporting the results of the Anderson-Darling test, the authors rely

exclusively on whether the p-values are above or below 5%. As emphasized by *Wasserstein and Lazar* [2016], the American Statistical Association explicitly warns against relying exclusively on p-values, and recommends additional metrics like effect sizes and confidence intervals. I think mapping effect sizes, possibly stippled to indicate whether the p-values are above or below 5%, would be better practice.

Finally, and though I would be the last reader to request that the paper get any lengthier, I am surprised that the authors did not focus on the most obvious Achille's heel of IFAs, as practiced, for instance, by *White et al.* [2018]. Contrary to the author's claim, there is nothing inherently wrong with using quantile-quantile plots to compare distributions – indeed it can be a fine idea. The one issue with QQ plots as applied in studies like *White et al.* [2018] is that it is a handful of extreme values that determine the slope, making the results extremely brittle to outliers. To my mind, this is the most urgent statistical issue to address about the way IFAs are currently presented.

**1.2 Structure**

The paper follows the classic structure of Introduction/Methods/Results/Discussion. The only issue here is that, because they consider 3 distinct questions, the methods are varied and lengthy, and by the time the reader gets to the end of Section 3, they have largely forgotten the relevant methods. It would seem more natural to me to structure one section per question, with relevant methods introduced where needed. This would look like: 1 introduction 2) distinguishing variance statistics 3) distinguishing distributions, 4) sensitivity to input parameters, 5) impacts of dissolution and bioturbation 6) discussion. One thing is for sure: the present structure is extremely indigestible, and squanders any goodwill that the reader might still have after reading that pompous introduction.

**1.3 Grammar**

I have reviewed dozens of papers over my career, but this one takes the prize for the most abstruse writing coming from native English speakers. A few times I had to look up whether some of the quirks might be differences between British and American English, but I could find no justification in any grammar book for spelling figures "figure's" (P14L30), for starting sentences by "Whilst" followed by a comma, for writing Proustian run-on sentences, or for being generally so incoherent that, after reviewing the paper on my iPad, the number one suggestion from my autocomplete is "incoherent grammar" (see annotated manuscript). I am surprised that experienced scientists like Didier Roche or Claire Waelbroeck let the paper be submitted once, let alone four times, with such flaws.

**1.4 Figures**

The figures are a piece of work. First, this is the first time I have seen figures so large that they make the text pages of the PDF look like microfilm. To add insult to injury, they are all of different sizes, making the document's navigation extremely tedious. The substance is no better than the style, unfortunately, as (apart from Fig 10), they are all so poorly designed that I would tell my students to redo them. It seems like the authors cannot decide what point to make, so they bombard the reader with lots of similar, overloaded figures. It is imperative to focus the design around the key points, and put the other figures in an appendix/supplement.

**2 Line by line Comments**

see annotated manuscript.

In summary, this paper is not appropriate for publication in present form, and it is unclear if, after this many trials, it can ever be brought up to that standard.

**2.2 Foraminifera as modelled entities (FAME)**

25  Foraminifera as modelled entities has been developed as a tool for translating, a climatic input (typically a reanalysis dataset or climate model output) into a (simulated-) climatic signal, a signal that aims to approximate the depth integrated growth of foraminifera (e.g., Pracht et al., 2019; Wilke et al., 2006; Steindhardt et al., 2015). Data-model comparison studies suffer from an inability to directly compare like with like so that there are differences in (i) the units used *i.e.*, most proxies reconstructing temperature do not give values of temperature in degrees °C or K but in their own proxy units (e.g., per mil ‰;

30  mmol/mol; species abundance or ratio) necessitating a conversion; and (ii) there is a reduction in scales, *i.e.*, models give a

[revised manuscript text omitted]

30   climb-down of actual El Niño and La Niña events. Therefore, these 'almost' El Niño or 'almost' La Niña are placed within their respective climatological pools as El Niño or La Niña.

Each time-step for the entirety of the Pacific was classified as one of three climate states (El Niño; Neutral; and La Niña), where after the resultant $\delta^{18}O_c$ and $T_c$ at each timestep produced by FAME for each grid-point were binned into their

respective categories. An Epanechnikov-kernel distribution was first fitted to the binned monthly output of a single climate state, the bandwidth varies between grid-points to provide for an optimal kernel distribution. The use of an Epanechnikov-kernel distribution to fit the data, as opposed to other types of distribution, represents a trade-off between keeping as many parameters constant whilst mimicking the underlying dataset for a large number of grid points. The conversion of the data

5   from dataset to distribution may induce some small error induced by: rounding to whole integers; the use of a $\delta^{18}O_{\text{mid-point}}$ which gives an error associated with the bin size ($\pm0.05$ ‰) that is symmetrical close to the distributions measures of central tendency but asymmetrical at the sides; and finally, the associated rounding error at the bin edges within a histogram ($\pm0.005$ ‰). Subsequently any two desired distributions can be compared for (dis)similarity using an Anderson-Darling test (1954). Here, all values, i.e., the population, associated with a climatological state are compared with the other populations

10   representing the different climatological state, the results plotted here are Neutral climate state vs. El Niño climate state.

**2.5 Test of input data (Temperature and calculated $\delta^{18}O_{eq}$)**

Foraminifera as modelled entities produces a modulated response that seeks to replicate how foraminifera modify the climate signal, several studies have approximated the foraminiferal signal in a different way (e.g., Thirmulai et al., 2013; Zhu et al., 2017a). In order to understand how FAME has altered the signal, and the degree to which the conclusions drawn depend

15   upon the modelled growth rates, the input datasets of the sea water properties (Temperature and calculated $\delta^{18}O_{eq}$), underwent a similar statistical test (Figure 4). Unlike FAME, which integrates over several depth levels using the computed growth rate, the test of the input datasets was with fixed depths without any growth rate weighting. These fixed depths are 5, 149 and 235 m, giving a Eulerian view (Zhu et al., 2017a) in which to observe the implications of FAME's dynamic depth habitat. As per the FAME output, each timestep value was placed into its climate state and an Anderson-Darling test

20   performed to compare the (dis)similarity of on the resultant distributions.

**2.6 Alternative statistical tests**

In order to compare our results with previously published studies using planktonic foraminifera we employed a series of simple statistical tests, mimicking those applied to sediment archives by the palaeoclimate community. A chief parameter that has been employed in previous ENSO proxy work using foraminiferal analysis (more specifically, individual

25   foraminiferal analysis; IFA) is the measure of individual foraminifera downcore standard deviation ($\sigma(\delta^{18}O_c)$). Increased $\sigma(\delta^{18}O_c)$ is considered to correlate to increased variation in SST and, in turn, increased ENSO incidence and/or magnitude (Leduc et al., 2009; Zhu et al., 2017a) or increased interannual variance (Thirmulai et al., 2013). The variance ($\sigma^2(\delta^{18}O_c)$) of the timeseries were computed both as the total variance and as the interannual variance, the latter is computed as outlined in Zhu et al. (2017a). For the interannual variance, the mean monthly climatology is subtracted from the dataset, producing

30   monthly anomalies and a linear trend removed (using the detrend function of MatLab 2019a) – the resultant data was left unfiltered (*i.e.*, Zhu et al., 2017a used a 1-2-1 filter). Four 'picking' experiments were performed, as FAME computes the average value for a given time step and given the single foraminiferal isotope variance for an equivalent time step (e.g.,

weeks: Steinhardt et al., 2015) it is more than likely that this computation reduces the real spread in values. Therefore, rather than use the terminology specimen we prefer to use months. Given the complexity in reconstructions of trace metal geochemistry (Elderfield and Ganssen, 2000; Nürnberg et al., 1996): the potential error associated with determining which carbonate phase is first used when foraminifera biomineralise (Jacob et al., 2017); growth-band integration; secondary factors (e.g., salinity, carbonate ion) the focus of the picking here has been on the $\delta^{18}O_c$. Irrespective of which experiment, 60 months were drawn, with replacement, and the number of Monte Carlo iterations is set at 10,000. We assume that the 'picker' is taxonomically well-trained and/or has a procedure in which species can be checked taxonomically post-analysis (e.g. photographing all specimens prior to analysis, Pracht et al. (2019)) and therefore do not include an error that deals with incorrect identification. Although we note that parameterisation of misidentification would be difficult, as it requires understanding of the variability in both standard deviation and absolute values for species co-occurring downcore (Feldmeijer et al., 2015; Metcalfe et al., 2015; 2019). For each run of experiment's (i) to (iii) the drawn months were saved in order to perform (iv):

(i) In Picking Experiment-I (Figure 3D), the months drawn for each iteration of the Monte Carlo were selected and each grid-point was sampled (i.e., there are 10000·60 selected months). This assumes that the same months are selected at grid point A as point B.

(ii) In Picking Experiment-II (Figure 3E), at each grid-point a Monte Carlo was run (*i.e.*, there are 170·40·10000·60 selected months). This assumes that different months could be selected between grid point A and point B.

(iii) In Picking Experiment-III (Figure 3F), at each grid-point a Monte Carlo was run using the growth rate weighting for each month (*i.e.*, there are 170·40·10000·60 selected months), this assumes that in periods of higher growth there will be a higher flux of the species and therefore a greater chance of selecting that month. The rationale being that not only are different months selected between grid point A and point B, but if A and B differ climatologically there may be an over subscription of ecologically beneficial habitats in one core location compared to the other.

(iv) In Picking Experiment-IV (Figures 3G to 3I), the experiment of (ii) was re-run but with the addition of two sources of error: The first error is based upon FAME producing the average value for a given time slice, therefore short-term variability in temperature and/or the spread in the population (*i.e.*, variance in depth of an individual; variance in chamber growth per individual), as evidenced by single foraminiferal analysis of sediment trap samples (*e.g.*, Steinhardt et al., 2015), is potentially lost. Therefore, for each picked month between -0.40 and 0.40 ‰ is added to the picked month value (in intervals of 0.02 ‰), this is approximately

±2° C (*i.e.*, ~4° C). The second error is the analytical error that an individual measurement will have. Machine measurement error is assumed to lie between -0.12 and 0.12 ‰ (in intervals of 0.005 ‰ – the 3rd decimal place is an exaggeration of machine capabilities although it will have repercussions for rounding) the 1σ of within run (as opposed to long-term average) of international stable isotope standards. The intervals of both errors (0.02 ‰ and 0.005 ‰) were chosen to give a similar number (n = 41 and 49) of potential randomly selected error for each picked month. Each picked month has their own randomly selected error for both of these errors, i.e., each value is the sum of the month picked and their own error. The values for within month variability (Figures 3G) and machine error (Figure 3H) are calculated separately and then combined (Figure 3I), as they may have a corresponding or conflicting signs, either 'cancelling' out each other or amplifying the difference.

An associated statistical methodology is the graphical summary (as opposed to a numerical summary via a test value) of plotting the quantiles of two probability or the quantiles of sample probability distribution against a theoretical distribution distributions also referred to as a Quantile-Quantile, or Q-Q plot (*e.g.*, Ford et al., 2015; White et al., 2018). A complimentary (*i.e.*, used in association with, not as replacement, Filliben 1975) test metric, the Probability plot correlation coefficient (Filliben, 1975) can be used as a numerical summation of this approach, which bases its rationale on near linearity between the two tested distributions. This graphical technique is not used here for the following reasons, (i) the climatic categories (*i.e.*, El Nino, Neutral, La Nina) imposed upon the data give uneven sized sample distributions requiring an interpolated quantile estimate; and (ii) the large graphical computation required (170·40).

**2.7 An approximation of sedimentary archives: Water depth & Sedimentation Rate**

Discrete sediment intervals retrieved from systematically bioturbated deep-sea sediment cores contain foraminifera with ages spanning many centuries (Lougheed et al., 2018; Peng et al., 1979).  The ambient signal following translation into a foraminiferal signal within the water is therefore further modulated by several post mortem processes, which include: the lat long shift in position of sinking foraminifera - the so-called 'funnel affect' (van Sebille et al., 2015; Deuser et al., 1981); dissolution of calcium carbonate either in the water (Schiebel et al., 2007), at the seafloor, or due to pore fluids; and bioturbation. As mentioned, mixing by bioturbation, results in an apparent smoothing of the downcore, discrete-depth multi-specimen signal (Hutson, 1980a; Löwemark, 2007; Löwemark et al., 2005, 2008; Löwemark and Grootes, 2004; Cole and Tudhope, 2017; Mix, 1987), thus leading to the possibility of interpreting single outlying foraminifera values within a specific depth as representing an 'extreme' climate, when they may in fact represent climate from a different time or epoch. This is especially apparent in $\delta^{18}O_c$ where there is a difference temporally of $\delta^{18}O_{sw}$ (e.g., the ice volume effect in glacial and interglacial cycles ~1.25 ‰) meaning that the same temperature can have radically different $\delta^{18}O_c$ values, a consequence of this is that a series of high magnitude, but low frequency El Niño events could be disturbed in a discrete-depth record.

Therefore, in order to reliably extract short-term environmental information from foraminiferal-based proxies, the signal that one is testing or aiming to recover must have a large enough magnitude, be largely unaffected by dissolution (*i.e.*, above the lysocline) so as not to adversely affect the population and the sedimentation rate must be high enough to give sufficient temporal coverage and rule out upwards bioturbation of single foraminifera from significantly different climate periods.

5    In our first step in consideration of post-mortem signal alteration we focus on dissolution. The lysocline, the depth at which dissolution first becomes apparent (Berger, 1968; 1970), and the Calcite (or Calcium Carbonate) Compensation Depth (CCD; Bramlette, 1961) vary between the different ocean basins; the Atlantic Ocean in which deep water forms has a relatively deep CCD as a by-product of 'young' well ventilated bottom waters whereas the Pacific Ocean the final section of the thermohaline circulation conveyor belt, has a shallower CCD. In order to highlight the potential for dissolution, the

10   bathymetry of the Pacific was extracted from the General Bathymetric Chart of the Oceans GEBCO 2014 30 arc-second grid (version 20150318, www.gebco.net) between -20°S to 20°N and 120°E to -70°W (Figure 8). Depths of 3500 m below sea-level (bsl), 4000 m bsl and 4500 m bsl are used here as cut-off values, these depths represent multiple possible depths under which there is the potential for noticeable dissolution (i.e., lysocline) or be dissolved (i.e., CCD). Whilst our intention here is a generalised view to be used as an approximate guide, it is important to note that the Pacific Ocean has the largest

15   proportion globally of >1 km tall seamounts that are smaller than <100 km (Wessel, 1997). Which may have important, relatively shallow-water sedimentary sequences, which may also be of sufficient sediment accumulation rate therefore we supplement the GEBCO bathymetric data with the locations of seamounts. However, whilst there are an estimated 50,000 seamounts in the Pacific that are taller than a km (Menard, 1964; Wessel and Lyons, 1997), only 12,000 have been documented on charts (Batiza, 1982), and approximately 291 have been dated (Koppers et al., 2003; Clouard and Bonneville,

20   2005; Hillier, 2007). It is these 291, <1% of the estimated seamounts, we have overlain onto the bathymetric data (Figure 8b), although this number is further reduced as we only plot between 20°S and 20°N.

The second step when considering post-mortem signal alteration is the sediment accumulation rate (SAR). We first plot the time-averaged deep-sea SAR (Olson et al., 2016), adapted by Lougheed *et al.* (2018) for the Tropical Pacific (Figure 9). New geochronological tools, such as dual $^{14}$C-$\delta^{18}$O measurements on single foraminifera (Lougheed et al., 2018), show that low

25   sedimentation rate cores can have large variances in age between individual foraminifera present within a discrete 1 cm depth interval (Berger and Heath, 1968; Lougheed et al., 2018). In order to model bioturbation, a number of papers have used a diffusion style approach that reduces the parameters down to sediment mixing intensity and sediment mixing depth (herein referred to as bioturbation depth, BD), although this may be an artificial division purely driven by mathematical need rather than biological constraints (Boudreau, 1998). The bioturbation depth has been shown to have a global average of 9.8

30   cm (1σ: ± 4.5 cm) that is independent of both water depth and sedimentation rate (Boudreau, 1998), though likely controlled as a result of the energy efficiency of foraging, *e.g.* deeper burrows may cost more energy to produce than can be offset in extracted food resources, and potential decay in labile food resources with sediment depth. It is not possible to carry out a transient bioturbation model upon the temperature and salinity ocean reanalysis data that we used for FAME, as it only covers half a century of data, whereas thousands of years of input data are required to force a transient bioturbation model.

To investigate how much temporal signal is integrated into discrete-depth intervals for typical tropical Pacific SAR, we, therefore, utilised the single foraminifera sediment accumulation simulator (SEAMUS, Lougheed, 2019) to bioturbate, as the input climate signal (Figure's 9 to 11), 0-40,000 year $\delta^{18}O_w$ of NGRIP (North Greenland Ice Core Project Members, 2004; Rasmussen et al., 2014; Seierstad et al., 2014). The ice core time series is an ideal input for a bioturbation simulator, as it

5 represents a highly temporally resolved climate input signal. SEAMUS simulates foraminifera in 10-year timesteps. The use of the NGRIP timeseries here is purely as an input parameter to investigate the effect of bioturbation upon a given climate signal - it is important to stress that by using NGRIP as an input signal for SEAMUS we are neither implying that tropical Pacific cores should have signal similar to NGRIP or inferring some kind of causal relationship. As we seek to investigate the effect of bioturbation, no attempt has been made to modulate the input signal's absolute values to mimic expected $\delta^{18}O_c$

10 values and this is why each plot of the synthetic down core time series retains the use of V-SMOW, despite carbonates being required to be V-PDB (Coplen 1995). Keeping all things constant, and varying a single parameter between experiments with SEAMUS, the sediment accumulation rate (SAR) was varied to fixed values of either 1, 2, 5 or 10 cm kyr$^{-1}$ (representative of typical Pacific SAR) and a bioturbation depth (BD) of either 5, 10 or 15cm based upon the global estimate and it's error bounds (Boudreau, 1998). For each experiment, the selected values of SAR and BD were kept constant for the entire

15 SEAMIS model run (i.e., the intensity and magnitude of bioturbation was not varied). In reality, SAR and BD may vary temporally depending on local conditions (*e.g.*, food, oxygen). Finally, the FAME results for the three species are overlaid with a water depth mask that highlights whether grid points are above or below 3500 m below sea-level (mbsl), to also show seafloor areas under the CCD depth, where carbonate material is not preserved (Berger, 1967, 1970b). A comparison between water depth and time-averaged deep-sea SAR (Olson et al., 2016), adapted by Lougheed *et al.* (2018) is shown in

20 Figure's 7 and 9.

**3. Results**

The results of the forward model (Figure 2 and 3) are compared with the input values (Figure 4) in order to identify regions in which the values are statistically distinct for different climate states (Figure's 5-7). These results are then shown against the water depth (Figure's 7 to 9) and the SAR (Figure's 9-11) for the region. The results utilise Foraminifera as Modelled

25 Entities (FAME; Figures 2, 3, 5, 6, and 7); the original Ocean Reanalysis data with computed $\delta^{18}O_{eq}$ (Figure 4); the General Bathymetric Chart of the Oceans (GEBCO; Figures 7 to 9); and the single foraminifera sediment accumulation simulator (SEAMUS; Figures 9 to 11).

**3.1 FAME Output: Variance**

We compute growth-weighted $\delta^{18}O_c$ (Figure 5 and 7) and temperature (Figure 6 and 7) distributions for each grid cell in the

30 fifty-eight year simulation using FAME (Roche et al., 2018), constraining the calculation to the Tropical Pacific Ocean (between -20°S to 20°N and 120°E to -70°W). Our model produces 696 individual monthly maps for all three species

(Figure 2). While two of the three species (*G. ruber* and *G. sacculifer*) have similar ecologies, they show differences in their resultant $\delta^{18}O_c$ for the same ocean conditions (Figure 2). A comparison of our computed variance with measured data (Supplementary Table 1) is made, we compare both the value of the nearest grid-cell and because of the size of the grid and drift of foraminifera (van Sebille et al., 2015) an average of a 3 by 3 grid in which the nearest grid-cell to the core location is

5    in the center. A comparison is made with both the iCESM model output and the core's that match this output (Zhu et al., 2017a). For the Late Holocene sample (~1.5 ka) MD02-2529 (08°12.33'N 84°07.32'W; 1619 m) in which *N. dutertrei* individual foraminifera were analysed from >250 µm (Leduc et al., 2009) giving a calculated standard deviation of measured foraminifera of 0.38 ‰. Whereas, the full ~60 year time series (n = 696) of FAME presented here, gives a standard deviation for all species, at depth cut off 60 m between 0.26 and 0.32 ‰; at depth cut off 100 m between 0.20 and 0.29 ‰; at depth cut

10   off 200 m between 0.20 and 0.25 ‰; and at depth cut off 400 m between 0.20 and 0.24 ‰ (see Table 1). Although these values vary if the average of the surrounding grid cells is used (see Table 1). In comparison the iCESM results have the following standard deviation values, for a Eulerian (fixed) depth of 50 m: 0.4 ‰; Eulerian 100 m: 0.6 ‰; and Lagrangian value of 0.49 ‰. There are three samples (Koutavas and Joanides, 2012; Sadekov et al., 2013) located south of core site MD02-2529, these are the Late Holocene (~1.6 ka) samples of V21-30 (01°13'S 89°41'W; 617 m) and (~1.1 ka) V21-29

15   (01°03'S 89°21'W; 712 m) in which *G. ruber* was measured individually (Sadekov et al., 2013). For these two sites the measured standard deviation is 0.507 ‰ and 0.510 ‰ for V21-30 and V21-29 respectively (Koutavas and Joanides, 2012). The third core site at a similar location is (~1.6ka) CD38-17P (01°36'04 S 90°25'32W; 2580 m) was not analysed individually, instead replicates of pooled samples of 2 or 3 shells of *N. dutertrei* (Sadekov et al., 2013) were made these measured values give a standard deviation of 0.28 ‰. The full ~60 year time series (n = 696) of FAME presented here, gives

20   a standard deviation for all species, at depth cut off 60 m between 0.33 and 0.41 ‰; at depth cut off 100 m between 0.27 and 0.40 ‰; at depth cut off 200 m between 0.25 and 0.35 ‰; and at depth cut off 400 m between 0.25 and 0.34 ‰ (see Table 1). Although these values vary if the average of the surrounding grid cells is used (see Table 1). In comparison the iCESM results have the following standard deviation values, for a Eulerian (fixed) depth of 50 m: 0.53 ‰; Eulerian 100 m: 0.75 ‰; and Lagrangian value of 0.35 ‰.

25   The study of ENSO has focused on whether the variability is entirely in response to ENSO or whether it is dominated by interannual variability (Xie, 1994, 1995; Wang et al 1994, 2010), here the interannual (Figure 3C) and total variance (Figure 3A) was computed and a ratio between the two calculated (Figure 3B; see Supplementary Table 1). Like the same analysis of interannual and total variance computed for iCESM and SODA reanalysis (Carton et al., 2000), outlined in Zhu et al. (2017a), there is also high ratio of interannual to total variance in our computed FAME dataset (Figure 3B). Although there

30   are regions in the Eastern Equatorial Pacific wherein this ratio reduces. Despite this reduction, the ratio between total and interannual variance is still above > 0.5.

The Monte-Carlo experiments (Figure 3D-I) highlight the variation in picking a subset of the months, here 60, from the full timeseries. The FAME-$\delta^{18}O_{eq}$ *G. sacculifer* with a depth cut-off of 60 m is plotted here, the values for each grid point is the range in standard deviation (*i.e.*, the maximum standard deviation minus the minimum standard deviation) between iterations

of the Monte-Carlo (n= 10,000). The range in standard deviations between iterations is plotted instead of the mean of the standard deviations; with increasing *n* the mean converges toward the sample mean, however as the point of the Monte-Carlo is to generate plausible 'samples' it is more important to take into account the range in possible values which would help to establish the potential variability of subsampling. For the most part, regions with high total variance (Figure 3A) also have a

5 larger range in standard deviations between the iterations 'picked'. It is interesting to note that by changing from the same months picked for each grid-point (Monte-Carlo I: Figure 3D) to varying the months picked between grid-points (Monte-Carlo II: Figure 3E or Monte-Carlo III: 3F) the range goes from 'smooth' to a more noisy dataset. Whilst the values plotted here are not the absolute values (as they are the range in standard deviation for a given grid point for the entire 10,000 iterations), it can be seen that some of the inter-core comparisons could in essence relate to differences in picking, *i.e.*

10 different 'months' picked between grid-points may exacerbate or accentuate differences. Likewise, adding random variability, between -0.4 and 0.4 ‰ (Figure 3G and 3I), may also reduce the differences between areas of high Total variance and low Total variance. Though the values associated with machine error (-0.12 to 0.12 ‰) appear to do little to affect the range (Figure 3H and 3I). Whilst again the values plotted are not the absolute values, the variability added in an attempt to mimic biological variation of a given time slice increases the range of possible standard deviations in regions with low Total

15 variance (Figure 3G and 3I). Therefore, understanding the biological variability on shorter timescale (e.g., Steinhardt et al., 2015; Mikis et al., 2019) which, maybe here over exaggerated, may be crucial for understanding discrepancies between cores.

**3.2 FAME Output: Anderson-Darling test**

Using a basin-wide statistical test, we examine whether the $\delta^{18}O_c$ values of a given El Niño foraminifera population ($FP_{EN}$)

20 and a given non-El Niño ('Neutral conditions') foraminifera population ($FP_{NEU}$) can be expected to be significantly different at any given specific location. Where $FP_{EN}$ and $FP_{NEU}$ exhibit significantly different distributions, ENSO events can potentially be detected by paleoceanographers. In cases where $FP_{EN}$ and $FP_{NEU}$ do not exhibit significantly different values, then the chosen species and/or location represent a poor choice to study ENSO dynamics. Each simulation time step was placed into a climate states: identification of timesteps that represent El Niño (EN), Neutral (NEU), and La Niña climate

25 conditions was done using the Oceanic Nino Index (ONI) derivative (Huang et al., 2017) (Figure 1). Comparison, for each species, FAME's predicted growth-weighted $\delta^{18}O_c$ and $T_c$ distributions associated with each climate event was done using an Anderson-Darling (AD) test. This statistical test can be used to determine whether or not two distributions can be said to come from the same population. The results of this test are presented in the following way, in which there are four criteria: areas where the population distributions of the two climate states are found to be statistically similar have black grid cells in

30 all panels referring to the Anderson-Darling test results (Figure's 4-7); the results in which the areas where the populations distributions of two climate states are found to be statistically distinct are shown with two distinct colour schemes depending on whether a computable error can be included (Grey and Hashed) or not (White).

For FAME-$\delta^{18}O_c$ the results where the populations are dissimilar are either plot as grey and hashed for *G. ruber* and *G. sacculifer* or white for *N. dutertrei* (Figure 5). This is because for these two species (*G. ruber* and *G. sacculifer*) we have the possibility to determine how robust these results are. We use the 1σ values of the observed (FAME) minus expected (MARGO), as computed by Roche et al. (2018) with the MARGO core top $\delta^{18}O_c$ database, as the potential error associated with the FAME model. Regions in which the difference between the two populations are larger than the potential error are associated with grey, whereas those less than the potential error as hashed regions (Figure 5A and B), these errors should be seen as a guide rather than a rejection of a site. Because the MARGO database does not contain *N. dutertrei* we have given the panels concerning this species a separate colour scheme, black represents grid-cells for which the two populations cannot be said to be statistically different, white grid-cells are those in which the two populations can be said to be statistically different (Figure 5C). As we do not have a similar way to calculate the error for $T_c$, FAME-$T_c$ results are shown (in Figure 6) with this same binary pattern (*i.e.*, white grid-cells are those in which the two populations can be said to be statistically different and black are those in which the two populations can be said to be similar). To reduce the complexity, the overlay of the species Anderson-Darling results (Figure 7) also uses the binary colour scheme (white or black).

Our results show that much of the Pacific Ocean can be considered to have statistically different population between $FP_{EN}$ and $FP_{NEU}$ for both $\delta^{18}O$ (Figure 5) and $T_c$ (Figure 6). We consider that the likely cause for such a remarkable result is due to FAME computing a weighted average and, therefore, the lack of a signal found exclusively within the regions demarked in Figure 1 as El Niño regions could represent how the temperature signal is integrated via an extension of the growth rate; growing season and depth habitat of distinct foraminiferal populations. Taking into account the FAME-$\delta^{18}O_c$ error for *G. ruber* and *G. sacculifer*, we have computed regions in which the difference in oxygen isotopes between the two populations ($\Delta\delta^{18}O_c$) compared with the AD-test is smaller than the aforementioned error (Hatching in Figure 5), *i.e.* where the mean difference between $FP_{EN}$ and $FP_{NEU}$ is within the error. The hatched regions in Figure 5 considerably reduce the areal extent of significant difference between $FP_{EN}$ and $FP_{NEU}$, with the remaining regions aligning with the El Niño 3.4 region (Figure 1). It is important to note that this error relates to the model and in reality, the difference between the climate states could be larger or smaller. No such test was performed on the *N. dutertrei* dataset, because of its absence from the MARGO dataset.

To further test the model-driven results and to assess if they are still consistent when the depth limitation is varied, the analysis was rerun with depths of 100, 200 and an extreme value of 400 m (Figure 5-7). Whilst it is possible to discern differences between the depths, it is important to note that a large percentage of the tropical Pacific remains accessible to palaeoclimate studies. A shallower depth limitation in the model increases the area for the 'warm' species, suggesting that the influence of a reduced variability in temperature or $\delta^{18}O_{eq}$ with a deeper depth limit causes the differences between $FP_{EN}$ and $FP_{NEU}$ to be reduced. Overlaying the results of the Anderson-Darling test for all three species (Figure 7) per depth for 60, 100 and 200 m highlights the areas where multi-species comparisons could be made. To account for potential differences in depth habitat we make a combination of shallower depth for *G. ruber* and deeper depths for *G. sacculifer* and *N. dutertrei* (Pracht et al., 2019) in the final panels (Figures 7D and 7H).

**3.3 Test of input parameters (fixed depth: temperature and $\delta^{18}O_{eq}$)**

The model-driven results were assessed with the underlying input dataset (temperature and $\delta^{18}O_{eq}$), these underwent the same statistical test (Figure 4), although with fixed depths of 5 m, 149 m and 235 m (see section 2.5). The results for each grid point are presented as either black, grey or hashed. Areas where the population distributions of the two climate states are found to be statistically similar have black grid cells. Regions in which the difference between the two populations are larger than the potential error are associated with grey, whereas those less than the potential error as hashed regions. The threshold error (i.e., the difference between the means of each distribution) is for temperature (Figure 4A-C) 0.5 °C and for $\delta^{18}O_{eq}$ (Figure 4D-F) 0.10 ‰, these errors should be seen as a guide rather than a rejection of a site. The results of this fixed depth, non-FAME, test show that the shallowest depths produce populations that are significantly different both in terms of their mean values and their distributions. In the upper panel of Figure 4, the canonical El Niño 3.4 region is clearly visible at 5 m depth. Whilst differences exist between Anderson-Darling results for the input data (Figure 4) and the FAME $\delta^{18}O$ (Figure 5) and $T_c$ (Figure 6), for instance close to the Panama isthmus, there are significant similarities between the plots. These plots also show that our FAME data (Figure 5-7), in which we allow foraminiferal growth down deeper than the depths in Roche et al. (2018), are a conservative estimate and thus are on the low-end (Figure 4), to account for potential discrepancies with depth habitats. In the original paper on depth habitats based upon temperatures derived from $\delta^{18}O_c$, Emiliani (1954) cautioned that the depth habitats obtained would represent a weighted average of the total population, and while foraminiferal depth habitats are likely to vary spatiotemporally, the average depth habitat is skewed toward the dominant signal (Mix, 1987).

**3.4 Water depth and SAR**

Our analysis uses reanalysis data for the time period 1958-2015, a hypothetical core that had a comparable resolution would essentially be analogous with a sediment core with a rapid sediment accumulation rate (SAR), representing 50 yr cm$^{-1}$ (or 20 cm kyr$^{-1}$). Based on our analysis, such a hypothetical core could allow for the possible disentanglement of El Niño related signals from the climatic signal, but only in a best-case scenario involving minimal bioturbation, which is unlikely in the case of oxygenated waters. Extracting the oxygen saturation (**SO₂**) state, of the Pacific Ocean bottom waters from the Annual Climatology WOA13 give values that are predominantly >40 % (Figure 9B). Oxygen saturation is the concentration of Oxygen in a medium against the maximum that can be dissolved in the same medium. Whilst annual variability may exist, it is unlikely that bioturbation would be prevented by low oxygen. Therefore, using a cut off value that has been considered sufficiently high enough to outpace bioturbation (*e.g.*, Koutavas and Lynch-Stieglitz, 2003) of 5 cm kyr$^{-1}$ (Figure 9A) it can be demonstrated that much of the Pacific has an inferred lower sedimentation rate (< 5 cm kyr$^{-1}$; Figure 9C) than this cut off value. To test the influence of bioturbation, the bioturbation simulator SEAMUS was run using the NGRIP time series. The results of SEAMUS highlight the potential single foraminifera depth displacement that low sedimentation rates can result in (Figure 9). Following the current available geochronological method (i.e., age-depth method) such specimens that are

displaced in depth are assigned the average age of the depth that they were displaced to, which could result in erroneous interpretations of climate variability when analysis such as IFA is applied (Lougheed et al., 2018). The results of SEAMUS are plotted both as time series of the bioturbated 'NGRIP' signal (Figure 10) and as histograms of the probability of finding a particularly pseudo-foraminifera with a given age in the bioturbation depth (Figure 11). As the bioturbation depth varies

5 between 5, 10 and 15 cm for the different simulations of SAR, the histogram in each panel (in Figure 11) represent different thicknesses of sediment, i.e., for Figure 11 panels a, d, g, and j histograms represent data with a BD thickness of 5 cm. Likewise, the timeseries is plotted with the discrete 1 cm depth median age; the median age of the bioturbation depth (Figure 11) is the reason why each timeseries does not 'start' at 0 age (Keigwin and Guilderson, 2009).

The variance within a single depth in a core largely represents the integrated time signal for that depth (Figure 11), as

10 opposed to the variance of a climatic signal for an inferred (or measured) average age for the depths in question. The proxy variance will be based both upon a non-uniform distribution in temporal frequency of specimens, i.e., older specimens are few compared to younger specimens. A large proportion of the specimens in the BD come from years that are 'proximal' (i.e., close to the youngest age) this may give undue confidence that the probability of picking a specimen from these years is higher, however the long-tail of the distribution means that there is an equally high chance of picking a specimen that has

15 come from several thousand years earlier than the discrete-depth's median age. If we consider for the moment this as picking specimens from a box, there is a high chance of picking from a single box that represents the age you want however there is an equally high chance of picking from numerous boxes with varying age. If the spread in the climatic variable is uniform throughout this time then it can be possible to reproduce a similar signal, although this would not by definition represent the actual spread in the actual climatic variable for a given time, however the spread in the climate variable is unlikely to be

20 constant. With a varying spread in the climatic signal bioturbation can introduce the possibility of spurious interpretations, but it is of course more obvious where the measured distributions over-exaggerate the climate signal (e.g., Wit et al., 2013). Furthermore, if we consider that researchers do not pick as randomly as they profess, there is both a size and preservation bias to specimens selected, and size is not constant down-core (e.g., Metcalfe et al., 2015) we can further introduce bias within the dataset. The SEAMUS output that corresponds with our chosen SAR cut-off value of 5 cm kyr $^{-1}$ (Figure 10 and

25 11), the lower limit of our mask (Figure 9), is shown in panels in Figures 10H to 10J and Figures 11G to 11I. It is important to note however, that much of the region for which FAME is calculated upon has inferred sedimentation rates lower than this cut-off value (Figure 9C to 9H).

An additional factor in the post-mortem preservation of the oceanographic signal in foraminiferal shells is whether the shells can be preserved. The GEBCO bathymetry data is binned into 250 m wide bins, and the data normalised to 1.0. As the data

30 contains both bathymetric and topographic (below and above 0 m), the grey area in each histogram represent > 0 m (Figure 8). Whilst, there are differences depending on the cut-off value (Figures 8C to 8E) much of the canonical El Niño 3.4 region (Wang et al., 2017) used in oceanography (Figure 1) is also excluded from these suitable areas. Overlaying the water depth and the SAR with the Anderson-Darling results (Figure 7) highlights that of the total area where $FP_{EN}$ is significantly different from $FP_{NEU}$ (i.e. those areas where planktonic foraminiferal flux is suitable for reconstructing past ENSO

dynamics), only a small proportion corresponds to areas where the sea floor is both above the CCD (< 3500 mbsl) and SAR is at least 5 cm/ka (Figure 9). However, at certain locations, near islands or seamounts, the SAR and water depth may be high enough to allow for a signal to be preserved (Figure 8B) that may not be represented here.

**4. Discussion**

**4.1 From Life to Sedimentary Assemblages**

Whilst we are principally interested in understanding whether living foraminifera can theoretically reconstruct ENSO, comparison with data requires an additional analysis. This is because data-model comparisons are subjective, nominally supposing that the data is the value to be achieved by the model. However, if the foraminifera modulate the original climate signal, then preservation selectively filters which specimens are conserved whereas bioturbation acts to reorder, transposing the order in which they are recovered from the depth domain. Once the sediment is recovered, the researcher acts as a final filter, which is in essence a random picking. Although technically most researchers will pick whole shells so alongside size selectivity (*e.g.*, Metcalfe et al., 2015) there is also preservation bias associated with picking of foraminifera (*e.g.* Koutavas and Lynch-Stieglitz, 2003). Whilst the presence of depths in the ocean whereupon calcite is absent from sediments was described in the earliest work (e.g., Murray and Renaud, 1891; Sverdrup, 1942), overlaying maps of measured surface sediment carbonate percentage with water depth in the Pacific Ocean led Bramlette (1961) to coin the term 'compensation depth' (Wise, 1978). This work highlighted the 'narrow' depths (4-5000 m) in the Central Pacific of the CCD. Conceptually Berger (1971) placed three levels in the Pacific ocean that were descriptive of the aspects (e.g., chemical, palaeontological and sedimentological) of the calcite budget; the saturation depth, demarking supersaturated from undersaturated; the lyscoline, the depth at which dissolution becomes noticeable (Berger 1968, 1971); and compensation depth (Bramlette, 1961), in which supply is compensated through dissolution. The aspects of the lysocline was estimated by the faunal assemblages of Parker and Berger (1971, figure's 14 and 15 of that publication), for much of the equatorial Pacific the lysocline is estimated at ~3800 m. As the lysocline is where dissolution becomes apparent, ergo it is a sample already visibly degraded, we therefore set the limit of the water depth mask shallower, at 3500 m bsl. In fact, in regions of high fertility, such as the Eastern Equatorial Pacific, the lysocline was estimated to be present at ~2800 m (Thunell et al., 1981) or ~3000 m (Berger, 1971; Parker and Berger, 1971). For instance, core V21-28 close to the Galapagos Islands (01°05'N, 87°17'W) has a shallower dissolution affect than either of these two values despite being collected from a water depth of 2714 m (Luz, 1973). A comparison between the hydrographic and sedimentary lyscoline, using a mooring in the Panama Basin showed that the sedimentary lyscocline is a product of where the hydrographic lyscocline meets the seafloor (Thunell et al., 1981), therefore, this could lead to dissolution within the water of the settling flux (e.g., Schiebel et al., 2007). In the EEP region the shallower lyscoline is accompanied by an equally shallower CCD (located at ~3600 m) for which the highly fertile is considered responsible for its shoaling, lowering the pH through increased $CO_2$ (Berger et al., 1976). The correspondence between lyscoline depth and CCD depth does not hold true for the entirety of the Pacific, plotting a N-S cross-section from

50°N to 50°S Berger (1971) noted that in the Central Equatorial Pacific, the high fertility region generates a larger zone of dissolution resistant facies even with a shoaled lysocline. If we factor in the sedimentation rate of the Pacific, which has been estimated to be considerably lower than 1 cm (Blackman and Somayajulu, 1966; Berger, 1969; Menard, 1964), then dissolution may become further exacerbated. The longer a shell remains at the sediment-water interface the greater the

5  prospects for it to be dissolved become, therefore low SAR increases the chance of dissolution (Bramlette, 1961). For instance, in 15 equatorial Pacific cores, below 4000 m, the average SAR was presented (Hays et al., 1969; here calculated) at 0.96 cm kyr$^{-1}$ (1σ ± 0.43 cm kyr$^{-1}$). Although there are regions and/or core locations in which the SAR is higher, for instance eight EEP cores shallower than the lysocline depth (Thunell et al., 1981) of ~2800 m were presented by Koutavas and Lynch-Stieglitz (2003) which have an average SAR, calculated at 7.20 cm kyr$^{-1}$ (1σ ± 2.82 cm kyr$^{-1}$). The average age for

10  these same core's 0 cm core depth is 2184 years (1σ ± 1521 yrs), whilst it cannot be assumed that there has been no loss during recovery (i.e., core top is not sediment-water interface),  a non-zero core top age is expected for both bioturbation (Keigwin and Guilderson, 2009) and dissolution. Alongside, the potential for dissolution there is the also the mixing of ocean sediments by the benthos (Bramlette and Bradley, 1942). For instance, Arrhenius (1961) noted that ash beds present in cores of the EEP (Worzel, 1959; Ewing et al., 1959) had a 2-3 cm layer above and below what should have originally been a

15  sharp boundary in which they estimated that ~50% of the material originated from the other side of the boundary. If one assumes 1 cm kyr$^{-1}$ sedimentation rates, then the range in age of the obviously 6 cm mixed sediments is minimally ~6000 years per cm, comparison with an analogous SEAMUS simulation (bioturbation depth 5 cm; SAR 1cm) highlights the considerable spread in age, placing the 95.45% range between 110 and 18954 years (Figure 11). Much of this temporal variability (either through bioturbation or dissolution) will be hidden, especially when proxy values correspond with the

20  expected values, and more obvious when the values are larger than expected (*e.g.*, Wit et al., 2013). Owing to the lack of absolute variability during the Holocene the apparent confirmation of similarity between proxy values and modern distributions of the '*to be reconstructed*' variable is not a confirmation of proxy reliability. Especially in the tropics wherein seasonal variability is limited. The effects of both bioturbation and dissolution are further amplified when combined with finite sampling strategies. Therefore, the results of the sedimentological features, presented here, imply that much of the

25  Pacific Ocean is not suitable for preserving (Figures 7-9) the ENSO signal, despite the possibility of the species of foraminifera having unique values for different climate states (Figures 4-7). ENSO studies using palaeoceanography have exposed shifts, one can, therefore, question what is being reconstructed in such studies.

**4.2 Palaeoceanographic Implications**

**4.2.1 Pacific climate reconstructions**

[revised manuscript text omitted]

5    ENSO conditions; as well as what the proxy is recording. There is also the presumption that a particular climate event should be recorded, our Anderson-Darling test for instance highlights that there are locations that cannot discern the difference between El Niño and other climate states whilst for the same time period there are locations where the different climate states can be differentiated. Whilst our analysis is a statistical treatment of the data, each species, and different types of phyto- or zooplankton preserved in ocean sediments, are likely to record the same set of environmental conditions differently

10    (Mix, 2006). This is, in brief, the rationale for the development of FAME, the same climate signal seen through the view of species-specific proxies will give a fractured view constrained by each species particular ecophysiological constraints (Mix, 1987; Roche et al., 2018). A dynamic depth habitat in which the environmental signal becomes a weighted average of the water column can further confound the original signal (Wilke et al., 2006). What can be seen as contradictory reconstructions can therefore be viewed as the prevailing or dominant conditions at a given location at the time when

15    environmental conditions overlap ecological constraints for a given species.

Terrestrial records suggest the number of El Niño events per century in the early Holocene (8-6 ka BP) was minimal (Moy et al., 2002), with between 0 and 10 events occurring per century. This dampened ENSO is observed within lake core colour intensity and records driven primarily by precipitation - although like other proxies this can also be interpreted differently, *i.e.* as a large change in the hydrological cycle shifting precipitation away regionally (Trenberth and Otto-Bliesner, 2003). If

20    we assume for now that the number and magnitude of ENSO events was reduced, the relatively low downcore resolution of marine records may not accurately capture the dynamics of such lower amplitude ENSO events using existing methods. The sensitivity and probability of detecting a change in IFA with changes in frequency and amplitude, has been dealt with before (Thirmulai et al., 2013), although without considering bioturbation. The synthesis of pseudo-timeseries to discern the potential distribution for different scenarios, whilst a necessary approximation, is nonetheless one that is free of cause and

25    causality. Modulating a timeseries for events with enhanced or weakened amplitude or fewer or greater number of events assumes in essence that there is limited feedback both regionally (between two sites) and internally within the timeseries (i.e., a process that operates on a higher level). Reconstructions of the past can benefit from inclusion within conceptual frameworks that incorporate both data and modelling studies (e.g., Trenberth and Otto-Bliesner, 2003; Rosenthal and Broccoli, 2004; McPhaden et al., 2006). The use of coupled ocean-atmosphere models (e.g., Clement et al., 1999; Zebiak

30    and Cane, 1987); isotope enabled Earth system models (e.g., iCESM; Zhu et al., 2017); or multi-model ensembles with prescribed boundary conditions can be used for the generation of timeseries in which the physics of atmospheric and oceanic circulation are constrained and feedbacks between sites can occur. The perceived failure of several climate models to resolve ENSO adequately, resulting in variable ENSO frequency and amplitude between models, could therefore be used to determine the proxy signal from model derived timeseries at different frequencies and intensities of ENSO. Albeit a

timeseries of variable ENSO that is grounded in ocean-atmosphere coupling. Such analysis could also provide information on a secondary assumption, in which time slices from the same core inherently assume that where a particular oceanographic feature exists now is also where it may have existed before. This gives a somewhat binary view. the feature either occurs or does not occur, and if it occurs then it has either enhanced or weakened. Yet this can (though not always) preclude a scenario

5    in which the feature has shifted. Analysis of the El Nino patterns suggests that there are two types of El Nino that are spatially delineated: the dateline Central Pacific El Nino and the Eastern Pacific El Nino. The expansion, contraction or shift of certain large scale oceanographic features (e.g., Polar Front, Upwelling) during periods of warmer than average (e.g., the last interglacial) or colder than average temperatures (e.g. the LGM) can complicate the comparison of two down core samples, i.e., a static core continuously recording a particular climate event as opposed to a shifting oceanographic regime

[revised manuscript text omitted]
 the integrating the $\delta^{18}O_{eq}$ values using a growth-rate based weighting (FAME; Roche et al., 2018). Values are in per mil (‰ V-PDB).**

**Figure 4. Anderson-Darling Results for Input datasets of Temperature and Equilibrium $\delta^{18}O$ ($\delta^{18}O_{eq}$). Results of the test in which input variables underwent the same statistical procedure (see section 2.0) as the modelled data for (A-C) temperature and (D-F) $\delta^{18}O_{eq}$ values. Here, model input data was extracted for three fixed depths ([A & D] 5 m; [B & E] 149 m; [C & F] and 235 m) without any growth weighting applied. Black regions are those grid points in which the null hypothesis ($H_0$), that the El Niño and Non- El Niño populations are not statistically different ($FP_{El\ Niño} = FP_{Non\text{-}El\ Niño}$), cannot be rejected. Gray regions represent grid points where the $H_1$ hypothesis is accepted, therefore the distributions of the foraminiferal population (FP) for El Niño and Non- El Niño can be said to be unique ($FP_{El\ Niño} \neq FP_{Non\text{-}El\ Niño}$). The hatched regions represent areas were the $H_1$ hypothesis can be accepted, therefore the distributions of the foraminiferal population (FP) for El Niño and Non- El Niño can be said to be unique ($FP_{El\ Niño} \neq FP_{Non\text{-}El\ Niño}$), though the difference between the means of tested distribution are less than (A-C) 0.5°C or (D-F) 0.1 ‰. Each panel represents a single depth (5, 149 and 235 m).**

**Figure 5. Anderson-Darling Results for modelled FAME-$\delta^{18}O_{eq}$:** Panels representing locations of where dissimilar and similar values of FAME modelled species $\delta^{18}O$ occur between climate states, for (columns) particular species and (rows) particular model depth cut-off limits. Each panel represents the Anderson-Darling test result, which are plotted with ([A] *Globigerinoides sacculifer* and [B] *Globigerinoides ruber*) and without ([C] *N. dutertrei*) model derived error. For all panels black areas reflect latitudinal and longitudinal grid points that failed to reject the null hypothesis ($H_0$) and therefore the foraminiferal population (FP) of the El Niño is similar to the Non-El Niño ($FP_{El\ Niño} = FP_{Non-El\ Niño}$). The results in which the $H_1$ hypothesis is accepted, in which the, therefore the distributions can be said to be different ($FP_{El\ Niño} \neq FP_{Non-El\ Niño}$), are plotted as either: (A – *G. sacculifer*, B – *G. ruber*) grey and hatched or (C – *N. dutertrei*) solely as white regions. For species with calculatable error, grey regions represent values where the difference between the two means of the population is greater than species-specific standard deviation of the FAME model and hatched regions represent those in which the means are less than this standard deviation (Roche et al., 2018). For species without a calculatable error, the regions are plotted in white. The rows represent the model runs with a depth cut-off limit at: (A-C) 60 m; (D) 100 m; (E) 200 m; and (F) 400 m.

.

**Figure 6. Anderson-Darling Results for modelled FAME-$T_c$:** Panels representing locations of where dissimilar and similar values of FAME modelled temperature recorded in the calcite shells ($T_c$) occur between climate states, for (columns) particular species and (rows) particular model depth cut-off limits. Each panel represents the Anderson-Darling test result, which are plotted with ([A] *Globigerinoides sacculifer* and [B] *Globigerinoides ruber*) and without ([C] *N. dutertrei*) model derived error. For all panels black areas reflect latitudinal and longitudinal grid points that failed to reject the null hypothesis ($H_0$) and therefore the foraminiferal population (FP) of the El Niño is similar to the Non-El Niño, and therefore the distribution between the neutral climate and El Nino cannot be said to be different ($FP_{El Niño} = FP_{Non-El Niño}$). The results in which the $H_1$ hypothesis is accepted, in which the distributions can be said to be different ($FP_{El Niño} \neq FP_{Non-El Niño}$), are plotted as white regions. The rows represent the model runs with a depth cut-off limit at: (A-C) 60 m; (D) 100 m; (E) 200 m; and (F) 400 m.

**Figure 7. Combined A-D plots. As figure 5 and figure 6, in that panels represent locations of where dissimilar and similar values for the two climate states for (a-d) FAME-$\delta^{18}O_{eq}$ modelled oxygen isotope values or (e-h) FAME-$T_c$ modelled temperature recorded in the calcite shells (Tc) occur. Each panel represents the Anderson-Darling test result, the results for *Globigerinoides sacculifer*, *Globigerinoides ruber* and *N. dutertrei* are overlaid. For all panels black areas reflect latitudinal and longitudinal grid points that failed to reject the null hypothesis ($H_0$) and therefore the foraminiferal population (FP) of the El Niño is similar to the Non-El Niño, and therefore the distribution between the neutral climate and El Nino cannot be said to be different ($FP_{El\ Niño}$ = $FP_{Non-El\ Niño}$). The results in which the $H_1$ hypothesis is accepted, in which the distributions can be said to be different ($FP_{El\ Niño} \neq FP_{Non-El\ Niño}$), are plotted as yellow where the depth is deeper than 3500 m bsl or purple where the depth is shallower than 3500 m bsl (see Figure 8). Purple locations are where our results suggest that the signal of ENSO has different values and the water depth allows for preservation – although this purple region will be smaller when inferred SAR is taken into account (see Figure 9). The rows represent the model runs with a depth cut-off limit at: (A and E) 60 m; (B and F) 100 m; (C and G) 200 m; and (D and H) where a combination of depths were utilised (Pracht et al., 2019).**

**Figure 8. Bathymetric map of the Tropical Pacific Ocean highlighting the areas above and below the Lysocline and/or Calcite compensation depth (CCD). (A) GEBCO map of height relative to 0 m; (B) same as (A) with location of seamounts plotted (white stars); (C-E) binary colour map of GEBCO data, yellow is values below cut-off depth value ([C] 3500 m below sea-level (bsl); [D] 4000 m bsl; and [E] 4500 m bsl respectively) and purple above the cut-off depth value. The histograms represent the normalised frequency of grid cell height in bins of 250 m wide, yellow is values below cut off value ([C] 3500 m below sea-level (bsl); [D] 4000 m bsl; and [E] 4500 m bsl respectively), purple above cut off value. The grey bins in each histogram are those above 0 m.**

**Figure 9. Map of the sedimentation rate and oxygen saturation for the Tropical Pacific. (A) Inferred sedimentation rate (Olsen et 2016). White regions represent continental shelf. (B) Oxygen saturation of the bottom grid layer of World Ocean Atlas 2013 (data from: https://www.nodc.noaa.gov/cgi-bin/OC5/woa13/woa13oxnu.pl ). (C, E, G) Overlay between water depth and inferred SAR, Red / Pink: Continental shelf sediments that are (Red) shallower or (Pink) deeper than 3500 mbsl; Gray / White: grid point SAR is lower than SAR threshold and the seafloor depth is (grey) shallower or (white) deeper than 3500 mbsl; Light Yellow/Gold: Light yellow represents areas where the SAR is above the threshold but the water depth is deeper than 3500 mbsl in comparison Gold represents areas where the SAR is above the threshold and the water depth is deeper than 3500 mbsl. The ideal locations are therefore plotted as Gold. Cut-off limits for SAR are (C) $\geq 1$ cm kyr$^{-1}$; (E) $\geq 2$ cm kyr$^{-1}$ and (G) $\geq 5$ cm kyr$^{-1}$, (D, F, H) alongside the maps the bioturbation simulations for the minimum SAR threshold is plotted (see Figure 10 and Figure 11 for the output of SEAMUS). Each plot gives the input values of NGRIP (grey) and for each SAR three analysis were performed with different bioturbation depths (BD) these are (Blue) 5 cm; (Green) 10 cm; and (Orange) 15 cm.**

**Figure 10. Output of the bioturbation model SEAMUS. (A) The unbioturbated input signal, NGRIP (North Greenland Ice Core Project Members, 2004; Rasmussen et al., 2014; Seierstad et al., 2014), used in our simulation of bioturbation for different SAR with SEAMUS (Lougheed, 2019). Sediment mixed layer referred to here as bioturbation depth (BD) is fixed at (B, E , H, K) 5 cm, (C, F, I, L) 10 cm and (D, G, J, M) 15 cm for sedimentation accumulation rates (SAR) of (B-D) 1 cm kyr$^{-1}$; (E-G) 2 cm kyr$^{-1}$; (H-J) 5 cm kyr$^{-1}$ and (K-M) 10 cm kyr$^{-1}$. The output is plotted as the discrete 1 cm depth median age. In (B-M) grey values represent the unbioturbated input signal, NGRIP. Note, we retain the original units (V-SMOW) of the original timeseries used, no inference between Pacific climate and Greenland is intended by the use of NGRIP (see section 2.7).**

**Figure 11. Histograms of simulated specimen age within the bioturbation depth. The simulated age distribution present within the sediment mixed layer, referred to here as bioturbation depth (BD). BD is fixed at (A, D, G, J) 5 cm, (B, E, H, K) 10 cm and (C, F, I, L) 15 cm for sedimentation accumulation rates (SAR) of (A-C) 1 cm kyr[-1]; (D-F) 2 cm kyr[-1]; (G-I) 5 cm kyr[-1] and (J-L) 10 cm kyr[-1]. The output is plotted as the discrete 1 cm depth median age. Note the size of the BD varies, therefore the simulated age distribution comes from a varying 'core depth'.**

[revised manuscript text omitted]

 the specific case of foraminifera populations in the water, it particularly arises from the species-specific ecological niche. The mapping of proxy value to climate value can therefore be skewed, a major factor governing the spatiotemporal distribution of a given planktonic foraminiferal species is the presence of an ideal water temperature. Proxies of past ENSO and Pacific SST (Ford et al., 2015; Koutavas et al., 2006; Koutavas and Joanides, 2012; Koutavas and Lynch-Stieglitz, 2003; Leduc et al., 2009; Sadekov et al., 2013; White et al., 2018) are based upon the biomineralisation of the calcite, or a polymorph such as verite (Jacob et al., 2017), shells of foraminifera (Emiliani, 1955; Evans et al., 2018; Zeebe and Wolf-Gladrow, 2001). In general, there are three major types of foraminifera-based palaeoceanographic proxies:

(1) those associated with the faunal composition and their abundance within deep-sea sediments that utilises either a qualitative approach (Phleger et al., 1953; Schott, 1952); a weighted average (Berger and Gardner, 1975; Jones, 1964; Lynts and Judd, 1971); a selected species approach (e.g. coiling direction, or warm-water species presence; Ericson et al., 1964; Ericson and Wollin, 1968; Hutson, 1980b; Parker, 1958; Peeters et al., 2004; Ruddiman, 1971; Schott, 1966); a regression analysis (Hecht, 1973; Imbrie and Kipp, 1971; Williams and Johnson, 1975); or, a transfer function (CLIMAP Project Members, 1976; McIntyre et al., 1976; Williams, 1976; Williams and Johnson, 1975) that compares the down-core records with a dataset of 'modern' values and their associated water column parameters (Hutson, 1977, 1978);

(2) those associated with the stable oxygen isotope composition of a whole shell analysed either individually (Ganssen et al., 2011; Koutavas et al., 2006; Koutavas and Joanides, 2012; Leduc et al., 2009) or pooled (Garidel-Thoron et al., 2007; Koutavas et al., 2002; Stott et al., 2002, 2004), herein $\delta^{18}O_c$ (c = calcite), which can be used to reconstruct SST and past oxygen isotope values in seawater $\delta^{18}O_{sw}$ (sw = seawater) when paired with a proxy that can either reconstruct temperature or salinity;

(3) those associated with trace metal geochemistry (e.g., Ford et al., 2015; Sadekov et al., 2013; Stott et al., 2002, 2004; White et al., 2018), more specifically the natural logarithm of the relative concentration of Mg and Ca (ln(Mg/Ca), of the shell, based upon the temperature dependent (Elderfield and Ganssen, 2000; Nürnberg et al., 1996) incorporation and substitution of a Mg cation into the calcite lattice (Branson et al., 2013, 2016).

The interpretation of these proxies, however, is not straightforward, for example, calibration of foraminiferal assemblage based transfer functions with surface temperatures as opposed to a deeper temperature signal may in fact skew the reconstructed temperature (Telford et al., 2013); $\delta^{18}O_c$ can be affected by species-specific size effects (Feldmeijer et al., 2015; Metcalfe et al., 2015; Pracht et al., 2018), disequilibria or vital effects, which clouds the accurate reconstruction of past SST and $\delta^{18}O_{sw}$.

5 There is also no simple bijective function between $\delta^{18}O_c$ and the oceanic variables $\delta^{18}O_{sw}$ and temperature used in its calculation, with variability in $\delta^{18}O_{sw}$ limiting the use of $\delta^{18}O_c$ as a pure temperature proxy. 
[revised manuscript text omitted]
), where after the resultant $\delta^{18}O_c$ and $T_c$ at each timestep produced by FAME for each grid-point were binned into their respective categories. An Epanechnikov kernel distribution was first fitted to the binned monthly output of a single climate state, the bandwidth varies between grid-points to provide for an optimal kernel distribution. The use of an Epanechnikov kernel

5    distribution to fit the data, as opposed to other types of distribution, represents a trade-off between keeping as many parameters constant whilst mimicking the underlying dataset for a large number of grid points. The conversion of the data from dataset to distribution may induce some small error induced by: rounding to whole integers; the use of a $\delta^{18}O_{mid-point}$ which gives an error associated with the bin size (±0.05 ‰) that is symmetrical close to the distributions measures of central tendency but asymmetrical at the sides; and finally, the associated rounding error at the bin edges within a histogram (±0.005 ‰).

10    Subsequently any two desired distributions can be compared for (dis)similarity using an Anderson-Darling test (1954). Here, all values, i.e., the population, associated with a climatological state are compared with the other populations representing the different climatological state, the results plotted here are Neutral climate state vs. El Niño climate state.

**2.5 Test of input data (Temperature and calculated $\delta^{18}O_{eq}$)**

Foraminifera as modelled entities produces a modulated response that seeks to replicate how foraminifera modify the climate
15    signal, several studies have approximated the foraminiferal signal in a different way (e.g., Thirmulai et al., 2013; Zhu et al., 2017a). In order to understand how FAME has altered the signal, and the degree to which the conclusions drawn depend upon the modelled growth rates, the input datasets of the sea water properties (Temperature and calculated $\delta^{18}O_{eq}$), underwent a similar statistical test (Figure 4). Unlike FAME, which integrates over several depth levels using the computed growth rate, the test of the input datasets was with fixed depths without any growth rate weighting. These fixed depths are 5, 149 and 235
20    m, giving a Eulerian view (Zhu et al., 2017a) in which to observe the implications of FAME's dynamic depth habitat. As per the FAME output, each timestep value was placed into its climate state and an Anderson-Darling test performed to compare the (dis)similarity of on the resultant distributions.

**2.6 Alternative statistical tests**

In order to compare our results with previously published studies using planktonic foraminifera we employed a series of simple
25    statistical tests, mimicking those applied to sediment archives by the palaeoclimate community. A chief parameter that has been employed in previous ENSO proxy work using foraminiferal analysis (more specifically, individual foraminiferal analysis; IFA) is the measure of individual foraminifera downcore standard deviation ($\sigma(\delta^{18}O_c)$). Increased $\sigma(\delta^{18}O_c)$ is considered to correlate to increased variation in SST and, in turn, increased ENSO incidence and/or magnitude (Leduc et al., 2009; Zhu et al., 2017a) or increased interannual variance (Thirmulai et al., 2013). The variance ($\sigma^2(\delta^{18}O_c)$) of the timeseries
30    were computed both as the total variance and as the interannual variance, the latter is computed as outlined in Zhu et al. (2017a). For the interannual variance, the mean monthly climatology is subtracted from the dataset, producing monthly anomalies and a linear trend removed (using the detrend function of MatLab 2019a) – the resultant data was left unfiltered (*i.e.*, Zhu et al., 2017a used a 1-2-1 filter). Four 'picking' experiments were performed, as FAME computes the average value

for a given time step and given the single foraminiferal isotope variance for an equivalent time step (e.g., weeks: Steinhardt et al., 2015) it is more than likely that this computation reduces the real spread in values. Therefore, rather than use the terminology specimen we prefer to use months. Given the complexity in reconstructions of trace metal geochemistry (Elderfield and Ganssen, 2000; Nürnberg et al., 1996): the potential error associated with determining which carbonate phase is first used when foraminifera biomineralise (Jacob et al., 2017); growth-band integration; secondary factors (e.g., salinity, carbonate ion) the focus of the picking here has been on the $\delta^{18}O_e$. Irrespective of which experiment, 60 months were drawn, with replacement, and the number of Monte Carlo iterations is set at 10,000. We assume that the 'picker' is taxonomically well-trained and/or has a procedure in which species can be checked taxonomically post-analysis (e.g. photographing all specimens prior to analysis, Pracht et al. (2019)) and therefore do not include an error that deals with incorrect identification. Although we note that parameterisation of misidentification would be difficult, as it requires understanding of the variability in both standard deviation and absolute values for species co-occurring downcore (Feldmeijer et al., 2015; Metcalfe et al., 2015; 2019). For each run of experiment's (i) to (iii) the drawn months were saved in order to perform (iv):

[revised manuscript text omitted]

An associated statistical methodology is the graphical summary (as opposed to a numerical summary via a test value) of plotting the quantiles of two probability or the quantiles of sample probability distribution against a theoretical distribution distributions also referred to as a Quantile-Quantile, or Q-Q plot (*e.g.*, Ford et al., 2015; White et al., 2018). A complimentary (*i.e.*, used in association with, not as replacement, Filliben 1975) test metric, the Probability plot correlation coefficient (Filliben, 1975) can be used as a numerical summation of this approach, which bases its rationale on near linearity between the two tested distributions. This graphical technique is not used here for the following reasons, (i) the climatic categories (*i.e.*, El Nino, Neutral, La Nina) imposed upon the data give uneven sized sample distributions requiring an interpolated quantile estimate; and (ii) the large graphical computation required (170·40).

2.

**7 An approximation of sedimentary archives: Water depth & Sedimentation Rate**

**Discrete. Experiment 5: Approximation of sediment** intervals retrieved from systematically bioturbated deep-sea sediment cores contain foraminifera with ages spanning many centuries (Lougheed et al., 2018; Peng et al., 1979). This is in contrast to other proxies such as corals (Cole and Tudhope, 2017), speleotherms (Chen et al., 2016) and molluscs (Butler et al., 2013; Milano et al., 2017), where distinct time-specific banding is present (true 'time-series' proxies). The ambient signal following translation into a foraminiferal signal within the water is therefore further modulated by several post mortem processes, which include: the latitudinal-longitudinal shift in position of sinking foraminifera - the so-called 'funnel affect' (van Sebille et al., 2015; Deuser et al., 1981);archives dissolution of calcium carbonate either in the water (Schiebel et al., 2007), at the seafloor, or due to pore fluids; and bioturbation. As mentioned, mixing by bioturbation, results in an apparent smoothing of the downcore, discrete-depth multi-specimen signal (Hutson, 1980a; Löwemark, 2007; Löwemark et al., 2005, 2008; Löwemark and Grootes, 2004; Cole and Tudhope, 2017; Mix, 1987), thus leading to the possibility of interpreting single outlying foraminifera values within a specific depth as representing an 'extreme' climate, when they may in fact represent climate from a different time or epoch. This is especially apparent in $\delta^{18}O_c$ where there is a difference temporally of $\delta^{18}O_{sw}$ (e.g., the ice volume effect in glacial and interglacial cycles ~1.25 ‰) meaning that the same temperature can have radically different $\delta^{18}O_c$ values, a consequence of this is that a series of high magnitude, but low frequency El Niño events could be disturbed in a discrete-depth record. Therefore, in order to reliably extract short-term environmental information from foraminiferal-based proxies, the signal that one is testing or aiming to recover must have a large enough magnitude, be largely unaffected by dissolution (*i.e.*, above the

**7.1 Objective**

5    In Experiment 5 we compare our FAME results with bathymetric and sedimentological features of the Tropical Pacific. The preceding analysis has focused upon ~60-year reanalysis data, such a comparable resolution would require a core to have a similar temporal resolution of ~60 years. The hypothetical core should also be above the lysocline to allow for the recovery of a proxy signal equivalent to the original climate signal. At lower sedimentation rates the modification of the original, ambient, climate signal is not limited to just its translation into a foraminiferal proxy signal and the shift in position of sinking

10   foraminifera (van Sebille et al., 2015; Deuser et al., 1981) but can also be affected by the dissolution of calcium carbonate either in the water (Schiebel et al., 2007), at the seafloor, or due to pore fluids; and bioturbation. Much of the deep-sea Pacific is both below the lysocline and has a SAR that is very low (e.g., Hays et al., 1969 at $0.96 \pm 0.43$ cm kyr$^{-1}$) although there are regions that satisfy both bathymetry and enhanced sedimentation (e.g., Koutavas and Lynch-Stieglitz, 2003 at $7.20 \pm 2.82$ cm kyr$^{-1}$). In the following section we investigate where in the tropical Pacific it is possible to extract environmental information

15   with short frequencies from foraminiferal-based proxies, we consider that a core site must be largely unaffected by dissolution (*i.e.*, above the lysocline) so as not to adversely affect the foraminifer population and the sedimentation rate must be high enough to minimise, as much as possible, the disturbance of the downcore temporal record by bioturbation.

[Figure]

**Figure 5 (A) Map of the sedimentation rate and bathymetry of the Tropical Pacific. (A) Inferred sedimentation rate (Olson et 2016). White regions represent continental shelf. (B) GEBCO map of height relative to 0 m with location of seamounts plotted (white stars). (C) A binary colour map of the GEBCO data, yellow is values below cut-off depth value (3500 m below sea-level (bsl)) and purple above the cut-off depth values. See Supplementary Figure 8 for variation in cut-off values.**

**7.2 Methodology**

**7.2.1 Dissolution: Cut-off depth rationale**

Whilst the presence of water depths in the ocean lacking calcite-rich sediment was described in the earliest work (e.g., Murray and Renaud, 1891; Sverdrup, 1942), overlaying maps of measured surface sediment carbonate percentage with water depth in the Pacific Ocean led Bramlette (1961) to coin the term 'compensation depth' (Wise, 1978). In our first step in consideration of post-mortem signal alteration we focus on dissolution. The lysocline, the depth at which dissolution first becomes apparent (Berger, 1968; 1970), and the Calcite (or Calcium Carbonate) Compensation Depth (This work highlighted the 'narrow' depths

of the CCD in the Central Pacific (4-5000 m). Conceptually Berger (1971) placed three levels in the Pacific ocean that were descriptive of the aspects (e.g., chemical, palaeontological and sedimentological) of the calcite budget; the saturation depth, demarking supersaturated from undersaturated; the lyscoline, the depth at which dissolution becomes noticeable (Berger 1968, 1971); and compensation depth (Bramlette, 1961), in which supply is compensated through dissolution. ; Bramlette, 1961The

5 lysocline and carbonate compensation depth (CCD) vary between the different ocean basins; the modern Atlantic Ocean in which deep water forms has a relatively deep CCD as a by-product of being 'young' well ventilated bottom waters whereas the Pacific Ocean (the final sectionportion of the global thermohaline circulation conveyor belt,) has a shallower CCD.

**7.2.2 Dissolution Approximation**

Dissolution is approximated by determining if each grid cells depth is above or below the prescribed cut-off value. For much
10 of the equatorial Pacific the lysocline is estimated by a foraminiferal assemblage methodology at ~3800 m (Parker and Berger, 1971), however as the lysocline is where dissolution becomes apparent, ergo it is a sample already visibly degraded, we first set the limit of the water depth mask shallower, at 3500 m bsl. In order to highlight the account for potential variability, two further depths were used as cut-off values: 4000 m bsl and 4500 m bsl these depths represent multiple possible depths under which there is the potential for noticeable dissolution (i.e., lysocline) or complete dissolution (i.e., CCD). for dissolution,
15 theThe bathymetry of the Pacific was extracted from the General Bathymetric Chart of the Oceans GEBCO 2014 30 arc-second grid (version 20150318, www.gebco.net) between -20°S to 20°N and 120°E to -70°W (Figure 8). Depths of 3500 m below sea-level (bsl), 4000 m bsl and 4500 m bsl are used here as cut-off values, these depths represent multiple possible depths under which there is the potential for noticeable dissolution (i.e., lysocline) or be dissolvedwww.gebco.net) between -20°S to 20°N and 120°E to -70°W (Figure 5B). A compilation of seamounts was also plotted, as these bathymetric features may
20 provide sufficient height to allow preservation of sediment alongside higher sediment accumulation rates (Batiza, 1982; Clouard and Bonneville, 2005; Hillier, 2007; Koppers et al., 2003; Menard, 1964; Wessel and Lyons, 1997). (i.e., CCD). Whilst our intention here is a generalised view to be used as an approximate guide, it is important to note that the Pacific Ocean has the largest proportion globally of >1 km tall seamounts that are smaller than <100 km (Wessel, 1997). Which may have important, relatively shallow-water sedimentary sequences, which may also be of sufficient sediment accumulation rate,
25 therefore we supplement the GEBCO bathymetric data with the locations of seamounts. However, whilst there are an estimated 50,000 seamounts in the Pacific that are taller than a km (Menard, 1964; Wessel and Lyons, 1997), only 12,000 have been documented on charts (Batiza, 1982), and approximately 291 have been dated (Koppers et al., 2003; Clouard and Bonneville, 2005; Hillier, 2007). It is these 291, <1% of the estimated seamounts, we have overlain onto the bathymetric data (Figure 8b), although this number is further reduced as we only plot between 20°S and 20°N.

30 **The second step when considering post-mortem signal alteration is the sediment accumulation rate (SAR). We first plot the time-averaged deep-sea SAR (Olson et al., 2016), adapted by Lougheed *et al.* (2018) for the Tropical Pacific (Figure 9). New geochronological tools, such as dual $^{14}$C-$\delta^{18}$O measurements on single foraminifera (Lougheed et al., 2018), show that low sedimentation rate cores can have large variances in age between individual foraminifera present within**

**a discrete 1 cm depth interval (Berger and Heath, 1968; Lougheed et al., 2018). In order to model bioturbation, a number of papers** 7.2.3 Bioturbation

If we factor in the sedimentation rate of the Pacific, which in some regions has been estimated to be lower than 1 cm/ka (Blackman and Somayajulu, 1966; Berger, 1969; Hays et al., 1969; Menard, 1964), then dissolution may become further

5   exacerbated. A secondary factor is bioturbation, systematically bioturbated deep-sea sediment cores can produce discrete sediment intervals with foraminifera that have ages spanning many centuries and/or millennia (Berger and Heath, 1968; Lougheed et al., 2018; Peng et al., 1979). In order to model the effect of bioturbation upon the age distribution of discrete core depths, a number of studies have used a diffusion style approach that reduces the parameters down to sediment  accumulation rate (SAR) and sediment mixing depth (herein referred to as bioturbation depth, BD), although this

10  may be an artificial division purely driven by mathematical need rather than biological constraints (Boudreau, 1998). The  BD has been shown to have a global average of 9.8  ± 4.5 cm (1σ) that is independent of both water depth and sedimentation rate (Boudreau, 1998),  likely controlled as a result of the energy efficiency of foraging, *e.g.* deeper burrows may cost more energy to produce than can be offset in extracted food resources, and potential decay in labile food resources with sediment depth.

15

Following the current available geochronological method (i.e., age-depth method) single specimens that are displaced in depth are assigned the average age of the depth that they were displaced to, which will result in erroneous interpretations of climate variability when analysis such as IFA is applied (Lougheed et al., 2018). To investigate how much temporal signal is integrated

20  variability when analysis such as IFA is applied (Lougheed et al., 2018). To investigate how much temporal signal is integrated into discrete-depth intervals for typical tropical Pacific SAR,  (Olson et al., 2016; adapted by Lougheed *et al.*, 2018) the single foraminifera sediment accumulation simulator (SEAMUS, Lougheed,  2020) was utilised to bioturbate,  a climate signal  carry out a transient bioturbation model with the SAR and BD of the Pacific with only half a century of data (such as the ORAS4 temperature and salinity ocean reanalysis

25  data) a longer highly temporally resolved climate input signal was used, to explore the effect of bioturbation upon a given climate signal. The 0-40,000-year δ¹⁸O_w of NGRIP (North Greenland Ice Core Project Members, 2004; Rasmussen et al., 2014; Seierstad et al., 2014 ) is considered to be a satisfactory replacement signal to simulate a foraminiferal signal in 10-year timesteps.  It must be stressed that the use of the NGRIP timeseries here

30  is purely as an input parameter to investigate the effect of bioturbation upon  an oxygen isotope-based climate signal   It is important to stress that by using NGRIP as an input signal for SEAMUS we are neither implying that tropical Pacific cores should have a signal similar to NGRIP, nor that we are translating the NGRIP signal to the tropical Pacific or inferring some kind of causal relationship. As we seek to investigate the effect of bioturbation, no attempt has been made to modulate the input signal's absolute values to mimic expected δ¹⁸O_w values and this is why each plot of the synthetic down core time

series retains the use of V-SMOW, despite carbonates being required to be V-PDB (Coplen 1995). Keeping all things constant, and varying a

A single parameter was varied whilst all others were kept constant between experiments with SEAMUS, the sediment accumulation rate (. Values of SAR) was were varied to fixed values of either 1, 2, 5 or 10 cm kyr$^{-1}$ (that are representative of typical Pacific SAR) and a bioturbation depth (BD) . As the oxygen saturation state of the Pacific Ocean bottom waters is above 40 % (Supplementary Figure 9), suggestive that oxygen may not be a limiting factor, values of BD of either 5, 10 or 15cm were used. These values are based upon the global estimate of BD and it's its error bounds (Boudreau, 1998). For each experiment, the selected values of SAR and BD were kept constant for the entire SEAMISSEAMUS model run (i.e., the intensity and magnitude of bioturbation was not varied). In) although in reality, SAR and BD may vary temporally depending on local conditions (e.g., food, oxygen). Finally, the FAME results for the three species are overlaid with a water depth mask that highlights whether grid points are above or below 3500 m below sea-level (mbsl), to also show seafloor areas under the CCD depth, where carbonate material is not preserved (Berger, 1967, 1970b). A comparison between water depth and time-averaged deep-sea SAR (Olson et al., 2016), adapted by Lougheed *et al.* (2018) is shown in Figure's 7 and 9.. Each experiment was plotted as a histogram of frequency of age of specimen in BD that represent different thicknesses of sediment (5, 10 and 15 cm) and a timeseries using the computed discrete 1 cm depth median age (Figure 9).

**7.3. Results**

**The results of the forward model (Figure 2 and 3) are compared with the input values (Figure 4)discussion**

A factor in order to identify regions in which the values are statistically distinct for different climate states (Figure's 5-7). These results are then shown against the water depth (Figure's 7 to 9) and the SAR (Figure's 9-11) for the region. The results utilise Foraminifera as Modelled Entities (FAME; Figures 2, 3, 5, 6, and 7); the original Ocean Reanalysis data with computed $\delta^{18}O_{eq}$ (Figure 4); the General Bathymetric Chart of the Oceans (GEBCO; Figures 7 to 9); and the single foraminifera sediment accumulation simulator (SEAMUS; Figures 9 to 11).

**3.1 FAME Output: Variance**

We compute growth-weighted $\delta^{18}O_e$ (Figure 5 and 7) and temperature (Figure 6 and 7) distributions for each grid cell in the fifty-eight year simulation using FAME (Roche et al., 2018), constraining the calculation to the Tropical Pacific Ocean (between -20°S to 20°N and 120°E to -70°W). Our model produces 696 individual monthly maps for all three species (Figure 2). While two of the three species (*G. ruber* and *G. sacculifer*) have similar ecologies, they show differences in their resultant $\delta^{18}O_e$ for the same ocean conditions (Figure 2). A comparison of our computed variance with measured data (Supplementary Table 1) is made, we compare both the value of the nearest grid-cell and because of the size of the grid and drift of foraminifera (van Sebille et al., 2015) an average of a 3 by 3 grid in which the nearest grid-cell to the core location is in the center. A

comparison is made with both the iCESM model output and the core's that match this output (Zhu et al., 2017a). For the Late Holocene sample (~1.5 ka) MD02-2529 (08°12.33'N 84°07.32'W; 1619 m) in which *N. dutertrei* individual foraminifera were analysed from >250 μm (Leduc et al., 2009) giving a calculated standard deviation of measured foraminifera of 0.38 ‰. Whereas, the full ~60 year time series (n = 696) of FAME presented here, gives a standard deviation for all species, at depth cut off 60 m between 0.26 and 0.32 ‰; at depth cut off 100 m between 0.20 and 0.29 ‰; at depth cut off 200 m between 0.20 and 0.25 ‰; and at depth cut off 400 m between 0.20 and 0.24 ‰ (see Table 1). Although these values vary if the average of the surrounding grid cells is used (see Table 1). In comparison the iCESM results have the following standard deviation values, for a Eulerian (fixed) depth of 50 m: 0.4 ‰; Eulerian 100 m: 0.6 ‰; and Lagrangian value of 0.49 ‰. There are three samples (Koutavas and Joanides, 2012; Sadekov et al., 2013) located south of core site MD02-2529, these are the Late Holocene (~1.6 ka) samples of V21-30 (01°13'S 89°41'W; 617 m) and (~1.1 ka) V21-29 (01°03'S 89°21'W; 712 m) in which *G. ruber* was measured individually (Sadekov et al., 2013). For these two sites the measured standard deviation is 0.507 ‰ and 0.510 ‰ for V21-30 and V21-29 respectively (Koutavas and Joanides, 2012). The third core site at a similar location is (~1.6ka) CD38-17P (01°36'04 S 90°25'32W; 2580 m) was not analysed individually, instead replicates of pooled samples of 2 or 3 shells of *N. dutertrei* (Sadekov et al., 2013) were made these measured values give a standard deviation of 0.28 ‰. The full ~60 year time series (n = 696) of FAME presented here, gives a standard deviation for all species, at depth cut off 60 m between 0.33 and 0.41 ‰; at depth cut off 100 m between 0.27 and 0.40 ‰; at depth cut off 200 m between 0.25 and 0.35 ‰; and at depth cut off 400 m between 0.25 and 0.34 ‰ (see Table 1). Although these values vary if the average of the surrounding grid cells is used (see Table 1). In comparison the iCESM results have the following standard deviation values, for a Eulerian (fixed) depth of 50 m: 0.53 ‰; Eulerian 100 m: 0.75 ‰; and Lagrangian value of 0.35 ‰.

The study of ENSO has focused on whether the variability is entirely in response to ENSO or whether it is dominated by interannual variability (Xie, 1994, 1995; Wang et al 1994, 2010), here the interannual (Figure 3C) and total variance (Figure 3A) was computed and a ratio between the two calculated (Figure 3B; see Supplementary Table 1). Like the same analysis of interannual and total variance computed for iCESM and SODA reanalysis (Carton et al., 2000), outlined in Zhu et al. (2017a), there is also high ratio of interannual to total variance in our computed FAME dataset (Figure 3B). Although there are regions in the Eastern Equatorial Pacific wherein this ratio reduces. Despite this reduction, the ratio between total and interannual variance is still above > 0.5.

[revised manuscript text omitted]

For FAME-$\delta^{18}O_c$ the results where the populations are dissimilar are either plot as grey and hashed for *G. ruber* and *G. sacculifer* or white for *N. dutertrei* (Figure 5). This is because for these two species (*G. ruber* and *G. sacculifer*) we have the possibility to determine how robust these results are. We use the 1σ values of the observed (FAME) minus expected (MARGO), as computed by Roche et al. (2018) with the MARGO core top $\delta^{18}O_c$ database, as the potential error associated with the FAME model. Regions in which the difference between the two populations are larger than the potential error are associated with grey, whereas those less than the potential error as hashed regions (Figure 5A and B), these errors should be seen as a guide rather than a rejection of a site. Because the MARGO database does not contain *N. dutertrei* we have given the

panels concerning this species a separate colour scheme, black represents grid-cells for which the two populations cannot be said to be statistically different, white grid-cells are those in which the two populations can be said to be statistically different (Figure 5C). As we do not have a similar way to calculate the error for $T_e$, FAME-$T_e$ results are shown (in Figure 6) with this same binary pattern (*i.e.*, white grid-cells are those in which the two populations can be said to be statistically different and

5 black are those in which the two populations can be said to be similar). To reduce the complexity, the overlay of the species Anderson-Darling results (Figure 7) also uses the binary colour scheme (white or black).

Our results show that much of the Pacific Ocean can be considered to have statistically different population between $FP_{EN}$ and $FP_{NEU}$ for both $\delta^{18}O$ (Figure 5) and $T_e$ (Figure 6). We consider that the likely cause for such a remarkable result is due to FAME computing a weighted average and, therefore, the lack of a signal found exclusively within the regions demarked in Figure 1

10 as El Niño regions could represent how the temperature signal is integrated via an extension of the growth rate; growing season and depth habitat of distinct in foraminiferal populations. Taking into account the FAME-$\delta^{18}O_e$ error for *G. ruber* and *G. sacculifer*, we have computed regions in which the difference in oxygen isotopes between the two populations ($\Delta\delta^{18}O_e$) compared with the AD-test is smaller than the aforementioned error (Hatching in Figure 5), *i.e.* where the mean difference between $FP_{EN}$ and $FP_{NEU}$ is within the error. The hatched regions in Figure 5 considerably reduce the areal extent of significant

15 difference between $FP_{EN}$ and $FP_{NEU}$, with the remaining regions aligning with the El Niño 3.4 region (Figure 1). It is important to note that this error relates to the model and in reality, the difference between the climate states could be larger or smaller. No such test was performed on the *N. dutertrei* dataset, because of its absence from the MARGO dataset. To further test the model-driven results and to assess if they are still consistent when the depth limitation is varied, the analysis was rerun with depths of 100, 200 and an extreme value of 400 m (Figure 5-7). Whilst it is possible to discern differences between the depths,

20 it is important to note that a large percentage of the tropical Pacific remains accessible to palaeoclimate studies. A shallower depth limitation in the model increases the area for the 'warm' species, suggesting that the influence of a reduced variability in temperature or $\delta^{18}O_{eq}$ with a deeper depth limit causes the differences between $FP_{EN}$ and $FP_{NEU}$ to be reduced. Overlaying the results of the Anderson-Darling test for all three species (Figure 7) per depth for 60, 100 and 200 m highlights the areas where multi-species comparisons could be made. To account for potential differences in depth habitat we make a combination

25 of shallower depth for *G. ruber* and deeper depths for *G. sacculifer* and *N. dutertrei* (Pracht et al., 2019) in the final panels (Figures 7D and 7H).

**3.3 Test of input parameters (fixed depth: temperature and $\delta^{18}O_{eq}$)**

The model-driven results were assessed with the underlying input dataset (temperature and $\delta^{18}O_{eq}$), these underwent the same statistical test (Figure 4), although with fixed depths of 5 m, 149 m and 235 m (see section 2.5). The results for each grid point

30 are presented as either black, grey or hashed. Areas where the population distributions of the two climate states are found to be statistically similar have black grid cells. Regions in which the difference between the two populations are larger than the potential error are associated with grey, whereas those less than the potential error as hashed regions. The threshold error (*i.e.*, the difference between the means of each distribution) is for temperature (Figure 4A-C) 0.5 °C and for $\delta^{18}O_{eq}$ (Figure 4D-F)

0.10 ‰, these errors should be seen as a guide rather than a rejection of a site. The results of this fixed depth, non-FAME, test show that the shallowest depths produce populations that are significantly different both in terms of their mean values and their distributions. In the upper panel of Figure 4, the shells is whether the shells can be preserved. Irrespective of the bathymetric cut-off value used for the GEBCO bathymetry data it is evident that much of the canonical El Niño 3.4 region is clearly visible at 5 m depth. Whilst differences exist between Anderson-Darling results for the input data (Figure 4) and the FAME $\delta^{18}O$ (Figure 5) and $T_c$ (Figure 6), for used in oceanography, as well as a large proportion of the Tropical Pacific, is excluded from suitability as a perspective core site (Figure 5B and 5C). Even in regions where bathymetry may be above the cut-off value dissolution may occur. For instance, in regions of high fertility, such as the Eastern Equatorial Pacific, the lysocline was estimated to be present at ~2800 m (Thunell et al., 1981) or ~3000 m (Berger, 1971; Parker and Berger, 1971).

In the EEP region the shallower lyscoline is accompanied by an equally shallower CCD (located at ~3600 m) for which the high fertility/primary production is considered responsible for its shoaling, lowering the $p$H through increased $CO_2$ (Berger et al., 1976). The correspondence between lyscoline depth and CCD depth does not hold true for the entirety of the Pacific, plotting a N-S cross-section from 50°N to 50°S Berger (1971) noted that in the Central Equatorial Pacific, the high fertility region generates a larger zone of dissolution resistant facies even with a shoaled lysocline. close to the Panama isthmus, there are significant similarities between the plots. These plots also show that our FAME data (Figure 5-7), in which we allow foraminiferal growth down deeper than the depths in Roche et al. (2018), are a conservative estimate and thus are on the low-end (Figure 4), to account for potential discrepancies with depth habitats. In the original paper on depth habitats based upon temperatures derived from $\delta^{18}O_c$, Emiliani (1954) cautioned that the depth habitats obtained would represent a weighted average of the total population, and while foraminiferal depth habitats are likely to vary spatiotemporally, the average depth habitat is skewed toward the dominant signal (Mix, 1987).

**3.4 Water depth and SAR**

Our analysis uses reanalysis data for the time period 1958-2015, a hypothetical core that had a comparable resolution would essentially be analogous with a sediment core with a rapid sediment accumulation rate (SAR), representing 50 yr cm$^{-1}$ (or 20 cm kyr$^{-1}$). Based on our analysis, such a hypothetical core could allow for the possible disentanglement of El Niño related signals from the climatic signal, but only in a best-case scenario involving minimal bioturbation, which is unlikely in the case of oxygenated waters. Extracting the oxygen saturation (SO$_2$) state, of the Pacific Ocean bottom waters from the Annual Climatology WOA13 give values that are predominantly >40 % (Figure 9B). Oxygen saturation is the concentration of Oxygen in a medium against the maximum that can be dissolved in the same medium. Whilst annual variability may exist, it is unlikely that bioturbation would be prevented by low oxygen. Therefore, using a cut-off value that has beenA second factor is the sedimentation rate, using a cut off value that has been previously considered sufficiently high enough to outpace bioturbation (*e.g.*, Koutavas and Lynch-Stieglitz, 2003) of 5 cm kyr$^{-1}$ (Figure 9A), it can be demonstrated that much of the Pacific has an inferred lower sedimentation rate (< 5 cm kyr$^{-1}$; Figure 9C) than this cut off value. To test the influence of bioturbation, the bioturbation simulator SEAMUS was run using the NGRIP time series. The results of SEAMUS highlight the potential single

foraminifera depth displacement that low sedimentation rates can result in (Figure 9). Following the current available geochronological method (i.e., age-depth method) such specimens that are displaced in depth are assigned the average age of the depth that they were displaced to, which could result in erroneous interpretations of climate variability when analysis such as IFA is applied (Lougheed et al., 2018). The results of SEAMUS are plotted both as time series of the bioturbated 'NGRIP' signal (Figure 10) and as histograms of the probability of finding a particularly pseudo-foraminifera with a given age in the bioturbation depth (Figure 11). As the bioturbation depth varies between 5, 10 and 15 cm for the different simulations of SAR, the histogram in each panel (in Figure 11) represent different thicknesses of sediment, i.e., for Figure 11 panels a, d, g, and j histograms represent data with a BD thickness of 5 cm. Likewise, the timeseries is plotted with the discrete 1 cm depth median age; the median age of the bioturbation depth (Figure 11) is the reason why each timeseries does not 'start' at 0 age (Keigwin and Guilderson, 20095A).

[Figure]

FAME - Bathymetry mask: Temperature (°C)

FAME - Bathymetry mask: Oxygen isotope equilibrium ($\delta^{18}O_{eq}$)

FAME & Depth = **possible** to discern El Nino values

Depth < 3500 m = above lysocline

FAME &/or Depth = not **possible** to discern El Nino values

Depth > 3500 m = below lysocline     Values similar

**Figure 6 Overlay between bathymetry and FAME results. The  results of the FAME Anderson-Darling test for (A) temperature and (B) oxygen isotope values as input. Locations where the $H_1$ hypothesis can be accepted, i.e. the distributions can be said to be different ($FP_{El\ Niño} \neq FP_{Non-El\ Niño}$), are plotted as yellow where the depth is deeper than 3500 m bsl or purple where the depth is shallower than 3500 m bsl (see Figure 2). Purple locations are where our results suggest that the signal of ENSO has different values and the water depth allows for preservation.**

Overlaying the water depth and the SAR with the Anderson-Darling results (Figure 6 and Supplementary Figure 7) highlights that of the total area where $FP_{EN}$ is significantly different from $FP_{NEU}$ (*i.e.* those areas where planktonic foraminiferal flux is suitable for reconstructing past ENSO dynamics), only a small proportion corresponds to areas where the sea floor is both above the CCD (< 3500 mbsl) and SAR is at least 5 cm/ka (Figure 7). However, at certain locations, near islands or seamounts,

the SAR and water depth may be high enough to allow for a signal to be preserved (Figure 5B) that may not be represented here.

[Figure]

Figure 7 Overlay between water depth and inferred SAR (Olson et al., 2016). Cut-off limits for bathymetry and SAR are 3500 m below sea-level and (A) ≥1 cm kyr⁻¹ and (B) ≥2 cm kyr⁻¹ respectively. The colours represent the following: Red / Pink: Continental shelf sediments that are (Red) shallower or (Pink) deeper than 3500 mbsl; Grey / White: grid point SAR is lower than SAR threshold and the seafloor depth is (grey) shallower or (white) deeper than 3500 mbsl; Light Yellow/Gold: Light yellow represents areas where the SAR is above the threshold but the water depth is deeper than 3500 mbsl in comparison Gold represents areas where the SAR is above the threshold and the water depth is deeper than 3500 mbsl. The ideal locations are therefore plotted as Gold.

[Figure]

Figure **8** Output of the bioturbation model SEAMUS. (A) The unbioturbated input signal, NGRIP (North Greenland Ice Core Project Members, 2004; Rasmussen et al., 2014; Seierstad et al., 2014), used in our simulation of bioturbation for different SAR with SEAMUS (Lougheed, 2019). Sediment mixed layer referred to here as bioturbation depth (BD) is fixed at (B, E , H, K) 5 cm, (C, F, I, L) 10 cm and (D, G, J, M) 15 cm for sedimentation accumulation rates (SAR) of (B-D) 1 cm kyr[-1]; (E-G) 2 cm kyr[-1]; (H-J) 5 cm kyr[-1] and (K-M) 10 cm kyr[-1]. The output is plotted as the discrete 1 cm depth median age. In (B-M) grey values represent the unbioturbated input signal, NGRIP. Note, we retain the original units (V-SMOW) of the original timeseries used, no inference between Pacific climate and Greenland is intended by the use of NGRIP.

The results of the bioturbation simulator SEAMUS, plotted as a time series of the bioturbated 'NGRIP' signal (Figure 8) and as histograms of the probability of finding a particularly pseudo-foraminifera with a given age within the bioturbation depth (Figure 9), highlight the potential single foraminifera depth displacement that occurs with low sedimentation rates (Figure 5). Within a single depth in a core, proxy values largely represents represent the integrated time signal for that depth (Figure 11),

5   as opposed to the variance of, the age of specimens within the bioturbation depth may vary from a few to tens of thousands of years (Figure 9). A data-model comparison without sufficient knowledge of bioturbation may equate an integrated proxy signal with, a climatic signal for an inferred (or measured) average age for the depths in question. The proxy variance For proxies that use an average values (i.e., a pooled foraminiferal signal) or a variance (i.e., individual foraminifera values), the individuals will be based both upon a non-uniform distribution in temporal frequency of specimens, i.e., older specimens are few compared

10  to younger specimens. A large proportion of the specimens in the BD come from years that are 'proximal' (i.e., close to the youngest age) this which may give undue confidence that the probability of picking a specimen from these years is higher, however the long-tail of the distribution means that there is an equally high chance of picking a specimen that has come from several thousand years earlier than the discrete-depth's median age. If we consider for the moment this as picking specimens from a box, there is a high chance of picking from a single box that represents the age you want however there is an equally

15  high chance of picking from numerous boxes with varying age. If the spread in the climatic variable is uniform throughout this time then it can be possible to reproduce a similar signal, although this would not by definition represent the actual spread in the actual climatic variable for a given time, however the spread in the climate variable is unlikely to be constant. With a varying spread in the climatic signal bioturbation can introduce the possibility of spurious interpretations, but it is of course more obvious where the measured distributions over-exaggerate the climate signal (e.g., Whilst the temporal integration

20  involved in bioturbation can be problematic for either age-depth modelling (e.g., Lougheed et al., 2018; Lougheed et al., 2020a) or discrete age measurements (e.g., Lougheed et al., 2020b) it will also integrate the climate signal carried by the individual foraminifera. Wit et al., 2013). Furthermore, if we consider that researchers do not pick as randomly as they profess, there is both a size and preservation bias to specimens selected, and size is not constant down-core (e.g., Metcalfe et al., 2015) we can further introduce bias within the dataset. The SEAMUS output that corresponds with our chosen SAR cut-off value of 5 cm

25  kyr $^{-1}$ (Figure 10 and 11), the lower limit of our mask (Figure 9), is shown in panels in Figures 10H to 10J and Figures 11G to 11I. It is important to note however, that much of the region for which FAME is calculated upon has inferred sedimentation rates lower than this cut-off value (Figure 9C to 9H).

If for example the spread in a climate variable, such as temperature, is uniform throughout the integrated time (and the abundance at each temperature value is also uniform) then it could be possible to reproduce a similar temperature distribution

30  in bioturbated cores. Although this would not by definition represent the actual spread in the actual climatic variable for a given time. However, the climate signal is unlikely to be constant, integrating a climatic signal bioturbation can therefore introduce artefacts inducing the possibility of spurious interpretations. Of course, identification of spurious datapoints are more obvious where the measured distributions over-exaggerate the climate signal (e.g., Wit et al., 2013). Our simulation of a climate signal reveals (Figure 8) the following: a reduction in signal amplitude with low SAR and/or increasing BD; loss of short

events at low SAR; a shift in the apparent timing of events with increasing BD; and an apparent increasing 'core-top' age with low SAR and increasing BD (Figure 9). The median age of the bioturbation depth (Figure 9) is the reason why each timeseries (Figure 8) does not 'start' at 0 age (Keigwin and Guilderson, 2009).

[Figure]

5 **Figure 9 Histograms of simulated specimen age within the bioturbation depth. The simulated age distribution present within the sediment mixed layer, referred to here as bioturbation depth (BD). BD is fixed at (A, D, G, J) 5 cm, (B, E, H, K) 10 cm and (C, F, I,**

**L) 15 cm for sedimentation accumulation rates (SAR) of (A-C) 1 cm kyr$^{-1}$; (D-F) 2 cm kyr$^{-1}$; (G-I) 5 cm kyr$^{-1}$ and (J-L) 10 cm kyr$^{-1}$. The output is plotted as the discrete 1 cm depth median age. Note the size of the BD varies, therefore the simulated age distribution comes from a varying 'core depth'.**

Whilst we are principally interested in understanding whether living foraminifera can theoretically reconstruct ENSO, the results of the sedimentological features, presented here, imply that much of the Pacific Ocean is not suitable for preserving (Figures 5-9) the ENSO signal, despite the possibility of the species of foraminifera in the water having unique values for different climate states (Section 4; Figure 6). In areas where preservation could occur, a hypothetical core could allow for the possible disentanglement of El Niño related signals from the climatic signal, but only in a best-case scenario involving minimal bioturbation, which is unlikely in the case of oxygenated waters. Combined with finite sampling strategies the effects of both dissolution and bioturbation can be further amplified.

**8. Discussion**

**8.1 Palaeoceanographic Implications**

Ecophysiological proxy system models are a mathematical approximation aimed at replicating the proxy signal both as its response to, and modification of, the original target climate signal (e.g., Dees et al., 2015). Linking ecophysiological models to coupled ocean-atmosphere models (e.g., Clement et al., 1999; Zebiak and Cane, 1987); isotope enabled Earth system models (e.g., iCESM; Zhu et al., 2017); or multi-model ensembles with prescribed boundary conditions could be used for the generation of timeseries for testing presumptions in proxy studies. Used a-priori, an explicit forward model can be used to test if it is plausible that the given recording system can record an oceanographic signal to allow robust reconstructions.

A critical presumption in proxy studies is embedded in site selection. Sites selected are presumed to be able to (or not) generate a climate signal, the presumptive answer in such studies is either the feature occurs or did not occur, and if it occurs then it has either enhanced or weakened. Such presumption precludes a scenario in which the feature or oceanographic regime has shifted, passing over or beyond a core site (Weyl, 1978), reacting to the expansion, contraction or shift of certain large scale oceanographic features (e.g., Polar Front, Upwelling) during periods of either warmer than average (e.g., the last interglacial) or colder than average temperatures (e.g. glacial maxima). The analysis of recent El Niño patterns suggests that there are two types of spatially delineated El Niño events: the dateline Central Pacific El Niño and the Eastern Pacific El Niño. Here we have highlighted a way of using models to determine the location where the different climate states could be differentiated. More explicit tests using climate models could be used to optimise sampling design, determine applicable core locations for comparison of proxy values with '*like with like*' oceanographic features (similar to the analysis of Evans et al. (1998) for predicting coral sites), without necessarily the cost of a time-slice project (e.g., CLIMAP, MARGO).

[revised manuscript text omitted]

5 the 'narrow' depths (4-5000 m) in the Central Pacific of the CCD. Conceptually Berger (1971) placed three levels in the Pacific ocean that were descriptive of the aspects (e.g., chemical, palaeontological and sedimentological) of the calcite budget; the saturation depth, demarking supersaturated from undersaturated; the lysocline, the depth at which dissolution becomes noticeable (Berger 1968, 1971); and compensation depth (Bramlette, 1961), in which supply is compensated through dissolution. The aspects of the lysocline was estimated by the faunal assemblages of Parker and Berger (1971, figure's 14 and

10 15 of that publication), for much of the equatorial Pacific the lysocline is estimated at ~3800 m. As the lysocline is where dissolution becomes apparent, ergo it is a sample already visibly degraded, we therefore set the limit of the water depth mask shallower, at 3500 m bsl. In fact, in regions of high fertility, such as the Eastern Equatorial Pacific, the lysocline was estimated to be present at ~2800 m (Thunell et al., 1981) or ~3000 m (Berger, 1971; Parker and Berger, 1971). For instance, core V21-28 close to the Galapagos Islands (01°05'N, 87°17'W) has a shallower dissolution affect than either of these two values despite

15 being collected from a water depth of 2714 m (Luz, 1973). A comparison between the hydrographic and sedimentary lyscocline, using a mooring in the Panama Basin showed that the sedimentary lyscocline is a product of where the hydrographic lyscocline meets the seafloor (Thunell et al., 1981), therefore, this could lead to dissolution within the water of the settling flux (e.g., Schiebel et al., 2007). In the EEP region the shallower lyscocline is accompanied by an equally shallower CCD (located at ~3600 m) for which the highly fertile is considered responsible for its shoaling, lowering the pH through increased $CO_2$ (Berger

20 et al., 1976). The correspondence between lyscocline depth and CCD depth does not hold true for the entirety of the Pacific, plotting a N-S cross-section from 50°N to 50°S Berger (1971) noted that in the Central Equatorial Pacific, the high fertility region generates a larger zone of dissolution resistant facies even with a shoaled lysocline. However, a number of the inferences are contentious, for instance the reduction in upwelling in this region (Koutavas and Lynch-Stieglitz, 2003) is contradicted by Dubois et al. (2009), who used alkenones (i.e., $U_{37}^{K'}$ ratios) to suggest an upwelling intensification. Whilst the

25 $U_{37}^{K'}$ proxy has problems within coastal upwelling sites (Kienast et al., 2012) it does not discount their claim, especially considering that $\delta^{18}O$ records can themselves be influenced by salinity upon the $\delta^{18}O_{sw}$ component (Rincón-Martínez et al., 2011) and the potential influence of carbonate ion concentration ($[CO_3^{2-}]$) upon foraminiferal $\delta^{18}O_c$ (de Nooijer et al., 2009; Spero et al., 1997; Spero and Lea, 1996). The discrepancies in reconstructed climate between marine cores are worth noting, as ultimately it is from proxies that inferences are made about past climate (Trenberth and Otto-Bliesner, 2003; Rosenthal and

30 Broccoli, 2004). Such inferences have suggested that the past climate of the Pacific region (from the geologically recent too deep time) has been in an: El Niño state (Koutavas et al., 2002; Stott et al., 2002; Koutavas and Lynch-Stieglitz, 2003); permanent El Niño state (Huber and Caballero, 2003); Super El Niño state (Stott et al., 2002); La Niña state (Andreasen et al., 2001; Beaufort et al., 2001; Martinez et al., 2003); or a different climatic state altogether (Pisias and Mix, 1997; Feldberg and Mix, 2003). Ultimately the possibility of a marine sediment archive being able to reconstruct ENSO dynamics comes down

to several fundamentals besides whether the signal can or cannot be preserved (*i.e.*, whether the core site has either too low SAR, too high BD or a water depth not conducive to calcite preservation): the time-period captured by the sediment intervals (a combination of SAR and bioturbation); the frequency and intensity of ENSO events; the foraminiferal abundance during ENSO and non-ENSO conditions; as well as what the proxy is recording. Reconstructions of the past can benefit from inclusion

5 within conceptual frameworks that incorporate both data and modelling studies (e.g., Trenberth and Otto-Bliesner, 2003; Rosenthal and Broccoli, 2004; McPhaden et al., 2006). If we factor in the sedimentation rate of the Pacific, which has been estimated to be considerably lower than 1 cm (Blackman and Somayajulu, 1966; Berger, 1969; Menard, 1964), then dissolution may become further exacerbated. The longer a shell remains at the sediment-water interface the greater the prospects for it to be dissolved become, therefore low SAR increases the chance of dissolution (Bramlette, 1961). For instance, in 15 equatorial

10 Pacific cores, below 4000 m, the average SAR was presented (

8 Hays et al., 1969; here calculated) at 0.96 cm kyr$^{-1}$ (1σ ± 0.43 cm kyr$^{-1}$). Although there are regions and/or core locations in which the SAR is higher, for instance eight EEP cores shallower than the lysocline depth (Thunell et al., 1981) of ~2800 m were presented by Koutavas and Lynch-Stieglitz (2003) which have an average SAR, calculated, at 7.20 cm kyr$^{-1}$ (1σ ± 2.82

15 cm kyr$^{-1}$). The average age for these same core's 0 cm core depth is 2184 years (1σ ± 1521 yrs), whilst it cannot be assumed that there has been no loss during recovery (i.e., core top is not sediment-water interface), a non-zero core top age is expected for both bioturbation (Keigwin and Guilderson, 2009) and dissolution. Alongside, the potential for dissolution there is the also the mixing of ocean sediments by the benthos (Bramlette and Bradley, 1942). For instance, Arrhenius (1961) noted that ash beds present in cores of the EEP (Worzel, 1959; Ewing et al., 1959) had a 2-3 cm layer above and below what should have

20 originally been a sharp boundary in which they estimated that ~50% of the material originated from the other side of the boundary. If one assumes 1 cm kyr$^{-1}$ sedimentation rates, then the range in age of the obviously 6 cm mixed sediments is minimally ~6000 years per cm, comparison with an analogous SEAMUS simulation (bioturbation depth 5 cm; SAR 1cm) highlights the considerable spread in age, placing the 95.45% range between 110 and 18954 years (Figure 11). Much of this temporal variability (either through bioturbation or dissolution) will be hidden, especially when proxy values correspond with

25 the expected values, and more obvious when the values are larger than expected (*e.g.*, Wit et al., 2013). Owing to the lack of absolute variability during the Holocene the apparent confirmation of similarity between proxy values and modern distributions of the '*to be reconstructed*' variable is not a confirmation of proxy reliability. Especially in the tropics wherein seasonal variability is limited. The effects of both bioturbation and dissolution are further amplified when combined with finite sampling strategies. Therefore, the results of the sedimentological features, presented here, imply that much of the Pacific Ocean is not

30 suitable for preserving (Figures 7-9) the ENSO signal, despite the possibility of the species of foraminifera having unique values for different climate states (Figures 4-7). ENSO studies using palaeoceanography have exposed shifts, one can, therefore, question what is being reconstructed in such studies.

**4.2 Palaeoceanographic Implications**

**4.2.1 Pacific climate reconstructions**

[revised manuscript text omitted]

5    The possibility of a marine sediment archive being able to reconstruct ENSO dynamics comes down to several fundamentals besides whether the signal can or cannot be preserved (*i.e.*, whether the core site has either too low SAR, too high BD or a water depth not conducive to calcite preservation): the time-period captured by the sediment intervals (a combination of SAR and bioturbation); the frequency and intensity of ENSO events; the foraminiferal abundance during ENSO and non-ENSO conditions; as well as what the proxy is recording. There is also the presumption that a particular climate event should be
10   recorded, our Anderson-Darling test for instance highlights that there are locations that cannot discern the difference between El Niño and other climate states whilst for the same time period there are locations where the different climate states can be differentiated. Whilst our analysis is a statistical treatment of the data, each species, and different types of phyto- or zooplankton preserved in ocean sediments, are likely to record the same set of environmental conditions differently (Mix, 2006). This is, in brief, the rationale for the development of FAME, the same climate signal seen through the view of species-
15   specific proxies will give a fractured view constrained by each species particular ecophysiological constraints (Mix, 1987; Roche et al., 2018). A dynamic depth habitat in which the environmental signal becomes a weighted average of the water column can further confound the original signal (Wilke et al., 2006). What can be seen as contradictory reconstructions can therefore be viewed as the prevailing or dominant conditions at a given location at the time when environmental conditions overlap ecological constraints for a given species.

20   Terrestrial records suggest the number of El Niño events per century in the early Holocene (8-6 ka BP) was minimal (Moy et al., 2002), with between 0 and 10 events occurring per century. This dampened ENSO is observed within lake core colour intensity and records driven primarily by precipitation – although like other proxies this can also be interpreted differently, *i.e.* as a large change in the hydrological cycle shifting precipitation away regionally (Trenberth and Otto-Bliesner, 2003). If we assume for now that the number and magnitude of ENSO events was reduced, the relatively low downcore resolution of marine
25   records may not accurately capture the dynamics of such lower amplitude ENSO events using existing methods. The sensitivity and probability of detecting a change in IFA with changes in frequency and amplitude, has been dealt with before (Thirmulai et al., 2013), although without considering bioturbation. The synthesis of pseudo-timeseries to discern the potential distribution for different scenarios, whilst a necessary approximation, is nonetheless one that is free of cause and causality. Modulating a timeseries for events with enhanced or weakened amplitude or fewer or greater number of events assumes in essence that there
30   is limited feedback both regionally (between two sites) and internally within the timeseries (i.e., a process that operates on a higher level). Reconstructions of the past can benefit from inclusion within conceptual frameworks that incorporate both data and modelling studies (e.g., Trenberth and Otto-Bliesner, 2003; Rosenthal and Broccoli, 2004; McPhaden et al., 2006). The use of coupled ocean-atmosphere models (e.g., Clement et al., 1999; Zebiak and Cane, 1987); isotope enabled Earth system models (e.g., iCESM; Zhu et al., 2017); or multi-model ensembles with prescribed boundary conditions can be used for the

~~generation of timeseries in which the physics of atmospheric and oceanic circulation are constrained and feedbacks between sites can occur. The perceived failure of several climate models to resolve ENSO adequately, resulting in variable ENSO frequency and amplitude between models, could therefore be used to determine the proxy signal from model derived timeseries at different frequencies and intensities of ENSO. Albeit a timeseries of variable ENSO that is grounded in ocean-atmosphere coupling. Such analysis could also provide information on a secondary assumption, in which time slices from the same core inherently assume that where a particular oceanographic feature exists now is also where it may have existed before. This gives a somewhat binary view, the feature either occurs or does not occur, and if it occurs then it has either enhanced or weakened. Yet this can (though not always) preclude a scenario in which the feature has shifted. Analysis of the El Nino patterns suggests that there are two types of El Nino that are spatially delineated: the dateline Central Pacific El Nino and the Eastern Pacific El Nino. The expansion, contraction or shift of certain large scale oceanographic features (e.g., Polar Front, Upwelling) during periods of warmer than average (e.g., the last interglacial) or colder than average temperatures (e.g. the LGM) can complicate the comparison of two down core samples, i.e., a static core continuously recording a particular climate event as opposed to a shifting oceanographic regime 
[revised manuscript text omitted]
 the integrating the $\delta^{18}O_{eq}$ values using a growth-rate based weighting (FAME; Roche et al., 2018). Values are in per mil (‰ V-PDB).

Figure Captions

Figure 10 Anderson-Darling Results for Input datasets of Temperature and Equilibrium $\delta^{18}O$ ($\delta^{18}O_{eq}$). Results of the test in which input variables underwent the same statistical procedure (see section 2.0) as the modelled data for (A) temperature and (B) $\delta^{18}O_{eq}$ values. Here, model input data was extracted for a single depth of ~5 m without any growth weighting applied. Black regions are those grid points in which the null hypothesis ($H_0$), that the El Niño and Non- El Niño (Neutral) foraminifera populations (FP) are not statistically different ($FP_{El Niño} = FP_{Non-El Niño}$), cannot be rejected. Grey regions represent grid points where the $H_1$ hypothesis is accepted, therefore the distributions of the foraminiferal population for El Niño and Non- El Niño can be said to be unique ($FP_{El Niño} \neq FP_{Non-El Niño}$). The hatched regions represent areas were the $H_1$ hypothesis can be accepted, therefore the distributions of the foraminiferal population for El Niño and Non- El Niño can be said to be unique ($FP_{El Niño} \neq FP_{Non-El Niño}$), though the difference between the means of tested distribution are less than (A) 0.5°C or (B) 0.1 ‰. For a comparison with three different fixed depths (5; 149; and 235 m) without any growth weighting applied see Supplementary Figure 2.

Figure 11 Anderson-Darling results plotted regionally in which species-specific results are overlain. Panels represent water depth locations where dissimilar and similar values for the two climate states for (a-b) FAME-T$_c$ modelled temperature (c-d) FAME-$\delta^{18}O_c$ modelled oxygen isotope values recorded in the calcite shells (Tc) occur. Each panel represents the Anderson-Darling test result, the results for *Globigerinoides sacculifer*, *Globigerinoides ruber* and *N. dutertrei* are overlain with (A and C) cut-off depth of 60 m and (B and D) species-specific cut-off values. For all panels black areas reflect latitudinal and longitudinal grid points that failed to reject the null hypothesis ($H_0$) and therefore the foraminiferal population (FP) of the El Niño is similar to the Non-El Niño, and therefore the distribution between the neutral climate and El Niño cannot be said to be different ($FP_{El Niño} = FP_{Non-El Niño}$).

Figure 12 Total variance and Interannual variance . (a) Total variance of *Globigerinoides sacculifer* $\delta^{18}O_c$, using FAME-$\delta^{18}O_{eq}$ for a cut-off value of 60 m. (b) The ratio of (a) and (c), where (c) is the Interannual variance of the timeseries of (a).

Figure 13 The range in standard deviation of the Monte-Carlo experiments using FAME-$\delta^{18}O_c$ *G. sacculifer* with a depth cut-off of 60 m. In (a-f) we plot the range in standard deviation obtained by picking 60 months with replacement with 10,000 iterations, the experiments are as follows: (a) the same months were chosen for all grid-points for each iteration of the Monte-Carlo; (b) each grid-point has its own randomly selected months for each iteration of the Monte-Carlo; (c) the same as (b) but we weight the values by the total amount of growth per month; (d) the months selected for (c) were re-run but a random variability is added to each month (between -0.4 and 0.4 ‰); (e) the months selected for (b) were re-run but a random measurement error is added to each month (between -0.12 and 0.12 ‰); and (f) the months selected for (b) were re-run but the (d) random variability and (e) measurement error were combined.

**Figure 4. Anderson-Darling Results for Input datasets of Temperature and Equilibrium $\delta^{18}O$ ($\delta^{18}O_{eq}$). Results of the test in which input variables underwent the same statistical procedure (see section 2.0) as the modelled data for (A-C) temperature and (D-F) $\delta^{18}O_{eq}$ values. Here, model input data was extracted for three fixed depths ([A & D] 5 m; [B & E] 149 m; [C & F] and 235 m) without any growth weighting applied. Black regions are those grid points in which the null hypothesis ($H_0$), that the El Niño and Non- El Niño populations are not statistically different ($FP_{El\ Niño} = FP_{Non-El\ Niño}$), cannot be rejected. Gray regions represent grid points where the $H_1$ hypothesis is accepted, therefore the distributions of the foraminiferal population (FP) for El Niño and Non- El Niño can be said to be unique ($FP_{El\ Niño} \neq FP_{Non-El\ Niño}$). The hatched regions represent areas were the $H_1$ hypothesis can be accepted, therefore the distributions of the foraminiferal population (FP) for El Niño and Non- El Niño can be said to be unique ($FP_{El\ Niño} \neq FP_{Non-El\ Niño}$), though the difference between the means of tested distribution are less than (A-C) 0.5°C or (D-F) 0.1 ‰. Each panel represents a single depth (5, 149 and 235 m).**

Figure 5. Anderson-Darling Results for modelled FAME $\delta^{18}O_{eq}$: Panels representing locations of where dissimilar and similar values of FAME modelled species $\delta^{18}O$ occur between climate states, for (columns) particular species and (rows) particular model depth cut-off limits. Each panel represents the Anderson-Darling test result, which are plotted with ([A] *Globigerinoides sacculifer* and [B] *Globigerinoides ruber*) and without ([C] *N. dutertrei*) model derived error. For all panels black areas reflect latitudinal and longitudinal grid points that failed to reject the null hypothesis ($H_0$) and therefore the foraminiferal population (FP) of the El Niño is similar to the Non-El Niño ($FP_{El\ Niño} = FP_{Non-El\ Niño}$). The results in which the $H_1$ hypothesis is accepted, in which the, therefore the distributions can be said to be different ($FP_{El\ Niño} \neq FP_{Non-El\ Niño}$), are plotted as either: (A — *G. sacculifer*, B — *G. ruber*) grey and hatched or (C — *N. dutertrei*) solely as white regions. For species with calculatable error, grey regions represent values where the difference between the two means of the population is greater than species-specific standard deviation of the FAME model and hatched regions represent those in which the means are less than this standard deviation (Roche et al., 2018). For species without a calculatable error, the regions are plotted in white. The rows represent the model runs with a depth cut-off limit at: (A-C) 60 m; (D) 100 m; (E) 200 m; and (F) 400 m.

.

Figure 6. Anderson-Darling Results for modelled FAME-$T_c$: Panels representing locations of where dissimilar and similar values of FAME modelled temperature recorded in the calcite shells (Tc) occur between climate states, for (columns) particular species and (rows) particular model depth cut-off limits. Each panel represents the Anderson-Darling test result, which are plotted with ([A] *Globigerinoides sacculifer* and [B] *Globigerinoides ruber*) and without ([C] *N. dutertrei*) model derived error. For all panels black areas reflect latitudinal and longitudinal grid points that failed to reject the null hypothesis ($H_0$) and therefore the foraminiferal population (FP) of the El Niño is similar to the Non-El Niño, and therefore the distribution between the neutral climate and El Nino cannot be said to be different ($FP_{El Niño} = FP_{Non-El Niño}$). The results in which the $H_1$ hypothesis is accepted, in which the distributions can be said to be different ($FP_{El Niño} \neq FP_{Non-El Niño}$), are plotted as white regions. The rows represent the model runs with a depth cut-off limit at: (A-C) 60 m; (D) 100 m; (E) 200 m; and (F) 400 m.

Figure 7. Combined A-D plots. As figure 5 and figure 6, in that panels represent locations of where dissimilar and similar values for the two climate states for (a-d) FAME-$\delta^{18}O_{eq}$ modelled oxygen isotope values or (

Figure 14 (A) Map of the sedimentation rate and bathymetry of the Tropical Pacific. (A) Inferred sedimentation rate (Olson et 2016). White regions represent continental shelf. (B) GEBCO map of height relative to 0 m with location of seamounts plotted (white stars). (C) A binary colour map of the GEBCO data, yellow is values below cut-off depth value (3500 m below sea-level (bsl)) and purple above the cut-off depth value. See Supplementary Figure 8 for variation in cut-off values.

Figure 15 Overlay between bathymetry and FAME results. The results of the FAME Anderson-Darling test for (A) temperature and (B) oxygen isotope values as input. Locations where the $H_1$ hypothesis can be accepted, i.e-h) FAME-$T_c$ modelled temperature recorded in the calcite shells (Tc) occur. Each panel represents the Anderson-Darling test result, the results for *Globigerinoides sacculifer*, *Globigerinoides ruber* and *N. dutertrei* are overlaid. For all panels black areas reflect latitudinal and longitudinal grid points that failed to reject the null hypothesis ($H_0$) and therefore the foraminiferal population (FP) of the El Niño is similar to the Non-El Niño, and therefore the distribution between the neutral climate and El Nino cannot be said to be different ($FP_{El Niño} = FP_{Non-El Niño}$). The results in which the $H_1$ hypothesis is accepted, in which, the distributions can be said to be different ($FP_{El Niño} \neq FP_{Non-El Niño}$), are plotted as yellow where the depth is deeper than 3500 m bsl or purple where the depth is shallower than 3500 m bsl (see Figure 82). Purple locations are where our results suggest that the signal of ENSO has different values and the water depth allows for preservation—although this purple region will be smaller when inferred SAR is taken into account (see Figure 9). The rows represent the model runs with a depth cut-off limit at: (A and E) 60 m; (B and F) 100 m; (C and G) 200 m; and (D and H) where a combination of depths were utilised (Pracht et al., 2019)...

**Figure** 8. Bathymetric map of the Tropical Pacific Ocean highlighting the areas above and below the Lysocline and/or Calcite compensation depth (CCD). (A) GEBCO map of height relative to 0 m; (B) same as (A) with location of seamounts plotted (white stars); (C-E) binary colour map of GEBCO data, yellow is values below cut-off depth value ([C] 3500 m below sea-level (bsl); [D] 4000 m bsl; and [E] 4500 m bsl respectively) and purple above the cut-off depth value. The histograms represent the normalised frequency of grid cell height in bins of 250 m wide, yellow is values below cut off value ([C] 3500 m below sea-level (bsl); [D] 4000 m bsl; and [E] 4500 m bsl respectively), purple above cut off value. The grey bins in each histogram are those above 0 m.

[Figure]

5 Figure 9. Map of the sedimentation rate and oxygen saturation for the Tropical Pacific. (A) Inferred sedimentation rate (Olsen et 2016).16 White regions represent continental shelf. (B) Oxygen saturation of the bottom grid layer of World Ocean Atlas 2013 (data from: https://www.node.noaa.gov/cgi-bin/OC5/woa13/woa13oxnu.pl ). (C, E, G) Overlay between water depth and inferred SAR, (Olson et al., 2016). Cut-off limits for bathymetry and SAR are 3500 m below sea-level and (A) ≥1 cm kyr$^{-1}$ and (B) ≥2 cm kyr$^{-1}$ respectively. The colours represent the following: Red / Pink: Continental shelf sediments that are (Red) shallower or (Pink) deeper than 3500 mbsl; Grey / White: grid point SAR is lower than SAR threshold and the seafloor depth is (grey) shallower or (white) 10 deeper than 3500 mbsl; Light Yellow/Gold: Light yellow represents areas where the SAR is above the threshold but the water depth is deeper than 3500 mbsl in comparison Gold represents areas where the SAR is above the threshold and the water depth is deeper than 3500 mbsl. The ideal locations are therefore plotted as Gold. Cut-off limits for SAR are (C) ≥1 cm kyr$^{-1}$; (E) ≥2 cm kyr$^{-1}$ and (G) ≥5 cm kyr$^{-1}$, (D, F, H) alongside the maps the bioturbation simulations for the minimum SAR threshold is plotted (see

15 Figure 10 and Figure 11 for the output of SEAMUS). Each plot gives the input values of NGRIP (grey) and for each SAR three analysis were performed with different bioturbation depths (BD) these are (Blue) 5 cm; (Green) 10 cm; and (Orange) 15 cm.

[revised manuscript text omitted]

Footer1

| Page 1: [2] Style Definition | Brett Metcalfe | 21/02/2020 18:26:00 |
|---|---|---|

Header: Line spacing:  single, Tab stops:  7.96 cm, Centered +  15.92 cm, Right + Not at  8 cm +  16 cm

| Page 1: [3] Style Definition | Brett Metcalfe | 21/02/2020 18:26:00 |
|---|---|---|

Footnote Text: Font: (Default) Times New Roman, Justified

| Page 1: [4] Style Definition | Brett Metcalfe | 21/02/2020 18:26:00 |
|---|---|---|

Default: Adjust space between Latin and Asian text, Adjust space between Asian text and numbers

| Page 1: [5] Style Definition | Brett Metcalfe | 21/02/2020 18:26:00 |
|---|---|---|

Bibliography

| Page 1: [6] Style Definition | Brett Metcalfe | 21/02/2020 18:26:00 |
|---|---|---|

Revision

| Page 1: [7] Style Definition | Brett Metcalfe | 21/02/2020 18:26:00 |
|---|---|---|

EndNote Bibliography: Check spelling and grammar

| Page 1: [8] Style Definition | Brett Metcalfe | 21/02/2020 18:26:00 |
|---|---|---|

EndNote Bibliography Title: Check spelling and grammar

| Page 1: [9] Style Definition | Brett Metcalfe | 21/02/2020 18:26:00 |
|---|---|---|

Comment Subject

| Page 1: [10] Style Definition | Brett Metcalfe | 21/02/2020 18:26:00 |
|---|---|---|

Comment Text

| Page 1: [11] Style Definition | Brett Metcalfe | 21/02/2020 18:26:00 |
|---|---|---|

Equation

| Page 1: [12] Style Definition | Brett Metcalfe | 21/02/2020 18:26:00 |
|---|---|---|

Balloon Text

| Page 1: [13] Style Definition | Brett Metcalfe | 21/02/2020 18:26:00 |
|---|---|---|

List Paragraph

| Page 1: [14] Style Definition | Brett Metcalfe | 21/02/2020 18:26:00 |
|---|---|---|

Copernicus_Word_template

| Page 1: [15] Style Definition | Brett Metcalfe | 21/02/2020 18:26:00 |
|---|---|---|

Name

| Page 1: [16] Style Definition | Brett Metcalfe | 21/02/2020 18:26:00 |
|---|---|---|

Kontakt

| Page 1: [17] Style Definition | Brett Metcalfe | 21/02/2020 18:26:00 |
|---|---|---|

Bullets:  No bullets or numbering

| Page 1: [18] Style Definition | Brett Metcalfe | 21/02/2020 18:26:00 |
|---|---|---|

Betreff

| Page 1: [19] Style Definition | Brett Metcalfe | 21/02/2020 18:26:00 |
|---|---|---|

Heading 4: Font: (Default) +Headings (Times New Roman), Not Bold, Italic, Font color: Accent 1, Space Before:

2 pt, Keep lines together

| Page 1: [20] Style Definition | Brett Metcalfe | 21/02/2020 18:26:00 |
|---|---|---|

Heading 3: Font: (Default) +Headings (Times New Roman), 12 pt, Not Bold, Font color: Accent 1, Space Before:

2 pt, After:  0 pt, Line spacing:  1.5 lines, Keep lines together

| Page 1: [21] Style Definition | Brett Metcalfe | 21/02/2020 18:26:00 |
|---|---|---|

Heading 2: Font: (Default) +Headings (Times New Roman), 13 pt, Not Bold, Font color: Accent 1, Space Before:

2 pt, After:  0 pt, Line spacing:  1.5 lines, Keep lines together

| Page 1: [22] Style Definition | Brett Metcalfe | 21/02/2020 18:26:00 |
|---|---|---|

Heading 1: Font: (Default) +Headings (Times New Roman), 16 pt, Not Bold, Font color: Accent 1, Space Before:

12 pt, After:  0 pt, Line spacing:  1.5 lines, Keep lines together

| Page 1: [23] Formatted | Brett Metcalfe | 21/02/2020 18:25:00 |
|---|---|---|

Width:  21 cm

| Page 1: [24] Formatted | Brett Metcalfe | 21/02/2020 18:25:00 |
|---|---|---|

Font: Times New Roman

| Page 1: [25] Formatted | Brett Metcalfe | 21/02/2020 18:25:00 |
|---|---|---|

Font: Times New Roman

| Page 1: [26] Formatted | Brett Metcalfe | 21/02/2020 18:25:00 |
|---|---|---|

Font: Times New Roman

| Page 1: [27] Formatted | Brett Metcalfe | 21/02/2020 18:25:00 |
|---|---|---|

Font: Times New Roman

| Page 1: [28] Formatted | Brett Metcalfe | 21/02/2020 18:25:00 |
|---|---|---|

Font: Times New Roman

| Page 1: [29] Formatted | Brett Metcalfe | 21/02/2020 18:25:00 |
|---|---|---|

Font: Times New Roman

| Page 1: [30] Formatted | Brett Metcalfe | 21/02/2020 18:25:00 |
|---|---|---|

Font: Times New Roman

| Page 1: [31] Formatted | Brett Metcalfe | 21/02/2020 18:25:00 |
|---|---|---|

Font: Times New Roman

| Page 1: [32] Formatted | Brett Metcalfe | 21/02/2020 18:25:00 |
|---|---|---|

Font: Times New Roman

| Page 1: [33] Formatted | Brett Metcalfe | 21/02/2020 18:25:00 |
|---|---|---|

Font: Times New Roman

| Page 1: [34] Formatted | Brett Metcalfe | 21/02/2020 18:25:00 |
|---|---|---|

Font: Times New Roman

| Page 1: [35] Formatted | Brett Metcalfe | 21/02/2020 18:25:00 |
|---|---|---|

Font: Times New Roman

| Page 1: [36] Formatted | Brett Metcalfe | 21/02/2020 18:25:00 |
|---|---|---|

Font: Times New Roman

| Page 1: [37] Formatted | Brett Metcalfe | 21/02/2020 18:25:00 |
|---|---|---|

Font: Times New Roman, Not Bold

| Page 1: [38] Formatted | Brett Metcalfe | 21/02/2020 18:25:00 |
|---|---|---|

Heading 11

| Page 1: [39] Formatted | Brett Metcalfe | 21/02/2020 18:25:00 |
|---|---|---|

Font: Times New Roman

---

## Author Response (AR3)

**TECHNICAL COMMENTS**

Thank you for your further revision of this paper. You have made substantial changes to the paper after the last set of comments, and in particular the reorganisation of the paper into separate experiments has made the paper much easier to digest and understand. I am sure that if I sent the paper for re-review there would be further comments, but i think now we have taken the process as far as peer review can go: there would still be criticisms of your methodology, but your paper now contains enough caveats and explanations that a reader will be aware of the strengths and limitations of your approach. For that reason I have now decided to accept the paper without further review. Thank you for sticking with the process; I realise that it has been one that neither authors, reviewers or editors have enjoyed.

I have a small number of minor remaining comments that I'd ask you to address in your final version but I have decided hat they only constitute "technical" level correction. Please do look at these points though.

Page 8, line 5: Section starting "Though" isn't a sentence, please edit.

*Changed*

Fig 2 (and S6). Please expand on what you mean by "species specific results are overlain". Do you mean that in white areas all 3 species show a significant difference, or that at least one shows a significant difference. I think it's probably the latter, but the wording needs to make this clear.

*Changed – it is that at least one species shows a significant difference. We have changed the caption AND added a sentence pg. 10 line 8-9*

Page 13, line 25 "studies reliant have used" doesn't make sense, please edit.

*Removed 'reliant'*

Page 14. This section is very hard to read (a long list of numbers). I don't have specific suggestion but can it be simplified at al?

*Unfortunately I don't know how to simplify it further (other than replacing it with 'see Table 1').*

Table 1 and Fig 3b. The values labelled as "% difference" are actually neither percent (values between 0 and 100) nor differences, rather they are ratios. Please re-label.

*Changed*

Section 6 (expt 4) is really not very clear and I suspect adds little to the paper. However in the spirit that in the end it's your paper, and that others may be able to build the idea in it, I am not proposing any change.

*We did raise a similar point in the first round of review (i.e., that a picking experiment is not necessarily needed)  and we did consider removing it following the second round of reviews. However, as it was a request of the reviewers to include it, we have decided to retain it.*

[revised manuscript text omitted]